

# Global climate forcing driven by altered BVOC fluxes from 1990–2010 land cover change in maritime Southeast Asia

Kandice L. Harper[1] and Nadine Unger[2]

[1]School of Forestry & Environmental Studies, Yale University, New Haven, Connecticut, 06511, USA
[2]College of Engineering, Mathematics and Physical Sciences, University of Exeter, Exeter, EX4 4QJ, UK

*Correspondence to*: Kandice L. Harper (kandice.harper@yale.edu)

**Abstract.** Over the period 1990–2010, maritime Southeast Asia experienced large-scale land cover changes, including expansion of high-isoprene-emitting oil palm plantations and contraction of low-isoprene-emitting natural forests. The ModelE2-Yale Interactive Terrestrial Biosphere global chemistry–climate model is used to quantify the atmospheric
composition changes and, for the first time, the associated radiative forcing induced by the land-cover-change-driven biogenic volatile organic compound (BVOC) emission changes (+6.5 TgC $y^{-1}$ isoprene, -0.5 TgC $y^{-1}$ monoterpenes). Regionally, surface-level ozone concentrations largely decreased (-3.8 to +0.8 ppbv). The tropical land cover changes occurred in a region of strong convective transport, providing a mechanism for the BVOC perturbations to affect the composition of the upper troposphere. Enhanced concentrations of isoprene and its degradation products are simulated in the
upper troposphere, and, on a global-mean basis, land cover change had a stronger impact on ozone in the upper troposphere (+0.6 ppbv) than in the lower troposphere (< 0.1 ppbv increase). The positive climate forcing from ozone changes (+9.2 mW $m^{-2}$) was partially offset by a negative forcing (-0.8 mW $m^{-2}$) associated with a regional enhancement in secondary organic aerosol concentrations. The global-mean ozone forcing per unit of regional oil palm expansion is +1 mW $m^{-2}$ $Mha^{-1}$. In light of expected continued expansion of oil palm plantations, regional land cover changes may play an increasingly important
role in driving future global ozone radiative forcing.

## 1 Introduction

Recent decades have witnessed large-scale land cover and land use changes on the maritime continent. More than 4.5 Mha of natural forest were cleared in Indonesia alone over 2000–2010 (Margono et al., 2014). Increasing demand for palm oil, produced by oil palm trees (*Elaeis guineensis*), has simultaneously driven widespread expansion of this agro-industrial tree
crop (USDA, 2010). Indonesia and Malaysia cumulatively produce 85 % of the current global palm oil supply (USDA, 2017). The amount of land area planted in oil palm in Indonesia and Malaysia nearly quadrupled over 1990–2010, reaching 13 Mha by 2010 (Gunarso et al., 2013).



Above-canopy flux measurements indicate that, compared to the natural forests of maritime Southeast Asia (MSEA), oil palm plantations are much stronger emitters of the biogenic volatile organic compound (BVOC) isoprene ($C_5H_8$) (Langford et al., 2010; Misztal et al., 2011). The simultaneous large-scale contraction of low-isoprene-emitting natural forest area and expansion of high-isoprene-emitting oil palm plantations suggests a land-cover-change-driven increase in regional isoprene

emissions over recent decades (Silva et al., 2016; Stavrakou et al., 2014). Isoprene is a precursor to the short-lived climate pollutants tropospheric ozone (Atkinson and Arey, 2003) and secondary organic aerosols (SOA) (Carlton et al., 2009). Previous investigations of atmospheric composition changes driven by land use change in MSEA have largely focused on the surface-level air quality impacts induced by BVOC emission changes from oil palm expansion (Ashworth et al., 2012; Silva et al., 2016; Warwick et al., 2013).

BVOC perturbations in the tropics may have a particularly powerful impact on longwave radiative forcing (Unger, 2014) because the strong vertical mixing prevalent in the tropics provides a mechanism for surface pollution perturbations to impact ozone concentrations in the upper troposphere (Thompson et al., 1997), where, on a per-molecule basis, ozone changes induce the strongest climate impact (Lacis et al., 1990). This study uses the ModelE2-Yale Interactive Terrestrial

Biosphere (ModelE2-YIBs) global chemistry–climate model, in conjunction with multiple observational datasets, to quantify the global atmospheric composition changes and, for the first time, the concomitant radiative forcings associated with BVOC emission changes from 1990–2010 land cover change in MSEA. The calculations consider changes in emissions of not only isoprene, but also monoterpenes ($C_{10}H_{16}$), which are important SOA precursors (Jokinen et al., 2015). The applied regional land cover changes are derived from a Landsat-based classification (Gunarso et al., 2013) and account for changes in eight

land covers that are prevalent in MSEA, including high-monoterpene-emitting (Baker et al., 2005; Klinger et al., 2002) rubber trees (*Hevea brasiliensis*).

## 2 Data and methods

### 2.1 ModelE2-YIBs description

Atmosphere-only simulations employ the NASA GISS ModelE2-YIBs global chemistry–climate model. The YIBs model

(Yue and Unger, 2015) is a land surface model embedded in the NASA GISS ModelE2 global chemistry–climate model (Schmidt et al., 2014). The model features 2°-latitude × 2.5°-longitude horizontal resolution, 40 vertical layers (surface to 0.1 hPa), and a physical and chemical time step of 30 minutes.

The chemical mechanism includes 156 reactions involving 51 chemical species with full coupling of tropospheric and

stratospheric chemistry (Schmidt et al., 2014; Shindell et al., 2006). The troposphere features $NO_X$-$O_X$-$HO_X$-CO-$CH_4$ chemistry and a lumped hydrocarbon scheme (Houweling et al., 1998) based on the Carbon Bond Mechanism-4 (Gery et al., 1989). Global annual-mean mixing ratios are prescribed for the well-mixed greenhouse gases carbon dioxide ($CO_2$), methane





(CH$_4$), nitrous oxide (N$_2$O), and chlorofluorocarbons (CFCs) (Meinshausen et al., 2011; Riahi et al., 2007). Prescribed monthly anthropogenic and biomass burning emissions of reactive gas and primary aerosol species follow the MACCity emissions pathway (Angiola et al., 2010; Granier et al., 2011) for all years, except for 2010, when the interpolated ACCMIP-RCP8.5 dataset (Heil and Schultz, 2014) is applied for biomass burning emissions. The interactive production of nitrogen

oxides (NO$_X$) from lightning is tied to the model's moist convective cloud scheme (Price and Rind, 1992; Price et al., 1997). Other interactive emissions in ModelE2-YIBs include soil NO$_X$ (Yienger and Levy II, 1995), dust (Miller et al., 2006), sea salt particles and marine dimethyl sulfide (Koch et al., 2006), and the BVOCs isoprene (Arneth et al., 2007; Unger et al., 2013) and monoterpenes (Lathière et al., 2006).

Leaf-level gas exchange in ModelE2 (Collatz et al., 1991) couples the Farquhar–von Caemmerer kinetic model of photosynthetic CO$_2$ uptake (Farquhar et al., 1980; Farquhar and von Caemmerer, 1982) to the Ball–Berry model of stomatal conductance (Ball et al., 1987). The environmental inputs used to drive the vegetation biophysics and interactive BVOC emissions are the online values simulated by the general circulation model. The standard YIBs vegetation comprises eight plant functional types (PFTs): C3-grassland, C4-grassland, crop, deciduous broadleaf forest, evergreen broadleaf forest,

evergreen needleleaf forest, shrubland, and tundra. The model code was modified to include four additional land cover types: (1) oil palm plantations; (2) rubber tree plantations; (3) other tree plantations; and (4) dipterocarp evergreen broadleaf forest. Relative to the forests of the Amazon, the tropical forests of MSEA contain a larger proportion of evergreen dipterocarp forests (Hewitt et al., 2010). The dipterocarp evergreen broadleaf forest PFT was added to account for the comparatively lower isoprene emission capacity (Langford et al., 2010; Stavrakou et al., 2014) of the natural forests of MSEA. In each

model grid cell, an individual canopy is simulated for each PFT. The canopy radiative transfer scheme divides each canopy into a flexible number of vertical layers (generally 2–16). In each layer, sunlit leaves use direct photosynthetically active radiation (PAR) for leaf-level photosynthesis, while shaded leaves use diffuse PAR (Spitters et al., 1986).

YIBs features a process-based biochemical model of isoprene emission in which the rate of isoprene production depends on

the online electron-transport-limited rate of photosynthetic carbon assimilation (Arneth et al., 2007; Unger et al., 2013). For each PFT, the isoprene emission rate depends linearly on the fraction of electrons available to undergo isoprene synthesis, calculation of which requires prescription of a PFT-specific leaf-level isoprene basal emission rate (BER) at standard conditions. The leaf-level isoprene emission rate additionally depends on the model's online values of atmospheric CO$_2$ concentration, canopy temperature, and PAR. The ability of the model to simulate isoprene emissions has been evaluated

extensively against multiple above-canopy flux datasets from tropical and temperate ecosystems (Unger et al., 2013). The model simulated the local flux magnitude within a factor of two at nine specific measurement sites, some of which correspond to short (weeks-long) measurement campaigns (Unger et al., 2013). Temperature-dependent monoterpene emissions, functionally α-pinene, likewise vary by ecosystem type (Lathière et al., 2006). SOA formation is driven by NO$_X$-



dependent oxidation of interactive emissions of isoprene, monoterpenes, and other reactive VOCs following a volatility-based two-product scheme (Tsigaridis and Kanakidou, 2007).

Aerosol and gas-phase chemistry are fully coupled, and the chemical mechanism is fully coupled to the climate modules
(e.g., radiation and dynamics). All simulations apply ozone and aerosol climatologies to the radiation code (Schmidt et al., 2014), but online changes in ozone and aerosols can impact online photolysis rates. In this study, aerosols do not affect cloud properties. Observation-based, monthly-varying, five-year-average sea surface temperature and sea ice fields are prescribed according to the Hadley Centre Sea Ice and Sea Surface Temperature dataset (Rayner et al., 2003). Meteorological nudging of the large-scale atmospheric circulation is performed through application of the horizontal winds from the National Centers
for Environmental Prediction/National Center for Atmospheric Research Reanalysis 1 dataset (Kalnay et al., 1996).

**2.2 Vegetation datasets**

The land cover distributions for MSEA that are applied to the simulations are built from the results of a visual classification of Landsat images that additionally used high-resolution Google Earth images and other datasets (e.g., maps of roads) to aid image interpretation (Gunarso et al., 2013). The Gunarso et al. (2013) classification encompasses 22 land cover types in 30
m × 30 m pixels for the principal oil-palm-producing regions of maritime Southeast Asia – Papua New Guinea, Malaysia (Peninsular Malaysia, Sabah, and Sarawak), and three regions of Indonesia (Kalimantan, Papua, and Sumatra) – nominally for 1990, 2000, 2005, and 2010. The year 2010 Gunarso et al. (2013) dataset relies on some 2009 Landsat images due to intense cloudiness in 2010. As Gunarso et al. (2013) did not perform classification for Papua New Guinea for 2005, the areal coverage for the grid cells in this region for this year is estimated here through linear interpolation of the year 2000 and 2010
Gunarso et al. (2013) datasets. The Gunarso et al. (2013) classification for 1990 for Indonesia is likewise incomplete; in this dataset, the pixels classified for 1990 are principally those that belong to or eventually become the oil palm class. The Gunarso et al. (2013) classification for 2000 was applied to all Indonesian pixels that were not classified in 1990. As such, the 1990 Indonesian land cover map that is applied to the ModelE2-YIBs simulations is a hybrid of 1990 and 2000 data. While Indonesian oil palm cover in 1990 is accurate within the limits of the classification methodology, Indonesian forest
cover is presumably lower in the hybrid dataset than it was in reality in 1990, which means that Indonesian forest loss over the period 1990–2010 is likely underestimated in this study.

Using the 30 m × 30 m pixels from the Gunarso et al. (2013) analysis, including the estimates described above for 2005 in Papua New Guinea and 1990 in Indonesia, the areal coverage of each of the 22 land cover types was quantified for each 2°-
latitude × 2.5°-longitude ModelE2-YIBs grid cell and distributed on the ModelE2-YIBs land surface map. Seventeen of the twenty-two land cover types of the Gunarso et al. (2013) dataset were aggregated into a set of eight cover types (Table S1). Four of the cover types in the aggregated set (shrubland, crops, C4-grassland, and dirt) belong to the standard set of YIBs



cover types, while the other four cover types (dipterocarp forest, oil palm plantations, rubber plantations, and other tree plantations) are new cover types added to YIBs for this study. The aggregation and subsequent calculation of fractional areal coverage of land for these eight cover types ignored five minor land cover types from the Gunarso et al. (2013) classification: water bodies, coastal fish ponds, mining areas, settlements, and pixels that are unclassified due to clouds.

In the land cover dataset applied to YIBs, 1990–2010 forest loss is likely underestimated because (1) the 1990 Indonesian forest extent is partially derived from the Gunarso et al. (2013) year 2000 map due to unclassified land area in the Gunarso et al. (2013) year 1990 map and (2) deforestation in this dataset represents complete removal of a forest patch and does not account for forest degradation. The ModelE2-YIBs dipterocarp forest PFT combines the "disturbed" and "undisturbed"

forest classes from the Gunarso et al. (2013) analysis (Table S1); thus, reduction in forest basal area from partial logging of a forest does not register as forest loss in the YIBs dataset as long as the affected patch still meets the Gunarso et al. (2013) classification criteria for the disturbed forest class. An analysis of the leaf area index (LAI) of rainforest plots in Central Sulawesi, Indonesia, under different land use regimes found that disturbance of the forest by selective logging reduced the LAI below the 6.2 value measured for the undisturbed natural forest (Dietz et al., 2007). Removal of "small-diameter" trees

reduced LAI to 5.3, while removal of "large-diameter" trees reduced LAI to 5.0 (Dietz et al., 2007). Thus, disturbed (selectively logged) forests maintain a high LAI, suggesting that combining the disturbed and undisturbed forest classes into a single PFT is well justified for the purposes of this study.

The Landsat-based YIBs-compatible land cover distributions are applied to the 57 model grid cells covering Peninsular

Malaysia, Sumatra, Borneo, and New Guinea. The simulations apply non-zero areal extents of the four new land cover types only in MSEA. Table S2 shows, for the new land cover types, the assigned physical (Dietz et al., 2007; Fan et al., 2015; Fowler et al., 2011; Krisnawati et al., 2011; Legros et al., 2009; Li et al., 2016; Misztal et al., 2011; Scales and Marsden, 2008; Suratman et al., 2004), photosynthetic (Kositsup et al., 2009; Meijide et al., 2017; Rakkibu, 2008; Ritchie and Runcie, 2014), and BVOC emission (Baker et al., 2005; Cronn and Nutmagul, 1982; Geron et al., 2006; Kesselmeier and Staudt,

1999; Klinger et al., 2002; Llusia et al., 2014; Singh et al., 2014) parameters.

The static land cover distribution applied to the rest of the world is the year 2000 land cover distribution developed for the Community Land Model (CLM; Oleson et al., 2010) using multiple satellite datasets, including retrievals from both MODIS (Hansen et al., 2003) and AVHRR (DeFries et al., 2000). The 16-PFT data were aggregated into the standard set of eight

YIBs PFTs (Yue and Unger, 2015). Gridded PFT-specific LAI and vegetation height parameters are prescribed. For the rest-of-world vegetation, prescribed LAI are derived from CLM (Oleson et al., 2010); and prescribed heights are the output of a 140-year ModelE2-YIBs simulation (Yue and Unger, 2015) that simulated dynamic carbon allocation, used the same CLM land cover distribution described here, and was forced with year 2000 meteorology from the WFDEI (WATCH Forcing Data methodology applied to ERA-Interim data; Weedon et al., 2014) dataset.



Table 1 shows the areal extents of YIBs land covers in MSEA for 1990, 2005, and 2010. Over 1990–2010, 11.3 Mha of natural rainforest was lost (-8% relative to 1990). Forest loss was widespread on Borneo, Sumatra, and Peninsular Malaysia (Figure S1). Contraction of rubber plantations (-1.4 Mha) was primarily confined to Sumatra and Peninsular Malaysia. The high-isoprene-emitting oil palm class experienced the largest absolute increase in areal extent over the study era (+9.6 Mha, +267%). Widespread expansion occurred on Sumatra, Borneo, and Peninsular Malaysia.

**Table 1**: Areal extents (Mha) of eight YIBs land cover types in maritime Southeast Asia. Extents encompass only the 57 grid cells that apply the land cover distributions derived from the Gunarso et al. (2013) analysis. The changes in areal extent (in Mha) relative to 1990 are listed in parentheses for 2005 and 2010.

| YIBs land cover | 1990 | 2005 | 2010 |
|---|---|---|---|
| Shrubland | 29.7 | 30.5 (+0.8) | 30.8 (+1.1) |
| Crops | 10.6 | 11.3 (+0.7) | 13.1 (+2.6) |
| C4-grassland | 3.2 | 2.9 (-0.3) | 2.9 (-0.3) |
| Bare | 2.8 | 3.3 (+0.5) | 3.6 (+0.8) |
| Oil palm plantations | 3.6 | 9.8 (+6.3) | 13.2 (+9.6) |
| Rubber plantations | 7.8 | 6.7 (-1.1) | 6.4 (-1.4) |
| Other tree plantations | 14.0 | 13.7 (-0.3) | 12.9 (-1.1) |
| Dipterocarp rainforest | 140.6 | 134.0 (-6.7) | 129.3 (-11.3) |

## 2.3 Simulation configurations

Table 2 summarizes the configurations of nine global chemistry–climate simulations. Two principal time-slice simulations – 2010land_base and 1990land_base – are used to diagnose the global-mean radiative perturbation associated with the atmospheric composition changes induced by 1990–2010 land cover change in MSEA. The two simulations differ only in terms of the year of the applied maritime Southeast Asian land cover distribution, which is indicated in the simulation names. The regional land cover changes are imposed on a background climate and atmosphere representative of year 2010; that is, both the 2010 land cover distribution in the perturbation simulation and the 1990 land cover distribution in the control simulation are exposed to the climate state, atmospheric $CO_2$ concentration, and background atmosphere representative of year 2010. The present-day rest-of-world land cover distribution is identical for all simulations. All simulations were run for 13 years, and averages over the final 10 years of output were used for analysis. The impact of 1990–2010 maritime Southeast Asian land cover change, denoted ΔLC, on atmospheric composition and radiative balance is diagnosed as 2010land_base minus 1990land_base (Table 3). Seven additional simulations (Tables 2 and 3) probe the sensitivity of the radiative forcing results to: (1) the applied background atmosphere; (2) the degree of regional land cover change; (3) the magnitude of the



isoprene BER assigned to the oil palm plantation PFT; and (4) the magnitude of the isoprene BER assigned to the dipterocarp forest PFT.

Because isoprene production in ModelE2-YIBs is interactively linked to photosynthetic carbon assimilation (Unger et al., 2013), isoprene emissions are sensitive to simulated changes in the parameters that affect photosynthesis, including the background climate state and atmospheric $CO_2$ concentration. Interactive monoterpene emissions are likewise sensitive to climate shifts (Lathière et al., 2006). Following emission, the atmospheric processing of the BVOCs is influenced by the background atmospheric composition; for example, the availability of $NO_X$ affects the VOC-driven production of both ozone (Sillman, 1999) and SOA (Tsigaridis and Kanakidou, 2007). The calculation 2010land_1990atm minus 1990land_1990atm, denoted ΔLC-1990atm, is designed to test the influence of the prescribed background state upon which the maritime Southeast Asian land cover changes are imposed (Table 3). Relative to the main set of simulations probing Southeast Asian land cover change (i.e., ΔLC), the ΔLC-1990atm simulations prescribe identical changes in Southeast Asian land cover, but impose the changes on a background state representative of year 1990 rather than year 2010 (Table 2). The different background states can lead to different BVOC-driven impacts on ozone and SOA concentrations from the identical prescribed land cover changes by influencing (1) the magnitude and distribution of BVOC emission changes associated with the land cover changes and (2) the atmospheric processing of the emitted BVOCs.

A third pair of simulations (2005land_base minus 1990land_base, denoted ΔLC-2005) quantifies the global impacts of 1990–2005 maritime Southeast Asian land cover change (Table 3). Because the land cover distributions in both the control and perturbation simulations experience 2010 background conditions, ΔLC-2005 represents the impacts in 2010 that would have been expected had land cover remained static after 2005. Because land cover in MSEA is rapidly changing and there is uncertainty associated with the classification procedures used to create the land cover maps (Sect. 2.2), such information provides insight into the degree to which the land-cover-change-driven forcing depends on the exact land cover distribution that is applied.

The fourth pair of simulations (2010land_OPber minus 1990land_OPber, denoted ΔLC-OPber) probes the sensitivity of the land-cover-change-induced impacts on atmospheric composition and climate to the magnitude of the isoprene BER applied to oil palm plantations (Table 3). Halving the isoprene BER applied to oil palm plantations brings the simulated 24-h mean isoprene emission rate from oil palm plantations for 2010 close to the rate measured using above-canopy fluxes (described in Sect. 3.3). Similarly, the fifth pair of simulations (2010land_DPTber minus 1990land_DPTber, denoted ΔLC-DPTber) probes the sensitivity of the radiative forcing from 1990–2010 land cover change to the magnitude of the isoprene BER applied to dipterocarp forests. In this sensitivity analysis, the dipterocarp forest BER is increased by a factor of 12, making it equivalent to the isoprene BER assigned to the standard evergreen broadleaf forest PFT in YIBs (Table S2).



**Table 2**: Simulation configurations.

| Simulation | Southeast Asian land cover | Other boundary conditions[a] | Isoprene BER for oil palm | Isoprene BER for dipterocarp forest |
|---|---|---|---|---|
| 2010land_base | 2010 | 2010 | Measured | Measured |
| 1990land_base | 1990 | 2010 | Measured | Measured |
| 2010land_1990atm | 2010 | 1990 | Measured | Measured |
| 1990land_1990atm | 1990 | 1990 | Measured | Measured |
| 2005land_base | 2005 | 2010 | Measured | Measured |
| 2010land_OPber | 2010 | 2010 | ½ × measured | Measured |
| 1990land_OPber | 1990 | 2010 | ½ × measured | Measured |
| 2010land_DPTber | 2010 | 2010 | Measured | 12 × measured |
| 1990land_DPTber | 1990 | 2010 | Measured | 12 × measured |

a) The non-land-cover variable boundary conditions include: well-mixed greenhouse gas ($CH_4$, $CO_2$, $N_2O$, and halocarbon) mixing ratios applied to model radiation; $CH_4$, $N_2O$, and halocarbon mixing ratios applied to atmospheric chemistry; $CO_2$ mixing ratio applied to the land biosphere; large-scale winds; reactive emissions of carbon monoxide (CO), ammonia ($NH_3$), $NO_X$, sulfur dioxide ($SO_2$), non-methane volatile organic compounds (NMVOCs), black carbon (BC), and organic carbon (OC) from anthropogenic and biomass burning sectors; sea surface temperatures; and sea ice fields. Monthly sea ice and sea surface temperature fields are five-year averages centered on the simulation year.

**Table 3**: Calculation methodology. MSEA LCC = maritime Southeast Asian land cover change. The control simulation varies between pairs of simulations.

| Identifier | Calculation | Description |
|---|---|---|
| ΔLC | 2010land_base – 1990land_base | Impact of 1990–2010 MSEA LCC imposed on 2010 background atmosphere |
| ΔLC-1990atm | 2010land_1990atm – 1990land_1990atm | Impact of 1990–2010 MSEA LCC imposed on 1990 background atmosphere |
| ΔLC-2005 | 2005land_base – 1990land_base | Impact of 1990–2005 MSEA LCC imposed on 2010 background atmosphere |
| ΔLC-OPber | 2010land_OPber – 1990land_OPber | Impact of 1990–2010 MSEA LCC imposed on 2010 background atmosphere using halved isoprene BER for oil palm plantations |
| ΔLC-DPTber | 2010land_DPTber – 1990land_DPTber | Impact of 1990–2010 MSEA LCC imposed on 2010 background atmosphere using enhanced (12×) isoprene BER for dipterocarp forests |



# 3 Results and discussion

## 3.1 Gross primary production and BVOC emissions

Simulated global gross primary production (GPP) for 2010 is 124 PgC $y^{-1}$ (simulation 2010land_base), which almost precisely matches an estimate derived from flux-tower measurements (123 ± 8 PgC $y^{-1}$; Beer et al., 2010). Guenther et al.

(2012) collated contemporary global annual BVOC emission estimates from the literature, finding ranges of 309–706 TgC $y^{-1}$ for isoprene and 26–156 TgC $y^{-1}$ for monoterpenes. The model estimates for 2010 (363 TgC $y^{-1}$ isoprene and 77 TgC $y^{-1}$ monoterpenes) fall well within these ranges. Using the same process-based, leaf-level isoprene production algorithm employed in ModelE2-YIBs, although driven with different forcing datasets, Hantson et al. (2017) predict contemporary isoprene emissions (385 TgC $y^{-1}$; 1971–2000 mean) that are only 6 % higher than those predicted here for 2010.

In MSEA, 2010 BVOC emission rates are higher on Sumatra, Peninsular Malaysia, and Borneo than on the island of New Guinea (Figure S2), which maintains high areal coverage of low-isoprene-emitting dipterocarp rainforests. Rubber plantations contributed 56 % (1.6 TgC $y^{-1}$) of the regional monoterpene emissions in 2010, while oil palm (8.8 TgC $y^{-1}$) and shrubs (5.8 TgC $y^{-1}$) dominated regional isoprene emissions (55 % and 36 %, respectively). The low-isoprene-emitting

dipterocarp rainforests, which covered 61 % of the region's land surface in 2010, were responsible for only 8 % of regional isoprene emissions. The strong contributions made by rubber and oil palm plantations to the regional BVOC budget underscore the importance of explicitly accounting for these high-emitting species in regional land use and land cover change analyses.

Regional 1990–2010 land cover change induced a negligible decrease in regional GPP (-0.1 PgC $y^{-1}$), which is principally attributed to an increase from oil palm (+0.3 PgC $y^{-1}$) and a decrease from dipterocarp rainforests (-0.4 PgC $y^{-1}$). Regional land cover change induced annual emissions changes of +6.5 TgC $y^{-1}$ isoprene and -0.5 TgC $y^{-1}$ monoterpenes. The land-cover-change-driven net increase in regional isoprene emissions is almost entirely due to expansion of industrial oil palm plantations (+6.4 TgC $y^{-1}$, +271 % relative to 1990 oil palm isoprene emissions). Regional isoprene emissions from shrubs

increased by 4.2 % (+0.2 TgC $y^{-1}$), associated with a 3.7 % increase in shrubland. The large loss of dipterocarp rainforest had little impact on isoprene emissions (-0.1 TgC $y^{-1}$), as this PFT is a weak isoprene emitter. Contraction of rubber plantation extent was largely responsible for the reduction in monoterpene emissions (-0.4 TgC $y^{-1}$).

## 3.2 Atmospheric composition

Low surface ozone concentrations are simulated for the pristine atmospheres of the tropical forests. In the Southeast Asia

study region in 2010 (2010land_base), the less disturbed landscapes of Borneo and New Guinea exhibit lower surface ozone concentrations than the comparatively more industrialized regions of Sumatra and Peninsular Malaysia (Figure S3). Considering all grid cells in the 57-grid-cell study area that are majority (> 50 %) forest by area, simulated annual-mean





surface ozone concentrations in 2010 are 9.0 ppbv in New Guinea (n=10) and 9.5 ppbv in Borneo (n=7). Considering only the grid cells that are > 85 % forest by area, simulated ozone concentrations are 7.8 ppbv in New Guinea (n=4) and 7.2 ppbv in Borneo (n=2). Measurements at a rainforest site in Malaysian Borneo in 2008 found daytime surface-level ozone mixing ratios of 5–8 ppbv (Hewitt et al., 2010), providing support for the low ozone concentrations simulated over the highly

forested regions of the study area.

1990–2010 land cover change in MSEA drove a reduction in annual-mean surface ozone concentrations over Borneo, Peninsular Malaysia, and Sumatra (Figure 1). Negligible changes occurred over New Guinea. Regionally, small surface ozone enhancements are simulated over the marine environment, with maximum enhancements occurring over the ocean to

the west of Sumatra. Because this region exhibits low surface ozone concentrations, the small absolute changes (-3.8 to +0.8 ppbv) translate into comparatively large relative changes (-18.3% to +4.3%).

Simulated reductions in surface ozone largely occur in the regions of enhanced isoprene emissions, specifically Peninsular Malaysia, Sumatra, and Borneo (Figure S4). Surface ozone reductions occurring in response to enhanced VOC emissions in

low-$NO_X$ conditions are associated with an increase in the relative importance of ozone destruction reactions (e.g., direct reaction of the VOC with ozone) (Ashworth et al., 2012; Sillman, 1999). $NO_X$ surface emissions in the analysis region were, on average, 0.036 mgN m$^{-2}$ h$^{-1}$ in 2010. Based on atmospheric chemical modeling in conjunction with aircraft and ground measurements in Borneo, $NO_X$ fluxes for 2008 were inferred as 0.009 mgN m$^{-2}$ h$^{-1}$ over a rainforest site and 0.019 mgN m$^{-2}$ h$^{-1}$ over an oil palm plantation (Hewitt et al., 2009). The applied $NO_X$ emissions are slightly higher than, but the same order

of magnitude as, the observation-based inferred fluxes, providing support for the "$NO_X$-sensitive regime" (Sillman, 1999) that is modeled in this study.

Previous studies have shown that the sign and strength of the regional surface ozone response to increased isoprene strongly depend on availability of $NO_X$ (Ashworth et al., 2012; Silva et al., 2016; Warwick et al., 2013). When applying

contemporary $NO_X$ emissions from published inventories, both Warwick et al. (2013) and Ashworth et al. (2012) simulated local surface ozone reductions in response to increased isoprene emissions from regional oil palm expansion. The Warwick et al. (2013) study was an idealized simulation that carpeted Borneo in oil palm, while the Ashworth et al. (2012) study applied future projections of oil palm expansion for biofuel production. Both studies found that inflating the $NO_X$ emissions in the region of land conversion enhanced surface ozone concentrations (Ashworth et al., 2012; Warwick et al., 2013). In a

study focused on estimating the air quality impacts associated with 2010 oil palm cover compared to a no-oil-palm landscape, Silva et al. (2016) simulated increased surface ozone concentrations over much of the region. They found that some low-$NO_X$ regions (e.g., parts of Borneo) exhibited surface ozone reductions in response to increased isoprene emissions (Silva et al., 2016). The differences in the simulated impacts on regional surface ozone between this study and the Silva et al. (2016) study are likely due to the magnitude of the applied regional $NO_X$ emissions. The simulated changes in





atmospheric composition might be a response not only to altered BVOC emissions, but also to land-cover-change-induced perturbations in the deposition of atmospheric species. Silva et al. (2016) investigated this possibility, but found that BVOC emission changes were almost exclusively responsible for the simulated surface ozone changes.

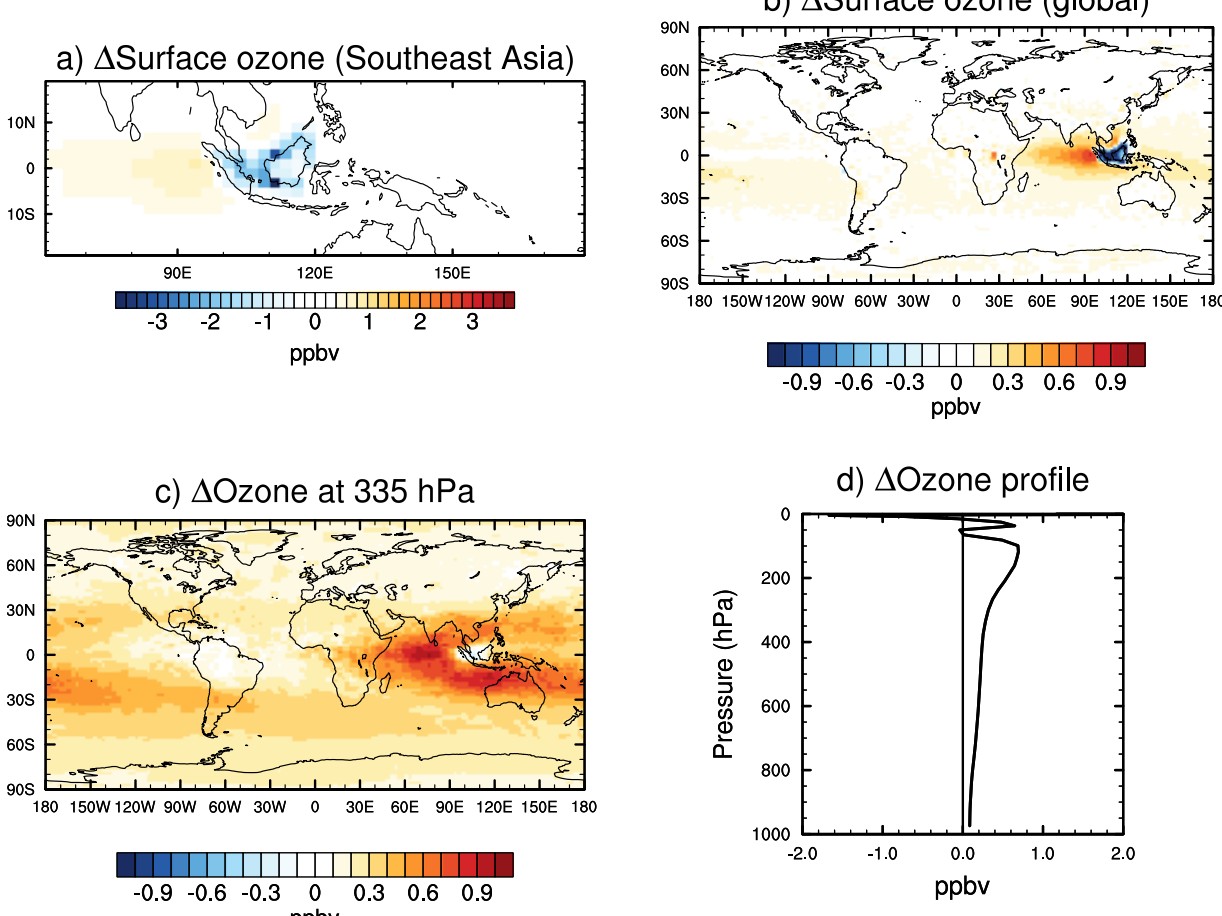

**Figure 1:** Changes in annual-mean ozone mixing ratio (ppbv) due to 1990–2010 maritime Southeast Asian land cover change: a) Southeast Asian surface-level ozone; b) global surface-level ozone; c) tropospheric ozone at 335 hPa; and d) global-mean ozone profile.





While the strongest impacts of regional land cover change on annual-mean surface ozone are confined to Southeast Asia, weak long-range enhancements are simulated, particularly over the tropical ocean (< 1 ppbv over Indian Ocean, < 0.5 ppbv elsewhere) (Figure 1). With decreasing atmospheric pressure, the long-range impact on ozone spreads beyond the tropics and generally grows in magnitude (Figure 1). The global-mean ozone enhancement increases in magnitude as altitude increases

from the surface to ~100 hPa (Figure 1). The global-mean ozone enhancement from regional land cover change is on the order of 0.5 ppbv in the upper troposphere, compared to < 0.1 ppbv in the lower troposphere. The maximum relative change in the global-mean ozone mixing ratio is +0.6 %, occurring at 160 hPa.

The land-cover-change-driven maritime Southeast Asian BVOC perturbations occur in a region of deep convection (Folkins

et al., 1997). The co-location of emissions perturbations and strong vertical transport results in changes in chemical composition throughout the atmosphere (Figure 2). The strongest zonal-mean enhancements in isoprene occur in the lower troposphere at pressures > 850 hPa (maximum = +88 %, occurring near the equator). As a doubly unsaturated hydrocarbon, isoprene is highly reactive in the oxidizing atmosphere. Most of the increased isoprene is rapidly oxidized in the lower atmosphere, yet some is transported to the middle and upper troposphere by strong tropical convection. The isoprene

enhancements are generally weaker in the middle and upper troposphere relative to the lower troposphere; considering pressures < 850 hPa, the relative change peaks at +37 % at 139 hPa, 5 °N.

Upper-tropospheric enhancements in isoprene have been predicted by global model simulations (Collins et al., 1999) and have been observed in convective events over Canada (Apel et al., 2012). Upper-tropospheric ozone enhancements are

predicted when transport-driven isoprene enhancements occur in the presence of lightning $NO_X$ emissions (Apel et al., 2012). Lightning $NO_X$ is prevalent in the tropical upper troposphere (Beirle et al., 2010; Boersma et al., 2005), and annual-mean upper-tropospheric ozone enhancements are simulated in this study. The largest zonal-mean tropospheric ozone changes induced by regional land cover change occur in the tropical near-tropopause region (Figure 2), which is particularly important from a climate perspective because the climate impact of a unit of tropospheric ozone change increases with

altitude and is maximized at the tropopause (Lacis et al., 1990). The maximum zonal-mean tropospheric ozone enhancement is +1.4 ppbv (99 hPa, -5 °N).

Formaldehyde (HCHO) is a high-yield oxidation product of isoprene (Wolfe et al., 2016). In response to maritime Southeast Asian land cover change, enhanced zonal-mean formaldehyde mixing ratios are simulated along the equator from the surface

to the upper troposphere (Figure 2). The strongest changes occur in the lower troposphere, with weaker enhancements in the middle and upper troposphere. Increased upper-tropospheric formaldehyde can result from direct convection of formaldehyde or from in situ production following convection of its precursors (e.g., isoprene).





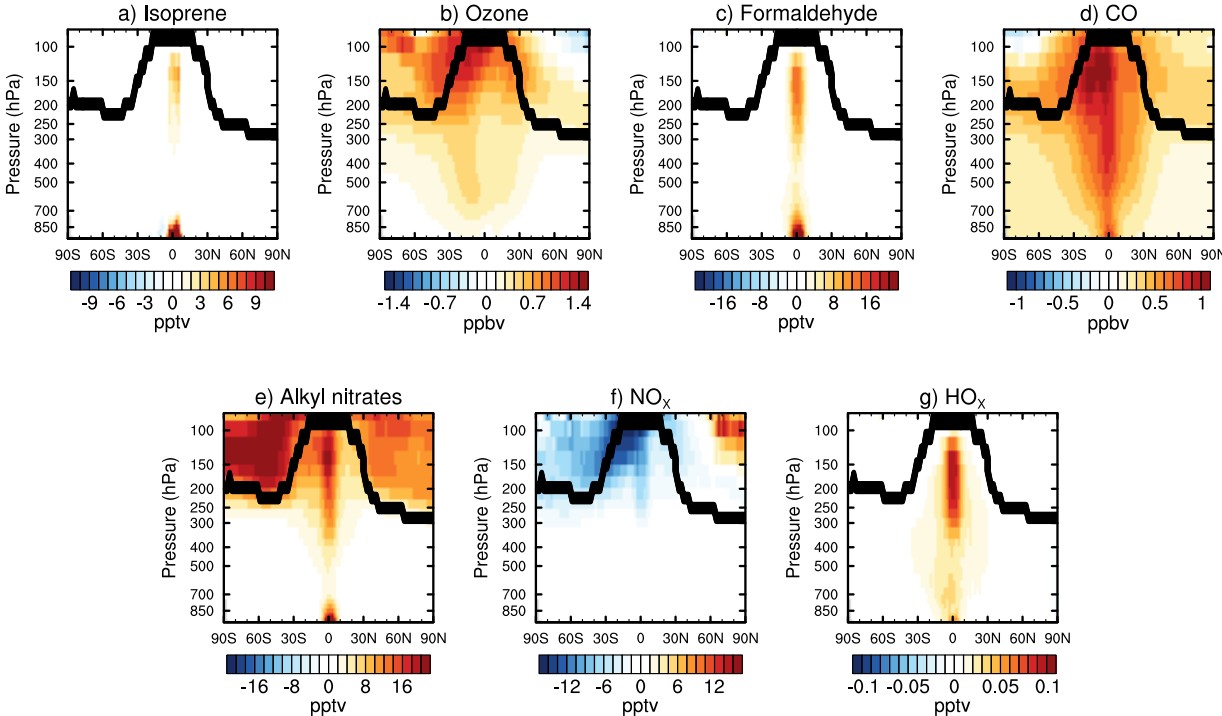

**Figure 2:** Changes in annual zonal-mean concentrations of a) isoprene, b) ozone, c) formaldehyde, d) CO, e) alkyl nitrates, f) $NO_X$, and g) $HO_X$ from 1990–2010 maritime Southeast Asian land cover change. Black traces indicate tropopause location.

Laboratory measurements suggest that, in high-$NO_X$ conditions, 60 % of isoprene-derived carbon in the atmosphere eventually becomes CO (Miyoshi et al., 1994). Zonal-mean CO enhancements are simulated for most of the atmosphere, with peak upper-tropospheric enhancements > 1 ppbv (Figure 2). CO is itself an ozone precursor, contributing to the simulated ozone enhancements. The isoprene and formaldehyde change signatures suggest vertical transport in the absence of significant horizontal mixing, which is expected given their short lifetimes on the order of hours. With a lifetime on the order of a few months, CO can undergo significant mixing before being destroyed, explaining why its perturbations are more widely distributed throughout the atmosphere.

In OH-initiated oxidation of isoprene in $NO_X$-rich environments, isoprene nitrates can form from direct reaction of isoprene peroxy radicals with nitric oxide (NO) (Lockwood et al., 2010). In ModelE2-YIBs, 15 % of isoprene + OH reactions in high-$NO_X$ conditions produce a primary alkyl nitrate molecule (alkyl nitrates can also be formed from other oxidation pathways)





(Shindell et al., 2003). Peak zonal-mean alkyl nitrate enhancements are simulated as +19 % in the lower troposphere, +5 % in the upper troposphere, and +4 % in the stratosphere (Figure 2). Stable alkyl nitrates are $NO_X$ reservoirs – they sequester reactive $NO_X$ upon formation and undergo long-range transport, eventually releasing the $NO_X$ far from the source region (Atherton, 1989). Thus, alkyl nitrates provide a mechanism for ozone formation far from the site of the initial hydrocarbon

perturbation (e.g., promoting enhanced surface ozone over the tropical ocean, Figure 1). Cold temperatures extend the lifetime of alkyl nitrates in the upper troposphere (Apel et al., 2012), resulting in alkyl nitrate perturbations that are more strongly mixed throughout the upper troposphere and lower stratosphere relative to the lower and middle troposphere (Figure 2). Sequestration of the reactive $NO_X$ into the alkyl nitrates contributes to the simulated reduction in $NO_X$ mixing ratios in the upper troposphere and lower stratosphere (Figure 2).

The land-cover-change-driven BVOC perturbations likewise impact the concentrations of the $HO_X$ (OH + $HO_2$) radical family. Collins et al. (1999) found that enhanced convection of isoprene and its oxidation products to the upper troposphere enhances $HO_X$ production, principally through increased photolysis of the oxidation products, including formaldehyde. Here, annual zonal-mean $HO_X$ concentrations are enhanced in the tropics from the surface to the upper troposphere in a pattern that

largely mimics that of the formaldehyde perturbations (Figure 2).

Cumulatively, these changes demonstrate that land cover change in MSEA has the capacity to alter the chemical composition of the upper troposphere. Increased BVOC emissions in a region of deep convection results in enhanced upper-tropospheric concentrations of isoprene and its degradation products, which contributes to enhanced ozone mixing ratios in the

climatically important near-tropopause region.

**3.3 Radiative forcing**

Instantaneous radiative forcings are reported as mean ± 1 standard deviation, calculated over 10 model years. The uncertainties represent internal model variability. In this study, the quantified radiative perturbations arise specifically from ozone and aerosol changes driven by regional land cover change. Only direct aerosol–radiation interactions are considered

for aerosols.

The atmospheric composition changes induced by 1990–2010 maritime Southeast Asian land cover change resulted in a positive annual global-mean radiative forcing of +8.4 ± 0.7 mW m$^{-2}$ (Table 4). The global ozone perturbation induced a positive forcing of +9.2 ± 0.7 mW m$^{-2}$, offset only slightly by a negative forcing (-0.8 ± 0.1 mW m$^{-2}$) induced by a regional

enhancement in largely reflective SOA particles (Figure S5). The ozone radiative forcing distribution (Figure 3; range: -10.4 to +37.6 mW m$^{-2}$) largely reflects the pattern of mid- and upper-tropospheric ozone changes (Figure 1), such that the magnitude of the positive forcing largely tracks the magnitude of the ozone enhancement, particularly in the tropics. On a per-molecule basis, enhancements in ozone have a stronger impact on longwave forcing the nearer they are to the tropopause



(Lacis et al., 1990), which explains the close association between the patterns of upper tropospheric ozone changes and the ozone forcing magnitude. The strongest ozone forcings occur over the tropical Indian Ocean.

5 **Table 4**: Global annual-mean radiative forcing (mW m$^{-2}$) from changes in atmospheric composition induced by maritime Southeast Asian land cover change. Mean ± 1 standard deviation, calculated over 10 model years.

| Radiative forcing (mW m$^{-2}$) | ΔLC | ΔLC-1990atm | ΔLC-OPber | ΔLC-DPTber | ΔLC-2005 |
|---|---|---|---|---|---|
| Ozone | +9.2 ± 0.7 | +8.8 ± 0.7 | +4.3 ± 0.7 | +6.2 ± 0.7 | +6.3 ± 0.7 |
| SOA | -0.8 ± 0.1 | -0.6 ± 0.1 | -0.3 ± 0.1 | -0.5 ± 0.1 | -0.5 ± 0.2 |
| Total | +8.4 ± 0.7 | +8.2 ± 0.7 | +4.0 ± 0.7 | +5.7 ± 0.7 | +5.8 ± 0.6 |

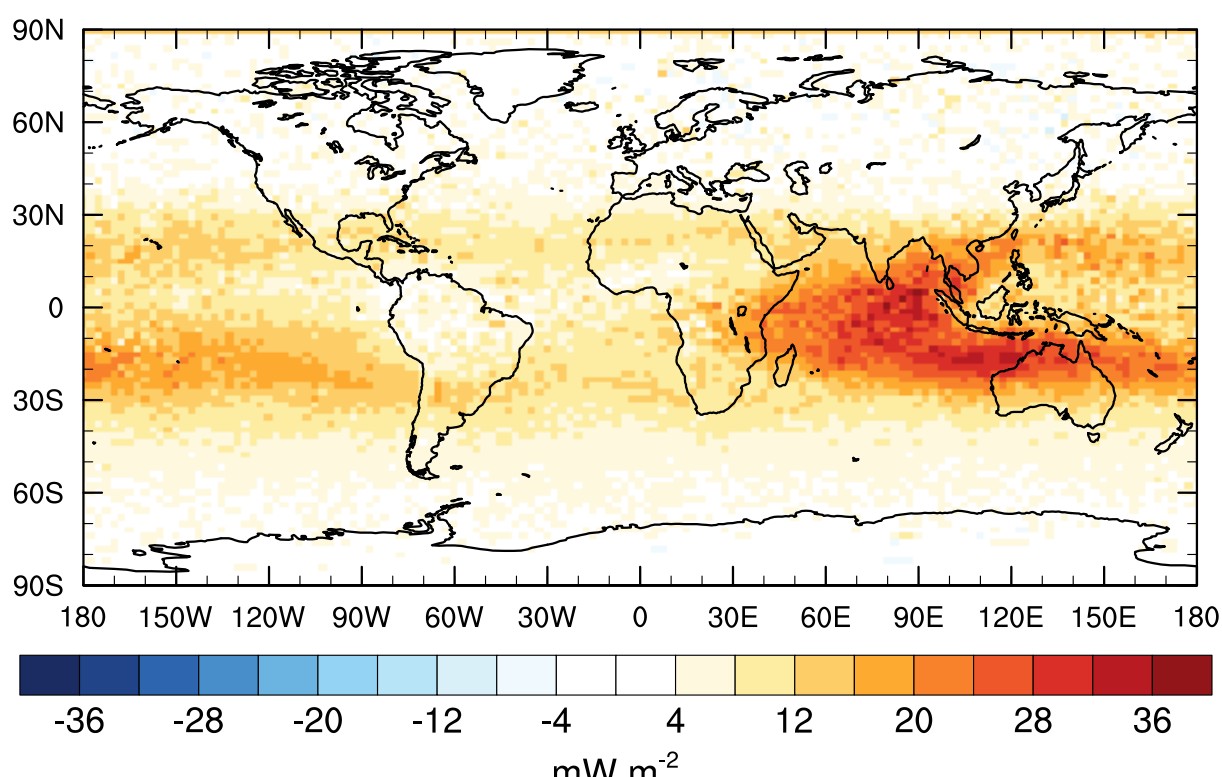

**Figure 3:** Annual-mean ozone forcing (mW m$^{-2}$) due to 1990–2010 maritime Southeast Asian land cover change.





The sensitivity studies investigate the uncertainty in the forcing magnitude. For all five drivers, the negative SOA forcing is much smaller in magnitude than the positive ozone forcing, resulting in a total forcing that is positive in sign (Table 4). This consistency provides confidence in the positive sign of the total forcing induced by land-cover-change-driven perturbations in atmospheric composition. Of all of the sensitivity analyses, ΔLC-OPber results in the smallest magnitude forcing for both

ozone and SOA (Table 4). The oil palm isoprene BER applied to the simulations used for the ΔLC-OPber calculations was half the magnitude of the published leaf-level BER (Table S2) applied to the simulations used for the ΔLC calculations.

Application of the halved BER brings the simulated 24-h mean isoprene emission rate from oil palm plantations for 2010 (3.8 mgC m$^{-2}$ h$^{-1}$ in simulation 2010land_OPber vs. 7.6 mgC m$^{-2}$ h$^{-1}$ in simulation 2010land_base) in line with the rate

measured in 2008 above an industrial oil palm estate in Malaysian Borneo (3.9 mgC m$^{-2}$ h$^{-1}$; Misztal et al., 2011). However, the measured flux, based on a short 12-day measurement window, is sensitive to the prevailing meteorological conditions (Misztal et al., 2011). Misztal et al. (2011) note that the isoprene flux would have been twice as high under the meteorological conditions (PAR and temperature) that they measured during the period prior to initiation of their flux measurements; such a rate would match the rate calculated for the simulation that applied the published leaf-level BER

(2010land_base). Thus, the isoprene flux measurements provide confidence in the order of magnitude of the simulated isoprene emissions from oil palm plantations. The ozone forcing calculated using the ΔLC-OPber sensitivity analysis (+4.0 mW m$^{-2}$) is considered to be the lower bound of a best estimate of the global-mean ozone forcing from maritime Southeast Asian land cover change.

Increasing the isoprene BER for the dipterocarp forest PFT by a factor of 12 increases the magnitude of the 1990–2010 isoprene emissions reduction associated with the large contraction in areal cover of this PFT (-1.3 TgC y$^{-1}$ for ΔLC-DPTber vs. -0.1 TgC y$^{-1}$ for ΔLC). This decreases the magnitude of the net enhancement in isoprene emissions from total land cover change in the study region (+5.3 TgC y$^{-1}$ for ΔLC-DPTber vs. +6.5 TgC y$^{-1}$ for ΔLC). The positive ozone forcing for ΔLC-DPTber is still two-thirds of the magnitude of the ozone forcing for ΔLC (Table 4), despite the factor-of-12 enhancement in

the assigned dipterocarp forest BER. Using the 2010 simulation that applies the dipterocarp forest isoprene BER taken from leaf-level measurements (2010land_base simulation), the simulated 24-h mean emission rate of isoprene from maritime Southeast Asian rainforests for 2010 (0.12 mgC m$^{-2}$ h$^{-1}$) is one-third of the eddy-covariance-based measured emission rate for 2008 from a natural forest in Malaysian Borneo (0.35 mgC m$^{-2}$ h$^{-1}$; Langford et al., 2010). The measured flux was shown to be highly sensitive to the meteorological conditions (e.g., wet season flux = 0.22 mgC m$^{-2}$ h$^{-1}$, dry season flux = 0.47 mgC

m$^{-2}$ h$^{-1}$; Langford et al., 2010). The results of ΔLC-DPTber suggest that increasing the dipterocarp forest BER by a factor of roughly three (to force alignment of the simulated isoprene emission magnitude with the 2008 measurements) would have little impact on the total forcing magnitude (because increasing the BER by a factor of twelve had only a slight impact on the forcing). Because the isoprene emissions capacity of oil palms is so strong relative to that of the dipterocarp forest, the isoprene emissions changes from 1990–2010 regional land cover change are dominated by the oil palm PFT. The total land-



cover-change-driven forcing is more sensitive to uncertainty in the magnitude of the oil palm BER than to uncertainty in the magnitude of the dipterocarp forest BER. The limited importance of the isoprene emission changes from dipterocarp forest loss additionally indicates that the potential underestimate in forest loss in the land cover dataset applied to these simulations (Sect. 2.2) is unlikely to have a strong impact on the magnitude of the quantified forcing.

The total forcing associated with 1990–2005 land cover change (ΔLC-2005) is 69% of the forcing associated with 1990–2010 land cover change (ΔLC), indicating that 31 % of the total 1990–2010 forcing is associated with land cover change that occurred over the short 2005–2010 period. This sensitivity study demonstrates that the climate forcing associated with regional land cover change is rapidly increasing. The ΔLC-2005 analysis investigates the sensitivity of the simulated forcing

to uncertainty in the land cover distribution maps. The ozone forcing is tightly linked to changes in the areal cover of oil palm, and the global-mean ozone forcing per Mha of regional expansion of oil palm plantations is +1 mW m$^{-2}$ Mha$^{-1}$ (the same value is obtained regardless of whether the output from the ΔLC analysis or the ΔLC-2005 analysis is used).

The land cover maps used in this study are based on the land cover classification of Gunarso et al. (2013), which identified

large (> 1,000 ha) patches of oil palm cover. Estimates suggest that around 40 % of Indonesian oil palm area in 2010 was associated with smallholders (Lee et al., 2014, citing the Indonesian Ministry of Agriculture), in contrast to state-owned or private companies. Smallholder plantings are either schemed (contiguous with a larger estate) or independent (Vermeulen and Goad, 2006). In the latter case, plantations are typically on the order of 2–50 ha (Vermeulen and Goad, 2006), far smaller than the patches identified by the Gunarso et al. (2013) analysis. Gunarso et al. (2013) note that it is unknown what

proportion of the total smallholder plantation area is independent and, therefore, excluded from the total oil palm areal cover quantified in their analysis. Nonetheless, it is likely that the oil palm areal extent from the Gunarso et al. (2013) remote sensing analysis is an underestimate of the true areal coverage of oil palm in maritime Southeast Asia.

The 1990–2010 change in oil palm cover dominates the land-cover-change-driven isoprene emissions changes in this region

(Sect. 3.1); thus, the underestimate in oil palm areal cover likewise represents an underestimate in the global-mean ozone forcing from regional land cover change. Extrapolating the smallholder estimate for Indonesia in 2010 (40 % of total) to year 1990 and to the entire region and assuming that this represents independent smallholders, the total regional expansion of oil palm cover for 1990–2010 increases to +16 Mha. Applying the land-area-based global-mean ozone forcing for regional oil palm expansion (+1 mW m$^{-2}$ Mha$^{-1}$), based on the ΔLC and ΔLC-2005 analyses, gives an estimate of +16 mW m$^{-2}$ for the

global-mean ozone forcing from regional land cover change. This value is considered to be the upper bound of a best estimate of the global-mean ozone forcing from maritime Southeast Asian land cover change.

There is little variability in the magnitude of the forcing associated with the prescribed background atmosphere as application of the year 1990 background atmosphere (ΔLC-1990atm) in lieu of the year 2010 background atmosphere (ΔLC)



had little impact on the ozone and SOA forcings induced by land cover change (Table 4). The fact that local surface ozone reductions, rather than enhancements, are simulated in response to enhanced BVOC emissions (Figure 1) suggests a $NO_X$-sensitive chemical environment. Application of higher regional surface $NO_X$ emissions to the simulations might be expected to result in an increase, rather than a decrease, in regional surface ozone in response to increased BVOC emissions, but an

increase in regional surface ozone concentrations is unlikely to have a significant impact on the induced ozone forcing since, as Lacis et al. (1990) find, changes in surface ozone have a much smaller effect on climate forcing relative to equivalent ozone changes in the upper troposphere. Thus, uncertainty in the magnitude of local surface $NO_X$ emissions is unlikely to be a significant contributor to uncertainty in the ozone forcing.

Taking into account the results of the sensitivity simulations, the best estimate of the global-mean forcing from ozone changes induced by regional 1990–2010 land cover change is +9 mW m$^{-2}$, with a range of +4 to +16 mW m$^{-2}$. The quantified range accounts only for uncertainties probed by the sensitivity studies.

**4 Conclusions and future work**

The best estimate of global-mean forcing from BVOC emission perturbations driven by regional land cover change –
quantified here using simulations that apply satellite-derived land cover distributions and measured leaf-level BVOC BERs –
indicates a positive forcing (+8.4 ± 0.7 mW m$^{-2}$), which is a warming impact. In absolute terms, the quantified forcing from 1990–2010 maritime Southeast Asian land cover change is small, particularly in comparison to the forcing associated with industrial-era perturbations of well-mixed greenhouse gases (e.g., Myhre et al., 2013). However, the ozone perturbations associated with changes in global anthropogenic emissions of non-methane VOCs over the industrial era (1750–2011)
induced a global-mean stratospherically adjusted forcing on the order of +30 mW m$^{-2}$ (Myhre et al., 2013), which is only around 3× the magnitude of the instantaneous ozone forcing associated with 1990–2010 land cover change in MSEA (+9.2 mW m$^{-2}$).

The climate forcing quantified here represents the forcing induced by atmospheric composition changes driven by 1990–
2010 land cover change in MSEA. Roughly 27 % of the 2010 oil palm plantation areal cover in this region already existed in 1990 (Table 1). Applying the land-area-based global-mean ozone forcing for regional oil palm expansion that was calculated here (+1 mW m$^{-2}$ Mha$^{-1}$), regional oil palm expansion over the modern era is responsible for a global-mean forcing of +12.7 mW m$^{-2}$ from induced ozone changes. Based on government-awarded leases, Carlson et al. (2012) project that at least 9.4 Mha of additional land in Indonesian Borneo alone will be converted to oil palm plantations over 2010–2020, indicating that
additional climate forcing is expected from regional land cover change in coming years. For comparison, oil palm expansion over 1990–2010 for the entirety of MSEA was +9.6 Mha (Table 1).





The sensitivity analyses indicate that important factors driving uncertainty in the forcing include (1) uncertainty in the magnitude of the isoprene BER for oil palm and (2) uncertainty in the areal extent of oil palm expansion. The simulations find that the expansion of oil palm plantations is responsible for almost all of the land-cover-change-driven net increase in regional isoprene emissions (Sect. 3.1). Because the total magnitude of isoprene emissions from oil palm plantations scales

linearly with the assigned leaf-level BER, the atmospheric composition changes and concomitant forcing from regional land cover change are strongly dependent on the magnitude of the assigned isoprene BER for oil palm plantations. An improved understanding of the isoprene BER for oil palm would strongly benefit the effort to quantify the environmental impacts of the recent large-scale changes in land cover in this sensitive region.

Based on above-canopy isoprene flux measurements, observed meteorological variables, and the isoprene emission algorithms of the Model of Emissions of Gases and Aerosols from Nature (MEGAN; Guenther et al., 2006) model, Hewitt et al. (2011) suggest that isoprene emissions from oil palm plantations are under circadian control at the canopy scale; that is, the isoprene BER exhibits diurnal variability, peaking in the early afternoon. Keenan and Niinemets (2012) subsequently argued that the apparent circadian control calculated by Hewitt et al. (2011) is likely caused by inappropriate assignment of

model parameters in the MEGAN model. One study reports apparent circadian control of oil palm isoprene emissions at the leaf level (Wilkinson et al., 2006). Like other global models, the YIBs model supports application of only a single invariant isoprene BER for each PFT or land cover type and, therefore, might misrepresent isoprene emission magnitudes if oil palm isoprene emissions are, in fact, under circadian control. Inclusion of a temporally variable BVOC BER in the global model would allow for an improved estimation of radiative forcing induced by land cover changes in this region.

Field measurements, including one campaign over Borneo (Stone et al., 2011), indicate that OH concentrations over the pristine tropical rainforest are much higher than predicted by known isoprene chemistry (Lelieveld et al., 2008; Martinez et al., 2010; Stone et al., 2011). A number of OH-recycling mechanisms associated with isoprene oxidation have been proposed (e.g., da Silva et al., 2010; Lelieveld et al., 2008; Paulot et al., 2009; Peeters et al., 2009). Despite the number and diversity

of proposed mechanisms, the degree of OH recycling associated with OH-initiated isoprene oxidation in low-$NO_X$ atmospheres remains a major uncertainty in the field of isoprene chemistry. Other researchers have argued that the high measured OH concentrations may be an artifact of the type of instrumentation employed, which results in an artificial inflation of the measured concentrations (Mao et al., 2012). Additional uncertainty surrounds the methodology of including enhanced OH recycling in models (Hofzumahaus et al., 2009; Kubistin et al., 2010; Pugh et al., 2010; Stone et al., 2011;

Whalley et al., 2011). Warwick et al. (2013) found that inclusion of OH recycling had little impact on the magnitude or distribution of surface ozone changes induced by total conversion of Bornean tropical forest to oil palm plantations. No OH recycling is applied in the OH-initiated isoprene oxidation pathway in the chemical mechanism of ModelE2-YIBs because of (1) the significant uncertainties associated with this aspect of isoprene chemistry and (2) its apparent inconsequence to the surface pollution impacts of regional land cover change. Future work would benefit from an exploration of the impact on





radiative forcing induced through application of different mechanisms of (1) isoprene photooxidation and (2) SOA formation (e.g., Surratt et al., 2010; Zhang et al., 2018).

Previous work has shown that land cover change in MSEA impacts the environment in numerous ways, including by threatening plant and animal diversity (Sodhi et al., 2004), driving large GHG emissions (FAO, 2014; WRI, 2015), and damaging air quality through vegetation and peat burning (Gaveau et al., 2014; Koplitz et al., 2016). The simulations presented here indicate that BVOC emission perturbations provide an additional mechanism by which regional land cover change impacts the environment. While the impact on global radiative forcing is small, the ozone radiative forcing exceeds +37 mW m$^{-2}$ in some localities. This forcing mechanism can be expected to grow in importance in future years if oil palm expansion continues at a high rate.

**Data availability**

Numerical data used to make the figures or used to otherwise support the analysis are included as supporting information. Raw model output is available from the authors. All other data used as input to the model are listed in the references.

**Author contribution**

K.H. and N.U. designed the study. K.H. modified the model source code, performed the model simulations, and analyzed the model output. K.H. prepared the manuscript with revisions from N.U.

**Competing interests**

The authors declare that they have no conflict of interest.

**Acknowledgements**

This project was supported in part by the facilities and staff of the Yale University Faculty of Arts and Sciences High Performance Computing Center. The authors thank Tim Killeen for providing land cover classification data for maritime Southeast Asia and Xu Yue for model de-bugging support.

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
