# Peer review of "Global climate forcing driven by altered BVOC fluxes from 1990– 2010 land cover change in maritime Southeast Asia"

_Atmospheric Chemistry and Physics, 2018_

## Referee Comment (RC1) · Anonymous Referee #1 · 12 Jun 2018

Global climate forcing driven by altered BVOC fluxes from 1990-2010 land cover change in maritime Southeast Asia

The authors present the findings of a global modeling study probing the impacts of historical land cover change on the islands of maritime SE Asia with a particular focus on the expansion of oil palm plantations at the expense of natural forest. They apply a chemistry-climate model with interactive land surface to investigate the resulting changes in BVOC emissions and atmospheric composition in the region. In line with previous studies they conclude that changes in surface concentrations of the air pollutants / short-lived climate forcers, ozone and secondary organic aerosols (SOA),

are negligible. However, they demonstrate that due to strong convection in the tropics, upper tropospheric concentrations are more strongly affected and calculate the radiative forcing associated with these changes, showing that land cover change over this 20-year period in this region may have resulted in local changes in radiative balance.

On the whole this is a carefully implemented study with a reasonable selection of simulations designed to probe some of sensitivities of the model to their assumptions of land cover and vegetation characteristics. Their findings are generally well-presented. There is no doubt that the issue of tropical forest loss and / or degradation is of major global importance with both air quality and climate, and maritime SE Asia is a region which is experiencing rapid and extensive changes in land use.

However, I do have a number of reservations regarding their methodology, some of the assumptions made and the style in which they have presented some of their results. At present I feel that these are of sufficient concern to preclude publication.

Principal among these is the coarse resolution of the model used; a global model at 2x2.5 degree is not sufficiently fine resolution to adequately resolve the complex terrain or the heterogeneity of land cover, emission sources and chemical background conditions. NOx emissions have also rapidly increased in this region and the land cover changes included in this study will also introduce further changes. Given the sensitivity of ozone production and loss and SOA formation to the relative abundance of NOx and VOCs finely resolved spatial distributions are required to correctly diagnose both the direction and the magnitude of the changes in ozone concentration in particular.

My second major concern is the chemistry mechanism included in ModelE2-YIBs which the authors describe as based on CBM-4. CBM-4 was developed in the late 1980s and early 1990s at a time when the atmospheric chemistry community was principally focused on urban air quality and inorganic pollutants. The limited detail that the authors provide here suggests that the mechanism has not been updated to include the recent (i.e. post-2008) improvements in our understanding of isoprene oxidation

under conditions of high BVOC and relatively low NOx concentrations, conditions that must apply to large parts of the region under study. The same applies to monoterpene chemistry and the formation of biogenic SOA from both isoprene and monoterpene oxidation products. Without these updates it is hard to have confidence in the modeled changes in atmospheric composition arising from changes in BVOC emissions and concentrations.

Finally, I find that the manuscript is highly skewed to changes in isoprene and ozone, with monoterpene and SOA impacts rarely mentioned in the main text. However the final figures of radiative forcing include the forcing due to changes in SOA. A full discussion of monoterpenes and SOA is therefore needed in the main text.

More detailed comments are given below.

p1

L22 - Could the authors provide a map of the region to indicate what they are describing as "maritime SE Asia" and "the maritime continent"

L22 - It would be useful if the authors could provide some sense of scale. What proportion of Indonesia as whole is 4.5Mha? Or perhaps more relevant, what proportion of the natural forest does this represent?

L26 - It may have quadrupled but it started from a very low base; this is one of a number of times that the authors have tended toward dramatising the results.

p2

L7-9 - In fact, Ashworth et al. reported the change in the total tropospheric burden of ozone and SOA before focusing on surface changes and Warwick et al. present altitudinal plots of the changes in some trace gases.

L20-21 - This is the first mention of monoterpenes (aside from the abstract). I suggest for the authors also to discuss the atmospheric chemistry and composition effects of

monoterpenes as per isoprene in the previous paragraph. For instance the surface flux measurements reported from the OP3 field study (Langford, Misztal) showed that natural forests are much stronger emitters of monoterpenes than oil palm plantations. And the previous investigations also included changes in monoterpene emissions which is not clear in L7-9 as the preceding lines had focused exclusively on isoprene.

The changes in monoterpenes and SOA seem to very much be of lesser importance to the authors than changes in isoprene and ozone here and throughout the manuscript. While I accept that the changes are smaller they still contribute to the overall radiative forcing reported by the authors and should be given full coverage in the main text and not just the SI

L24 - Does this mean that the authors have only conducted a series of atmosphere-only model simulations? So there are no climate / land surface feedbacks included on-line?

L24-27 - Actually I am now confused as to exactly what model simulations were performed. The authors have referred to atmosphere-only, chemistry-climate, and land surface models here. Exactly what configuration is being used?

L27 - 2 deg x 2.5 deg is too coarse to adequately resolve the complexity and heterogeneity of the land mass and land cover in this region particularly given the sensitivity of BVOC oxidation and ozone production rate to VOC:NOx ratio.

L31 - I have reservations whether the chemistry mechanism employed here is suitable for the conditions encountered in this region. Although it is rapidly developing with the concomitant increases in anthropogenic emissions, much of island areas of the region are still low-NOx, high-VOC regimes. Older chemical mechanisms were designed for the typical chemistry encountered in mid- to high-NOx urban / industrial areas and considerable understanding has been gained of the very different oxidation pathways of (particularly) isoprene under lower NOx conditions. Have any of these updates been included in the chemistry here?

p3

L8 - Are the monoterpenes emitted as a single lumped monoterpene species? Or at least in part speciated (e.g to specifically include a-pinene, b-pinene, d-limonene and others as is often done)? It should be noted that the monoterpene emissions algorithms included in Lathiere et al are in fact the Guenther et al. algorithms from 1995; at the very least this paper should be referenced here. Also, these algorithms assume that monoterpene emissions are entirely temperature dependent whereas more recent field measurements have shown that many species emit a proportion of monoterpenes directly (i.e. monoterpene emissions exhibit both light and temperature dependency, see e.g. Steinbrecher et al, Guenther et al. 2012). Are the authors confident that this is not the case for SE Asian plant species?

L15-19 - How were emission factors assigned to these additional land cover types? How do they differ from the standard land cover in this region in the default land surface map? Again this is critical to the results and should be included in the main text and not just the SI.

L31-32 - Of real importance to this study is the previous performance of the YIBs model in this region; the 2013 study was global. Do the referenced comparisons include field sites in maritime SE Asia?

p4

L14-15 - It seems odd to go the lengths of using 30m x 30m resolution land cover data in a model running at 2 deg/ x 2.5 deg L20-26 - Please clarify. Are you saying that in 1990 the only land cover data available shows natural forest (or whatever) in locations that were shown as oil palm in 2000 and 2010? What fraction of data is missing? Of the 1990 data what fraction is converted to oil palm by 2000 and 2010? Of the missing data what fraction is "converted" to oil palm by 2000 and 2010? It would be useful to have a feel for how substantial the "likely underestimation" might be.

L30 - Is dirt equivalent to the "bare ground" classification included in other land surface schemes?

p5

L3 - The authors say that these are "minor land cover types". Again it would be useful to be provided with sufficient information to judge just how minor. What fraction of land is included in these 5 types in Gunsaro et al classification?

L8-13 - Again it would be good to have an idea of the likely underestimation, and this should be relatively easy for the authors to achieve by applying an LAI reduction, as described, to the areas classified as "disturbed" in Gunsaro.

L21 - As noted above, Table S2 should be in the main text as these parameter values are critical to the results. The notes regarding their derivation can be left in SI. The values for the "standard" PFTs in YIBs for this region should also be shown in this table for comparison.

L27-34 - There is a real mishmash of years for the various datasets. As the simulations are being conducted for a nominal 2010 (i.e. that is the climatology) with 1990 or 2010 SE Asia landcover why introduce further limitations / discrepancies by using Y2000 landcover for the rest of the world with vegetation characteristics derived using Y2000 meteorology only to change to 2010?

p6

L33-34 - Could the authors please clarify how the simulations were driven with the meteorology? Was the same climate / meteorology applied for 13 years? Because there will be an effect of inter-annual variability on emissions, chemistry and therefore O3 and SOA formation; how has this been accounted for? Is this what the authors have attempted to do via the additional simulations?

p7

L4-6 - This is the case for all current isoprene emissions models which are linked to PAR, T, CO2, soil moisture, etc. Please clarify what aspect the authors mean is the case "because" it is interactively linked OR remove the word "because"

L6-7 - Please give more detail of how monoterpene emissions are sensitive to climate as per isoprene

L9-11 - But changing the landcover will also affect e.g. NOx emissions, either due to changes in fertiliser application or to changes in natural soil emissions. How have the authors accounted for this?

Because again the resolution of the global model will not be sufficient to pick up changes such as this simply by running a sensitivity test with a different background atmosphere.

L28-30 - Why have the authors not used the measured emission rate in the first instance?

L30-32 - Why? The authors specifically introduced this land class because measurements had shown that the global emission factors were not suitable. The work reported in Langford and Misztal suggested that emission factors were out by a factor of 3 so using 12 seems rather extreme.

p8

Table 2 - this seems to imply that the isoprene emission factor applied to oil palm is as measured in the standard run but half measured in the OPber sensitivity tests which appears to contradict what the authors have described in the previous page.

p9

L3-4 - It would also be good to see how well 1990land_base and 1990land_1990atm GPP compare with measured GPP

L5 - please define contemporary, because Table 5 in the Guenther paper contains

estimates from early 90s to around 2008. Again it would be useful to see the emissions estimates for both 1990 and 2010 land cover and climates here.

L10-19 - Here and throughout, although the authors describe this as a study of how BVOC emissions changes have affected the region the manuscript is entirely dominated by consideration of isoprene. While this is understandable given that total regional isoprene emissions are more than 5x those of monoterpenes I think the paper would benefit from more consideration of monoterpene emissions and impacts as monoterpenes and isoprene have different effects on atmospheric composition and chemistry. I suggest the authors also pay careful consideration to their use of the catch-all BVOC as this study appears only to include isoprene and monoterpenes.

L16-18 - Similarly to the previous comment, rubber plantations don't make such a strong contribution to total BVOC emissions (e.g. compared to shrubs). However they do appear to dominate the monoterpene budget and might therefore have a strong role in SOA formation rate and yield. I suggest re-phrasing this sentence to make this distinction clearer.

L20-21 - but as previously noted by the authors, their assumptions in assigning land cover has likely led to an underestimation of deforestation. Please could the authors make some attempt to quantify the uncertainty in the changes in GPP. In addition to the method applied to fill data gaps and the non-inclusion of changes in LAI due to disturbance, the authors have used 2010 climatology in both cases which will affect the estimated GPP. I am assuming that the figure quoted here is based on difference between the two base simulations.

L25-26 - this could perhaps be better phrased as the loss of dipterocarp forest is due to its replacement by another land cover; as this is often oil palm so overall isoprene emissions go up.

L31 - I think the authors mean "Considering only the grid-cells that are majority . . ." as they then go on to give surface O3 concentrations in 2 sub-regions rather than ALL of

the study area.

p10

L3-5 - These measurements would seem to support the low ozone concentrations simulated by the model over Malaysian Borneo; however I am not sure they provide evidence of ozone concentrations over ALL forested areas in the region particularly as the authors are comparing annual mean concentrations with measurements made for one particular (short) period.

And related to this, are the authors intending to imply that they simulate much higher levels of ozone over the non-forested areas in the region? In which case, what are the average ozone concentrations for e.g. peninsular Malaysia which is far more industrialised? i.e. it is likely the case for many of the included grid cells that the proportion of the grid cell not occupied by forest in Borneo and New Guinea is ocean, whereas in peninsular Malaysia, etc many more will contain urban / industrial areas with higher NOx emissions.

L7-8 - This would be a suitable place for the authors to emphasise the difference between the effects of monoterpenes vs isoprene, with isoprene oxidation more implicated in ozone production and loss rather than SOA formation.

L8-9 - Please comment on the possible reasons for the observed enhancements over the ocean.

L7-20 - The authors present and discuss only annual average surface concentrations. This masks seasonal changes in magnitude, sign and distribution. For example, Ashworth et al reported different patterns of ozone and SOA changes depending on wind direction between the two monsoon wind periods. And given the NOx-sensitivity of the region, the relative position of NOx and VOC sources can become even more important at different times depending on wind speed and direction. This is likely to become increasingly important as the region continues to industrialise and oil palm plantations

continue to expand into areas that are currently urban / industrial rather than remote. Please discuss these limitations in the study.

L16-19 - What is the resolution of the NOx emissions input? As previously noted I do have concerns over the capability of the model to resolve the heterogeneity of this region and Hewitt et al. 2010 demonstrated the sensitivity of the atmospheric chemistry in this region to NOx levels over a range of BVOC emissions

L28 - Please re-phrase; "inflated" sounds as if the authors applied an arbitrary increase whereas in both cases the scenarios in which NOx emissions were increased were based on the differences observed between forest and plantation.

L34 - A likely key difference between the work of Silva and that presented here is that Silva applied the GEOS-Chem model at a resolution of 0.5deg x 0.667deg, a far more appropriate resolution for this highly complex region.

p11

L1-3 - But how does deposition change in YIBs which unlike GEOS-Chem couples the atmosphere to a process-based parameterisation of stomatal conductance? Otherwise I'm not sure what point the authors are trying to make here

Figure 1 - Please add a panel showing clearly where the changes in land cover were made. In the SI the figures imply that the changes were made only to the islands of SE Asia; here panel (a) shows a wider SE Asia than this. I suggest it would be useful for the authors to add a bounding box in each of (a) to (c) to show where the LCC occurred. I also suggest that panels should be added to show typical absolute concentrations (perhaps best done with 2010 base) of O3, SOA, isoprene and monoterpene emissions and changes in these emissions.

P12

L3-5 - These 2 sentences appear to be saying the same thing; are both needed?

L5-6 - Please state the heights/pressures being used define upper and lower tropo-sphere; as evident from Figure 1 (d) the reported average changes in ozone will be critically dependent on this boundary.

L6-7 - This seems somewhat negligible (?)

L19-21 - Would suggest that the authors re-order this sentence to aid readability; the previous sentence considered isoprene so would be more logical to start with isoprene here: e.g. "When transport-driven isoprene . . ."

L26 - "-5N"? Please use "5S" for consistency with Figure 2.

L28 - Wolfe et al seems an odd choice of primary reference for the formation of HCHO from isoprene as HCHO columns have been used as a proxy since Palmer et al. 2001, 2006 and several authors since (including Palmer et al 2003) have considered the rel-ative contributions of other VOCs to HCHO which seems of real relevance here as this study considers monoterpenes as well as isoprene, although as previously commented the text seems rather skewed toward isoprene.

p13

Figure 2 - Why are monoterpenes and SOA not included here? As previously noted, monoterpene emission changes and atmospheric composition impacts are barely cov-ered.

L10-11 - but atmospheric concentrations of CO are of the order of 60-120 so this is a small relative change.

L17-19 - Previously the authors have strongly made the case that this region is low-NOx. What is the yield of alkyl nitrates in ModelE2-YIBs from BVOC oxidation under low-NOx conditions as this seems more pertinent than commenting on yields in high-nix environments?

p14

[Figure]

L8-9 - Although there is a clear mis-match between the spatial distribution of the enhancement in nitrates and reduction in NOx concentrations. And in particular, the formation of alkyl nitrates appears particularly increased at the surface where no change in NOx is evident. Could the authors comment on the reason for this?

L11-15 - However, unless the authors have incorporated the "new" isoprene oxidation pathways under low-NOx conditions in the chemistry mechanism (in which case this needs to be made clear in the model description section) this is more a model artefact reflecting the atmospheric chemistry community's understanding of HOx chemistry in 1999. Please clarify the isoprene oxidation scheme in ModelE2-YIBs. Is it really still CBM-4?

L18-19 - Again the issue of BVOC or isoprene or isoprene+monoterpenes; which is being considered here?

L22-23 - Please justify why the model is being driven with a single (repeated) year of meteorology. While it is useful to know the uncertainties associated with internal model variability it would be of far more use to know how inter-annual variability in climate / meteorology affects the radiative forcings calculated here as ultimately what is of real interest is how future LCC in the region might play out.

L23-24 - Now we come back to one of my chief concerns with the manuscript itself. This is now the first mention of aerosol changes. Although the abstract and introduction mentioned SOA as well as ozone the changes in SOA were not presented or discussed anywhere in the results section. If the authors wish to include the effects of aerosol on radiative forcing it is essential that the changes in tropospheric aerosol concentrations are introduced and discussed prior to this; likewise monoterpenes. Is the authors' reluctance to fully consider monoterpenes and SOA due to deficiencies in the chemistry mechanism and/or gas-to-particle partitioning in the ModelE2-YIBs model?

L27-30 - Given the levels of uncertainty in calculations of radiative forcing, and the limitations previously identified with this study, it is hard to see a net forcing of ∼0.008

Wm-2 as globally significant. The more substantial (but highly localised changes over the Indian Ocean) could be of real interest in terms of how they affect the Indian monsoon which has seen significant changes in recent years, but this is not explored by the authors. Again, however, these are temporally "averaged" results whereas the "interesting" effects are likely to be temporally as well as spatially localised. This would be another extremely interesting avenue to explore. Do the changes peak at times and locations when small changes in climate-relevant atmospheric components matter?

p16

L1-2 - See previous comments regarding monoterpene and SOA results and discussions. But interesting to note that O3 and SOA forcings seem to scale, presumably also with isoprene emissions changes. But this comes back to my previous questions regarding the fitness of the chemistry mechanism for the conditions encountered in this region and the ability of the model to capture the heterogeneity of land and chemical climatology given its coarse resolution.

L16-18 - See previous questions and comments regarding inter-annual variability; as noted by the authors in L10-12 the isoprene flux is critically dependent on meteorology and so presumably the impact on radiative forcing would be similarly sensitive.

L20-32 - I do not understand why the authors chose a x12 enhancement in isoprene emission factor from the dipterocarp forest given the x3 enhancement observed during in-situ measurements. Further, given the incredibly low isoprene emissions from these forests relative to both other natural tropical ecosystems and oil palm plantations why there was a need for this sensitivity test. Monoterpene emissions are relatively strong from dipterocarp forests so I would have expected to see a sensitivity test involving increased monoterpene emission rates instead. Perhaps the authors could comment on why this was not done?

p17 L2-4 - However, this forest loss is likely to have affected monoterpene emissions more substantially, coming back to my previous questions regarding the importance

of monoterpenes and SOA contributions to the LCC-induced radiative forcing in the region and whether this is well captured in the model used.

L6-9 - Here and in many other sections of the discussion section I feel that the authors have lost objectivity and are attempting to over-emphasise aspects of their results to fit a particular narrative. The study covers a 20-year period. If the annual changes were constant you would expect to see 75% of the forcing associated with 1990-2005; 69% is not so far from that. As the authors have only broken the 20-year period down in one way rather than into 5-year blocks throughout they do not have sufficient evidence that the 5-year period from 2005-2010 is worse than every other 5-year period which would be needed to support the statement that the forcing is "rapidly increasing".

L19-22 - Presumably this underestimation is likely to affect both the 1990 and 2010 land cover maps. Have the authors attempted to find out from other sources (e.g. FOA, Malaysian Oil Palm Board, Round Table for Sustainable Palm Oil) whether the proportion of smallholder plantings to industrial plantations has remained constant during the rapid expansion of the oil palm industry or whether the number of smallholder plantings has remained closer to the 1990 figures?

L26 - Can the authors clarify what they mean by "extrapolating the smallholder estimate"? Do they mean they assume that 40% of reported oil palm area across the whole of SE Asia and through the entire time period represents smallholdings? So they assume that the area they have taken from Gunarso represents only 60% of the actual extent of oil palm cultivation? See above regarding justification of this assumption.

L33-34 - To what extent is this insensitivity the result of the coarse resolution and outdated chemistry scheme?

p18 L5-10 - Please explain why an increase in surface ozone concentrations could not be concomitant with increases in upper tropospheric ozone?

L11-12 - Please could the authors attempt to list some of the uncertainties not considered by the sensitivity tests that might be expected to be substantial?

L21-22 - The authors have just stated that their best estimate is +8.4; that is therefore the figure that should be quoted here, in which case we are looking at 3.5x

L27 - As highlighted previously, the authors are over-emphasising the magnitude and implications of their findings. Using 8.4 suggests a figure of 11.5 rather than 12.7 mW m-2.

L28-31 - It would have been of real interest if the authors had looked at future projections rather than confining the study to historical LCC and radiative forcing.

p19

L1-2 - Just out of curiosity, how much more uncertainty is associated with isoprene BERs from oil palm in comparison with isoprene emissions from other tropical species / ecosystems OR monoterpene BERs from rubber palms which the authors earlier highlight as important factors in the changes in BVOC emissions in the region.

L10-19 - Equally, many previous studies (e.g. Grate et al 2007) have shown strong apparent seasonality in BERs that are not adequately accounted for by consideration of leaf age. Inclusion of seasonally varying BERs might also be argued as improving the estimated radiative forcing in this study.

L18-19 - The isoprene vs BVOC issue again. Presumably given the focus of the paragraph to this point the authors are referring to the isoprene BER in YIBs.

L18-19 - And again, the issue of misrepresentation. In the title and throughout the manuscript the authors refer to BVOC emissions and emission changes yet YIBs includes only a very limited number of BVOCs, and here the authors have only altered isoprene and monoterpene BERs. I suggest that the authors remove the term BVOCs from the title and discussions as it is not an accurate reflection of the study performed. Likewise the authors need to devote far more attention to the changes in monoterpenes

and SOA throughout the main text.

L21-34 - This discussion of OH recycling in the conclusions section is disturbing on a number of counts. (1) This is the first and only discussion of the apparent limitations of the chemical mechanism in ModelE2-YIBs, as previously commented above. Can the authors please describe exactly what BVOC oxidation chemistry is included in the "based on Carbon Bond Mechanism-4" scheme? As previously noted, CBM-4 was developed for high NOx anthropogenic VOC-rich urban environments and such schemes have been found to inadequately capture observed concentrations / chemistry / oxidation products in low-NOx high BVOC environments such as those in SE Asia. (2) The field of isoprene oxidation chemistry has moved on considerably since the sensitivity studies employing crude "OH-recycling" schemes referenced here with new pathways identified leading to the regeneration of HOx in low-NOx environments. (3) While surface ozone concentrations might be only negligibly affected the authors have repeatedly argued elsewhere that these are not the changes that are significant in terms of radiative forcing. (4) The new understanding of isoprene chemistry gained in trying to reconcile the apparent differences between modeled and observed gas-phase chemistry has also identified mechanisms driving high yields of isoprene-derived SOA via MACR oxidation. The points made in this paragraph bring into question the validity of the modeled changes in ozone and SOA presented here. It also raises questions regarding the monoterpene oxidation scheme and gas-to-particle phase partitioning included.

p20

L1-2 - Actually this should have been included in this work as the chemistry seems to be a critical source of uncertainty that has not been adequately considered.

L8-9 - Concluding with the absolute maximum single pixel increase is not scientifically balanced. Please maintain objectivity rather than cherry-picking the results to fit a particular narrative.

---

## Referee Comment (RC2) · Anonymous Referee #2 · 13 Jun 2018

Harper and Unger present a study of the radiative forcing brought about via differences in isoprene emission under different land use configurations in the maritime Southeast Asia (MSEA) region. These land use changes comprise the move towards more oil palm plantations, which emit more biogenic volatile organic compounds than the native natural forests. The changes in isoprene emitted to the atmosphere as a result of the increased oil palm leads to changes in ozone. Of particular interest is that the Enhanced BVOCs caused bigger changes globally to ozone in the upper troposphere (0.6 ppb) than lower troposphere (>0.1 ppb), which would seem an important result. The novelty of this study is that the authors then go on to calculate the radiative forcing expected by these ozone changes, finding a small increase of +1 mW m-2 Mha-1.

This shows that impacts of land use changes in tropical regions, which are subject to stronger convective patterns that elsewhere, are very important.

My feeling is that this is a really nice idea, but the wrong tool has been used to carry out the study. The small changes in ozone seen at the top of the troposphere are probably lost in the noise of uncertainty of the chemical scheme chosen, and thus I question the impacts on radiative forcing.

The authors use the carbon bond 4 chemical mechanism to represent the oxidation of isoprene in the atmosphere. This scheme is very old and does not include some of the recent discoveries brought about via questioning the discrepancies between isoprene predicted by models, and observed mixing ratios. These particularly relate to additional OH recycling, which directly impact the influence of isoprene on O3 (eg Lelieveld et al., 2008; Peeters et al., 2009).

The authors do mention the uncertainty in the isoprene chemistry regarding increased oxidant cycling, right at the end of the paper in the conclusions, but I think there are other problems with this choice of chemistry scheme. High isoprene atmospheres, such as that found in this MSEA region, have caused more differences in chemical mechanisms than most others. Unfortunately, the carbon bond scheme has never fared well when tested alongside other chemistry schemes under similar isoprene rich atmospheres. I wonder why there has been no model development in the chemistry scheme in this work when the science behind this paper depends so highly upon it?

For example Knote et al (2015) tested two variants of the newer carbon bond 5 (CB05) scheme (neither of which contained updates to the isoprene chemistry) and found they "tended to be biased low in O3 under low NOx/high VOC conditions (e.g. biogenic emissions rich) as well as under very high NOx conditions. In general, the CB05 schemes produced 'lower than average 8 hourly O3' produced by other schemes. Mechanisms were 'found to differ more strongly in their predictions of O3 levels and other pollutants in regions with strong biogenic VOC emissions".

Archibald et al (2010) tested 8 chemical schemes in isoprene rich regions and found that the CB05 mechanism was 'unable to generate/recycle HOx at the rates needed to match recently reported observations at locations characterized by low levels of NOx.'

An older study - Emmerson and Evans (2009) tested the carbon bond 4 scheme against 6 other schemes. However the carbon bond 4 results disagreed with the majority of the other schemes, in even the sign of the changes in ozone (e.g. loss instead of production - see figure 3 panel e). Differences (and thus uncertainty) of 14 ppb were found between the resulting ozone from the Master Chemical Mechanism and the carbon bond 4 scheme, which is 14 times more than the ~1 ppb of ozone changes found in Harper and Unger's study at the top of the troposphere, and upon which the radiative forcing calculations are based.

Thus I don't agree with the authors' comment that no updates to the chemistry have occurred because of "its apparent inconsequence to the surface pollution impacts of regional land cover change". I think if a different chemistry scheme had been implemented that the changes in ozone found by Harper and Unger as a result of including more oil palm plantations in the model would lead to more significant differences in the radiative forcing than found by their study.

I'd recommend updating the chemistry scheme. Perhaps even to include a sensitivity study with a more up to date representation of just the isoprene chemistry – particularly one that agrees with the sign of ozone changes driven by our current understanding. The chemical aspect of Harper and Unger's work is my only criticism, which if rectified I would then recommend publication in ACP.

General comments

A map figure would be good, showing the study area with the areal extent of regions growing oil palm in 1990 and where/how these regions have increased by 2010.

Page 2 line 2. "Compared to natural forests oil palm plantations are much stronger

emitters of BVOCs" Some numbers would be good here. How much stronger?

Page 2 line 20. Try placing the (Baker et al., 2005; Klinger et al., 2002) references at the end of sentence to avoid breaking the flow of the sentence up too much.

Page 2 line 29. How is photolysis treated in the model?

Page 3 line 27. 'the' calculation

Page 5 line 12. It is not clear where this LAI dataset has come from?

Page 5 line 14 (onwards in this paragraph). LAI has units of m2 m-2

Page 5 line 21. Table S2 – mention this is in the supplementary section.

Page 19 line 21. This whole discussion of uncertainties in the chemistry scheme would be better placed in section 2.1 which introduces the method used.

References

Archibald, A.T., Jenkin, M.E., Shallcross, D.E., 2010. An isoprene mechanism intercomparison. Atmos. Environ. 44 (40), 5356e5364.

Emmerson, K. M., and Evans, M. J.: Comparison of tropospheric gas-phase chemistry schemes for use within global models, Atmos Chem Phys, 9, 1831-1845, DOI 10.5194/acp-9-1831-2009, 2009.

Knote, C., Tuccella, P., Curci, G., Emmons, L., Orlando, J. J., Madronich, S., Baro, R., Jimenez-Guerrero, P., Luecken, D., Hogrefe, C., Forkel, R., Werhahn, J., Hirtl, M., Perez, J. L., San Jose, R., Giordano, L., Brunner, D., Yahya, K., and Zhang, Y.: Influence of the choice of gas-phase mechanism on predictions of key gaseous pollutants during the AQMEII phase-2 intercomparison, Atmos Environ, 115, 553-568, 10.1016/j.atmosenv.2014.11.066, 2015.

Lelieveld, J., Butler, T.M., Crowley, J.N., Dillon, T.J., Fischer, H., Ganzeveld, L., Harder, H., Lawrence, M.G., Martinez, M., Taraborrelli, D., and Williams, J.: Atmospheric oxidation capacity sustained by a tropical forest, Nature, 452, 737–740,doi: 10.1038/nature06870, 2008.

Peeters, J., Nguyen, T.L., and Vereecken, L.: HOx radical regeneration in the oxidation of isoprene, Phys. Chem. Chem. Phys., 11, 5935–5939, doi: 10.1039/B908511D, 2009.

---

## Referee Comment (RC3) · Anonymous Referee #3 · 25 Jun 2018

Overall comments:

This paper examines the impacts of land cover change in maritime Southeast Asia induced mostly by oil palm expansion and the associated changes in BVOC emissions on surface ozone concentrations and tropospheric ozone profiles, and the subsequent impacts on radiative forcing. This is a novel piece of work that highlights the importance of considering atmospheric chemistry-mediated climate forcing in climate and land use change studies. The data integration and modeling approach are all scientifically sound, rigorous and valid. There are, however, insufficient or unclear exposition and explanation of the results at various places of the paper, as well as inadequate

discussion of the results in relation to previous works. I recommend the publication of this paper, if the concerns raised below are addressed.

Specific comments:

P1 L21: The introduction section appears too short, and do not set up a context nuanced enough to motivate the work (the findings of which are exciting). I recommend the authors to expand the introduction (by 30-50%) by discussing at greater lengths the various references cited. More suggestions in relation to this are given below.

P1 L25: The "%" sign usually immediately follows the number without space.

P1 L27: Can there be a sentence or two describing why we are concerned with oil palm planation from an environmental or ecological perspective (not just a climate perspective as included in the current second paragraph)?

P2 L8-9: Please expand this paragraph by discussing briefly the key findings of these few papers (Ashworth et al., 2012; Silva et al., 2016; Warwick et al., 2013). How large or in what ranges are the concentration changes? Is the surface air quality changes significant relatively to, e.g., the impacts of anthropogenic emissions or warming?

P2 L14: Why does upper tropospheric ozone have a larger climate impact than surface ozone?

P2 L19: How does the land cover change derived from this data source differ from or compare with that used by Silva et al. (2016)?

P3 L4: Please explain and justify whether the discontinuity created by using two different biomass burning datasets is acceptable, especially considering that biomass burning emissions are an important source of ozone there.

P3 L4-6: "Interactive" is a modeler's jargon, and even for modelers can mean different things for different purposes. I recommend avoiding it and state more clearly that these emission schemes are "semi-empirical", "mechanistic" or "dynamic functions of x, y, z,

[Figure]

...", especially for those that are not described more below.

P3 L13: Avoid the use of "online".

P3 L28: Avoid "online" and "model's".

P3 L33: Please explain and justify the single chemical representation of monoterpene. Can all monoterpenes really be modeled as $\alpha$-pinene?

P5 L30: What about the LAI values for the new PFTs used for this study for MSEA? They are not described above. Are dynamic but grid-level LAI observed from, e.g., MODIS, used, or are PFT-level LAI values used for these new PFTs? If so, where are these values from? As LAI is so important for atmospheric chemistry, these need to be better stated and explained.

P10 L1: In the methodology section above, the authors have only discussed about model validity and model-observation comparison for the vegetation aspects (e.g., GPP, biogenic emissions). What about an evaluation of the ozone simulations by the model? How does the model's simulated ozone globally compare with observations and with other models? Is the general high biases of simulated ozone in many climate-chemistry models also seen in this model? Since ozone concentration is crucial to this paper, I strongly recommend having a paragraph somewhere (preferably in the methodology section) discussing these.

P10 L14: Wong et al. (2018) also examined and quantified the factors behind the sensitivity of surface ozone to vegetation changes including isoprene emission and dry deposition. They also found a large impact of background NOx. See reference list below.

P11 L1: Dry deposition definitely also plays a role, and have you quantified the relative importance of isoprene emission vs. dry deposition to surface ozone in your model simulations? This appears to be a major missing part of this analysis and should be better addressed or discussed, even if the authors have already found that dry deposition plays only a minor role. For instance, Wong et al. (2018) found it necessary and developed a method to formally disentangle the contributions from isoprene emission and dry deposition when leaf density changes.

P12 L2: The physical reasons for the enhancements (as opposed to reductions) of ozone over the ocean have to be explained. Can these enhancements be explained by, e.g., the mechanisms suggested by Hollaway et al. (2017)? A discussion in relation to this paper is recommended. See reference list below.

P12 L16: In Fig. 2a) and 2c), why is there a second peak for isoprene and HCHO enhancement near the tropopause?

P14 L5: Now I see that the oceanic enhancements are explained. But this explanation, with reference to Hollaway et al. (2017), should be mentioned early (see comment to P12 L2).

P17 L8-9: "This sensitivity study demonstrates that the climate forcing associated with regional land cover change is rapidly increasing." I feel that this is too strong a statement. All the results are showing is that 2005-2010 as a 5-year period is responsible for a noticeably large fraction of the total RF compared to other possible 5-year periods, but without breaking down the other years into incremental 5-year periods (e.g., 1990-1995, 1996-2000, 2001-2005), we can't really say there is a rapidly rising trend in RF.

P18 L5-7: "increase in regional surface ozone concentrations is unlikely to have a significant impact on the induced ozone forcing since, as Lacis et al. (1990) find, changes in surface ozone have a much smaller effect on climate forcing relative to equivalent ozone changes in the upper troposphere." This is contingent upon the assumption that the formation and long-range transport of isoprene nitrate will respond in the same way even as the surface environment becomes more high-NOx. This needs to be justified.

P18 L24-31: I think one major missing discussion is to compare the ozone-mediated

[Figure]

RF with the biogeophysical RF (e.g., changing albedo, latent heat, sensible heat, etc.) and biogeochemical ($CO_2$ exchange) associated with oil palm expansion. Indeed, most climatologists are still just concerned with the biogeophysical or biogeochemical RF, and having a comparison between those and the ozone-mediated forcing would give much insight into the importance of considering atmospheric chemistry in climate/land use change studies.

P19 L18-19: "Inclusion of a temporally variable BVOC BER in the global model would allow for an improved estimation of radiative forcing induced by land cover changes in this region." I think the current debate is exactly that we are not sure about the circadian control or not, and thus this statement is not necessarily true.

P19 L33-34: "(2) its apparent inconsequence to the surface pollution impacts of regional land cover change" Is there really no OH titration problem in MSEA in ModelE2-YIBs? Is that because the BER is low to begin with, compared to, say, the Amazon?

References:

Hollaway, M. J., S. R. Arnold, W. J. Collins, G. Folberth, and A. Rap (2017), Sensitivity of mid-nineteenth century tropospheric ozone to atmospheric chemistry-vegetation interactions, J. Geophys. Res. Atmos., 122, 2452–2473, doi:10.1002/2016JD025462.

Wong, A. Y. H., A. P. K. Tai, and Y.-Y. Ip (2018), Attribution and statistical parameterization of the sensitivity of surface ozone to changes in leaf area index based on a chemical transport model. J. Geophys. Res. Atmos., 123, 1883–1898, doi:10.1002/2017JD027311.

---

## Author Comment (AC1) · 26 Aug 2018

We thank the reviewers for their helpful comments, which have led us to a substantially improved version of the paper. Here, the reviewers' comments are shown in boldfaced black text, and our responses are shown in non-boldfaced blue text. The page and line numbers to which we refer in our responses correspond to the updated manuscript (the comments of all reviewers are taken into account in this updated manuscript).

First and foremost, we confirm that the tropospheric chemical mechanism in GISS ModelE2 is not CBM04. The original manuscript version contained an incorrect oversimplified description of the tropospheric chemistry scheme in GISS ModelE2 that has caused our Reviewers confusion and understandable concerns. We understand that using an old-fashioned chemical mechanism developed 25 years ago for urban polluted high-NOx environments would be an inappropriate tool to apply to a study of large-scale isoprene emission perturbation in the tropics. The chemical mechanism in GISS ModelE2 has been substantially updated and improved over the past 15 years, for example, to account for important reactions, pathways, and species under low-NOx conditions (e.g. Shindell et al., 2003; 2006; 2013; Schmidt et al., 2014).

We now include a more detailed description of the chemical mechanism in Section 2.1 (ModelE2-YIBs description) (Page 4, Line 32):

"The troposphere features $NO_X$-$O_X$-$HO_X$-CO-$CH_4$ chemistry; an explicit representation of isoprene; and a lumped hydrocarbon scheme involving terpenes, peroxyacyl nitrates (PANs), alkyl nitrates, aldehydes, alkenes, and alkanes. The representation of hydrocarbons generally follows Houweling et al. (1998), which is originally derived from the Carbon Bond Mechanism-4 (Gery et al., 1989) and the Regional Atmospheric Chemistry Model (RACM; Stockwell et al., 1997), but includes several modifications aimed at representing the wide range of chemical conditions found in Earth's atmosphere, such as the addition of reactions important in low-$NO_X$ conditions including representation of organic peroxy radical chemistry under low-$NO_X$ conditions and introduction of organic nitrate chemistry. Shindell et al. (2013) describe in detail the recent updates to the tropospheric chemistry scheme, including the incorporation of acetone chemistry (Houweling et al., 1998) and the addition of terpene oxidation (Tsigaridis and Kanakidou, 2007). SOA formation is driven by $NO_X$-dependent oxidation of emissions of isoprene, monoterpenes, and other reactive VOCs following a volatility-based two-product scheme (Tsigaridis and Kanakidou, 2007). The formation of secondary inorganic aerosols, including sulfate (Bell et al., 2005; Koch et al., 2006) and nitrate (Bauer et al., 2007a), depend on both modeled oxidant levels and the availability of source gases. Primary aerosol types include dust (which provides a surface for heterogeneous chemistry; Bauer and Koch, 2005; Bauer et al., 2007b), black carbon, organic carbon, and sea salt (Koch et al., 2006). Stratospheric chemistry, introduced to the chemical mechanism by Shindell et al. (2006), includes nitrous oxide ($N_2O$) and halogen (bromine and chlorine) chemistry. Recent updates to stratospheric chemistry are summarized by Shindell et al. (2013) and include changes in the representations of polar stratospheric cloud formation (Hanson and Mauersberger, 1988) and heterogeneous hydrolysis of $N_2O_5$ on sulfate (Hallquist et al., 2003; Kane et al., 2001)."

Interdisciplinary work is challenging. We would like to emphasize the novel aspects of this project. (1) The land cover dataset for maritime Southeast Asia that we use in our study is built from an existing classification based on Landsat images (Gunarso et al., 2013). This dataset represents a wall-to-wall mapping of land cover in this region, including explicit representation of plantations of oil palm (high isoprene emitter) and rubber (high monoterpene emitter). Gunarso et al. (2013) used a consistent classification methodology for each year of their analysis, which has provided an internally consistent set of land cover maps for this period for this region. Other studies have investigated the atmospheric composition impacts from land cover change in this region: Ashworth et al. (2012) considered a projection of forest to oil palm conversion based on meeting future demand for biofuels; Warwick et al. (2013) considered the total conversion of Borneo to oil palm from forest; and Silva et al. (2016) considered the impact of 2010 oil palm cover relative to an oil-palm-free landscape in addition to considering future projections of oil palm. We consider the impacts of actual historical land cover change, which is clearly different than Ashworth et al. (2012) and Warwick et al. (2013). The Silva et al. (2016) study imposes oil palm expansion by overlaying an oil palm map for 2010 on a separate 16-PFT land cover distribution; this is a different methodology than we apply here, where we apply an internally consistent set of maps developed using a wall-to-wall classification methodology. (2) We have developed the global climate model code to add four additional land cover type PFTs, focusing on land covers that are pervasive in maritime Southeast Asia, including oil palm and rubber plantations. Previous studies have focused only on the impacts of oil palm expansion. (3) We consider the impacts of land-cover-change-driven changes in emissions of both isoprene and monoterpenes. The study by Silva et al. (2016) presumably includes dynamic changes in monoterpene emissions for the land covers that are displaced by oil palm, but their one new land cover type – oil palm – only has the isoprene emission capacity altered relative to the forest land cover type. Ashworth et al. (2012) and Warwick et al. (2013) consider only isoprene emission changes. (4) We directly quantify the global radiative forcing induced by ozone and SOA changes driven by historical land cover change in this region using a coupled global land-chemistry-climate model framework with the embedded radiative transfer model developed by A. Lacis and J. Hansen in GISS ModelE2 (e.g. Schmidt et al., 2014). (5) We provide new climate policy metrics for global ozone radiative forcing per Mha land cover change in the tropics. (6) We quantitatively identify that important factors driving uncertainty in the forcing include (a) uncertainty in the magnitude of the isoprene BER for oil palm and (b) uncertainty in the areal extent of oil palm expansion. Using an analysis based on fixed SOA yields, we additionally show that the sign of the net forcing is sensitive to uncertainty in the SOA yield from BVOCs.

**Responses to Reviewer #1**

**The authors present the findings of a global modeling study probing the impacts of historical land cover change on the islands of maritime SE Asia with a particular focus on the expansion of oil palm plantations at the expense of natural forest. They apply a chemistry-climate model with interactive land surface to investigate the resulting changes in BVOC emissions and atmospheric composition in the region. In line with previous studies they conclude that changes in surface concentrations of the air pollutants / short-lived climate forcers, ozone and secondary organic aerosols (SOA), are negligible. However, they demonstrate that due to**

strong convection in the tropics, upper tropospheric concentrations are more strongly affected and calculate the radiative forcing associated with these changes, showing that land cover change over this 20-year period in this region may have resulted in local changes in radiative balance.

On the whole this is a carefully implemented study with a reasonable selection of simulations designed to probe some of sensitivities of the model to their assumptions of land cover and vegetation characteristics. Their findings are generally well-presented. There is no doubt that the issue of tropical forest loss and / or degradation is of major global importance with both air quality and climate, and maritime SE Asia is a region which is experiencing rapid and extensive changes in land use.

However, I do have a number of reservations regarding their methodology, some of the assumptions made and the style in which they have presented some of their results. At present I feel that these are of sufficient concern to preclude publication.

Principal among these is the coarse resolution of the model used; a global model at 2x2.5 degree is not sufficiently fine resolution to adequately resolve the complex terrain or the heterogeneity of land cover, emission sources and chemical background conditions. NOx emissions have also rapidly increased in this region and the land cover changes included in this study will also introduce further changes. Given the sensitivity of ozone production and loss and SOA formation to the relative abundance of NOx and VOCs finely resolved spatial distributions are required to correctly diagnose both the direction and the magnitude of the changes in ozone concentration in particular.

My second major concern is the chemistry mechanism included in ModelE2-YIBs which the authors describe as based on CBM-4. CBM-4 was developed in the late 1980s and early 1990s at a time when the atmospheric chemistry community was principally focused on urban air quality and inorganic pollutants. The limited detail that the authors provide here suggests that the mechanism has not been updated to include the recent (i.e. post-2008) improvements in our understanding of isoprene oxidation under conditions of high BVOC and relatively low NOx concentrations, conditions that must apply to large parts of the region under study. The same applies to monoterpene chemistry and the formation of biogenic SOA from both isoprene and monoterpene oxidation products. Without these updates it is hard to have confidence in the modeled changes in atmospheric composition arising from changes in BVOC emissions and concentrations.

Finally, I find that the manuscript is highly skewed to changes in isoprene and ozone, with monoterpene and SOA impacts rarely mentioned in the main text. However the final figures of radiative forcing include the forcing due to changes in SOA. A full discussion of monoterpenes and SOA is therefore needed in the main text.

More detailed comments are given below.

**1. p1 L22 - Could the authors provide a map of the region to indicate what they are describing as "maritime SE Asia" and "the maritime continent"**

We provide Figure S2 (previously known as Figure S1), which shows the applied land cover changes. In Figure 1a, we analyze the surface ozone impacts over a wider region. Additional maps of the region are shown in Figures S3–S6 .

**2. L22 - It would be useful if the authors could provide some sense of scale. What proportion of Indonesia as whole is 4.5Mha? Or perhaps more relevant, what proportion of the natural forest does this represent?**

Good suggestion. We have modified this sentence (addition is bolded; Page 1, Line 24): "More than 4.5 Mha of natural forest were cleared in Indonesia alone over 2000–2010, **which is a loss of 4.6 % of year 2000 Indonesian natural forest cover** (Margono et al., 2014)."

Reference:
Margono, B.A., Potapov, P.V., Turubanova, S., Stolle, F., and Hansen, M.C.: Primary forest cover loss in Indonesia over 2000–2012, Nat. Clim. Change, 4, 730–735, doi: 10.1038/nclimate2277, 2014. (Their Table 1 reports total natural forest area in Indonesia in 2000 as 98.4 Mha.)

**3. L26 - It may have quadrupled but it started from a very low base; this is one of a number of times that the authors have tended toward dramatising the results.**

The reviewer appears to be somewhat missing the point here. Firstly, the areal cover in 1990 is implicit in the sentence: "The amount of land area planted in oil palm in Indonesia and Malaysia nearly quadrupled over 1990–2010, reaching 13 Mha by 2010 (Gunarso et al., 2013)." We modify the sentence to make the areal cover in 1990 now explicit (Page 1, Line 27): "The amount of land area planted in oil palm in Indonesia and Malaysia increased from 3.5 Mha in 1990 to 13 Mha by 2010 (Gunarso et al., 2013)." Secondly, we don't agree that this statement can be "one of a number of times that the authors have tended toward dramatizing the results" because (1) we are not discussing any project results in the Introduction Section and (2) in the Introduction section, we are describing the background motivation for the study as an opportunity for a real world case study during which a large-scale human-induced land cover change happened in the Earth system that has driven a regional-scale increase in isoprene emission.

**4. p2 L7-9 - In fact, Ashworth et al. reported the change in the total tropospheric burden of ozone and SOA before focusing on surface changes and Warwick et al. present altitudinal plots of the changes in some trace gases.**

We have expanded the Introduction Section, including highlighting the important findings of both the Ashworth et al. (2012) and Warwick et al. (2013) studies, in addition to another relevant study (Silva et al., 2016).

The Ashworth et al. (2012) study reports tropospheric burden changes for ozone and OH, but does not report changes for any specific altitude other than the surface. We do not find any mention of non-surface-level changes in SOA in Ashworth et al. (2012). The Warwick et al. (2013) study plots PAN and OH changes as a vertical cross-section at the equator from the surface to 90 hPa and the spatial distribution of PAN changes at 500 hPa; in addition, they report the peak ozone change at 500 hPa over Borneo, but they do not report any other non-surface-level ozone changes, which are particularly important for our study.

We have added:

(1) Page 4, Line 7: "In response to isoprene emission enhancements associated with total conversion of vegetated land to oil palm on the island of Borneo, Warwick et al. (2013) simulate a 20% increase in ozone at 500 hPa over Borneo and a 20% increase in peroxyacetyl nitrate (PAN) at 500 hPa downwind of Borneo over the Pacific Ocean. PAN is an organic nitrate that can undergo long-range transport before releasing its reactive $NO_X$ moiety (Moxim et al., 1996), providing a means for ozone formation in remote environments (Kotchenruther et al., 2001). The results of Warwick et al. (2013) suggest that regional isoprene emission changes have the capacity to alter ozone concentrations in the free troposphere and, therefore, induce a radiative forcing."

(2) Page 3, Line 33: "Ashworth et al. (2012) speculated a small global forcing impact from the increased isoprene emissions in their land-use change scenario, based on the small simulated global changes in the tropospheric burdens of ozone and the hydroxyl radical (OH). However, no study has directly quantified the global radiative impacts associated with the induced changes in atmospheric composition."

**5. L20-21 - This is the first mention of monoterpenes (aside from the abstract). I suggest for the authors also to discuss the atmospheric chemistry and composition effects of monoterpenes as per isoprene in the previous paragraph. For instance the surface flux measurements reported from the OP3 field study (Langford, Misztal) showed that natural forests are much stronger emitters of monoterpenes than oil palm plantations. And the previous investigations also included changes in monoterpene emissions which is not clear in L7-9 as the preceding lines had focused exclusively on isoprene.**

**The changes in monoterpenes and SOA seem to very much be of lesser importance to the authors than changes in isoprene and ozone here and throughout the manuscript. While I accept that the changes are smaller they still contribute to the overall radiative forcing reported by the authors and should be given full coverage in the main text and not just the SI**

The reviewer is correct in that we mainly focus on isoprene emission changes because of the larger change in isoprene (+6.5 TgC $y^{-1}$) relative to the change in monoterpenes (-0.5 TgC $y^{-1}$). Likewise, we deliberately focus on ozone more than SOA because of the stronger simulated radiative forcing from the ozone perturbations. Hence, the paper is not skewed. For example, the paper would be skewed if the primary focus was monoterpenes-SOA.

That said, we agree with the reviewer that the monoterpenes and SOA need to be given appropriate coverage in the main text and not just the SI (additions in bold, deletions crossed out; Page 2, Line 11): "Above-canopy flux measurements **taken in Borneo in 2008** indicate that, compared to the natural forests of maritime Southeast Asia (MSEA), oil palm plantations are much stronger emitters of the biogenic volatile organic compound (BVOC) isoprene ($C_5H_8$), **with mean midday fluxes about five times stronger from oil palm** (Langford et al., 2010; Misztal et al., 2011). The simultaneous large-scale contraction of low-isoprene-emitting natural forest area and expansion of high-isoprene-emitting oil palm plantations suggests a land-cover-change-driven increase in regional isoprene emissions over recent decades (Silva et al., 2016; Stavrakou et al., 2014). **Measurements indicate that the forests of MSEA emit monoterpenes, a class of BVOCs with chemical formula $C_{10}H_{16}$, but find negligible monoterpene emissions from oil palm (Langford et al., 2010; Misztal et al., 2011). Both** isoprene **and monoterpenes are**  precursors to the short-lived climate pollutants tropospheric ozone (Atkinson and Arey, 2003) and secondary organic aerosols (SOA) (Carlton et al., 2009; **Friedman and Farmer, 2018**); **as such, perturbations in regional isoprene and monoterpene emissions serve as an additional mechanism by which regional land cover change can affect air quality and climate**."

Added reference:
Friedman, B. and Farmer, D.K.: SOA and gas phase organic acid yields from the sequential photooxidation of seven monoterpenes, Atmos. Env., 187, 335–345, doi: 10.1016/j.atmosenv.2018.06.003, 2018.

With respect to the previous studies, it is our understanding that: Ashworth et al. (2012) only consider emission changes for isoprene, but do consider the impact that the resulting changes in atmospheric composition have on monoterpene processing; Warwick et al. (2013) likewise consider only isoprene emission changes for forest to oil palm conversion (their paper does not explicitly state how the atmospheric composition changes from the isoprene emission perturbations impact the simulated monoterpene chemistry, although this is presumably taken into account); and Silva et al. (2016) alter only the isoprene emission capacity (and not the monoterpene emission capacity) of their new oil palm land cover type relative to the forest PFT, but their results presumably take into account the effect of monoterpene emission changes associated with the loss of the various land covers to oil palm.

**6. L24 - Does this mean that the authors have only conducted a series of atmosphere-only model simulations? So there are no climate / land surface feedbacks included on-line?**

Atmosphere-only run is a standard technical term widely used by the World Climate Research Program (WCRP) Coupled Model Intercomparison Project (CMIP). It means that the global climate model uses prescribed observed sea surface temperatures and sea ice fields. Thus, the climate model does not have a fully coupled dynamic ocean. The term is in common usage in the global climate modeling community. Atmosphere-only simulations can be dynamically coupled to atmospheric chemistry and the land surface, as in our work.

**7. L24-27 - Actually I am now confused as to exactly what model simulations were performed. The authors have referred to atmosphere-only, chemistry-climate, and land surface models here. Exactly what configuration is being used?**

The reviewer is unfamiliar with standard terminology in the global climate and atmospheric chemistry modeling communities. See response to Point (6). Atmosphere-only means prescribed sea surface temperatures and sea ice fields. Atmosphere-only simulations can be dynamically coupled to atmospheric chemistry and the land surface, as in our work. Our description of the model set-up and configuration is clear, complete, and appropriate. No further changes are needed here.

**8. L27 - 2 deg x 2.5 deg is too coarse to adequately resolve the complexity and heterogeneity of the land mass and land cover in this region particularly given the sensitivity of BVOC oxidation and ozone production rate to VOC:NOx ratio.**

The reviewer is raising a longstanding query that concerns the entire large-scale global climate and chemistry mathematical modeling communities, way beyond the scope of this study, regarding what is actually needed in a global model (with associated limited computational resources) to reproduce large-scale composition impacts versus a highly localized mathematical representation of every real process on the ground tending to continuous resolution, many of which do not actually matter to the global radiative impact of ozone and SOA. This conflict commonly emerges between communities engaged in large-scale mathematical modeling versus communities engaged in local ecosystem-scale measurements. Nobody is ever surprised when it comes up in interdisciplinary work.

The reviewer's comment applies to the use of all global chemistry-climate (CCM) and global chemistry-transport (CTM) models for the study of the global radiative impacts of regional-scale changes in short-lived emission precursors. The "complexity and heterogeneity of the land mass and land cover and the sensitivity of BVOC oxidation and ozone production rate to VOC:NOx ratio" are NOT issues unique to the MSEA region. Undeniably, these issues are important in all chemical regimes and regions of the world where the large-scale atmospheric responses to short-lived emission precursor perturbations are being studied. State-of-the-science global CCMs and CTMs typically have horizontal resolution 1-2° latitude/longitude. The model horizontal and vertical resolution applied in this study is comparable to global CCMs and CTMs currently being used in the WCRP CMIP6 Aerchem-MIP and RF-MIP in support of the

forthcoming IPCC AR6 assessment, and the Task Force on Hemispheric Transport of Air Pollutants (HTAP). These international assessment programs each employ dozens of global models with 1-2° latitude/longitude resolution to quantify the impacts of local and regional short-lived precursor emission changes, including VOCs, in very different regions and regimes. It is a moot point that global CCMs and CTMs do not fully resolve the complexity and heterogeneity of land mass and land cover and other sub-grid phenomena. The models parameterize these conditions and processes.

By the reviewer's own logic, thousands of peer-reviewed publications in high-caliber journals, HTAP, Aerchem-MIP and RF-MIP, and short-lived climate forcers in IPCC AR6/AR5/AR4 are invalid. We do not agree. The goals of this work are to quantify the global radiative forcing of ozone and SOA changes due to the isoprene emission injection and altered BVOC fluxes as a result of recent human-induced land cover change in MSEA. Therefore, we have used a model framework that has been designed to simulate the global radiative forcing impacts from local and regional short-lived emission precursor perturbations, including, but not limited to, assessments by HTAP, Aerchem-MIP, RF-MIP, and a wide range of other international multi-model assessment programs over the past 20 years.

**9. L31 - I have reservations whether the chemistry mechanism employed here is suitable for the conditions encountered in this region. Although it is rapidly developing with the concomitant increases in anthropogenic emissions, much of island areas of the region are still low-NOx, high-VOC regimes. Older chemical mechanisms were designed for the typical chemistry encountered in mid- to high-NOx urban / industrial areas and considerable understanding has been gained of the very different oxidation pathways of (particularly) isoprene under lower NOx conditions. Have any of these updates been included in the chemistry here?**

Please see comment at the top of this document. In the original manuscript version, we neglected to provide a detailed enough description of the current chemical mechanism. Certainly, we too would have major reservations about a study using CBM04 to quantify composition impacts of a large isoprene emission injection in the tropics. The revised manuscript now includes a more detailed and accurate description of the chemical mechanism.

**10. p3 L8 - Are the monoterpenes emitted as a single lumped monoterpene species? Or at least in part speciated (e.g to specifically include a-pinene, b-pinene, d-limonene and others as is often done)? It should be noted that the monoterpene emissions algorithms included in Lathiere et al are in fact the Guenther et al. algorithms from 1995; at the very least this paper should be referenced here. Also, these algorithms assume that monoterpene emissions are entirely temperature dependent whereas more recent field measurements have shown that many species emit a proportion of monoterpenes directly (i.e. monoterpene emissions exhibit both light and temperature dependency, see e.g. Steinbrecher et al, Guenther et al. 2012). Are the authors confident that this is not the case for SE Asian plant species?**

We have added the Guenther et al. (1995) reference and a brief discussion of light-dependency. ModelE2 applies a lumped monoterpene species (Page 6, Line 28): "Temperature-dependent leaf-level monoterpene emissions, functionally $\alpha$-pinene, likewise vary by ecosystem type, similarly through prescription of PFT-specific basal emission rates (Guenther et al., 1995; Lathière et al., 2006). Recent work suggests that tropical monoterpene emissions exhibit both light and temperature dependency (Guenther et al., 2012; Jardine et al., 2015, 2017) that is not included in the emission algorithm here but may be explored in future work."

Added references:

Guenther, A., Hewitt, C.N., Erickson, D., Fall, R., Geron, C., Graedel, T., Harley, P., Klinger, L., Lerdau, M., McKay, W.A., Pierce, T., Scholes, B., Steinbrecher, R., Tallamraju, R., Taylor, J., and Zimmerman, P.: A global model of natural volatile organic compound emissions, J. Geophys. Res.-Atmos., 100, 8873–8892, doi: 10.1029/94JD02950, 1995.

Jardine, A.B., Jardine, K.J., Fuentes, J.D., Martin, S.T., Martins, G., Durgante, F., Carneiro, V., Higuchi, N., Manzi, A.O., and Chambers, J.Q.: Highly reactive light-dependent monoterpenes in the Amazon, Geophys. Res. Lett., 42, 1576–1583, doi: 10.1002/2014GL062573, 2015.

Jardine, K.J., Jardine, A.B., Holm, J.A., Lombardozzi, D.L., Negron-Juarez, R.I., Martin, S.T., Beller, H.R., Gimenez, B.O., Higuchi, N., and Chambers, J.Q.: Monoterpene 'thermometer' of tropical forest-atmosphere response to climate warming, Plant Cell Environ., 40, 441–452, doi: 10.1111/pce.12879, 2017.

**11. L15-19 - How were emission factors assigned to these additional land cover types? How do they differ from the standard land cover in this region in the default land surface map? Again this is critical to the results and should be included in the main text and not just the SI.**

On (Page 6, Line 22), we state: "For each PFT, the isoprene emission rate depends linearly on the fraction of electrons available to undergo isoprene synthesis, **the calculation of which requires prescription of a PFT-specific leaf-level isoprene basal emission rate (BER) at standard conditions.**"

We have moved the Table of model parameters (previously known as Table S2 in the Supplement) to the main text (now known as Table 1).

On Page 9, Line 20, we removed the list of references from this sentence (as this information is found in the footnotes of Table 1, now in the main text) and re-phrased to better describe what is found in the table: "Table 1 shows, for the new land cover types, the assigned physical parameters (including LAI and vegetation height), photosynthetic parameters, and leaf-level basal emission rates of isoprene and monoterpenes."

We describe the relationship between the isoprene BERs for the standard and new rainforest PFTs where this information is critical (Page 13, Line 10): "In this sensitivity analysis, the dipterocarp forest **isoprene** BER is increased by a factor of 12, **making it equivalent to the isoprene BER assigned to the standard evergreen broadleaf forest PFT in YIBs**."

**12. L31-32 - Of real importance to this study is the previous performance of the YIBs model in this region; the 2013 study was global. Do the referenced comparisons include field sites in maritime SE Asia?**

The 2013 global evaluation paper did include time-varying OP3 Borneo field measurements (e.g. Langford et al., 2010). The point of referencing the global-scale evaluation against a wide range of different ecosystems and regions is to demonstrate that the model has reasonable isoprene emission performance over a range of different ecosystems and regions. The present manuscript provides a comparison of oil palm and forest isoprene fluxes to those from the OP3 campaign in Borneo (Page 22, Line 8; Page 22, Line 25).

**13. p4 L14-15 - It seems odd to go the lengths of using 30m x 30m resolution land cover data in a model running at 2 deg/ x 2.5 deg**

The 30 m x 30 m resolution land cover dataset (i.e., the dataset of Gunarso et al., 2013) is re-gridded to the model resolution of 2° latitude x 2.5° longitude before application to the model (Page 8, Line 27). The purpose of applying the high-resolution dataset is because, as far as we know, this is the only land cover dataset available for this region that employs a wall-to-wall classification methodology that provides multiple years of data (using a consistent classification methodology for each year) and includes several land covers that are prevalent in Southeast Asia (e.g., oil palm and rubber plantations). The availability of such a dataset prevents us from needing to build a single dataset out of multiple datasets that were potentially derived using different classification methodologies or are from different time periods.

**14. L20-26 - Please clarify. Are you saying that in 1990 the only land cover data available shows natural forest (or whatever) in locations that were shown as oil palm in 2000 and 2010? What fraction of data is missing? Of the 1990 data what fraction is converted to oil palm by 2000 and 2010? Of the missing data what fraction is "converted" to oil palm by 2000 and 2010? It would be useful to have a feel for how substantial the "likely underestimation" might be.**

We have clarified the language (modifications bolded, Page 8, Line 19): "The Gunarso et al. (2013) classification for 1990 for Indonesia is likewise incomplete; in this dataset, the pixels classified for 1990 are principally those that are **oil palm in 1990 or eventually become oil palm**."

Thus (Page 8, Line 23): "Indonesian oil palm cover in 1990 is accurate within the limits of the classification methodology."

The rest of Indonesia is largely unclassified in 1990, as described, which is why we apply the year 2000 land cover to these pixels.

**15. L30 - Is dirt equivalent to the "bare ground" classification included in other land surface schemes?**

Fixed. We have amended the text to state "bare ground" rather than "dirt."

**16. p5 L3 - The authors say that these are "minor land cover types". Again it would be useful to be provided with sufficient information to judge just how minor. What fraction of land is included in these 5 types in Gunsaro et al classification?**

These minor land cover types account for 2.8% of pixels in both 1990 and 2010.

**17. L8-13 - Again it would be good to have an idea of the likely underestimation, and this should be relatively easy for the authors to achieve by applying an LAI reduction, as described, to the areas classified as "disturbed" in Gunsaro.**

For the forest class, around 43% is disturbed, while around 57% is undisturbed (these fractions are largely consistent across years). Dietz et al. (2007) report LAIs for various disturbance levels: undisturbed (6.2 $m^2$ $m^{-2}$), removal of small-diameter trees (5.3 $m^2$ $m^{-2}$), and removal of large-diameter trees (5.0 $m^2$ $m^{-2}$). Based on these LAIs, the mean LAI for the forest class (47% disturbed, 57% undisturbed) would be: (1) 0.57 x 6.2 $m^2$ $m^{-2}$ + 0.43 x 5.3 $m^2$ $m^{-2}$ = 5.8 $m^2$ $m^{-2}$ (assuming that the disturbed forest falls closer to the small-diameter removal category) and (2) 0.57 x 6.2 $m^2$ $m^{-2}$ + 0.43 x 5 $m^2$ $m^{-2}$ = 5.7 $m^2$ $m^{-2}$ (assuming that the disturbed forest falls closer to the large-diameter removal category). In our simulations, we assign a forest LAI of 6.0 $m^2$ $m^{-2}$, based on measurement of a natural forest plot in Malaysian Borneo (Fowler et al., 2011), which is an area included in our land cover change analysis. Thus, our assigned value is only about 3–5% higher than these rough estimates, which is a good approximation considering that we do not have any information about the level of disturbance of the "disturbed" forest patches in the land cover change dataset that we apply (that is, the classification of Gunarso et al. (2013)). In the Gunarso et al. (2013) classification, a "disturbed" forest patch has a reduced basal area with evidence of clearing or logging. Such classifications are not uncommon; for example, Margono et al., Nature Climate Change, 2014, use a "primary degraded forest" class, in which the forest has been fragmented or experienced selective logging or other disturbance.

**18. L21 - As noted above, Table S2 should be in the main text as these parameter values are critical to the results. The notes regarding their derivation can be left in SI. The values for the "standard" PFTs in YIBs for this region should also be shown in this table for comparison.**

We have moved this table and the footnotes, which are an integral part of the table, to the main text (now known as Table 1).

**19. L27-34 - There is a real mishmash of years for the various datasets. As the simulations are being conducted for a nominal 2010 (i.e. that is the climatology) with 1990 or 2010 SE Asia landcover why introduce further limitations / discrepancies by using Y2000 landcover for the rest of the world with vegetation characteristics derived using Y2000 meteorology only to change to 2010?**

We assume that the global radiative forcing impacts by ozone and SOA due to oil palm expansion in MSEA are insensitive to changes in the background land cover state outside of the MSEA region over the 1990–2010 period. Unfortunately, we do not always have available all observational or modeled data for each year for each boundary condition for model runs, which means that we sometimes need to combine datasets in appropriate ways to run simulations. Here, we use a dataset for 2000, which falls within the era of interest (1990–2010). We use the year 2000 rest-of-world land cover dataset specifically because we already had available the PFT-specific vegetation height parameters for the set of PFTs used in ModelE2-YIBs at the resolution used in ModelE2-YIBs. As we describe in the paper, we obtained the PFT-specific height parameters applied to the rest of the world vegetation from an existing 140-year simulation run with our model. This 140-year simulation was run using dynamic carbon allocation and applied the same rest-of-world land cover distribution that we apply here. Using this configuration, a 140-year simulation requires at least a few months of run time, which accounts only for the actual time that the model is integrating and does not include time spent in the simulation queue between re-submissions (since our cluster allows only one week of run time before the simulation must be re-submitted, the additional time spent in the run queue can be substantial). With unlimited computational resources, we could run an additional century-long simulation for year 1990 or 2010, but we don't have access to these resources, and, more importantly, it is unlikely that switching to 1990 and 2010 background land cover datasets has any meaningful influence on the results here. The benefit of doing so is not clear because we hold static the rest-of-world land cover map and physical vegetation characteristics (because we need to isolate the impacts of MSEA regional land cover change) and this dataset is a reasonable approximation of land cover and vegetation characteristics for this 1990–2010 era.

**20. p6 L33-34 - Could the authors please clarify how the simulations were driven with the meteorology? Was the same climate / meteorology applied for 13 years? Because there will be an effect of inter-annual variability on emissions, chemistry and therefore O3 and SOA formation; how has this been accounted for? Is this what the authors have attempted to do via the additional simulations?**

We have removed the incorrect description of the nudged winds. The quantified standard deviations (e.g., radiative forcing in Table 5 on Page 21) are based on internal interannual variability in the climate model. We have additionally performed a sensitivity simulation to assess the impact of using a different background climate (including emissions year) on the forcing results (Table 5 on Page 21).

**21. p7 L4-6 - This is the case for all current isoprene emissions models which are linked to PAR, T, CO2, soil moisture, etc. Please clarify what aspect the authors mean is the case "because" it is interactively linked OR remove the word "because"**

To improve clarity, we have re-phrased this (additions bolded, deletions crossed out; Page 12, Line 14): " Isoprene production in ModelE2-YIBs is  **calculated as a semi-mechanistic function of** photosynthetic carbon assimilation (Unger et al., 2013). Isoprene emissions are sensitive to simulated changes in the parameters that affect photosynthesis, including the background climate state (**e.g., temperature, PAR, and soil moisture**) and **the** atmospheric $CO_2$ concentration."

**22. L6-7 - Please give more detail of how monoterpene emissions are sensitive to climate as per isoprene**

We have changed "climate" to "temperature" (Page 12, Line 17): "Simulated monoterpene emissions are likewise sensitive to temperature shifts (Lathière et al., 2006)."

**23. L9-11 - But changing the landcover will also affect e.g. NOx emissions, either due to changes in fertiliser application or to changes in natural soil emissions. How have the authors accounted for this?**

**Because again the resolution of the global model will not be sufficient to pick up changes such as this simply by running a sensitivity test with a different background atmosphere.**

The reviewer raises a good point. We account for anthropogenic changes in NOx emissions and all other short-lived emission precursors by applying the MACCity inventory for anthropogenic emissions of carbonaceous aerosols and reactive gases (Granier et al., 2011). The MACCity inventory is partially based on the ACCMIP inventory, which is based on a multitude of global- and regional-scale emission inventories (Lamarque et al., 2010). MACCity includes agricultural NOx emissions. Climate-sensitive lightning NOx and soil NOx emissions are included in the simulations. Atmospheric NOx measurements in the region are extremely limited. A possible future work direction beyond the scope here is to exploit satellite NOx data to learn more about the NOx levels in the region.

**24. L28-30 - Why have the authors not used the measured emission rate in the first instance?**

Our model requires a leaf-level BER (YIBs has its own canopy up-scaling scheme consistent for carbon, water, energy, and BVOCs). Therefore, we adopted a strategy that maximizes use of the

limited available BVOC flux data in the region. First, we implemented published leaf-level isoprene BERs to all PFTs including oil palm (Table 1; now in the main text). Then, we used the raw measured fluxes (not canopy-level BERs) from the OP3 campaign to evaluate and validate the model's simulated above-canopy fluxes (Page 22, Line 8; Page 22, Line 25). This strategy is more physically realistic, and provides a better benchmark than, for example, artificially forcing the model to reproduce the OP3 canopy-scale BERs as a boundary condition.

**25. L30-32 - Why? The authors specifically introduced this land class because measurements had shown that the global emission factors were not suitable. The work reported in Langford and Misztal suggested that emission factors were out by a factor of 3 so using 12 seems rather extreme.**

The rationale is simply a sensitivity study to examine the impacts when the forest emits with a default isoprene BER for tropical rainforest, that is where the factor of 12 comes from. See response to point (11).

**26. p8 Table 2 - this seems to imply that the isoprene emission factor applied to oil palm is as measured in the standard run but half measured in the OPber sensitivity tests which appears to contradict what the authors have described in the previous page.**

The values reflect the difference between the canopy-scale and leaf-level BERs for oil palm isoprene emission.

**27. p9 L3-4 - It would also be good to see how well 1990land_base and 1990land_1990atm GPP compare with measured GPP**

We amended the text to state (Page 14, Line 9): "Simulated global gross primary production (GPP) for 2010 is 124 PgC $y^{-1}$ (simulation 2010land_base), which almost precisely matches an estimate derived from flux-tower measurements that is representative of 1998–2005: 123 ± 8 PgC $y^{-1}$ (mean ± 1 standard deviation; Beer et al., 2010). The simulated 1990 global GPP of 108 PgC $y^{-1}$ (simulation 1990land_1990atm) is outside of the 1-standard-deviation range of the observation-based mean, but falls within the 95% confidence interval (102–135 PgC $y^{-1}$; Beer et al., 2010)."

The small change in GPP from 1990–2010 land cover change (2010land_base minus 1990land_base) is described later in the text, so providing the value for 1990land_base here would be repetitive.

**28. L5 - please define contemporary, because Table 5 in the Guenther paper contains estimates from early 90s to around 2008. Again it would be useful to see the emissions estimates for both 1990 and 2010 land cover and climates here.**

The estimates from Guenther et al. (2012) Table 5 are from references that were published over the period 1995–2011. The table does not indicate the year represented by each estimate. However, the forcing datasets (e.g., those for "weather" and "LAI") listed in the table for each of the estimates suggest that the estimates are from the contemporary (modern day) period as opposed to future projections or the pre-industrial era. We use these estimates to show that the global emissions of isoprene and monoterpenes that are simulated in our study are reasonable. We also compare our 2010 isoprene estimate to another estimate representative of the 1971–2000 mean.

We have expanded the text to include our 1990 estimates (Page 14, Line 15): "The model estimates for 1990 (325 TgC $y^{-1}$ isoprene and 90 TgC $y^{-1}$ monoterpenes for simulation 1990land_1990atm) and 2010 (363 TgC $y^{-1}$ isoprene and 77 TgC $y^{-1}$ monoterpenes for simulation 2010land_base) fall within these ranges. Using the same process-based, leaf-level isoprene production algorithm employed in ModelE2-YIBs, although driven with different forcing datasets, Hantson et al. (2017) predict contemporary isoprene emissions (385 TgC $y^{-1}$; 1971–2000 mean) that are 18% higher than those predicted here for 1990 and only 6% higher than those predicted here for 2010."

**29. L10-19 - Here and throughout, although the authors describe this as a study of how BVOC emissions changes have affected the region the manuscript is entirely dominated by consideration of isoprene. While this is understandable given that total regional isoprene emissions are more than 5x those of monoterpenes I think the paper would benefit from more consideration of monoterpene emissions and impacts as monoterpenes and isoprene have different effects on atmospheric composition and chemistry. I suggest the authors also pay careful consideration to their use of the catch-all BVOC as this study appears only to include isoprene and monoterpenes.**

See response to Point (5). We will continue with the use of "BVOC" to describe isoprene and monoterpenes. There is a growing body of literature on the impacts of isoprene and monoterpenes on regional and global radiation budgets and short-lived climate forcers (e.g., Heald and Geddes, 2016; Hollaway et al., 2017; Scott et al., 2017, 2018; Unger, 2014a,b). To our knowledge, there is no current published evidence that any other BVOC species have statistically significant large-scale global and regional radiative effects. Because of their extremely short-lifetimes, it is likely that other highly reactive emitted compounds have much more localized impacts.

References:

Heald, C.L. and Geddes, J.A.: The impact of historical land use change from 1850 to 2000 on secondary particulate matter and ozone, Atmos. Chem. Phys., 16, 14997–15010, doi: 10.5194/acp-16-14997-2016, 2016.

Hollaway, M.J., Arnold, S.R., Collins, W.J., Folberth, G., and Rap, A.: (2017), Sensitivity of midnineteenth century tropospheric ozone to atmospheric chemistry-vegetation interactions, J. Geophys. Res.-Atmos., 122, 2452–2473, doi:10.1002/2016JD025462, 2017.

Scott, C.E., Monks, S.A., Spracklen, D.V., Arnold, S.R., Forster, P.M., Rap, A., Äijälä, M., Artaxo, P., Carslaw, K.S., Chipperfield, M.P., Ehn, M., Gilardoni, S., Heikkinen, L., Kulmala, M., Petäjä, T., Reddington, C.L.S., Rizzo, L.V., Swietlicki, E., Vignati, E., and Wilson, C.: Impact on short-lived climate forcers increases projected warming due to deforestation, Nat. Commun., 9:157, 1–9, doi: 10.1038/s41467-017-02412-4, 2018.

Scott, C.E., Monks, S.A., Spracklen, D.V., Arnold, S.R., Forster, P.M., Rap, A., Carslaw, K.S., Chipperfield, M.P., Reddington, C.L.S., and Wilson, C.: Impact on short-lived climate forcers (SLCFs) from a realistic land-use change scenario via changes in biogenic emissions, Faraday Discuss., 200, 101–120, doi: 10.1039/c7fd00028f, 2017.

Unger, N.: Human land-use-driven reduction of forest volatiles cools global climate, Nat. Clim. Change, 4, 907–910, doi: 10.1038/NCLIMATE2347, 2014a.

Unger, N.: On the role of plant volatiles in anthropogenic global climate change, Geophys. Res. Lett., 41, 8563–8569, doi: 10.1002/2014GL061616, 2014b.

**30. L16-18 - Similarly to the previous comment, rubber plantations don't make such a strong contribution to total BVOC emissions (e.g. compared to shrubs). However they do appear to dominate the monoterpene budget and might therefore have a strong role in SOA formation rate and yield. I suggest re-phrasing this sentence to make this distinction clearer.**

This is a good point, and we have re-phrased the sentence (Page 15, Line 3): "The strong contributions made by rubber and oil palm plantations to the regional monoterpene and isoprene budgets, respectively, underscore the importance of explicitly accounting for these land covers in regional land use and land cover change analyses."

**31. L20-21 - but as previously noted by the authors, their assumptions in assigning land cover has likely led to an underestimation of deforestation. Please could the authors make some attempt to quantify the uncertainty in the changes in GPP. In addition to the method applied to fill data gaps and the non-inclusion of changes in LAI due to disturbance, the authors have used 2010 climatology in both cases which will affect the estimated GPP. I am assuming that the figure quoted here is based on difference between the two base simulations.**

We purposefully apply the 2010 physical climate state in order to isolate the impacts of the human-induced land cover change only on global radiative forcing by ozone and SOA. For example, the physical climate changes themselves between 1990 and 2010 induce changes in ozone and SOA forcing.

**32. L25-26 - this could perhaps be better phrased as the loss of dipterocarp forest is due to its replacement by another land cover; as this is often oil palm so overall isoprene emissions go up.**

We keep the original phrase that is focusing on the changes in dipterocarp forest alone and useful to quantify (Page 15, Line 12): "The large loss of dipterocarp rainforest had little impact on isoprene emissions (-0.1 TgC y-1), as this PFT is a weak isoprene emitter."

**33. L31 - I think the authors mean "Considering only the grid-cells that are majority . . ." as they then go on to give surface O3 concentrations in 2 sub-regions rather than ALL of the study area.**

Fixed.

**34. p10 L3-5 - These measurements would seem to support the low ozone concentrations simulated by the model over Malaysian Borneo; however I am not sure they provide evidence of ozone concentrations over ALL forested areas in the region particularly as the authors are comparing annual mean concentrations with measurements made for one particular (short) period.**

Good point. However, we only have measurements for Malaysian Borneo, so we use them as a proxy for the forested parts of the entire region.

**And related to this, are the authors intending to imply that they simulate much higher levels of ozone over the non-forested areas in the region? In which case, what are the average ozone concentrations for e.g. peninsular Malaysia which is far more industrialised? i.e. it is likely the case for many of the included grid cells that the proportion of the grid cell not occupied by forest in Borneo and New Guinea is ocean, whereas in peninsular Malaysia, etc many more will contain urban / industrial areas with higher NOx emissions.**

Yes, this is clearly visible in Figure S4.

**35. L7-8 - This would be a suitable place for the authors to emphasise the difference between the effects of monoterpenes vs isoprene, with isoprene oxidation more implicated in ozone production and loss rather than SOA formation.**

We added this statement (Page 15, Line 27): "Isoprene oxidation is more implicated in ozone production and loss rather than SOA formation (whereas monoterpenes are more implicated in SOA formation)."

**36. L8-9 - Please comment on the possible reasons for the observed enhancements over the ocean.**

This explanation was originally provided later in the paper, but we have now moved it to this location.

**37. L7-20 - The authors present and discuss only annual average surface concentrations. This masks seasonal changes in magnitude, sign and distribution. For example, Ashworth et al reported different patterns of ozone and SOA changes depending on wind direction between the two monsoon wind periods. And given the NOx-sensitivity of the region, the relative position of NOx and VOC sources can become even more important at different times depending on wind speed and direction. This is likely to become increasingly important as the region continues to industrialise and oil palm plantations continue to expand into areas that are currently urban / industrial rather than remote. Please discuss these limitations in the study.**

The reviewer's comments would be more relevant to a surface air quality study rather than a study focused on global radiative forcing. The global mean annual average radiative forcing metric is used because it is a linear predictor of global mean surface air temperature response at steady state. Therefore, we focus on annual average analyses in this study (e.g. IPCC AR5, Myhre et al., 2013). Global and regional radiative forcing effects of perturbations to short-lived precursor emissions are typically reported on an annual-mean basis. The paper is already getting rather too long and therefore we do not include seasonal surface changes in the manuscript.

The PhD thesis "Forcings and feedbacks in the climate system: The role of reactive compounds in the atmosphere, Yale University, K. L. Harper, 2018" reports seasonal changes: "Surface ozone reductions are simulated over Peninsular Malaysia, Sumatra, and Borneo in all seasons for $\Delta$LC (Figure 3.12). The changes in circulation and precipitation associated with the boreal winter (DJF) and boreal summer (JJA) monsoons are the likely sources of the small seasonal variations in the distribution of the surface ozone changes. The location of peak ozone enhancement over the marine environment shifts from west of Sumatra (in DJF and MAM) to north of Borneo (in JJA). Negligible changes are simulated over New Guinea in all seasons."

[Figure]

**Figure 3.12.** Changes in seasonal-average surface ozone mixing ratios (ppbv) for ΔLC.

**38. L16-19 - What is the resolution of the NOx emissions input? As previously noted I do have concerns over the capability of the model to resolve the heterogeneity of this region and Hewitt et al. 2010 demonstrated the sensitivity of the atmospheric chemistry in this region to NOx levels over a range of BVOC emissions**

The emissions input resolution corresponds to the global model resolution. See response to Point (23). Hewitt et al. (2010) used a box model. We have reservations about using a box model to simulate regional ozone air quality changes. Box models are appropriate tools to use to understand reactive radicals unaffected by transport processes and can be assumed to be in steady state in the atmosphere. Ozone has a relatively long lifetime and is strongly determined by transport and physical processes in the atmosphere. Using a box model designed to understand radical reaction pathways and kinetics to project changes in regional ozone surface air quality is just plain wrong. One may obtain some insights into key reaction pathways and important chemical species, but the projected changes in ozone concentrations aren't particularly useful in the absence of atmospheric physics and transport.

**39. L28 - Please re-phrase; "inflated" sounds as if the authors applied an arbitrary increase whereas in both cases the scenarios in which NOx emissions were increased were based on the differences observed between forest and plantation.**

We have re-phrased this sentence (Page 16, Line 16): "Both studies found that increasing the $NO_X$ emissions in the region of land conversion (to account for enhanced fertilizer application and industrial processing of the oil palm) enhanced surface ozone concentrations (Ashworth et al., 2012; Warwick et al., 2013)."

Our extended introduction (which is detailed in our response to comment 1 for reviewer #3) also mentions the reason for the enhanced $NO_X$ emissions in these studies.

As a point of clarification, the increased NO$_X$ emissions in the Warwick et al. (2013) and Ashworth et al. (2012) studies were not entirely based on the observed differences between forest and plantation. In both studies, they increase NO$_X$ emissions to represent increased fertilizer application of the oil palm (it appears that this is based on the observations), but they also include emissions based on increased industrial processing and, in the case of the Warwick et al. (2013) study, transportation. The transportation and processing emissions are based on estimates of energy requirements for these activities. In the Warwick et al. (2013) study, in the simulation where they apply enhanced NO$_X$ emissions in the oil palm landscape, the applied NO$_X$ emissions are more than 3.5x those inferred by Hewitt et al. (2009) for the oil palm landscape (0.07 mg(N) m$^{-2}$ h$^{-1}$ in their simulation vs. 0.019 mg(N) m$^{-2}$ h$^{-1}$ from the Hewitt et al. (2009) study). Hewitt et al. (2009) inferred fluxes for the forest landscape of 0.009 mg(N) m$^{-2}$ h$^{-1}$, indicating that the NO$_X$ fluxes were only about 2x as high for oil palm relative to forest. Warwick et al. (2013) apply a factor of 7 increase in NO$_X$ emissions relative to their baseline case.

**40. L34 - A likely key difference between the work of Silva and that presented here is that Silva applied the GEOS-Chem model at a resolution of 0.5deg x 0.667deg, a far more appropriate resolution for this highly complex region.**

Perhaps more appropriate if the goal is to quantify regional surface air quality impacts associated with regional oil palm expansion. Again, that is not our goal here. Our study is not a regional air quality study, rather, we quantify the global radiative perturbation associated with atmospheric composition changes. As such, we apply a model with the typical resolution used by IPCC CMIP6 and HTAP for studying the global radiative impacts of regional perturbations to the short-lived precursor emissions. See response to point (8). The reviewer may consider that simply increasing horizontal resolution without changing the model's sub-grid parameterizations, processes and mechanisms does not imply an improvement in simulation accuracy. The reviewer seems to assume an automatic increase in accuracy. It depends on the linearity of the processes and impacts involved. For example, in the NOx-limited regime, ozone production has a linear dependence on NOx concentrations (Introduction to Atmospheric Chemistry, Daniel J. Jacob). Therefore, the coarse resolution grid is simply an average of the higher resolution version. Certainly, increased resolution does give more output information because the grid cell numbers have increased and that is important for regional air quality applications. GEOS-Chem is an excellent model to study regional air quality and large-scale composition changes at all the horizontal resolutions at which is it available.

Reference:
Jacob, D.J.: Introduction to Atmospheric Chemistry, Princeton University Press, 1999.

**41. p11 L1-3 - But how does deposition change in YIBs which unlike GEOS-Chem couples the atmosphere to a process-based parameterisation of stomatal conductance? Otherwise I'm not sure what point the authors are trying to make here**

We have updated this description (Page 16, Line 25): "The simulated changes in atmospheric composition might be a response not only to altered isoprene and monoterpene emissions, but also to changes in the deposition of atmospheric species induced by changes in leaf density (Wong et al., 2018) or related changes, such as surface roughness, stomatal conductance, and evapotranspiration, that are affected by the applied changes in land cover distribution. Here, the relative changes in regional ozone deposition rates (-19.7 to +4.3%) are similar to the relative changes in regional surface-level ozone concentrations (-18.3 to +4.3%) from 1990–2010 regional land cover change, in part because the ozone deposition rate depends on the atmospheric concentration change. While increased isoprene emission leading to increased isoprene ozonolysis drives ozone losses near the surface, a formal quantitative attribution analysis disentangling the relative roles of emission and deposition changes requires further complex sensitivity simulations that are beyond the scope of this analysis. In their analysis of Southeast Asian oil palm expansion, Silva et al. (2016) used sensitivity studies to determine that the induced BVOC emission changes, rather than altered deposition rates from LAI changes, were almost exclusively responsible for the simulated surface ozone changes."

The sentences are simply describing that isoprene oxidation under low NOx conditions leads to ozone loss (by direct reaction). That is the dominant effect determining the ozone reductions near the surface in the large-scale models.

Added reference:

Wong, A.Y.H., Tai, A.P.K., and Ip, Y.-Y.: Attribution and statistical parameterization of the sensitivity of surface ozone to changes in leaf area index based on a chemical transport model, J. Geophys. Res.-Atmos., 123, 1883–1898, doi: 10.1002/2017JD027311, 2018.

**(42) Figure 1 - Please add a panel showing clearly where the changes in land cover were made. In the SI the figures imply that the changes were made only to the islands of SE Asia; here panel (a) shows a wider SE Asia than this. I suggest it would be useful for the authors to add a bounding box in each of (a) to (c) to show where the LCC occurred. I also suggest that panels should be added to show typical absolute concentrations (perhaps best done with 2010 base) of O3, SOA, isoprene and monoterpene emissions and changes in these emissions.**

Figure S2 (previously known as Figure S1) shows clearly where the land cover changes were made (also see new Figure S1). We also describe in the text (Page 8, Line 14) that the land cover classification that is applied encompasses Papua New Guinea, Malaysia (Peninsular Malaysia, Sabah, and Sarawak), and three regions of Indonesia (Kalimantan, Papua, and Sumatra). In Figure 1, we show a wider region than this because we are interested in the surface ozone changes in the broader region of Southeast Asia (i.e., not only where the land cover changes occur).

In the Supplement, we show a number of plots for the 2010 base case, including Figure S3 and Figure S4. We reference these plots at relevant places in the text.

**43. P12 L3–5 These 2 sentences appear to be saying the same thing; are both needed?**

We agree that these two sentences are closely related, but we have retained both as they are describing slightly different things and refer to different panels of Figure 1: (1) the change in the horizontal distribution of ozone with decreasing atmospheric pressure (Figure 1c) and (2) the change in the global-mean ozone enhancement with decreasing pressure (Figure 1d).

**44. L5-6 - Please state the heights/pressures being used define upper and lower troposphere; as evident from Figure 1 (d) the reported average changes in ozone will be critically dependent on this boundary.**

We have added this information to this sentence (Page 18, Line 9): "Considering the troposphere, the global-mean ozone enhancement from regional land cover change is on the order of 0.5 ppbv in the upper troposphere (e.g., at 237 hPa), compared to < 0.1 ppbv in the lower troposphere (at pressures > 875 hPa)."

**45. L6-7 - This seems somewhat negligible (?)**

Yes.

**46. L19-21 - Would suggest that the authors re-order this sentence to aid readability; the previous sentence considered isoprene so would be more logical to start with isoprene here: e.g. "When transport-driven isoprene . . ."**

Fixed.

**47. L26 - "-5N"? Please use "5S" for consistency with Figure 2.**

Fixed.

**48. L28 - Wolfe et al seems an odd choice of primary reference for the formation of HCHO from isoprene as HCHO columns have been used as a proxy since Palmer et al. 2001, 2006 and several authors since (including Palmer et al 2003) have considered the relative contributions of other VOCs to HCHO which seems of real relevance here as this study considers monoterpenes as well as isoprene, although as previously commented the text seems rather skewed toward isoprene.**

We disagree. We prefer to keep the Wolfe et al. (2016) reference here because it is an important and insightful analysis based on USA field measurement data characterizing the HCHO-isoprene relationship under different NOx. There is no need to start talking about satellite HCHO columns as a proxy for isoprene emission here at this point in the paper; it would be a distraction. However, a really interesting future study could examine long-term changes in HCHO columns in the MSEA region along with satellite NOx.

**49. p13 Figure 2 - Why are monoterpenes and SOA not included here? As previously noted, monoterpene emission changes and atmospheric composition impacts are barely covered.**

Please see response to point (5) above. In the revised manuscript we discuss the monoterpenes and SOA more upfront in the manuscript.

We discuss the land-cover-change-driven monoterpene emissions changes in Sect. 3.1, and these are plotted in Figure S3. Regional changes in surface SOA are plotted in Figure S6.

We now state the change in the global SOA burden (Page 20, Line 32): "The global ozone perturbation induced a positive forcing of $+9.2 \pm 0.7$ mW m$^{-2}$, offset only slightly by a negative forcing ($-0.8 \pm 0.1$ mW m$^{-2}$) induced by a 1.4% enhancement ($+6.5$ Gg) in the global burden of largely reflective SOA particles. (The regional change in SOA is plotted in Figure S6.)"

The simulated global annual-mean burden of biogenic SOA is 0.46 Tg in the 2010 base simulation (2010land_base). A recent study using the UKCA model calculates the annual-mean SOA burden, considering isoprene and monoterpene precursors, as 0.41 Tg (Kelly et al., ACP 2018). In the MSEA region (here, the region shown in Figure 1a), the maximum surface SOA concentration in 4.1 µg m$^{-3}$, occurring over central Sumatra, with most grid cells showing concentrations of < 2 µg m$^{-3}$. Previous global model simulations have reported SOA concentrations of the same order of magnitude in this region (Hoyle et al., 2007; Yu, 2011). Yu (2011) simulated regional SOA concentrations of < 2 µg m$^{-3}$, similar to the results of the 2010land_base simulation.

References:

Hoyle, C.R., Berntsen, T., Myhre, G., and Isaksen, I.S.A.: Secondary organic aerosol in the global aerosol–chemical transport model Oslo CTM2, Atmos. Chem. Phys., 7, 5675–5694, doi: 10.5194/acp-7-5675-2007, 2007.

Kelly, J.M., Doherty, R.M., O'Connor, F., and Mann, G.W.: The impact of biogenic, anthropogenic, and biomass burning volatile organic compound emissions on regional and seasonal variations in secondary organic aerosol, Atmos. Chem. Phys., 18, 7393–7422, doi: 10.5194/acp-18-7393-2018, 2018.

Yu, F.: A secondary organic aerosol formation model considering successive oxidation aging and kinetic condensation of organic compounds: Global scale implications, Atmos. Chem. Phys., 11, 1083–1099, doi: 10.5194/acp-11-1083-2011, 2011.

**50. L10-11 - but atmospheric concentrations of CO are of the order of 60-120 so this is a small relative change.**

Yes.

**51. L17-19 - Previously the authors have strongly made the case that this region is low-NOx. What is the yield of alkyl nitrates in ModelE2-YIBs from BVOC oxidation under low-NOx conditions as this seems more pertinent than commenting on yields in high- nix environments?**

Fixed. This is a typo error. We have modified the sentence to: "In OH-initiated oxidation of isoprene in the presence of NOx,…."

**52. p14 L8-9 - Although there is a clear mis-match between the spatial distribution of the enhancement in nitrates and reduction in NOx concentrations. And in particular, the for-mation of alkyl nitrates appears particularly increased at the surface where no change in NOx is evident. Could the authors comment on the reason for this?**

We decided to show changes in concentrations in this plot in recognizable commonly used units, rather than fractional percentage changes. There are large differences in absolute concentrations between the different species in this plot. The NOx does decrease near the surface corresponding the isoprene-induced alkyl nitrate formation, but it does not show up on the plot because the absolute changes are so small on this color bar (relative to the changes in the upper troposphere).

**53. L11-15 - However, unless the authors have incorporated the "new" isoprene oxidation pathways under low-NOx conditions in the chemistry mechanism (in which case this needs to be made clear in the model description section) this is more a model artefact reflecting the atmospheric chemistry community's understanding of HOx chemistry in 1999. Please clarify the isoprene oxidation scheme in ModelE2-YIBs. Is it really still CBM-4?**

Defunct. No, it is not. Please see comment in top of document at response to point (9).

**54. L18-19 - Again the issue of BVOC or isoprene or isoprene+monoterpenes; which is being considered here?**

We altered about 20 places in the text where we previously used the term "BVOCs," now providing more specificity regarding which BVOCs are considered in the statements.

**55. L22-23 - Please justify why the model is being driven with a single (repeated) year of meteorology. While it is useful to know the uncertainties associated with internal model variability it would be of far more use to know how inter-annual variability in climate / meteorology affects the radiative forcings calculated here as ultimately what is of real interest is how future LCC in the region might play out.**

Addressed in point (20).

**56. L23-24 - Now we come back to one of my chief concerns with the manuscript itself. This is now the first mention of aerosol changes. Although the abstract and introduction mentioned SOA as well as ozone the changes in SOA were not presented or discussed anywhere in the results section. If the authors wish to include the effects of aerosol on radiative forcing it is essential that the changes in tropospheric aerosol concentrations are introduced and discussed prior to this; likewise monoterpenes. Is the authors' reluctance to fully consider monoterpenes and SOA due to deficiencies in the chemistry mechanism and/or gas-to-particle partitioning in the ModelE2-YIBs model?**

Defunct. Please see responses to point (5) and point (49). No, the initial "reluctance" to devote large sections of text in the manuscript is because the regional monoterpene emissions change and global radiative impacts from SOA are tiny, especially compared to the isoprene and ozone changes. Furthermore, as the reviewer states, understanding of SOA production mechanisms is rapidly changing and associated with large uncertainties. The global SOA modeling community has concerns about the validity of the 2-product scheme (e.g., Tsigaridis et al., 2014). Many recently published global SOA model studies in the high impact magazines use fixed yield approaches to SOA production (i.e., the original 1990s approach), for example: Rap et al., Nature Geoscience, 2018; Scott et al., Faraday Discussions, 2017; Scott et al., Nature Geoscience, 2017; Scott et al., Nature Communications, 2018. To address this issue, we have added a new uncertainty analysis in the Conclusions section based on fixed yields for SOA (Page 25, Line 10): "Our study has several limitations. The radiative forcing results are likely sensitive to the isoprene chemical mechanism, SOA production scheme, and convective transport and atmospheric transport schemes in the model. For example, this study applies the two-product scheme for SOA production (Tsigaridis and Kanakidou, 2007), but the appropriateness of using such schemes in global models is still under debate (e.g., Tsigaridis et al., 2014). Many recent global SOA model studies use fixed SOA yields for calculating SOA production from isoprene and monoterpene oxidation (e.g., Rap et al., 2018; Scott et al., 2017, 2018). For the ΔLC analysis, the global-mean SOA radiative forcing per unit of SOA burden change is -115 mW m$^{-2}$ Tg$^{-1}$. This value is largely consistent across the sensitivity analyses, ranging from -112 mW m$^{-2}$ Tg$^{-1}$ to -119 mW m$^{-2}$ Tg$^{-1}$. This metric can be used to estimate the SOA radiative forcing induced by the simulated isoprene and monoterpene emission changes under the assumption of fixed SOA yields. Assuming fixed SOA yields of 10% from the simulated monoterpene emission changes (e.g., Tsigaridis et al., 2014) and 1% from the simulated isoprene emission changes (lower end of range suggested by Kroll et al., 2005), in conjunction with the SOA forcing per burden metric, results in an SOA forcing of -2.5 mW m$^{-2}$ from 1990–2010 land cover change

(i.e., ΔLC analysis). The SOA radiative forcing based on fixed SOA yields is more than three times stronger than, but of the same sign as, the SOA radiative forcing calculated by the global model; in both cases, the SOA radiative forcing is negligible and partially offsets the positive forcing from ozone. For the ΔLC analysis, the cumulative radiative forcing, considering impacts of both ozone and SOA changes, is 8.4 mW m$^{-2}$ computed by the model and 6.7 mW m$^{-2}$ computed using the simulated ozone forcing plus the SOA forcing computed here using fixed SOA yields. That is, using fixed SOA yields, the total radiative forcing would be slightly smaller in magnitude than, but the same sign as, the forcing simulated by the model. Several recent studies have applied slightly larger SOA yields: +14.3% from monoterpenes and +3.3% from isoprene (by mass; Rap et al., 2018; Scott et al., 2017, 2018). Using these larger SOA yields for the ΔLC analysis results in an SOA forcing of -19.4 mW m$^{-2}$ and a total radiative forcing, taking into account the ozone forcing, of -10.2 mW m$^{-2}$, which is the opposite sign of that simulated by the model (+8.4 mW m$^{-2}$). This analysis indicates that uncertainty associated with biogenic SOA yields from isoprene and monoterpene oxidation has a strong influence on the quantified forcing."

We then added to the abstract (Page 1, Line 18): "The sign of the net forcing is sensitive to uncertainty in the SOA yield from BVOCs."

References:

Kroll, J.H., Ng, N.L., Murphy, S.M., Flagan, R.C., and Seinfeld, J.H.: Secondary organic aerosol formation from isoprene photooxidation under high-NO$_X$ conditions, Geophys. Res. Lett., 32, L18808, doi: 10.1029/2005GL023637, 2005.

Rap, A., Scott, C.E., Reddington, C.L., Mercado, L., Ellis, R.J., Garraway, S., Evans, M.J., Beerling, D.J., MacKenzie, A.R., Hewitt, C.N., and Spracklen, D.V.: Enhanced global primary production by biogenic aerosol via diffuse radiation fertilization, Nat. Geosci., et al., Nat Geoscience, doi: 10.1038/s41561-018-0208-3, 2018.

Scott, C.E., Arnold, S.R., Monks, S.A., Asmi, A., Paasonen, P., and Spracklen, D.V.: Substantial large-scale feedbacks between natural aerosols and climate, Nat Geoscience, 11, 44–48, doi: 10.1038/s41561-017-0020-5, 2018.

Scott, C.E., Monks, S.A., Spracklen, D.V., Arnold, S.R., Forster, P.M., Rap, A., Äijälä, M., Artaxo, P., Carslaw, K.S., Chipperfield, M.P., Ehn, M., Gilardoni, S., Heikkinen, L., Kulmala, M., Petäjä, T., Reddington, C.L.S., Rizzo, L.V., Swietlicki, E., Vignati, E., and Wilson, C.: Impact on short-lived climate forcers increases projected warming due to deforestation, Nat. Commun., 9:157, 1–9, doi: 10.1038/s41467-017-02412-4, 2018.

Scott, C.E., Monks, S.A., Spracklen, D.V., Arnold, S.R., Forster, P.M., Rap, A., Carslaw, K.S., Chipperfield, M.P., Reddington, C.L.S., and Wilson, C.: Impact on short-lived climate forcers

(SLCFs) from a realistic land-use change scenario via changes in biogenic emissions, Faraday Discuss., 200, 101–120, doi: 10.1039/c7fd00028f, 2017.

Tsigaridis, K., Daskalakis, N., Kanakidou, M., Adams, P.J., Artaxo, P., Bahadur, R., Balkanski, Y., Bauer, S.E., Bellouin, N., Benedetti, A., Bergman, T., Berntsen, T.K., Beukes, J.P., Bian, H., Carslaw, K.S., Chin, M., Curci, G., Diehl, T., Easter, R.C., Ghan, S.J., Gong, S.L., Hodzic, A., Hoyle, C.R., Iversen, T., Jathar, S., Jimenez, J.L., Kaiser, J.W., Kirkevåg, A., Koch, D., Kokkola, H., Lee, Y.H., Lin, G., Liu, X., Luo, G., Ma, X., Mann, G.W., Mihalopoulos, N., Morcrette, J.-J., Müller, J.-F., Myhre, G., Myriokefalitakis, S., Ng, N.L., O'Donnell, D., Penner, J.E., Pozzoli, L., Pringle, K.J., Russell, L.M., Schulz, M., Sciare, J., Seland, Ø., Shindell, D.T., Sillman, S., Skeie, R. B., Spracklen, D., Stavrakou, T., Steenrod, S.D., Takemura, T., Tiitta, P.,Tilmes, S., Tost, H., van Noije, T., van Zyl, P.G., von Salzen, K., Yu, F., Wang, Z., Wang, Z., Zaveri, R. A., Zhang, H., Zhang, K., Zhang, Q., and Zhang, X.: The AeroCom evaluation and intercomparison of organic aerosol in global models, Atmos. Chem. Phys, 14, 10845–10895, doi: 10.5194/acp-14-10845-2014, 2014.

**57. L27-30 - Given the levels of uncertainty in calculations of radiative forcing, and the limitations previously identified with this study, it is hard to see a net forcing of 0.008 Wm-2 as globally significant. The more substantial (but highly localised changes over the Indian Ocean) could be of real interest in terms of how they affect the Indian monsoon which has seen significant changes in recent years, but this is not explored by the authors. Again, however, these are temporally "averaged" results whereas the "interesting" effects are likely to be temporally as well as spatially localised. This would be another extremely interesting avenue to explore. Do the changes peak at times and locations when small changes in climate-relevant atmospheric components matter?**

The reviewer misuses the term "significant" above. We do provide uncertainty estimate for the net global climate impact ($+8.4 \pm 0.7$ mW m$^{-2}$). We have rebutted the reviewer's previous concerns and misunderstandings (please see all comments above). We agree with the reviewer that the global ozone radiative forcing from the oil palm expansion in MSEA is small. That is the main finding of this study and stated several times in the Conclusions section. We did not know before we launched the experiments what would be the final results.

In the Conclusions section we have added (Page24, Line 17): "For comparison, the global ozone forcing driven by the 1990–2010 land cover change in MSEA is at the low end of the range of estimates for ozone forcing from global anthropogenic emission source sectors in year 2000 (+5 to +80 mW m$^{-2}$): for example, industry = +15 mW m$^{-2}$; household biofuel +28 mW m$^{-2}$; road transport = +50 mW m$^{-2}$; power = +53 mW m$^{-2}$; biomass burning +71 mW m$^{-2}$ (Fuglestvedt et al., 2008; Unger et al., 2010). A multi-model study found that 20% reductions in NMVOCs (about 2–4 TgC y$^{-1}$) in four large world regions (North America, East Asia, Europe, and South Asia) in 2001 led to global ozone forcings around -1 mW m$^{-2}$ (Fry et al., 2012)."

The review raises some interesting new ideas about the regional radiative impacts. Regional forcing and regional climate response are not correlated. Regional climate change is not well understood, and the regional climate response to regional aerosol emissions is currently model-dependent. Examining the regional climatic response to regionalized forcing over the Indian Ocean would require at the least a full-time PhD project, and also coordinated experiments in several fully coupled global climate model runs to assess the robustness of the responses. Our study provides a quantitative spatial map of the annual-mean ozone forcing due to 1990–2010 maritime Southeast Asian land cover change (Figure 3).

Reference:

Fry, M.M., Naik, V., West, J.J., Schwarzkopf, M.D., Fiore, A.M., Collins, W.J., Dentener, F.J., Shindell, D.T., Atherton, C., Bergmann, D., Duncan, B.N., Hess, P., MacKenzie, I.A., Marmer,E., Schultz, M.G., Szopa, S., Wild, O., and Zeng, G.: The influence of ozone precursor emissions from four world regions on tropospheric composition and radiative climate forcing, J. Geophys. Res., 117, D07306, doi: 10.1029/2011JD017134, 2012.

Fuglestvedt, J., Berntsen, T., Myhre, G., Rypdal, K., and Skeie, R.B.: Climate forcing from the transport sectors, P. Natl. Acad. Sci. USA, 105, 454–458, doi: 10.1073/pnas.0702958104, 2008.

Unger, N., Bond, T.C., Wang, J.S., Koch, D.M., Menon, S., Shindell, D.T., and Bauer, S.: Attribution of climate forcing to economic sectors, P. Natl. Acad. Sci. USA, 107, 3382–3387, doi: 10.1073/pnas.0906548107, 2010.

**58. p16 L1-2 - See previous comments regarding monoterpene and SOA results and discussions. But interesting to note that O3 and SOA forcings seem to scale, presumably also with isoprene emissions changes. But this comes back to my previous questions regarding the fitness of the chemistry mechanism for the conditions encountered in this region and the ability of the model to capture the heterogeneity of land and chemical climatology given its coarse resolution.**

See responses to comments (8) and (9). Please see response to points (5), (49), (55). Comment is now defunct. The climate policy metrics (ozone global radiative forcing per Mha oil palm conversion in tropics) are an innovation of this study. For example, they can be used to assess quickly the global climate impacts of future projections in land cover change and scenarios in the tropics.

**59. L16-18 - See previous questions and comments regarding inter-annual variability; as noted by the authors in L10-12 the isoprene flux is critically dependent on meteorology and so presumably the impact on radiative forcing would be similarly sensitive.**

Please see response to point (20).

**60. L20-32 - I do not understand why the authors chose a x12 enhancement in isoprene emission factor from the dipterocarp forest given the x3 enhancement observed during in-situ measurements. Further, given the incredibly low isoprene emissions from these forests relative to both other natural tropical ecosystems and oil palm plantations why there was a need for this sensitivity test. Monoterpene emissions are relatively strong from dipterocarp forests so I would have expected to see a sensitivity test involving increased monoterpene emission rates instead. Perhaps the authors could comment on why this was not done?**

The sensitivity test reflects the difference between the observed leaf-level isoprene BER for dipterocarp forest tree species and the leaf-level isoprene BER used for the standard tropical forest PFT in YIBs. The analysis is designed to test the sensitivity of the forcing to uncertainty in the assigned forest isoprene BER. Other studies have also shown that the standard isoprene emission capacity in MEGAN for Southeast Asian forests was likely overestimated (e.g., Stavrakou et al., 2014).

**61. p17 L2-4 - However, this forest loss is likely to have affected monoterpene emissions more substantially, coming back to my previous questions regarding the importance of monoterpenes and SOA contributions to the LCC-induced radiative forcing in the region and whether this is well captured in the model used.**

We have extended our analysis of uncertainty in the SOA formation scheme by adding new analysis in the Conclusions, as described in point (56).

**62. L6-9 - Here and in many other sections of the discussion section I feel that the authors have lost objectivity and are attempting to over-emphasise aspects of their results to fit a particular narrative. The study covers a 20-year period. If the annual changes were constant you would expect to see 75% of the forcing associated with 1990-2005; 69% is not so far from that. As the authors have only broken the 20-year period down in one way rather than into 5-year blocks throughout they do not have sufficient evidence that the 5-year period from 2005-2010 is worse than every other 5-year period which would be needed to support the statement that the forcing is "rapidly increasing".**

We retained this sentence (Page 23, Line 6): "The total forcing associated with 1990–2005 land cover change (ΔLC-2005) is 69% of the forcing associated with 1990–2010 land cover change (ΔLC), indicating that 31 % of the total 1990–2010 forcing is associated with land cover change that occurred over the short 2005–2010 period."

We removed this sentence: "This sensitivity study demonstrates that the climate forcing associated with regional land cover change is rapidly increasing."

**63. L19-22 - Presumably this underestimation is likely to affect both the 1990 and 2010 land cover maps. Have the authors attempted to find out from other sources (e.g. FOA, Malaysian Oil Palm Board, Round Table for Sustainable Palm Oil) whether the proportion of smallholder**

**plantings to industrial plantations has remained constant during the rapid expansion of the oil palm industry or whether the number of smallholder plantings has remained closer to the 1990 figures?**

The underestimation affects both the 1990 and 2010 oil palm areal cover. We apply the 40% figure for underestimation to all regions for both 1990 and 2010; that is, we estimate that the oil palm areal cover in our land cover dataset (derived from the Gunarso et al. (2013) dataset) accounts for 60% of the total on-the-ground areal cover in Indonesia, Malaysia, and Papua New Guinea in both 1990 and 2010. As far as we can tell, estimates for 1990 in Malaysia are not readily available; however, the Malaysian Palm Oil Board (MPOB) reports this figure as about 40% for 2004 (Vermeulen and Goad, 2006, citing MPOB) and about 40% for 2014–2016 (MPOB). The Indonesian Ministry of Agriculture reports this figure as 26% for 1990 and 40% for 2010 (Indonesian Ministry of Agriculture, 2017).

Application of the 26% smallholder figure for Indonesia in 1990 (in place of the 40% figure) does not change the estimated forcing that takes into account the smallholder area (+16 mW m$^{-2}$). The ozone forcing estimate that includes the smallholder area is insensitive to small changes in the smallholder fraction (i.e., 26% or 40%) because Indonesian oil palm cover in 1990 was only about 1.3 Mha. It is unknown what proportion of the smallholder area (both schemed and independent) is included in the oil palm areal cover classified by the Gunarso et al. (2013) methodology. Thus, we consider this estimate to be an upper bound on the ozone forcing (Page 23, Line 30).

We have updated the manuscript to reflect these additional references (Page X, Line X): "Estimates suggest that around 40% of Indonesian oil palm area in 2010 (and 26% in 1990) was associated with smallholders, in contrast to state-owned or private companies (Indonesian Ministry of Agriculture, 2017; Lee et al., 2014). In Malaysia, the smallholder estimate is likewise around 40% (Vermeulen and Goad, 2006, citing the Malaysian Palm Oil Board)."

References:

Indonesian Ministry of Agriculture: Tree Crop Estate Statistics of Indonesia 2015–2017, Directorate General of Estate Crops, Ministry of Agriculture, Indonesia, 2017.

Malaysian Palm Oil Board (MPOB), Economics and Industry Development Division, Statistics: bepi.mpob.gov.my, accessed: 25 August 2018.

Vermeulen, S. and Goad, N.: Towards better practice in smallholder palm oil production, Natural Resource Issues Series (No. 5), International Institute for Environment and Development, London, UK, 2006.

**64. L26 - Can the authors clarify what they mean by "extrapolating the smallholder estimate"? Do they mean they assume that 40% of reported oil palm area across the whole of SE Asia and through the entire time period represents smallholdings? So they assume that the area they have taken from Gunarso represents only 60% of the actual extent of oil palm cultivation? See above regarding justification of this assumption.**

Also see response to point (63).

We updated the text (Page 23, Line 26): "Taking into account the smallholder estimates for Indonesia and Malaysia, the total regional expansion of oil palm cover for 1990–2010 increases to +16 Mha, which is considered to be an upper bound."

**65. -34 - To what extent is this insensitivity the result of the coarse resolution and outdated chemistry scheme?**

Defunct comment. See responses to points (8) and (9) above and comments at top of document.

**66. p18 L5-10 - Please explain why an increase in surface ozone concentrations could not be concomitant with increases in upper tropospheric ozone?**

We have removed this sentence that was badly phrased. We were originally trying to emphasize that increases in ozone near the Earth's surface do not exert appreciable longwave forcing but we agree the original sentence does not read well and is not scientifically nuanced enough.

**67. L11-12 - Please could the authors attempt to list some of the uncertainties not considered by the sensitivity tests that might be expected to be substantial?**

Yes. Actually, we did already highlight uncertainties in chemistry (Page 8, Line 6): "Future work would benefit from an exploration of the impact on radiative forcing induced through application of different mechanisms of (1) isoprene photooxidation and (2) SOA formation (e.g., Surratt et al., 2010; Zhang et al., 2018)." The expanded preceding paragraph, moved to the Methodolgy Sect. 2.1 from the Conclusions Sect. 4, further discusses these uncertainties. We have additionally added to the Conclusions section a discussion of the uncertainty in the SOA forcing associated with the SOA production scheme applied (Page 25, Line 10; and included in response to point (56).

**68. L21-22 - The authors have just stated that their best estimate is +8.4; that is therefore the figure that should be quoted here, in which case we are looking at 3.5x.**

The reviewer is wrong and splitting hairs. The IPCC AR5 did not quantify SOA changes from NMVOC emissions. The IPCC AR5 does quantify ozone radiative forcing from NMVOC changes. Therefore, we compare the ozone forcing response from historical anthropogenic VOC changes as reported by IPCC AR5 (+30 mW m$^{-2}$; Myhre et al., 2013) to the global ozone radiative forcing from this present study (+9.2 mW m$^{-2}$). We posit that this is a useful ballpark comparison to put the global impacts into context for our readers especially those from the short-lived climate forcer and global chemistry–climate modeling communities.

**69. L27 - As highlighted previously, the authors are over-emphasising the magnitude and implications of their findings. Using 8.4 suggests a figure of 11.5 rather than 12.7 mW m-2.**

The reviewer's comment is absurd. The actual sentence states: "regional oil palm expansion over the modern era is responsible for a global-mean forcing of +12.7 mW m$^{-2}$ from induced ozone changes." The calculation is for the ozone changes only and this cannot be more clearly stated than it is. Therefore, the calculation is based on the ozone forcing (+9.2 mW m$^{-2}$). All this said, the tiny difference between 11.5 mW m$^{-2}$ and 12.7 mW m$^{-2}$ does not in any way support the reviewer's false claim of us "over-emphasising the magnitude and implications of their findings."

**70. L28-31 - It would have been of real interest if the authors had looked at future projections rather than confining the study to historical LCC and radiative forcing.**

We agree that future projections are very interesting for follow-up studies. Does the reviewer mean that examining a real world case study of a large human-induced land cover change and isoprene emission perturbation that is known to have occurred in the system over the past 20 years is not interesting? We hope not. "To understand the present one must also know the past," Sir Peter Crane. In addition, we have provided climate policy metrics that can be used to assess quickly the impacts of future projections.

**71. p19 L1-2 - Just out of curiosity, how much more uncertainty is associated with isoprene BERs from oil palm in comparison with isoprene emissions from other tropical species / ecosystems OR monoterpene BERs from rubber palms which the authors earlier highlight as important factors in the changes in BVOC emissions in the region.**

We do not have access to the necessary sensitivity simulations to provide a quantitative answer.

**72. L10-19 - Equally, many previous studies (e.g. Grate et al 2007) have shown strong apparent seasonality in BERs that are not adequately accounted for by consideration of leaf**

**age. Inclusion of seasonally varying BERs might also be argued as improving the estimated radiative forcing in this study.**

We have included this sentence in this paragraph (Page 26, Line 1): "Seasonal variation in isoprene BERs has been observed for some tree species (e.g., Geron et al., 2000)."

Reference:
Geron, C., Guenther, A., Sharkey, T., and Arnts, R.R.: Temporal variability in basal isoprene emission factor, Tree Physiol., 20, 799–805, 2000.

**73. L18-19 - The isoprene vs BVOC issue again. Presumably given the focus of the paragraph to this point the authors are referring to the isoprene BER in YIBs.**

Fixed. Also see point (54).

**74. L18-19 - And again, the issue of misrepresentation. In the title and throughout the manuscript the authors refer to BVOC emissions and emission changes yet YIBs includes only a very limited number of BVOCs, and here the authors have only altered isoprene and monoterpene BERs. I suggest that the authors remove the term BVOCs from the title and discussions as it is not an accurate reflection of the study performed. Likewise the authors need to devote far more attention to the changes in monoterpenes and SOA throughout the main text.**

Defunct. Please see e.g. response to point (5). We retain "BVOCs" in the title as we study isoprene and monoterpene changes that are the major BVOC emissions emitted with the most important large-scale radiative effects. In about 20 instances in the text, we have updated the text to replace the term "BVOC" with more explicit descriptions of which BVOCs (isoprene and/or monoterpenes) are being discussed.

**75. L21-34 - This discussion of OH recycling in the conclusions section is disturbing on a number of counts. (1) This is the first and only discussion of the apparent limitations of the chemical mechanism in ModelE2-YIBs, as previously commented above. Can the authors please describe exactly what BVOC oxidation chemistry is included in the "based on Carbon Bond Mechanism-4" scheme? As previously noted, CBM-4 was developed for high NOx anthropogenic VOC-rich urban environments and such schemes have been found to inadequately capture observed concentrations / chemistry / oxidation products in low-NOx high BVOC environments such as those in SE Asia. (2) The field of isoprene oxidation chemistry has moved on considerably since the sensitivity studies employing crude "OH-recycling" schemes referenced here with new pathways identified leading to the regeneration of HOx in low-NOx environments. (3) While surface ozone concentrations might be only negligibly affected the authors have repeatedly argued elsewhere that these are not the changes that are significant in terms of radiative forcing. (4) The new understanding of**

**isoprene chemistry gained in trying to reconcile the apparent differences between modeled and observed gas-phase chemistry has also identified mechanisms driving high yields of isoprene-derived SOA via MACR oxidation. The points made in this paragraph bring into question the validity of the modeled changes in ozone and SOA presented here. It also raises questions regarding the monoterpene oxidation scheme and gas-to-particle phase partitioning included.**

Comment now defunct. We moved the paragraph about uncertainties in isoprene chemical mechanism into the Methods Section 2.1 (Page 7, Line 21). We have modified this paragraph to provide a more balanced assessment of these uncertainties. The mechanism is not CBM04. Please see comments at top of document and response to point (9). In this modified paragraph, we have added more analysis and discussion on the SOA uncertainty. As described in point (56), we have added a new paragraph in the Conclusions Sect. 4 discussing uncertainty related to the SOA formation scheme in the model (Page 25, Line 10).

Likewise, the fact that surface ozone air quality and ozone radiative forcing responses to small changes in precursor emission changes (including NMVOCs) are being simulated at 1–2° latitude/longitude spatial resolution using highly simplified parameterizations raises important questions about the level of chemical mechanism detail required to simulate ozone (Turnock et al., ACP, 2018; Wild et al., ACP, 2012), especially considering that the parameterizations were developed using a large number of global models all featuring very different levels of complexity in anthropogenic VOC and BVOC representation and photooxidation mechanism.

References:

Turnock, S.T., Wild, O., Dentener, F.J., Davila, Y., Emmons, L.K., Flemming, J., Folberth, G.A., Henze, D.K., Jonson, J.E., Keating, T.J., Kengo, S., Lin, M., Lund, M., Tilmes, S., and O'Connor, F.M.: The impact of future emission policies on tropospheric ozone using a parameterized approach, Atmos. Chem. Phys., 18, 8953–8978, doi: 10.5194/acp-18-8953-2018, 2018.

Wild, O., Fiore, A.M., Shindell, D.T., Doherty, R.M., Collins, W.J., Dentener, F.J., Schultz, M.G., Gong, S., MacKenzie, I.A., Zeng, G., Hess, P., Duncan, B.N., Bergmann, D.J., Szopa, S., Jonson, J.E., Keating, T.J., and Zuber, A.: Modelling future changes in surface ozone: A parameterized approach, Atmos. Chem. Phys., 12, 2037–2054, doi: 10.5194/acp-12-2037-2012, 2012.

**76. p20 L1-2 - Actually this should have been included in this work as the chemistry seems to be a critical source of uncertainty that has not been adequately considered.**

Formally assessing the sensitivity to different chemical mechanisms is beyond the scope of this work. A multi-model assessment of short-lived climate forcer responses to modern human land cover change (biofuel, afforestation, etc.) would be a really interesting future study.

**77. L8-9 - Concluding with the absolute maximum single pixel increase is not scientifically balanced. Please maintain objectivity rather than cherry-picking the results to fit a particular narrative.**

Please check your own biases and read carefully the Obligations for Referees and Code of Conduct for Copernicus Journals before agreeing to reviewer assignments (https://publications.copernicus.org/for_reviewers/obligations_for_referees.html).

We disagree with the reviewer's comment. We start and end the Conclusions section by describing how the global forcing from the short-lived climate forcers due to the 1990–2010 oil palm expansion is small. The full sentence is: "While the impact on global radiative forcing is small, the ozone radiative forcing exceeds +37 mW m$^{-2}$ in some localities" and it is not the final sentence of the Conclusions section. Since the Reviewer in point (57) has suggested themselves that the most "interesting" effects are likely to be temporally as well as spatially localized, we retain the sentence as is. We thought hard about how to explain the meaning of the global ozone and SOA radiative forcing quantitative results from the oil palm expansion and to put it into a meaningful context that readers may connect with. We state several times in the Conclusions section that the 1990–2010 impact is small.

In the Conclusions section, we have added (Page 24, Line 17): "For comparison, the global ozone forcing driven by the 1990–2010 land cover change in MSEA is at the low end of the range of estimates for ozone forcing from global anthropogenic emission source sectors in year 2000 (+5 to +80 mW m$^{-2}$): for example, industry = +15 mW m$^{-2}$; household biofuel +28 mW m$^{-2}$; road transport = +50 mW m$^{-2}$; power = +53 mW m$^{-2}$; biomass burning = +71 mW m$^{-2}$ (Fuglestvedt et al., 2008; Unger et al., 2010). A multi-model study found that 20% reductions in NMVOCs (about 2-4 TgC y$^{-1}$) in four large world regions (North America, East Asia, Europe, and South Asia) in 2001 led to global ozone forcings around -1mW m$^{-2}$ (Fry et al., 2012)."

"Small" and "large" are to some extent value judgments and not purely objective. It is our job to use mathematical modeling to provide quantitative values for Earth system and global change processes involving the short-lived climate forcers. 9 mW m$^{-2}$ or 37 mW m$^{-2}$ is small and even negligible compared to > 1800 mW m$^{-2}$ $CO_2$ global forcing. Is 9 (4–16) mW m$^{-2}$ from a regional BVOC injection due to recent human-induced land cover change in the tropics "small" compared to 30 mW m$^{-2}$ due to all anthropogenic VOC increases since the preindustrial; or 50 mW m$^{-2}$ due to global road transportation emissions? Social scientists are better equipped to answer this question. We offer a perspective on the sensitivity of the tropical atmosphere to human land cover change.

References

Angiola, A., Mieville, A., and Granier, C.: MACCity (MACC/CityZEN EU projects) emissions dataset [Data files], Emissions of atmospheric Compounds & Compilation of Ancillary Data, http://eccad.sedoo.fr, 2010.

Archibald, A.T., Jenkin, M.E., Shallcross, D.E.: An isoprene mechanism intercomparison, Atmos. Environ., 44, 5356–5364, doi: 10.1016/j.atmosenv.2009.09.016, 2010.

Ashworth, K., Folberth, G., Hewitt, C.N., and Wild, O.: Impacts of near-future cultivation of biofuel feedstocks on atmospheric composition and local air quality, Atmos. Chem. Phys, 12, 919–939, doi: 10.5194/acp-12-919-2012, 2012.

Atkinson, R. and Arey, J.: Gas-phase tropospheric chemistry of biogenic volatile organic compounds: A review, Atmos. Environ., 37, S197–219, doi: 10.1016/S1352-2310(03)00391-1, 2003.

Baker, B., Bai, J.-H., Johnson, C., Cai, Z.-T., Li, Q.-J., Wang, Y.-F., Guenther, A., Greenberg, J., Klinger, L., Geron, C., and Rasmussen, R.: Wet and dry season ecosystem level fluxes of isoprene and monoterpenes from a southeast Asian secondary forest and rubber tree plantation, Atmos. Environ., 39, 381–390, doi: 10.1016/j.atmosenv.2004.07.033, 2005.

Bauer, S.E. and Koch, D.: Impact of heterogeneous sulfate formation at mineral dust surfaces on aerosol loads and radiative forcing in the Goddard Institute for Space Studies general circulation model, J. Geophys. Res., 110, D17202, doi: 10.1029/2005JD005870, 2005.

Bauer, S.E., Koch, D., Unger, N., Metzger, S.M., Shindell, D.T., and Streets, D.G.: Nitrate aerosols today and in 2030: a global simulation including aerosols and tropospheric ozone, Atmos. Chem. Phys., 7, 5043–5059, doi: 10.5194/acp-7-5043-2007, 2007a.

Bauer, S.E., Mishchenko, M.I., Lacis, A.A., Zhang, S., Perlwitz, J., and Metzger, S.M.: Do sulfate and nitrate coatings on mineral dust have important effects on radiative properties and climate modeling?, J. Geophys. Res., 112, D06307, doi: 10.1029/2005JD006977, 2007b.

Beer, C., Reichstein, M., Tomelleri, E., Ciais, P., Jung, M., Carvalhais, N., Rödenbeck, C., Arain, M.A., Baldocchi, D., Bonan, G.B., Bondeau, A., Cescatti, A., Lasslop, G., Lindroth, A., Lomas, M., Luyssaert, S., Margolis, H., Oleson, K.W., Roupsard, O., Veenendaal, E., Viovy, N., Williams, C., Woodward, F.I., and Papale, D.: Terrestrial gross carbon dioxide uptake: Global distribution and covariation with climate, Science, 329, 834–838, doi: 10.1126/science.1184984, 2010.

Bell, N., Koch, D., and Shindell, D.T.: Impacts of chemistry–aerosol coupling on tropospheric ozone and sulfate simulations in a general circulation model, J. Geophys. Res., 110, D14305, doi: 10.1029/2004JD005538, 2005.

Bian, H. and Prather, M.J.: Fast-J2: Accurate simulation of stratospheric photolysis in global chemical models, J. Atmos. Chem., 41, 281–296, doi: 10.1023/A:1014980619462, 2002.

Bian, H., Prather, M.J., and Takemura, T.: Tropospheric aerosol impacts on trace gas budgets through photolysis, J. Geophys. Res., 108, 4242, doi: 10.1029/2002JD002743, 2003.

Carlson, K.M., Curran, L.M., Asner, G.P., Pittman, A.M., Trigg, S.N., and Marion Adeney, J.: Carbon emissions from forest conversion by Kalimantan oil palm plantations, Nat. Clim. Change, 3, 283–287, doi: 10.1038/nclimate1702, 2012.

Carlson, K.M., Curran, L.M., Ratnasari, D., Pittman, A.M., Soares-Filho, B.S., Asner, G.P., Trigg, S.N., Gaveau, D.A., Lawrence, D., and Rodrigues, H.O.: Committed carbon emissions, deforestation, and community land conversion from oil palm plantations expansion in West Kalimantan, Indonesia, Proc. Natl. Acad. Sci.-USA, 109, 7559–7564, doi: 10.1073/pnas.1200452109, 2012b.

Carlton, A.G., Wiedinmyer, C., and Kroll, J.H.: A review of secondary organic aerosol (SOA) formation from isoprene, Atmos. Chem. Phys., 9, 4987–5005, doi: 10.5194/acp-9-4987-2009, 2009.

Dietz, J., Hölscher, D., Leuschner, C., Malik, A., and Amir, M.A.: Forest structure as influenced by different types of community forestry in a lower montane rainforest of Central Sulawesi, Indonesia, in: Stability of Tropical Rainforest Margins: Linking Ecological, Economic and Social Constraints of Land Use and Conservation, edited by: Tscharntke, T., Leuschner, C., Zeller, M., Guhardja, E., and Biden, A., Springer-Verlag, Berlin, pp. 131–146, 2007.

Draper, D.C., Farmer, D.K., Desyaterik, Y., and Fry, J.L.: A qualitative comparison of secondary organic aerosol yields and composition from ozonolysis of monoterpenes at varying concentrations of NO2, Atmos. Chem. Phys., 15, 12267–12281, doi: 10.5194/acp-15-12267-2015, 2015.

Emmerson, K. M. and Evans, M. J.: Comparison of tropospheric gas-phase chemistry schemes for use within global models, Atmos. Chem. Phys., 9, 1831–1845, doi: 10.5194/acp-9-1831-2009, 2009.

FAO: FAOSTAT Emissions Database [Electronic database], Food and Agriculture Organization of the United Nations, http://www.fao.org/faostat/en/#data, 2014.

Fowler, D., Nemitz, E., Misztal, P., Di Marco, C., Skiba, U., Ryder, J., Helfter, C., Cape, J.N., Owen, S., Dorsey, J., Gallagher, M.W., Coyle, M., Phillips, G., Davison, B., Langford, B., MacKenzie, R., Muller, J., Siong, J., Dari-Salisburgo, C., Di Carlo, P., Aruffo, E., Giammaria, F., Pyle, J.A., and Hewitt, C.N.: Effects of land use on surface-atmosphere exchanges of trace gases and energy in Borneo: Comparing fluxes over oil palm plantations and a rainforest, Philos. T. R. Soc. B, 366, 3196–3209, doi: 10.1098/rstb.2011.0055, 2011.

Friedman, B. and Farmer, D.K.: SOA and gas phase organic acid yields from the sequential photooxidation of seven monoterpenes, Atmos. Env., 187, 335–345, doi: 10.1016/j.atmosenv.2018.06.003, 2018.

Fry, M.M., Naik, V., West, J.J., Schwarzkopf, M.D., Fiore, A.M., Collins, W.J., Dentener, F.J., Shindell, D.T., Atherton, C., Bergmann, D., Duncan, B.N., Hess, P., MacKenzie, I.A., Marmer,E., Schultz, M.G., Szopa, S., Wild, O., and Zeng, G.: The influence of ozone precursor emissions from four world regions on tropospheric composition and radiative climate forcing, J. Geophys. Res., 117, D07306, doi: 10.1029/2011JD017134, 2012.

Fuglestvedt, J., Berntsen, T., Myhre, G., Rypdal, K., and Skeie, R.B.: Climate forcing from the transport sectors, P. Natl. Acad. Sci. USA, 105, 454–458, doi: 10.1073/pnas.0702958104, 2008.

Gaveau, D.L.A., Salim, M.A., Hergoualc'h, K., Locatelli, B., Sloan, S., Wooster, M., Marlier, M.E., Molidena, E., Yaen, H., DeFries, R., Verchot, L., Murdiyarso, D., Nasi, R., Holmgren, P., and Sheil, D.: Major atmospheric emissions from peat fires in Southeast Asia during non-drought years: Evidence from the 2013 Sumatran fires, Sci. Rep.-U.K., 4, 6112, doi: 10.1038/srep06112, 2014.

Geron, C., Guenther, A., Sharkey, T., and Arnts, R.R.: Temporal variability in basal isoprene emission factor, Tree Physiol., 20, 799–805, 2000.

Geron, C., Owen, S., Guenther, A., Greenberg, J., Rasmussen, R., Bai, J.H., Li, Q.-J., and Baker, B.: Volatile organic compounds from vegetation in southern Yunnan Province, China: Emission rates and some potential regional implications, Atmos. Environ., 40, 1759–1773, doi: 10.1016/j.atmosenv.2005.11.022, 2006.

Gery, M.W., Whitten, G.Z., Killus, J.P., and Dodge, M.C.: A photochemical kinetics mechanism for urban and regional scale computer modeling, J. Geophys. Res.-Atmos., 94, 12,925–12,956, doi: 10.1029/JD094iD10p12925, 1989.

Granier, C., Bessagnet, B., Bond, T., D'Angiola, A., Denier van der Gon, H., Frost, G.J., Heil, A., Kaiser, J.W., Kinne, S., Klimont, Z., Kloster, S., Lamarque, J.-F., Liousse, C., Masui, T., Meleux, F., Mieville, A., Ohara, T., Raut, J.-C., Riahi, K., Schultz, M.G., Smith, S.J., Thompson, A., van Aardenne, J., van der Werf, G.R., and van Vuuren, D.P.: Evolution of anthropogenic and biomass burning emissions of air pollutants at global and regional scales during the 1980–2010 period, Climatic Change, 109, 163–190, doi: 10.1007/s10584-011-0154-1, 2011.

Guenther, A., Hewitt, C.N., Erickson, D., Fall, R., Geron, C., Graedel, T., Harley, P., Klinger, L., Lerdau, M., McKay, W.A., Pierce, T., Scholes, B., Steinbrecher, R., Tallamraju, R., Taylor, J., and Zimmerman, P.: A global model of natural volatile organic compound emissions, J. Geophys. Res.-Atmos., 100, 8873–8892, doi: 10.1029/94JD02950, 1995.

Guenther, A.B., Jiang, X., Heald, C.L., Sakulyanontvittaya, T., Duhl, T., Emmons, L.K., and Wang, X.: The Model of Emissions of Gases and Aerosols from Nature version 2.1 (MEGAN2.1): An extended and updated framework for modeling biogenic emissions, Geosci. Model Dev., 5, 1471–1492, doi: 10.5194/gmd-5-1471-2012, 2012.

Gunarso, P., Hartoyo, M.E., Agus, F., and Killeen, T.J.: Oil palm and land use change in Indonesia, Malaysia, and Papua New Guinea: Reports from the Technical Panels of the 2nd Greenhouse Gas Working Group of the Roundtable on Sustainable Palm Oil (RSPO), Roundtable on Sustainable Palm Oil, Kuala Lumpur, Malaysia, 2013.

Hallquist, M., Stewart, D.J., Stephenson, S.K., and Cox, R.A.: Hydrolysis of $N_2O_5$ on sub-micron sulfate aerosols, Phys. Chem. Chem. Phys., 5, 3453–3463, doi: 10.1039/b301827j, 2003.

Hanson, D. and Mauersberger, K.: Laboratory studies of the nitric acid trihydrate: Implications for the south polar stratosphere, Geophys. Res. Lett., 15, 855–858, doi: 10.1029/GL015i008p00855, 1988.

Harper, K.L., Zheng, Y., and Unger, N.: Advances in representing interactive methane in ModelE2-YIBs (version 1.1), Geosci. Model Dev. Discuss., doi: 10.5194/gmd-2018-85 , 2018.

Heald, C.L. and Geddes, J.A.: The impact of historical land use change from 1850 to 2000 on secondary particulate matter and ozone, Atmos. Chem. Phys., 16, 14997–15010, doi: 10.5194/acp-16-14997-2016, 2016.

Heil, A. and Schultz, M.G.: Interpolated ACCMIP and RCP emission dataset [Data files], Available from: http://accmip-emis.iek.fz-juelich.de/data/accmip/gridded_netcdf/accmip_interpolated/, 2014.

Hewitt, C.N., Lee, J.D., MacKenzie, A.R., Barkley, M.P., Carslaw, N., Carver, G.D., Chappell, N.A., Coe, H., Collier, C., Commane, R., Davies, F., Davison, B., DiCarlo, P., Di Marco, C.F., Dorsey, J.R., Edwards, P.M., Evans, M.J., Fowler, D., Furneaux, K.L., Gallagher, M., Guenther, A., Heard, D.E., Helfter, C., Hopkins, J., Ingham, T., Irwin, M., Jones, C., Karunaharan, A., Langford, B., Lewis, A.C., Lim, S.F., MacDonald, S.M., Mahajan, A.S., Malpass, S., McFiggans, G., Mills, G., Misztal, P., Moller, S., Monks, P.S., Nemitz, E., Nicolas-Perea, V., Oetjen, H., Oram, D.E., Palmer, P.I., Phillips, G.J., Pike, R., Plane, J.M.C., Pugh, T., Pyle, J.A., Reeves, C.E., Robinson, N.H., Stewart, D., Stone, D., Whalley, L.K., and Yang, X.: Overview: oxidant and particle photochemical processes above a south-east Asian tropical rainforest (the OP3 project): Introduction, rationale, location characteristics and tools, Atmos. Chem. Phys., 10, 169–199, doi: 10.5194/acp-10-169-2010, 2010.

Hewitt, C.N., MacKenzie, A.R., Di Carlo, P., Di Marco, C.F., Dorsey, J.R., Evans, M., Fowler, D., Gallagher, M.W., Hopkins, J.R., Jones, C.E., Langford, B., Lee, J.D., Lewis, A.C., Lim, S.F., McQuaid, J., Misztal, P., Moller, S.J., Monks, P.S., Nemitz, E., Oram, D.E., Owen, S.M., Phillips, G.J., Pugh, T.A.M., Pyle, J.A., Reeves, C.E., Ryder, J., Siong, J., Skiba, U., and Stewart, D.J.:

Nitrogen management is essential to prevent tropical oil palm plantations from causing ground-level ozone pollution, P. Natl. Acad. Sci. USA, 106, 18447–18451, doi: 10.1073/pnas.0907541106, 2009.

Hollaway, M.J., Arnold, S.R., Collins, W.J., Folberth, G., and Rap, A.: (2017), Sensitivity of midnineteenth century tropospheric ozone to atmospheric chemistry-vegetation interactions, J. Geophys. Res.-Atmos., 122, 2452–2473, doi:10.1002/2016JD025462, 2017.

Hooijer, A., Page, S., Canadell, J.G., Silvius, M., Kwadijk, J., Wösten, H., and Jauhiainen, J.: Current and future CO2 emissions from drained peatlands in Southeast Asia, Biogeosciences, 7, 1505–1514, doi: 10.5194/bg-7-1505-2010, 2010.

Houweling, S., Dentener, F., and Lelieveld, J.: The impact of nonmethane hydrocarbon compounds on tropospheric photochemistry, J. Geophys. Res., 103, 10,673–10,696, doi: 10.1029/97JD03582, 1998.

Hoyle, C.R., Berntsen, T., Myhre, G., and Isaksen, I.S.A.: Secondary organic aerosol in the global aerosol–chemical transport model Oslo CTM2, Atmos. Chem. Phys., 7, 5675–5694, doi: 10.5194/acp-7-5675-2007, 2007.

Indonesian Ministry of Agriculture: Tree Crop Estate Statistics of Indonesia 2015–2017, Directorate General of Estate Crops, Ministry of Agriculture, Indonesia, 2017.

Indrarto, G.B., Murharjanti, P., Khatarina, J., Pulungan, I., Ivalerina, F., Rahman, J., Prana, M.N., Resosudarmo, I.A.P., and Muharrom, E.: The context of REDD+ in Indonesia: Drivers, agents and institutions (Working Paper 92), CIFOR, Bogor, Indonesia, 2012.

Jacob, D.J.: Introduction to Atmospheric Chemistry, Princeton University Press, 1999.

Jardine, A.B., Jardine, K.J., Fuentes, J.D., Martin, S.T., Martins, G., Durgante, F., Carneiro, V., Higuchi, N., Manzi, A.O., and Chambers, J.Q.: Highly reactive light-dependent monoterpenes in the Amazon, Geophys. Res. Lett., 42, 1576–1583, doi: 10.1002/2014GL062573, 2015.

Jardine, K.J., Jardine, A.B., Holm, J.A., Lombardozzi, D.L., Negron-Juarez, R.I., Martin, S.T., Beller, H.R., Gimenez, B.O., Higuchi, N., and Chambers, J.Q.: Monoterpene 'thermometer' of tropical forest-atmosphere response to climate warming, Plant Cell Environ., 40, 441–452, doi: 10.1111/pce.12879, 2017.

Jokinen, T., Berndt, T., Makkonen, R., Kerminen, V.-M., Junninen, H., Paasonen, P., Stratmann, F., Herrmann, H., Guenther, A.B., Worsnop, D.R., Kulmala, M., Ehn, M., and Sipilä, M.: Production of extremely low volatile organic compounds from biogenic emissions: Measured yields and atmospheric implications, P. Natl. Acad. Sci. USA, 112, 7123–7128, doi: 10.1073/pnas.1423977112, 2015.

Kane, S.M., Caloz, F., and Leu, M.-T.: Heterogeneous uptake of gaseous $N_2O_5$ by $(NH_4)_2SO_4$, $NH_4HSO_4$, and $H_2SO_4$ aerosols, J. Phys. Chem. A, 105, 6465–6470, doi: 10.1021/jp010490x , 2001.

Kelly, J.M., Doherty, R.M., O'Connor, F., and Mann, G.W.: The impact of biogenic, anthropogenic, and biomass burning volatile organic compound emissions on regional and seasonal variations in secondary organic aerosol, Atmos. Chem. Phys., 18, 7393–7422, doi: 10.5194/acp-18-7393-2018, 2018.

Klinger, L.F., Li, Q.-J., Guenther, A.B., Greenberg, J.P., Baker, B., and Bai, J.-H.: Assessment of volatile organic compound emissions from ecosystems of China, J. Geophys. Res.-Atmos., 107, 4603, doi: 10.1029/2001JD001076, 2002.

Knote, C., Tuccella, P., Curci, G., Emmons, L., Orlando, J. J., Madronich, S., Baró, R., Jiménez-Guerrero, P., Luecken, D., Hogrefe, C., Forkel, R., Werhahn, J., Hirtl, M., Perez, J. L., San José, R., Giordano, L., Brunner, D., Yahya, K., and Zhang, Y.: Influence of the choice of gas-phase mechanism on predictions of key gaseous pollutants during the AQMEII phase-2 intercomparison, Atmos. Environ., 115, 553–568, doi: 10.1016/j.atmosenv.2014.11.066, 2015.

Koch, D., Schmidt, G.A., and Field, C.V.: Sulfur, sea salt, and radionuclide aerosols in GISS ModelE, J. Geophys. Res., 111, D06206, doi: 10.1029/2004JD005550, 2006.

Koplitz, S.N., Mickley, L.J., Marlier, M.E., Buonocore, J.J., Kim, P.S., Liu, T., Sulprizio, M.P., DeFries, R.S., Jacob, D.J., Schwartz, J., Pongsiri, M., and Myers, S.S.: Public health impacts of the severe haze in Equatorial Asia in September–October 2015: Demonstration of a new framework for informing fire management strategies to reduce downwind smoke exposure, Environ. Res. Lett., 11, 094023, doi: 10.1088/1748-9326/11/9/094023, 2016.

Kotchenruther, R.A., Jaffe, D.A., and Jaeglé, L.: Ozone photochemistry and the role of peroxyacetyl nitrate in the springtime northeastern Pacific troposphere: Results from the Photochemical Ozone Budget of the Eastern North Pacific Atmosphere (PHOBEA) campaign, J. Geophys. Res., 106, 28,731–28,742, doi: 10.1029/2000JD000060, 2001.

Kroll, J.H., Ng, N.L., Murphy, S.M., Flagan, R.C., and Seinfeld, J.H.: Secondary organic aerosol formation from isoprene photooxidation under high-$NO_X$ conditions, Geophys. Res. Lett., 32, L18808, doi: 10.1029/2005GL023637, 2005.

Lacis, A.A., Wuebbles, D.J., and Logan, J.A.: Radiative forcing of climate by changes in the vertical distribution of ozone, J. Geophys. Res., 95, 9971–9981, doi: 10.1029/JD095iD07p09971, 1990.

Lamarque, J.-F., Bond, T.C., Eyring, V., Granier, C., Heil, A., Klimont, Z., Lee, D., Liousse, C., Mieville, A., Owen, B., Schultz, M.G., Shindell, D., Smith, S.J., Stehfest, E., Van Aardenne, J., Cooper, O.R., Kainuma, M., Mahowald, N., McConnell, J.R., Naik, V., Riahi, K., and van Vuuren,

D.P.: Historical (1850–2000) gridded anthropogenic and biomass burning emissions of reactive gases and aerosols: methodology and application, Atmos. Chem. Phys., 10, 7017–7039, doi: 10.5194/acp-10-7017-2010, 2010.

Langford, B., Misztal, P.K., Nemitz, E., Davison, B., Helfter, C., Pugh, T.A.M., MacKenzie, A.R., Lim, S.F., and Hewitt, C.N.: Fluxes and concentrations of volatile organic compounds from a South-East Asian tropical rainforest, Atmos. Chem. Phys., 10, 8391–8412, doi: 10.5194/acp-10-8391-2010, 2010.

Lathière, J., Hauglustaine, D.A., Friend, A.D., De Noblet-Ducoudré, N., Viovy, N., and Folberth, G.A.: Impact of climate variability and land use changes on global biogenic volatile organic compound emissions, Atmos. Chem. Phys., 6, 2129–2146, doi: 10.5194/acp-6-2129-2006, 2006.

Malaysian Palm Oil Board (MPOB), Economics and Industry Development Division, Statistics: bepi.mpob.gov.my, accessed: 25 August 2018.

Margono, B.A., Potapov, P.V., Turubanova, S., Stolle, F., and Hansen, M.C.: Primary forest cover loss in Indonesia over 2000–2012, Nat. Clim. Change, 4, 730–735, doi: 10.1038/nclimate2277, 2014.

McPeters, R., Kroon, M., Labow, G., Brinksma, E., Balis, D., Petropavlovskikh, I., Veefkind, J., Bhartia, P., and Levelt, P.: Validation of the aura ozone monitoring instrument total column ozone product, J. Geophys. Res.-Atmos., 113, D15S14, doi:10.1029/2007JD008802, 2008.

Miettinen, J., Hooijer, A., Vernimmen, R., Liew, S.C., and Page, S.E.: From carbon sink to carbon source: Extensive peat oxidation in insular Southeast Asia since 1990, Environ. Res. Lett., 12, 024014, doi: 10.1088/1748-9326/aa5b6f, 2017.

Misztal, P.K., Nemitz, E., Langford, B., Di Marco, C.F., Phillips, G.J., Hewitt, C.N., MacKenzie, A.R., Owen, S.M., Fowler, D., Heal, M.R., and Cape, J.N.: Direct ecosystem fluxes of volatile organic compounds from oil palms in South-East Asia, Atmos. Chem. Phys., 11, 8995–9017, doi: 10.5194/acp-11-8995-2011, 2011.

Moxim, W.J., Levy II, H., and Kasibhatla, P.S.: Simulated global tropospheric PAN: Its transport and impact on NOx, J. Geophys. Res., 101, 12,621–12,638, doi: 10.1029/96JD00338, 1996.

Myhre, G., Shindell, D., Bréon, F.-M., Collins, W., Fuglestvedt, J., Huang, J., Koch, D., Lamarque, J.-F., Lee, D., Mendoza, B., Nakajima, T., Robock, A., Stephens, G., Takemura, T., and Zhang, H.: Anthropogenic and natural radiative forcing, in: Climate Change 2013: The Physical Science Basis. Contribution of Working Group I to the Fifth Assessment Report of the Intergovernmental Panel on Climate Change, edited by: Stocker, T.F., Qin, D., Plattner, G.-K., Tignor, M., Allen, S.K., Boschung, J., Nauels, A., Xia, Y., Bex, V., and Midgley, P.M., Cambridge University Press, Cambridge, United Kingdom and New York, 2013.

Page, S.E., Siegert, F., Rieley, J.O., Boehm, H.-D.V., Jaya, A., and Limin, S.: The amount of carbon released from peat and forest fires in Indonesia during 1997, Nature, 420, 61–65, doi: 10.1038/nature01131, 2002.

Rap, A., Scott, C.E., Reddington, C.L., Mercado, L., Ellis, R.J., Garraway, S., Evans, M.J., Beerling, D.J., MacKenzie, A.R., Hewitt, C.N., and Spracklen, D.V.: Enhanced global primary production by biogenic aerosol via diffuse radiation fertilization, Nat. Geosci., et al., Nat Geoscience, doi: 10.1038/s41561-018-0208-3, 2018.

Schmidt, G.A., Kelley, M., Nazarenko, L., Ruedy, R., Russell, G.L., Aleinov, I., Bauer, M., Bauer, S.E., Bhat, M.K., Bleck, R., Canuto, V., Chen, Y.-H., Cheng, Y., Clune, T.L., Del Genio, A., de Fainchtein, R., Faluvegi, G., Hansen, J.E., Healy, R.J., Kiang, N.Y., Koch, D., Lacis, A.A., LeGrande, A.N., Lerner, J., Lo, K.K., Matthews, E.E., Menon, S., Miller, R.L., Oinas, V., Oloso, A.O., Perlwitz, J.P., Puma, M.J., Putman, W.M., Rind, D., Romanou, A., Sato, M., Shindell, D.T., Sun, S., Syed, R.A., Tausnev, N., Tsigaridis, K., Unger, N., Voulgarakis, A., Yao, M.-S., and Zhang, J.: Configuration and assessment of the GISS ModelE2 contributions to the CMIP5 archive, J. Adv. Model. Earth Sy., 6, 141–184, doi: 10.1002/2013MS000265, 2014.

Schultz, M.G., Heil, A., Hoelzemann, J.J., Spessa, A., Thonicke, K., Goldammer, J.G., Held, A.C., Pereira, J.M.C., and van het Bolscher, M.: Global wildland fire emissions from 1960 to 2000, Global Biogeochem. Cy., 22, GB2002, doi: 10.1029/2007GB003031, 2008.

Scott, C.E., Arnold, S.R., Monks, S.A., Asmi, A., Paasonen, P., and Spracklen, D.V.: Substantial large-scale feedbacks between natural aerosols and climate, Nat Geoscience, 11, 44–48, doi: 10.1038/s41561-017-0020-5, 2018.

Scott, C.E., Monks, S.A., Spracklen, D.V., Arnold, S.R., Forster, P.M., Rap, A., Äijälä, M., Artaxo, P., Carslaw, K.S., Chipperfield, M.P., Ehn, M., Gilardoni, S., Heikkinen, L., Kulmala, M., Petäjä, T., Reddington, C.L.S., Rizzo, L.V., Swietlicki, E., Vignati, E., and Wilson, C.: Impact on short-lived climate forcers increases projected warming due to deforestation, Nat. Commun., 9:157, 1–9, doi: 10.1038/s41467-017-02412-4, 2018.

Scott, C.E., Monks, S.A., Spracklen, D.V., Arnold, S.R., Forster, P.M., Rap, A., Carslaw, K.S., Chipperfield, M.P., Reddington, C.L.S., and Wilson, C.: Impact on short-lived climate forcers (SLCFs) from a realistic land-use change scenario via changes in biogenic emissions, Faraday Discuss., 200, 101–120, doi: 10.1039/c7fd00028f, 2017.

Shindell, D.T., Faluvegi, G., and Bell, N.: Preindustrial-to-present-day radiative forcing by tropospheric ozone from improved simulations with the GISS chemistry-climate GCM, Atmos. Chem. Phys., 3, 1675–1702, doi: 10.5194/acp-3-1675-2003, 2003.

Shindell, D.T., Faluvegi, G., Unger, N., Aguilar, E., Schmidt, G.A., Koch, D.M., Bauer, S.E., and Miller, R.L.: Simulations of preindustrial, present-day, and 2100 conditions in the NASA GISS

composition and climate model G-PUCCINI, Atmos. Chem. Phys., 6, 4427–4459, doi: 10.5194/acp-6-4427-2006, 2006.

Shindell, D.T., Pechony, O., Voulgarakis, A., Faluvegi, G., Nazarenko, L., Lamarque, J.-F., Bowman, K., Milly, G., Kovari, B., Ruedy, R., and Schmidt, G.A.: Interactive ozone and methane chemistry in GISS-E2 historical and future climate simulations, Atmos. Chem. Phys., 13, 2653–2689, doi: 10.5194/acp-13-2653-2013, 2013.

Silva, S.J., Heald, C.L., Geddes, J.A., Austin, K.G., Kasibhatla, P.S., and Marlier, M.E.: Impacts of current and projected oil palm plantation expansion on air quality over Southeast Asia, Atmos. Chem. Phys., 16, 10621–10635, doi: 10.5194/acp-16-10621-2016, 2016.

Sodhi, N.S., Koh, L.P., Brook, B.W., and Ng, P.K.L.: Southeast Asian biodiversity: An impending disaster, Trends Ecol. Evol., 19, 654–660, doi: 10.1016/j.tree.2004.09.006, 2004.

Stavrakou, T., Müller, J.-F., Bauwens, M., De Smedt, I., Van Roozendael, M., Guenther, A., Wild, M., and Xia, X.: Isoprene emissions over Asia 1979-2012: Impact of climate and land-use changes, Atmos. Chem. Phys., 14, 4587–4605, doi: 10.5194/acp-14-4587-2014, 2014.

Stockwell, W.R., Kirchner, F., Kuhn, M., and Seefeld, S.: A new mechanism for regional atmospheric chemistry modeling, J. Geophys. Res., 102, 25,847–25,879, doi: 10.1029/97JD00849, 1997.

Surratt, J.D., Chan, A.W.H., Eddingsaas, N.C., Chan, M., Loza, C.L., Kwan, A.J., Hersey, S.P., Flagan, R.C., Wennberg, P.O., and Seinfeld, J.H.: Reactive intermediates revealed in secondary organic aerosol formation from isoprene, P. Natl. Acad. Sci. USA, 107, 6640–6645, doi: 10.1073/pnas.0911114107, 2010.

Thompson, A.M., Tao, W.-K., Pickering, K.E., Scala, J.R., and Simpson, J.: Tropical deep convection and ozone formation, B. Am. Meteorol. Soc., 78, 1043–1054, doi: 10.1175/1520-0477(1997)078<1043:TDCAOF>2.0.CO;2, 1997.

Tilmes, S., Lamarque, J.-F., Emmons, L.K., Conley, A., Schultz, M.G., Saunois, M., Thouret, V., Thompson, A.M., Oltmans, S.J., Johnson, B., and Tarasick, D.: Technical Note: Ozonesonde climatology between 1995 and 2011: Description, evaluation and applications, Atmos. Chem. Phys., 12, 7475–7497, doi: 10.5194/acp-12-7475-2012, 2012.

Tsigaridis, K., Daskalakis, N., Kanakidou, M., Adams, P.J., Artaxo, P., Bahadur, R., Balkanski, Y., Bauer, S.E., Bellouin, N., Benedetti, A., Bergman, T., Berntsen, T.K., Beukes, J.P., Bian, H., Carslaw, K.S., Chin, M., Curci, G., Diehl, T., Easter, R.C., Ghan, S.J., Gong, S.L., Hodzic, A., Hoyle, C.R., Iversen, T., Jathar, S., Jimenez, J.L., Kaiser, J.W., Kirkevåg, A., Koch, D., Kokkola, H., Lee, Y.H., Lin, G., Liu, X., Luo, G., Ma, X., Mann, G.W., Mihalopoulos, N., Morcrette, J.-J., Müller, J.-F., Myhre, G., Myriokefalitakis, S., Ng, N.L., O'Donnell, D., Penner, J.E., Pozzoli, L., Pringle, K.J., Russell, L.M., Schulz, M., Sciare, J., Seland, Ø., Shindell, D.T., Sillman, S., Skeie, R. B., Spracklen,

D., Stavrakou, T., Steenrod, S.D., Takemura, T., Tiitta, P.,Tilmes, S., Tost, H., van Noije, T., van Zyl, P.G., von Salzen, K., Yu, F., Wang, Z., Wang, Z., Zaveri, R. A., Zhang, H., Zhang, K., Zhang, Q., and Zhang, X.: The AeroCom evaluation and intercomparison of organic aerosol in global models, Atmos. Chem. Phys, 14, 10845–10895, doi: 10.5194/acp-14-10845-2014, 2014.

Tsigaridis, K. and Kanakidou, M.: Secondary organic aerosol importance in the future atmosphere, Atmos. Environ., 41, 4682–4692, doi: 10.1016/j.atmosenv.2007.03.045, 2007.

Turnock, S.T., Wild, O., Dentener, F.J., Davila, Y., Emmons, L.K., Flemming, J., Folberth, G.A., Henze, D.K., Jonson, J.E., Keating, T.J., Kengo, S., Lin, M., Lund, M., Tilmes, S., and O'Connor, F.M.: The impact of future emission policies on tropospheric ozone using a parameterized approach, Atmos. Chem. Phys., 18, 8953–8978, doi: 10.5194/acp-18-8953-2018, 2018.

Unger, N.: Human land-use-driven reduction of forest volatiles cools global climate, Nat. Clim. Change, 4, 907–910, doi: 10.1038/NCLIMATE2347, 2014a.

Unger, N.: On the role of plant volatiles in anthropogenic global climate change, Geophys. Res. Lett., 41, 8563–8569, doi: 10.1002/2014GL061616, 2014b.

Unger, N., Bond, T.C., Wang, J.S., Koch, D.M., Menon, S., Shindell, D.T., and Bauer, S.: Attribution of climate forcing to economic sectors, P. Natl. Acad. Sci. USA, 107, 3382–3387, doi: 10.1073/pnas.0906548107, 2010.

van der Werf, G.R., Morton, D.C., DeFries, R.S., Olivier, J.G., Kasibhatla, P.S., Jackson, R.B., Collatz, G.J., and Randerson, J.T.: CO2 emissions from forest loss, Nat. Geosci., 2, 737–738, doi: 10.1038/ngeo671, 2009.

van der Werf, G.R., Randerson, J.T., Giglio, L., Collatz, G.J., Kasibhatla, P.S., and Arellano Jr., A.F.: Interannual variability in global biomass burning emissions from 1997 to 2004, Atmos. Chem. Phys., 6, 3423–3441, doi: 10.5194/acp-6-3423-2006, 2006.

Vermeulen, S. and Goad, N.: Towards better practice in smallholder palm oil production, Natural Resource Issues Series (No. 5), International Institute for Environment and Development, London, UK, 2006.

Warwick, N.J., Archibald, A.T., Ashworth, K., Dorsey, J., Edwards, P.M., Heard, D.E., Langford, B., Lee, J., Misztal, P.K., Whalley, L.K., and Pyle, J.A.: A global model study of the impact of land-use change in Borneo on atmospheric composition, Atmos. Chem. Phys., 13, 9183–9194, doi: 10.5194/acp-13-9183-2013, 2013.

Wild, O., Fiore, A.M., Shindell, D.T., Doherty, R.M., Collins, W.J., Dentener, F.J., Schultz, M.G., Gong, S., MacKenzie, I.A., Zeng, G., Hess, P., Duncan, B.N., Bergmann, D.J., Szopa, S., Jonson, J.E., Keating, T.J., and Zuber, A.: Modelling future changes in surface ozone: A parameterized approach, Atmos. Chem. Phys., 12, 2037–2054, doi: 10.5194/acp-12-2037-2012, 2012.

Wolfe, G.M., Kaiser, J., Hanisco, T.F., Keutsch, F.N., de Gouw, J.A., Gilman, J.B., Graus, M., Hatch, C.D., Holloway, J., Horowitz, L.W., Lee, B.H., Lerner, B.M., Lopez-Hilifiker, F., Mao, J., Marvin, M.R., Peischl, J., Pollack, I.B., Roberts, J.M., Ryerson, T.B., Thornton, J.A., Veres, P.R., and Warneke, C.: Formaldehyde production from isoprene oxidation across NOx regimes, Atmos. Chem. Phys., 16, 2597–2610, doi: 10.5194/acp-16-2597-2016, 2016.

Wong, A.Y.H., Tai, A.P.K., and Ip, Y.-Y.: Attribution and statistical parameterization of the sensitivity of surface ozone to changes in leaf area index based on a chemical transport model, J. Geophys. Res.-Atmos., 123, 1883–1898, doi: 10.1002/2017JD027311, 2018.

WRI: CAIT Climate Data Explorer [Electronic database], World Resources Institute, http://cait.wri.org, 2015.

Yu, F.: A secondary organic aerosol formation model considering successive oxidation aging and kinetic condensation of organic compounds: Global scale implications, Atmos. Chem. Phys., 11, 1083–1099, doi: 10.5194/acp-11-1083-2011, 2011.

Zhang, H., Yee, L.D., Lee, B.H., Curtis, M.P., Worton, D.R., Isaacman-VanWertz, G., Offenberg, J.H., Lewandowski, M., Kleindienst, T.E., Beaver, M.R., Holder, A.L., Lonneman, W.A., Docherty, K.S., Jaoui, M., Pye, H.O.T., Hu, W., Day, D.A., Campuzano-Jost, P., Jimenez, J.L., Guo, H., Weber, R.J., de Gouw, J., Koss, A.R., Edgerton, E.S., Brune, W., Mohr, C., Lopez-Hilfiker, F.D., Lutz, A., Kreisberg, N.M., Spielman, S.R., Hering, S.V., Wilson, K.R., Thornton, J.A., and Goldstein, A.H.: Monoterpenes are the largest source of summertime organic aerosol in the southeastern United States, P. Natl. Acad. Sci. USA, 115, 2038–2043, doi: 10.1073/pnas.1717513115, 2018.

---

## Author Comment (AC2) · 26 Aug 2018

We thank the reviewers for their helpful comments, which have led us to a substantially improved version of the paper. Here, the reviewers' comments are shown in boldfaced black text, and our responses are shown in non-boldfaced blue text. The page and line numbers to which we refer in our responses correspond to the updated manuscript (the comments of all reviewers are taken into account in this updated manuscript).

First and foremost, we confirm that the tropospheric chemical mechanism in GISS ModelE2 is not CBM04. The original manuscript version contained an incorrect oversimplified description of the tropospheric chemistry scheme in GISS ModelE2 that has caused our Reviewers confusion and understandable concerns. We understand that using an old-fashioned chemical mechanism developed 25 years ago for urban polluted high-NOx environments would be an inappropriate tool to apply to a study of large-scale isoprene emission perturbation in the tropics. The chemical mechanism in GISS ModelE2 has been substantially updated and improved over the past 15 years, for example, to account for important reactions, pathways, and species under low-NOx conditions (e.g. Shindell et al., 2003; 2006; 2013; Schmidt et al., 2014).

We now include a more detailed description of the chemical mechanism in Section 2.1 (ModelE2-YIBs description) (Page 4, Line 32):

"The troposphere features $NO_X$-$O_X$-$HO_X$-CO-$CH_4$ chemistry; an explicit representation of isoprene; and a lumped hydrocarbon scheme involving terpenes, peroxyacyl nitrates (PANs), alkyl nitrates, aldehydes, alkenes, and alkanes. The representation of hydrocarbons generally follows Houweling et al. (1998), which is originally derived from the Carbon Bond Mechanism-4 (Gery et al., 1989) and the Regional Atmospheric Chemistry Model (RACM; Stockwell et al., 1997), but includes several modifications aimed at representing the wide range of chemical conditions found in Earth's atmosphere, such as the addition of reactions important in low-$NO_X$ conditions including representation of organic peroxy radical chemistry under low-$NO_X$ conditions and introduction of organic nitrate chemistry. Shindell et al. (2013) describe in detail the recent updates to the tropospheric chemistry scheme, including the incorporation of acetone chemistry (Houweling et al., 1998) and the addition of terpene oxidation (Tsigaridis and Kanakidou, 2007). SOA formation is driven by $NO_X$-dependent oxidation of emissions of isoprene, monoterpenes, and other reactive VOCs following a volatility-based two-product scheme (Tsigaridis and Kanakidou, 2007). The formation of secondary inorganic aerosols, including sulfate (Bell et al., 2005; Koch et al., 2006) and nitrate (Bauer et al., 2007a), depend on both modeled oxidant levels and the availability of source gases. Primary aerosol types include dust (which provides a surface for heterogeneous chemistry; Bauer and Koch, 2005; Bauer et al., 2007b), black carbon, organic carbon, and sea salt (Koch et al., 2006). Stratospheric chemistry, introduced to the chemical mechanism by Shindell et al. (2006), includes nitrous oxide ($N_2O$) and halogen (bromine and chlorine) chemistry. Recent updates to stratospheric chemistry are summarized by Shindell et al. (2013) and include changes in the representations of polar stratospheric cloud formation (Hanson and Mauersberger, 1988) and heterogeneous hydrolysis of $N_2O_5$ on sulfate (Hallquist et al., 2003; Kane et al., 2001)."

Interdisciplinary work is challenging. We would like to emphasize the novel aspects of this project. (1) The land cover dataset for maritime Southeast Asia that we use in our study is built from an existing classification based on Landsat images (Gunarso et al., 2013). This dataset represents a wall-to-wall mapping of land cover in this region, including explicit representation of plantations of oil palm (high isoprene emitter) and rubber (high monoterpene emitter). Gunarso et al. (2013) used a consistent classification methodology for each year of their analysis, which has provided an internally consistent set of land cover maps for this period for this region. Other studies have investigated the atmospheric composition impacts from land cover change in this region: Ashworth et al. (2012) considered a projection of forest to oil palm conversion based on meeting future demand for biofuels; Warwick et al. (2013) considered the total conversion of Borneo to oil palm from forest; and Silva et al. (2016) considered the impact of 2010 oil palm cover relative to an oil-palm-free landscape in addition to considering future projections of oil palm. We consider the impacts of actual historical land cover change, which is clearly different than Ashworth et al. (2012) and Warwick et al. (2013). The Silva et al. (2016) study imposes oil palm expansion by overlaying an oil palm map for 2010 on a separate 16-PFT land cover distribution; this is a different methodology than we apply here, where we apply an internally consistent set of maps developed using a wall-to-wall classification methodology. (2) We have developed the global climate model code to add four additional land cover type PFTs, focusing on land covers that are pervasive in maritime Southeast Asia, including oil palm and rubber plantations. Previous studies have focused only on the impacts of oil palm expansion. (3) We consider the impacts of land-cover-change-driven changes in emissions of both isoprene and monoterpenes. The study by Silva et al. (2016) presumably includes dynamic changes in monoterpene emissions for the land covers that are displaced by oil palm, but their one new land cover type – oil palm – only has the isoprene emission capacity altered relative to the forest land cover type. Ashworth et al. (2012) and Warwick et al. (2013) consider only isoprene emission changes. (4) We directly quantify the global radiative forcing induced by ozone and SOA changes driven by historical land cover change in this region using a coupled global land-chemistry-climate model framework with the embedded radiative transfer model developed by A. Lacis and J. Hansen in GISS ModelE2 (e.g. Schmidt et al., 2014). (5) We provide new climate policy metrics for global ozone radiative forcing per Mha land cover change in the tropics. (6) We quantitatively identify that important factors driving uncertainty in the forcing include (a) uncertainty in the magnitude of the isoprene BER for oil palm and (b) uncertainty in the areal extent of oil palm expansion. Using an analysis based on fixed SOA yields, we additionally show that the sign of the net forcing is sensitive to uncertainty in the SOA yield from BVOCs.

**Responses to Reviewer #2**

**Harper and Unger present a study of the radiative forcing brought about via differences in isoprene emission under different land use configurations in the maritime Southeast Asia (MSEA) region. These land use changes comprise the move towards more oil palm plantations, which emit more biogenic volatile organic compounds than the native natural forests. The changes in isoprene emitted to the atmosphere as a result of the increased oil palm leads to changes in ozone. Of particular interest is that the Enhanced BVOCs caused**

bigger changes globally to ozone in the upper troposphere (0.6 ppb) than lower troposphere (>0.1 ppb), which would seem an important result. The novelty of this study is that the authors then go on to calculate the radiative forcing expected by these ozone changes, finding a small increase of +1 mW m-2 Mha-1.

This shows that impacts of land use changes in tropical regions, which are subject to stronger convective patterns that elsewhere, are very important.

My feeling is that this is a really nice idea, but the wrong tool has been used to carry out the study. The small changes in ozone seen at the top of the troposphere are probably lost in the noise of uncertainty of the chemical scheme chosen, and thus I question the impacts on radiative forcing.

The authors use the carbon bond 4 chemical mechanism to represent the oxidation of isoprene in the atmosphere. This scheme is very old and does not include some of the recent discoveries brought about via questioning the discrepancies between isoprene predicted by models, and observed mixing ratios. These particularly relate to additional OH recycling, which directly impact the influence of isoprene on O3 (eg Lelieveld et al., 2008; Peeters et al., 2009).

The authors do mention the uncertainty in the isoprene chemistry regarding increased oxidant cycling, right at the end of the paper in the conclusions, but I think there are other problems with this choice of chemistry scheme. High isoprene atmospheres, such as that found in this MSEA region, have caused more differences in chemical mechanisms than most others. Unfortunately, the carbon bond scheme has never fared well when tested alongside other chemistry schemes under similar isoprene rich atmospheres. I wonder why there has been no model development in the chemistry scheme in this work when the science behind this paper depends so highly upon it?

For example Knote et al (2015) tested two variants of the newer carbon bond 5 (CB05) scheme (neither of which contained updates to the isoprene chemistry) and found they "tended to be biased low in O3 under low NOx/high VOC conditions (e.g. biogenic emissions rich) as well as under very high NOx conditions. In general, the CB05 schemes produced 'lower than average 8 hourly O3' produced by other schemes. Mechanisms were 'found to differ more strongly in their predictions of O3 levels and other pollutants in regions with strong biogenic VOC emissions".

Archibald et al (2010) tested 8 chemical schemes in isoprene rich regions and found that the CB05 mechanism was 'unable to generate/recycle HOx at the rates needed to match recently reported observations at locations characterized by low levels of NOx.'

An older study - Emmerson and Evans (2009) tested the carbon bond 4 scheme against 6 other schemes. However the carbon bond 4 results disagreed with the majority of the other schemes, in even the sign of the changes in ozone (e.g. loss instead of production - see figure

**3 panel e). Differences (and thus uncertainty) of 14 ppb were found between the resulting ozone from the Master Chemical Mechanism and the carbon bond 4 scheme, which is 14 times more than the ~1 ppb of ozone changes found in Harper and Unger's study at the top of the troposphere, and upon which the radiative forcing calculations are based.**

**Thus I don't agree with the authors' comment that no updates to the chemistry have occurred because of "its apparent inconsequence to the surface pollution impacts of regional land cover change". I think if a different chemistry scheme had been implemented that the changes in ozone found by Harper and Unger as a result of including more oil palm plantations in the model would lead to more significant differences in the radiative forcing than found by their study.**

**I'd recommend updating the chemistry scheme. Perhaps even to include a sensitivity study with a more up to date representation of just the isoprene chemistry – particularly one that agrees with the sign of ozone changes driven by our current understanding. The chemical aspect of Harper and Unger's work is my only criticism, which if rectified I would then recommend publication in ACP.**

Thank you for the thoughtful comments and guidance. We confirm that the tropospheric chemical mechanism in GISS ModelE2 is not CBM04. Please see response at the top of this document at response to Reviewer #1 point (9). Certainly, we too would have major concerns about a study using CBM04 to quantify composition impacts of a large isoprene emission injection in the tropics. The revised manuscript now includes a more detailed and accurate description of the chemical mechanism.

We have removed this sentence: "its apparent inconsequence to the surface pollution impacts of regional land cover change." We now provide a more balanced assessment of uncertainties due to isoprene oxidation chemistry, which we have moved to the methodology (Sect. 2.1) as advised in point (9) below. In our expanded assessment, we have included reference to the studies noted above (Archibald et al., 2010; Emmerson and Evans (2009); Knote et al. (2015)).

**General comments**

**(1) A map figure would be good, showing the study area with the areal extent of regions growing oil palm in 1990 and where/how these regions have increased by 2010.**

The original version of the manuscript includes Figure S1 in the Supplementary Information; this figure is now labeled Figure S2 in the updated manuscript. This figure shows, for each of eight land cover types (including oil palm), the regional change in land cover for (1) 1990–2005 and (2) 1990–2010. In the manuscript, we refer to this figure on Page 11, Line 16.

We have included a new figure (known as Figure S1 in the updated manuscript) that shows the areal extent of these same eight land cover types in year 1990. We have added the following

sentence to the manuscript to point readers to this figure (Page 16, Line 14): "Figure S1 in the Supplement shows the regional land cover distribution for 1990."

**(2) Page 2 line 2. "Compared to natural forests oil palm plantations are much stronger emitters of BVOCs" Some numbers would be good here. How much stronger?**

Updated text (additions in bold; Page 2, Line 11): "Above-canopy flux measurements **taken in Borneo in 2008** indicate that, compared to the natural forests of maritime Southeast Asia (MSEA), oil palm plantations are much stronger emitters of the biogenic volatile organic compound (BVOC) isoprene ($C_5H_8$), **with mean midday fluxes about five times stronger from oil palm** (Langford et al., 2010; Misztal et al., 2011)."

The factor difference may be even stronger if comparing the canopy-level BERs, but the values of the BERs can depend on the model parameterization applied (e.g., Misztal et al., 2011), so we use the above comparison instead.

**(3) Page 2 line 20. Try placing the (Baker et al., 2005; Klinger et al., 2002) references at the end of sentence to avoid breaking the flow of the sentence up too much.**

Fixed. (We introduce the high-monoterpene-emitting capacity of rubber trees earlier in the revised introduction.)

**(4) Page 2 line 29. How is photolysis treated in the model?**

We have provided an expanded description of photolysis in the manuscript (Page 5, Line 18): "Photolysis rate calculations follow the Fast-J2 scheme of Bian and Prather (2002). At each 30 minute time step, the simulated distributions of clouds, ozone, and aerosols are passed to the photolysis code, providing a mechanism for simulated changes in aerosols to impact atmospheric chemistry through modification of photolysis rates (Bian et al., 2003)."

We have added these references:

Bian, H. and Prather, M.J.: Fast-J2: Accurate simulation of stratospheric photolysis in global chemical models, J. Atmos. Chem., 41, 281–296, doi: 10.1023/A:1014980619462, 2002.

Bian, H., Prather, M.J., and Takemura, T.: Tropospheric aerosol impacts on trace gas budgets through photolysis, J. Geophys. Res., 108, 4242, doi: 10.1029/2002JD002743, 2003.

**(5) Page 3 line 27. 'the' calculation**

Fixed.

**(6) Page 5 line 12. It is not clear where this LAI dataset has come from?**

For the four new land cover types that have been added to ModelE2-YIBs for this study, the assigned LAI values are from published literature and are shown in Table 1 in the main text (this table was previously Table S2 in the Supplement), with the references noted in the footnotes of that table. To better point readers to this information, we have added the phrase (Page 9, Line 20) "including LAI and vegetation height" to the sentence in the manuscript that describes the information that can be found in Table 1; in this sentence, we additionally replace the phrase "BVOC emissions" with the more specific phrase "leaf-level basal emission rates for isoprene and monoterpenes." (The assigned LAI applied to other vegetation is described in the following paragraph.)

**(7) Page 5 line 14 (onwards in this paragraph). LAI has units of m2 m-2**

Fixed in all three instances in this paragraph (Page 9, Line 11): "An analysis of the leaf area index (LAI) of rainforest plots in Central Sulawesi, Indonesia, under different land use regimes found that disturbance of the forest by selective logging reduced the LAI below the 6.2 $m^2$ [leaf] $m^{-2}$ [ground] value measured for the undisturbed natural forest (Dietz et al., 2007). Removal of "small-diameter" trees reduced LAI to 5.3 $m^2$ $m^{-2}$, while removal of "large-diameter" trees reduced LAI to 5.0 $m^2$ $m^{-2}$ (Dietz et al., 2007)."

We also added the units to two places in the new Table 1 (previously Table S2 in the Supplement) – once in the table and once in the footnote.

**(8) Page 5 line 21. Table S2 – mention this is in the supplementary section.**

We have moved this table and its footnotes to the main text (now known as Table 1).

**(9) Page 19 line 21. This whole discussion of uncertainties in the chemistry scheme would be better placed in section 2.1 which introduces the method used.**

Fixed. We expanded this discussion and moved it to Section 2.1.

**References**

Archibald, A.T., Jenkin, M.E., Shallcross, D.E., 2010. An isoprene mechanism intercomparison. Atmos. Environ. 44 (40), 5356e5364.

Emmerson, K. M., and Evans, M. J.: Comparison of tropospheric gas-phase chemistry schemes for use within global models, Atmos Chem Phys, 9, 1831-1845, DOI 10.5194/acp-9-1831-2009, 2009.

Knote, C., Tuccella, P., Curci, G., Emmons, L., Orlando, J. J., Madronich, S., Baro, R., Jimenez-Guerrero, P., Luecken, D., Hogrefe, C., Forkel, R., Werhahn, J., Hirtl, M., Perez, J. L., San Jose, R., Giordano, L., Brunner, D., Yahya, K., and Zhang, Y.: Influence of the choice of gas-phase mechanism on predictions of key gaseous pollutants during the AQMEII phase-2 intercomparison, Atmos Environ, 115, 553-568, 10.1016/j.atmosenv.2014.11.066, 2015.

Lelieveld, J., Butler, T.M., Crowley, J.N., Dillon, T.J., Fischer, H., Ganzeveld, L., Harder, H., Lawrence, M.G., Martinez, M., Taraborrelli, D., and Williams, J.: Atmospheric oxidation capacity sustained by a tropical forest, Nature, 452, 737–740, doi: 10.1038/na- ture06870, 2008.

Peeters, J., Nguyen, T.L., and Vereecken, L.: HOx radical regeneration in the oxidation of isoprene, Phys. Chem. Chem. Phys., 11, 5935–5939, doi: 10.1039/B908511D, 2009.

**References**

Angiola, A., Mieville, A., and Granier, C.: MACCity (MACC/CityZEN EU projects) emissions dataset [Data files], Emissions of atmospheric Compounds & Compilation of Ancillary Data, http://eccad.sedoo.fr, 2010.

Archibald, A.T., Jenkin, M.E., Shallcross, D.E.: An isoprene mechanism intercomparison, Atmos. Environ., 44, 5356–5364, doi: 10.1016/j.atmosenv.2009.09.016, 2010.

Ashworth, K., Folberth, G., Hewitt, C.N., and Wild, O.: Impacts of near-future cultivation of biofuel feedstocks on atmospheric composition and local air quality, Atmos. Chem. Phys, 12, 919–939, doi: 10.5194/acp-12-919-2012, 2012.

Atkinson, R. and Arey, J.: Gas-phase tropospheric chemistry of biogenic volatile organic compounds: A review, Atmos. Environ., 37, S197–219, doi: 10.1016/S1352-2310(03)00391-1, 2003.

Baker, B., Bai, J.-H., Johnson, C., Cai, Z.-T., Li, Q.-J., Wang, Y.-F., Guenther, A., Greenberg, J., Klinger, L., Geron, C., and Rasmussen, R.: Wet and dry season ecosystem level fluxes of isoprene and monoterpenes from a southeast Asian secondary forest and rubber tree plantation, Atmos. Environ., 39, 381–390, doi: 10.1016/j.atmosenv.2004.07.033, 2005.

Bauer, S.E. and Koch, D.: Impact of heterogeneous sulfate formation at mineral dust surfaces on aerosol loads and radiative forcing in the Goddard Institute for Space Studies general circulation model, J. Geophys. Res., 110, D17202, doi: 10.1029/2005JD005870, 2005.

Bauer, S.E., Koch, D., Unger, N., Metzger, S.M., Shindell, D.T., and Streets, D.G.: Nitrate aerosols today and in 2030: a global simulation including aerosols and tropospheric ozone, Atmos. Chem. Phys., 7, 5043–5059, doi: 10.5194/acp-7-5043-2007, 2007a.

Bauer, S.E., Mishchenko, M.I., Lacis, A.A., Zhang, S., Perlwitz, J., and Metzger, S.M.: Do sulfate and nitrate coatings on mineral dust have important effects on radiative properties and climate modeling?, J. Geophys. Res., 112, D06307, doi: 10.1029/2005JD006977, 2007b.

Beer, C., Reichstein, M., Tomelleri, E., Ciais, P., Jung, M., Carvalhais, N., Rödenbeck, C., Arain, M.A., Baldocchi, D., Bonan, G.B., Bondeau, A., Cescatti, A., Lasslop, G., Lindroth, A., Lomas, M., Luyssaert, S., Margolis, H., Oleson, K.W., Roupsard, O., Veenendaal, E., Viovy, N., Williams, C., Woodward, F.I., and Papale, D.: Terrestrial gross carbon dioxide uptake: Global distribution and covariation with climate, Science, 329, 834–838, doi: 10.1126/science.1184984, 2010.

Bell, N., Koch, D., and Shindell, D.T.: Impacts of chemistry–aerosol coupling on tropospheric ozone and sulfate simulations in a general circulation model, J. Geophys. Res., 110, D14305, doi: 10.1029/2004JD005538, 2005.

Bian, H. and Prather, M.J.: Fast-J2: Accurate simulation of stratospheric photolysis in global chemical models, J. Atmos. Chem., 41, 281–296, doi: 10.1023/A:1014980619462, 2002.

Bian, H., Prather, M.J., and Takemura, T.: Tropospheric aerosol impacts on trace gas budgets through photolysis, J. Geophys. Res., 108, 4242, doi: 10.1029/2002JD002743, 2003.

Carlson, K.M., Curran, L.M., Asner, G.P., Pittman, A.M., Trigg, S.N., and Marion Adeney, J.: Carbon emissions from forest conversion by Kalimantan oil palm plantations, Nat. Clim. Change, 3, 283–287, doi: 10.1038/nclimate1702, 2012.

Carlson, K.M., Curran, L.M., Ratnasari, D., Pittman, A.M., Soares-Filho, B.S., Asner, G.P., Trigg, S.N., Gaveau, D.A., Lawrence, D., and Rodrigues, H.O.: Committed carbon emissions, deforestation, and community land conversion from oil palm plantations expansion in West Kalimantan, Indonesia, Proc. Natl. Acad. Sci.-USA, 109, 7559–7564, doi: 10.1073/pnas.1200452109, 2012b.

Carlton, A.G., Wiedinmyer, C., and Kroll, J.H.: A review of secondary organic aerosol (SOA) formation from isoprene, Atmos. Chem. Phys., 9, 4987–5005, doi: 10.5194/acp-9-4987-2009, 2009.

Dietz, J., Hölscher, D., Leuschner, C., Malik, A., and Amir, M.A.: Forest structure as influenced by different types of community forestry in a lower montane rainforest of Central Sulawesi, Indonesia, in: Stability of Tropical Rainforest Margins: Linking Ecological, Economic and Social Constraints of Land Use and Conservation, edited by: Tscharntke, T., Leuschner, C., Zeller, M., Guhardja, E., and Biden, A., Springer-Verlag, Berlin, pp. 131–146, 2007.

Draper, D.C., Farmer, D.K., Desyaterik, Y., and Fry, J.L.: A qualitative comparison of secondary organic aerosol yields and composition from ozonolysis of monoterpenes at varying concentrations of NO2, Atmos. Chem. Phys., 15, 12267–12281, doi: 10.5194/acp-15-12267-2015, 2015.

Emmerson, K. M. and Evans, M. J.: Comparison of tropospheric gas-phase chemistry schemes for use within global models, Atmos. Chem. Phys., 9, 1831–1845, doi: 10.5194/acp-9-1831-2009, 2009.

FAO: FAOSTAT Emissions Database [Electronic database], Food and Agriculture Organization of the United Nations, http://www.fao.org/faostat/en/#data, 2014.

Fowler, D., Nemitz, E., Misztal, P., Di Marco, C., Skiba, U., Ryder, J., Helfter, C., Cape, J.N., Owen, S., Dorsey, J., Gallagher, M.W., Coyle, M., Phillips, G., Davison, B., Langford, B., MacKenzie, R., Muller, J., Siong, J., Dari-Salisburgo, C., Di Carlo, P., Aruffo, E., Giammaria, F., Pyle, J.A., and Hewitt, C.N.: Effects of land use on surface-atmosphere exchanges of trace gases and energy in Borneo: Comparing fluxes over oil palm plantations and a rainforest, Philos. T. R. Soc. B, 366, 3196–3209, doi: 10.1098/rstb.2011.0055, 2011.

Friedman, B. and Farmer, D.K.: SOA and gas phase organic acid yields from the sequential photooxidation of seven monoterpenes, Atmos. Env., 187, 335–345, doi: 10.1016/j.atmosenv.2018.06.003, 2018.

Fry, M.M., Naik, V., West, J.J., Schwarzkopf, M.D., Fiore, A.M., Collins, W.J., Dentener, F.J., Shindell, D.T., Atherton, C., Bergmann, D., Duncan, B.N., Hess, P., MacKenzie, I.A., Marmer,E., Schultz, M.G., Szopa, S., Wild, O., and Zeng, G.: The influence of ozone precursor emissions from four world regions on tropospheric composition and radiative climate forcing, J. Geophys. Res., 117, D07306, doi: 10.1029/2011JD017134, 2012.

Fuglestvedt, J., Berntsen, T., Myhre, G., Rypdal, K., and Skeie, R.B.: Climate forcing from the transport sectors, P. Natl. Acad. Sci. USA, 105, 454–458, doi: 10.1073/pnas.0702958104, 2008.

Gaveau, D.L.A., Salim, M.A., Hergoualc'h, K., Locatelli, B., Sloan, S., Wooster, M., Marlier, M.E., Molidena, E., Yaen, H., DeFries, R., Verchot, L., Murdiyarso, D., Nasi, R., Holmgren, P., and Sheil,

D.: Major atmospheric emissions from peat fires in Southeast Asia during non-drought years: Evidence from the 2013 Sumatran fires, Sci. Rep.-U.K., 4, 6112, doi: 10.1038/srep06112, 2014.

Geron, C., Guenther, A., Sharkey, T., and Arnts, R.R.: Temporal variability in basal isoprene emission factor, Tree Physiol., 20, 799–805, 2000.

Geron, C., Owen, S., Guenther, A., Greenberg, J., Rasmussen, R., Bai, J.H., Li, Q.-J., and Baker, B.: Volatile organic compounds from vegetation in southern Yunnan Province, China: Emission rates and some potential regional implications, Atmos. Environ., 40, 1759–1773, doi: 10.1016/j.atmosenv.2005.11.022, 2006.

Gery, M.W., Whitten, G.Z., Killus, J.P., and Dodge, M.C.: A photochemical kinetics mechanism for urban and regional scale computer modeling, J. Geophys. Res.-Atmos., 94, 12,925–12,956, doi: 10.1029/JD094iD10p12925, 1989.

Granier, C., Bessagnet, B., Bond, T., D'Angiola, A., Denier van der Gon, H., Frost, G.J., Heil, A., Kaiser, J.W., Kinne, S., Klimont, Z., Kloster, S., Lamarque, J.-F., Liousse, C., Masui, T., Meleux, F., Mieville, A., Ohara, T., Raut, J.-C., Riahi, K., Schultz, M.G., Smith, S.J., Thompson, A., van Aardenne, J., van der Werf, G.R., and van Vuuren, D.P.: Evolution of anthropogenic and biomass burning emissions of air pollutants at global and regional scales during the 1980–2010 period, Climatic Change, 109, 163–190, doi: 10.1007/s10584-011-0154-1, 2011.

Guenther, A., Hewitt, C.N., Erickson, D., Fall, R., Geron, C., Graedel, T., Harley, P., Klinger, L., Lerdau, M., McKay, W.A., Pierce, T., Scholes, B., Steinbrecher, R., Tallamraju, R., Taylor, J., and Zimmerman, P.: A global model of natural volatile organic compound emissions, J. Geophys. Res.-Atmos., 100, 8873–8892, doi: 10.1029/94JD02950, 1995.

Guenther, A.B., Jiang, X., Heald, C.L., Sakulyanontvittaya, T., Duhl, T., Emmons, L.K., and Wang, X.: The Model of Emissions of Gases and Aerosols from Nature version 2.1 (MEGAN2.1): An extended and updated framework for modeling biogenic emissions, Geosci. Model Dev., 5, 1471–1492, doi: 10.5194/gmd-5-1471-2012, 2012.

Gunarso, P., Hartoyo, M.E., Agus, F., and Killeen, T.J.: Oil palm and land use change in Indonesia, Malaysia, and Papua New Guinea: Reports from the Technical Panels of the 2nd Greenhouse Gas Working Group of the Roundtable on Sustainable Palm Oil (RSPO), Roundtable on Sustainable Palm Oil, Kuala Lumpur, Malaysia, 2013.

Hallquist, M., Stewart, D.J., Stephenson, S.K., and Cox, R.A.: Hydrolysis of $N_2O_5$ on sub-micron sulfate aerosols, Phys. Chem. Chem. Phys., 5, 3453–3463, doi: 10.1039/b301827j, 2003.

Hanson, D. and Mauersberger, K.: Laboratory studies of the nitric acid trihydrate: Implications for the south polar stratosphere, Geophys. Res. Lett., 15, 855–858, doi: 10.1029/GL015i008p00855, 1988.

Harper, K.L., Zheng, Y., and Unger, N.: Advances in representing interactive methane in ModelE2-YIBs (version 1.1), Geosci. Model Dev. Discuss., doi: 10.5194/gmd-2018-85 , 2018.

Heald, C.L. and Geddes, J.A.: The impact of historical land use change from 1850 to 2000 on secondary particulate matter and ozone, Atmos. Chem. Phys., 16, 14997–15010, doi: 10.5194/acp-16-14997-2016, 2016.

Heil, A. and Schultz, M.G.: Interpolated ACCMIP and RCP emission dataset [Data files], Available from:  http://accmip-emis.iek.fz-juelich.de/data/accmip/gridded_netcdf/accmip_interpolated/, 2014.

Hewitt, C.N., Lee, J.D., MacKenzie, A.R., Barkley, M.P., Carslaw, N., Carver, G.D., Chappell, N.A., Coe, H., Collier, C., Commane, R., Davies, F., Davison, B., DiCarlo, P., Di Marco, C.F., Dorsey, J.R., Edwards, P.M., Evans, M.J., Fowler, D., Furneaux, K.L., Gallagher, M., Guenther, A., Heard, D.E., Helfter, C., Hopkins, J., Ingham, T., Irwin, M., Jones, C., Karunaharan, A., Langford, B., Lewis, A.C., Lim, S.F., MacDonald, S.M., Mahajan, A.S., Malpass, S., McFiggans, G., Mills, G., Misztal, P., Moller, S., Monks, P.S., Nemitz, E., Nicolas-Perea, V., Oetjen, H., Oram, D.E., Palmer, P.I., Phillips, G.J., Pike, R., Plane, J.M.C., Pugh, T., Pyle, J.A., Reeves, C.E., Robinson, N.H., Stewart, D., Stone, D., Whalley, L.K., and Yang, X.: Overview: oxidant and particle photochemical processes above a south-east Asian tropical rainforest (the OP3 project): Introduction, rationale, location characteristics and tools, Atmos. Chem. Phys., 10, 169–199, doi: 10.5194/acp-10-169-2010, 2010.

Hewitt, C.N., MacKenzie, A.R., Di Carlo, P., Di Marco, C.F., Dorsey, J.R., Evans, M., Fowler, D., Gallagher, M.W., Hopkins, J.R., Jones, C.E., Langford, B., Lee, J.D., Lewis, A.C., Lim, S.F., McQuaid, J., Misztal, P., Moller, S.J., Monks, P.S., Nemitz, E., Oram, D.E., Owen, S.M., Phillips, G.J., Pugh, T.A.M., Pyle, J.A., Reeves, C.E., Ryder, J., Siong, J., Skiba, U., and Stewart, D.J.: Nitrogen management is essential to prevent tropical oil palm plantations from causing ground-level ozone pollution, P. Natl. Acad. Sci. USA, 106, 18447–18451, doi: 10.1073/pnas.0907541106, 2009.

Hollaway, M.J., Arnold, S.R., Collins, W.J., Folberth, G., and Rap, A.: (2017), Sensitivity of midnineteenth century tropospheric ozone to atmospheric chemistry-vegetation interactions, J. Geophys. Res.-Atmos., 122, 2452–2473, doi:10.1002/2016JD025462, 2017.

Hooijer, A., Page, S., Canadell, J.G., Silvius, M., Kwadijk, J., Wösten, H., and Jauhiainen, J.: Current and future CO2 emissions from drained peatlands in Southeast Asia, Biogeosciences, 7, 1505–1514, doi: 10.5194/bg-7-1505-2010, 2010.

Houweling, S., Dentener, F., and Lelieveld, J.: The impact of nonmethane hydrocarbon compounds on tropospheric photochemistry, J. Geophys. Res., 103, 10,673–10,696, doi: 10.1029/97JD03582, 1998.

Hoyle, C.R., Berntsen, T., Myhre, G., and Isaksen, I.S.A.: Secondary organic aerosol in the global aerosol–chemical transport model Oslo CTM2, Atmos. Chem. Phys., 7, 5675–5694, doi: 10.5194/acp-7-5675-2007, 2007.

Indonesian Ministry of Agriculture: Tree Crop Estate Statistics of Indonesia 2015–2017, Directorate General of Estate Crops, Ministry of Agriculture, Indonesia, 2017.

Indrarto, G.B., Murharjanti, P., Khatarina, J., Pulungan, I., Ivalerina, F., Rahman, J., Prana, M.N., Resosudarmo, I.A.P., and Muharrom, E.: The context of REDD+ in Indonesia: Drivers, agents and institutions (Working Paper 92), CIFOR, Bogor, Indonesia, 2012.

Jacob, D.J.: Introduction to Atmospheric Chemistry, Princeton University Press, 1999.

Jardine, A.B., Jardine, K.J., Fuentes, J.D., Martin, S.T., Martins, G., Durgante, F., Carneiro, V., Higuchi, N., Manzi, A.O., and Chambers, J.Q.: Highly reactive light-dependent monoterpenes in the Amazon, Geophys. Res. Lett., 42, 1576–1583, doi: 10.1002/2014GL062573, 2015.

Jardine, K.J., Jardine, A.B., Holm, J.A., Lombardozzi, D.L., Negron-Juarez, R.I., Martin, S.T., Beller, H.R., Gimenez, B.O., Higuchi, N., and Chambers, J.Q.: Monoterpene 'thermometer' of tropical forest-atmosphere response to climate warming, Plant Cell Environ., 40, 441–452, doi: 10.1111/pce.12879, 2017.

Jokinen, T., Berndt, T., Makkonen, R., Kerminen, V.-M., Junninen, H., Paasonen, P., Stratmann, F., Herrmann, H., Guenther, A.B., Worsnop, D.R., Kulmala, M., Ehn, M., and Sipilä, M.: Production of extremely low volatile organic compounds from biogenic emissions: Measured yields and atmospheric implications, P. Natl. Acad. Sci. USA, 112, 7123–7128, doi: 10.1073/pnas.1423977112, 2015.

Kane, S.M., Caloz, F., and Leu, M.-T.: Heterogeneous uptake of gaseous $N_2O_5$ by $(NH_4)_2SO_4$, $NH_4HSO_4$, and $H_2SO_4$ aerosols, J. Phys. Chem. A, 105, 6465–6470, doi: 10.1021/jp010490x , 2001.

Kelly, J.M., Doherty, R.M., O'Connor, F., and Mann, G.W.: The impact of biogenic, anthropogenic, and biomass burning volatile organic compound emissions on regional and seasonal variations in secondary organic aerosol, Atmos. Chem. Phys., 18, 7393–7422, doi: 10.5194/acp-18-7393-2018, 2018.

Klinger, L.F., Li, Q.-J., Guenther, A.B., Greenberg, J.P., Baker, B., and Bai, J.-H.: Assessment of volatile organic compound emissions from ecosystems of China, J. Geophys. Res.-Atmos., 107, 4603, doi: 10.1029/2001JD001076, 2002.

Knote, C., Tuccella, P., Curci, G., Emmons, L., Orlando, J. J., Madronich, S., Baró, R., Jiménez-Guerrero, P., Luecken, D., Hogrefe, C., Forkel, R., Werhahn, J., Hirtl, M., Perez, J. L., San José, R., Giordano, L., Brunner, D., Yahya, K., and Zhang, Y.: Influence of the choice of gas-phase

mechanism on predictions of key gaseous pollutants during the AQMEII phase-2 intercomparison, Atmos. Environ., 115, 553–568, doi: 10.1016/j.atmosenv.2014.11.066, 2015.

Koch, D., Schmidt, G.A., and Field, C.V.: Sulfur, sea salt, and radionuclide aerosols in GISS ModelE, J. Geophys. Res., 111, D06206, doi: 10.1029/2004JD005550, 2006.

Koplitz, S.N., Mickley, L.J., Marlier, M.E., Buonocore, J.J., Kim, P.S., Liu, T., Sulprizio, M.P., DeFries, R.S., Jacob, D.J., Schwartz, J., Pongsiri, M., and Myers, S.S.: Public health impacts of the severe haze in Equatorial Asia in September–October 2015: Demonstration of a new framework for informing fire management strategies to reduce downwind smoke exposure, Environ. Res. Lett., 11, 094023, doi: 10.1088/1748-9326/11/9/094023, 2016.

Kotchenruther, R.A., Jaffe, D.A., and Jaeglé, L.: Ozone photochemistry and the role of peroxyacetyl nitrate in the springtime northeastern Pacific troposphere: Results from the Photochemical Ozone Budget of the Eastern North Pacific Atmosphere (PHOBEA) campaign, J. Geophys. Res., 106, 28,731–28,742, doi: 10.1029/2000JD000060, 2001.

Kroll, J.H., Ng, N.L., Murphy, S.M., Flagan, R.C., and Seinfeld, J.H.: Secondary organic aerosol formation from isoprene photooxidation under high-$NO_X$ conditions, Geophys. Res. Lett., 32, L18808, doi: 10.1029/2005GL023637, 2005.

Lacis, A.A., Wuebbles, D.J., and Logan, J.A.: Radiative forcing of climate by changes in the vertical distribution of ozone, J. Geophys. Res., 95, 9971–9981, doi: 10.1029/JD095iD07p09971, 1990.

Lamarque, J.-F., Bond, T.C., Eyring, V., Granier, C., Heil, A., Klimont, Z., Lee, D., Liousse, C., Mieville, A., Owen, B., Schultz, M.G., Shindell, D., Smith, S.J., Stehfest, E., Van Aardenne, J., Cooper, O.R., Kainuma, M., Mahowald, N., McConnell, J.R., Naik, V., Riahi, K., and van Vuuren, D.P.: Historical (1850–2000) gridded anthropogenic and biomass burning emissions of reactive gases and aerosols: methodology and application, Atmos. Chem. Phys., 10, 7017–7039, doi: 10.5194/acp-10-7017-2010, 2010.

Langford, B., Misztal, P.K., Nemitz, E., Davison, B., Helfter, C., Pugh, T.A.M., MacKenzie, A.R., Lim, S.F., and Hewitt, C.N.: Fluxes and concentrations of volatile organic compounds from a South-East Asian tropical rainforest, Atmos. Chem. Phys., 10, 8391–8412, doi: 10.5194/acp-10-8391-2010, 2010.

Lathière, J., Hauglustaine, D.A., Friend, A.D., De Noblet-Ducoudré, N., Viovy, N., and Folberth, G.A.: Impact of climate variability and land use changes on global biogenic volatile organic compound emissions, Atmos. Chem. Phys., 6, 2129–2146, doi: 10.5194/acp-6-2129-2006, 2006.

Malaysian Palm Oil Board (MPOB), Economics and Industry Development Division, Statistics: bepi.mpob.gov.my, accessed: 25 August 2018.

Margono, B.A., Potapov, P.V., Turubanova, S., Stolle, F., and Hansen, M.C.: Primary forest cover loss in Indonesia over 2000–2012, Nat. Clim. Change, 4, 730–735, doi: 10.1038/nclimate2277, 2014.

McPeters, R., Kroon, M., Labow, G., Brinksma, E., Balis, D., Petropavlovskikh, I., Veefkind, J., Bhartia, P., and Levelt, P.: Validation of the aura ozone monitoring instrument total column ozone product, J. Geophys. Res.-Atmos., 113, D15S14, doi:10.1029/2007JD008802, 2008.

Miettinen, J., Hooijer, A., Vernimmen, R., Liew, S.C., and Page, S.E.: From carbon sink to carbon source: Extensive peat oxidation in insular Southeast Asia since 1990, Environ. Res. Lett., 12, 024014, doi: 10.1088/1748-9326/aa5b6f, 2017.

Misztal, P.K., Nemitz, E., Langford, B., Di Marco, C.F., Phillips, G.J., Hewitt, C.N., MacKenzie, A.R., Owen, S.M., Fowler, D., Heal, M.R., and Cape, J.N.: Direct ecosystem fluxes of volatile organic compounds from oil palms in South-East Asia, Atmos. Chem. Phys., 11, 8995–9017, doi: 10.5194/acp-11-8995-2011, 2011.

Moxim, W.J., Levy II, H., and Kasibhatla, P.S.: Simulated global tropospheric PAN: Its transport and impact on NOx, J. Geophys. Res., 101, 12,621–12,638, doi: 10.1029/96JD00338, 1996.

Myhre, G., Shindell, D., Bréon, F.-M., Collins, W., Fuglestvedt, J., Huang, J., Koch, D., Lamarque, J.-F., Lee, D., Mendoza, B., Nakajima, T., Robock, A., Stephens, G., Takemura, T., and Zhang, H.: Anthropogenic and natural radiative forcing, in: Climate Change 2013: The Physical Science Basis. Contribution of Working Group I to the Fifth Assessment Report of the Intergovernmental Panel on Climate Change, edited by: Stocker, T.F., Qin, D., Plattner, G.-K., Tignor, M., Allen, S.K., Boschung, J., Nauels, A., Xia, Y., Bex, V., and Midgley, P.M., Cambridge University Press, Cambridge, United Kingdom and New York, 2013.

Page, S.E., Siegert, F., Rieley, J.O., Boehm, H.-D.V., Jaya, A., and Limin, S.: The amount of carbon released from peat and forest fires in Indonesia during 1997, Nature, 420, 61–65, doi: 10.1038/nature01131, 2002.

Rap, A., Scott, C.E., Reddington, C.L., Mercado, L., Ellis, R.J., Garraway, S., Evans, M.J., Beerling, D.J., MacKenzie, A.R., Hewitt, C.N., and Spracklen, D.V.: Enhanced global primary production by biogenic aerosol via diffuse radiation fertilization, Nat. Geosci., et al., Nat Geoscience, doi: 10.1038/s41561-018-0208-3, 2018.

Schmidt, G.A., Kelley, M., Nazarenko, L., Ruedy, R., Russell, G.L., Aleinov, I., Bauer, M., Bauer, S.E., Bhat, M.K., Bleck, R., Canuto, V., Chen, Y.-H., Cheng, Y., Clune, T.L., Del Genio, A., de Fainchtein, R., Faluvegi, G., Hansen, J.E., Healy, R.J., Kiang, N.Y., Koch, D., Lacis, A.A., LeGrande, A.N., Lerner, J., Lo, K.K., Matthews, E.E., Menon, S., Miller, R.L., Oinas, V., Oloso, A.O., Perlwitz, J.P., Puma, M.J., Putman, W.M., Rind, D., Romanou, A., Sato, M., Shindell, D.T., Sun, S., Syed, R.A., Tausnev, N., Tsigaridis, K., Unger, N., Voulgarakis, A., Yao, M.-S., and Zhang, J.:

Configuration and assessment of the GISS ModelE2 contributions to the CMIP5 archive, J. Adv. Model. Earth Sy., 6, 141–184, doi: 10.1002/2013MS000265, 2014.

Schultz, M.G., Heil, A., Hoelzemann, J.J., Spessa, A., Thonicke, K., Goldammer, J.G., Held, A.C., Pereira, J.M.C., and van het Bolscher, M.: Global wildland fire emissions from 1960 to 2000, Global Biogeochem. Cy., 22, GB2002, doi: 10.1029/2007GB003031, 2008.

Scott, C.E., Arnold, S.R., Monks, S.A., Asmi, A., Paasonen, P., and Spracklen, D.V.: Substantial large-scale feedbacks between natural aerosols and climate, Nat Geoscience, 11, 44–48, doi: 10.1038/s41561-017-0020-5, 2018.

Scott, C.E., Monks, S.A., Spracklen, D.V., Arnold, S.R., Forster, P.M., Rap, A., Äijälä, M., Artaxo, P., Carslaw, K.S., Chipperfield, M.P., Ehn, M., Gilardoni, S., Heikkinen, L., Kulmala, M., Petäjä, T., Reddington, C.L.S., Rizzo, L.V., Swietlicki, E., Vignati, E., and Wilson, C.: Impact on short-lived climate forcers increases projected warming due to deforestation, Nat. Commun., 9:157, 1–9, doi: 10.1038/s41467-017-02412-4, 2018.

Scott, C.E., Monks, S.A., Spracklen, D.V., Arnold, S.R., Forster, P.M., Rap, A., Carslaw, K.S., Chipperfield, M.P., Reddington, C.L.S., and Wilson, C.: Impact on short-lived climate forcers (SLCFs) from a realistic land-use change scenario via changes in biogenic emissions, Faraday Discuss., 200, 101–120, doi: 10.1039/c7fd00028f, 2017.

Shindell, D.T., Faluvegi, G., and Bell, N.: Preindustrial-to-present-day radiative forcing by tropospheric ozone from improved simulations with the GISS chemistry-climate GCM, Atmos. Chem. Phys., 3, 1675–1702, doi: 10.5194/acp-3-1675-2003, 2003.

Shindell, D.T., Faluvegi, G., Unger, N., Aguilar, E., Schmidt, G.A., Koch, D.M., Bauer, S.E., and Miller, R.L.: Simulations of preindustrial, present-day, and 2100 conditions in the NASA GISS composition and climate model G-PUCCINI, Atmos. Chem. Phys., 6, 4427–4459, doi: 10.5194/acp-6-4427-2006, 2006.

Shindell, D.T., Pechony, O., Voulgarakis, A., Faluvegi, G., Nazarenko, L., Lamarque, J.-F., Bowman, K., Milly, G., Kovari, B., Ruedy, R., and Schmidt, G.A.: Interactive ozone and methane chemistry in GISS-E2 historical and future climate simulations, Atmos. Chem. Phys., 13, 2653–2689, doi: 10.5194/acp-13-2653-2013, 2013.

Silva, S.J., Heald, C.L., Geddes, J.A., Austin, K.G., Kasibhatla, P.S., and Marlier, M.E.: Impacts of current and projected oil palm plantation expansion on air quality over Southeast Asia, Atmos. Chem. Phys., 16, 10621–10635, doi: 10.5194/acp-16-10621-2016, 2016.

Sodhi, N.S., Koh, L.P., Brook, B.W., and Ng, P.K.L.: Southeast Asian biodiversity: An impending disaster, Trends Ecol. Evol., 19, 654–660, doi: 10.1016/j.tree.2004.09.006, 2004.

Stavrakou, T., Müller, J.-F., Bauwens, M., De Smedt, I., Van Roozendael, M., Guenther, A., Wild, M., and Xia, X.: Isoprene emissions over Asia 1979-2012: Impact of climate and land-use changes, Atmos. Chem. Phys., 14, 4587–4605, doi: 10.5194/acp-14-4587-2014, 2014.

Stockwell, W.R., Kirchner, F., Kuhn, M., and Seefeld, S.: A new mechanism for regional atmospheric chemistry modeling, J. Geophys. Res., 102, 25,847–25,879, doi: 10.1029/97JD00849, 1997.

Surratt, J.D., Chan, A.W.H., Eddingsaas, N.C., Chan, M., Loza, C.L., Kwan, A.J., Hersey, S.P., Flagan, R.C., Wennberg, P.O., and Seinfeld, J.H.: Reactive intermediates revealed in secondary organic aerosol formation from isoprene, P. Natl. Acad. Sci. USA, 107, 6640–6645, doi: 10.1073/pnas.0911114107, 2010.

Thompson, A.M., Tao, W.-K., Pickering, K.E., Scala, J.R., and Simpson, J.: Tropical deep convection and ozone formation, B. Am. Meteorol. Soc., 78, 1043–1054, doi: 10.1175/1520-0477(1997)078<1043:TDCAOF>2.0.CO;2, 1997.

Tilmes, S., Lamarque, J.-F., Emmons, L.K., Conley, A., Schultz, M.G., Saunois, M., Thouret, V., Thompson, A.M., Oltmans, S.J., Johnson, B., and Tarasick, D.: Technical Note: Ozonesonde climatology between 1995 and 2011: Description, evaluation and applications, Atmos. Chem. Phys., 12, 7475–7497, doi: 10.5194/acp-12-7475-2012, 2012.

Tsigaridis, K., Daskalakis, N., Kanakidou, M., Adams, P.J., Artaxo, P., Bahadur, R., Balkanski, Y., Bauer, S.E., Bellouin, N., Benedetti, A., Bergman, T., Berntsen, T.K., Beukes, J.P., Bian, H., Carslaw, K.S., Chin, M., Curci, G., Diehl, T., Easter, R.C., Ghan, S.J., Gong, S.L., Hodzic, A., Hoyle, C.R., Iversen, T., Jathar, S., Jimenez, J.L., Kaiser, J.W., Kirkevåg, A., Koch, D., Kokkola, H., Lee, Y.H., Lin, G., Liu, X., Luo, G., Ma, X., Mann, G.W., Mihalopoulos, N., Morcrette, J.-J., Müller, J.-F., Myhre, G., Myriokefalitakis, S., Ng, N.L., O'Donnell, D., Penner, J.E., Pozzoli, L., Pringle, K.J., Russell, L.M., Schulz, M., Sciare, J., Seland, Ø., Shindell, D.T., Sillman, S., Skeie, R. B., Spracklen, D., Stavrakou, T., Steenrod, S.D., Takemura, T., Tiitta, P.,Tilmes, S., Tost, H., van Noije, T., van Zyl, P.G., von Salzen, K., Yu, F., Wang, Z., Wang, Z., Zaveri, R. A., Zhang, H., Zhang, K., Zhang, Q., and Zhang, X.: The AeroCom evaluation and intercomparison of organic aerosol in global models, Atmos. Chem. Phys, 14, 10845–10895, doi: 10.5194/acp-14-10845-2014, 2014.

Tsigaridis, K. and Kanakidou, M.: Secondary organic aerosol importance in the future atmosphere, Atmos. Environ., 41, 4682–4692, doi: 10.1016/j.atmosenv.2007.03.045, 2007.

Turnock, S.T., Wild, O., Dentener, F.J., Davila, Y., Emmons, L.K., Flemming, J., Folberth, G.A., Henze, D.K., Jonson, J.E., Keating, T.J., Kengo, S., Lin, M., Lund, M., Tilmes, S., and O'Connor, F.M.: The impact of future emission policies on tropospheric ozone using a parameterized approach, Atmos. Chem. Phys., 18, 8953–8978, doi: 10.5194/acp-18-8953-2018, 2018.

Unger, N.: Human land-use-driven reduction of forest volatiles cools global climate, Nat. Clim. Change, 4, 907–910, doi: 10.1038/NCLIMATE2347, 2014a.

Unger, N.: On the role of plant volatiles in anthropogenic global climate change, Geophys. Res. Lett., 41, 8563–8569, doi: 10.1002/2014GL061616, 2014b.

Unger, N., Bond, T.C., Wang, J.S., Koch, D.M., Menon, S., Shindell, D.T., and Bauer, S.: Attribution of climate forcing to economic sectors, P. Natl. Acad. Sci. USA, 107, 3382–3387, doi: 10.1073/pnas.0906548107, 2010.

van der Werf, G.R., Morton, D.C., DeFries, R.S., Olivier, J.G., Kasibhatla, P.S., Jackson, R.B., Collatz, G.J., and Randerson, J.T.: CO2 emissions from forest loss, Nat. Geosci., 2, 737–738, doi: 10.1038/ngeo671, 2009.

van der Werf, G.R., Randerson, J.T., Giglio, L., Collatz, G.J., Kasibhatla, P.S., and Arellano Jr., A.F.: Interannual variability in global biomass burning emissions from 1997 to 2004, Atmos. Chem. Phys., 6, 3423–3441, doi: 10.5194/acp-6-3423-2006, 2006.

Vermeulen, S. and Goad, N.: Towards better practice in smallholder palm oil production, Natural Resource Issues Series (No. 5), International Institute for Environment and Development, London, UK, 2006.

Warwick, N.J., Archibald, A.T., Ashworth, K., Dorsey, J., Edwards, P.M., Heard, D.E., Langford, B., Lee, J., Misztal, P.K., Whalley, L.K., and Pyle, J.A.: A global model study of the impact of land-use change in Borneo on atmospheric composition, Atmos. Chem. Phys., 13, 9183–9194, doi: 10.5194/acp-13-9183-2013, 2013.

Wild, O., Fiore, A.M., Shindell, D.T., Doherty, R.M., Collins, W.J., Dentener, F.J., Schultz, M.G., Gong, S., MacKenzie, I.A., Zeng, G., Hess, P., Duncan, B.N., Bergmann, D.J., Szopa, S., Jonson, J.E., Keating, T.J., and Zuber, A.: Modelling future changes in surface ozone: A parameterized approach, Atmos. Chem. Phys., 12, 2037–2054, doi: 10.5194/acp-12-2037-2012, 2012.

Wolfe, G.M., Kaiser, J., Hanisco, T.F., Keutsch, F.N., de Gouw, J.A., Gilman, J.B., Graus, M., Hatch, C.D., Holloway, J., Horowitz, L.W., Lee, B.H., Lerner, B.M., Lopez-Hilifiker, F., Mao, J., Marvin, M.R., Peischl, J., Pollack, I.B., Roberts, J.M., Ryerson, T.B., Thornton, J.A., Veres, P.R., and Warneke, C.: Formaldehyde production from isoprene oxidation across NOx regimes, Atmos. Chem. Phys., 16, 2597–2610, doi: 10.5194/acp-16-2597-2016, 2016.

Wong, A.Y.H., Tai, A.P.K., and Ip, Y.-Y.: Attribution and statistical parameterization of the sensitivity of surface ozone to changes in leaf area index based on a chemical transport model, J. Geophys. Res.-Atmos., 123, 1883–1898, doi: 10.1002/2017JD027311, 2018.

WRI: CAIT Climate Data Explorer [Electronic database], World Resources Institute, http://cait.wri.org, 2015.

Yu, F.: A secondary organic aerosol formation model considering successive oxidation aging and kinetic condensation of organic compounds: Global scale implications, Atmos. Chem. Phys., 11, 1083–1099, doi: 10.5194/acp-11-1083-2011, 2011.

Zhang, H., Yee, L.D., Lee, B.H., Curtis, M.P., Worton, D.R., Isaacman-VanWertz, G., Offenberg, J.H., Lewandowski, M., Kleindienst, T.E., Beaver, M.R., Holder, A.L., Lonneman, W.A., Docherty, K.S., Jaoui, M., Pye, H.O.T., Hu, W., Day, D.A., Campuzano-Jost, P., Jimenez, J.L., Guo, H., Weber, R.J., de Gouw, J., Koss, A.R., Edgerton, E.S., Brune, W., Mohr, C., Lopez-Hilfiker, F.D., Lutz, A., Kreisberg, N.M., Spielman, S.R., Hering, S.V., Wilson, K.R., Thornton, J.A., and Goldstein, A.H.: Monoterpenes are the largest source of summertime organic aerosol in the southeastern United States, P. Natl. Acad. Sci. USA, 115, 2038–2043, doi: 10.1073/pnas.1717513115, 2018.

---

## Author Comment (AC3) · 26 Aug 2018

We thank the reviewers for their helpful comments, which have led us to a substantially improved version of the paper. Here, the reviewers' comments are shown in boldfaced black text, and our responses are shown in non-boldfaced blue text. The page and line numbers to which we refer in our responses correspond to the updated manuscript (the comments of all reviewers are taken into account in this updated manuscript).

First and foremost, we confirm that the tropospheric chemical mechanism in GISS ModelE2 is not CBM04. The original manuscript version contained an incorrect oversimplified description of the tropospheric chemistry scheme in GISS ModelE2 that has caused our Reviewers confusion and understandable concerns. We understand that using an old-fashioned chemical mechanism developed 25 years ago for urban polluted high-NOx environments would be an inappropriate tool to apply to a study of large-scale isoprene emission perturbation in the tropics. The chemical mechanism in GISS ModelE2 has been substantially updated and improved over the past 15 years, for example, to account for important reactions, pathways, and species under low-NOx conditions (e.g. Shindell et al., 2003; 2006; 2013; Schmidt et al., 2014).

We now include a more detailed description of the chemical mechanism in Section 2.1 (ModelE2-YIBs description) (Page 4, Line 32):

"The troposphere features $NO_X$-$O_X$-$HO_X$-CO-$CH_4$ chemistry; an explicit representation of isoprene; and a lumped hydrocarbon scheme involving terpenes, peroxyacyl nitrates (PANs), alkyl nitrates, aldehydes, alkenes, and alkanes. The representation of hydrocarbons generally follows Houweling et al. (1998), which is originally derived from the Carbon Bond Mechanism-4 (Gery et al., 1989) and the Regional Atmospheric Chemistry Model (RACM; Stockwell et al., 1997), but includes several modifications aimed at representing the wide range of chemical conditions found in Earth's atmosphere, such as the addition of reactions important in low-$NO_X$ conditions including representation of organic peroxy radical chemistry under low-$NO_X$ conditions and introduction of organic nitrate chemistry. Shindell et al. (2013) describe in detail the recent updates to the tropospheric chemistry scheme, including the incorporation of acetone chemistry (Houweling et al., 1998) and the addition of terpene oxidation (Tsigaridis and Kanakidou, 2007). SOA formation is driven by $NO_X$-dependent oxidation of emissions of isoprene, monoterpenes, and other reactive VOCs following a volatility-based two-product scheme (Tsigaridis and Kanakidou, 2007). The formation of secondary inorganic aerosols, including sulfate (Bell et al., 2005; Koch et al., 2006) and nitrate (Bauer et al., 2007a), depend on both modeled oxidant levels and the availability of source gases. Primary aerosol types include dust (which provides a surface for heterogeneous chemistry; Bauer and Koch, 2005; Bauer et al., 2007b), black carbon, organic carbon, and sea salt (Koch et al., 2006). Stratospheric chemistry, introduced to the chemical mechanism by Shindell et al. (2006), includes nitrous oxide ($N_2O$) and halogen (bromine and chlorine) chemistry. Recent updates to stratospheric chemistry are summarized by Shindell et al. (2013) and include changes in the representations of polar stratospheric cloud formation (Hanson and Mauersberger, 1988) and heterogeneous hydrolysis of $N_2O_5$ on sulfate (Hallquist et al., 2003; Kane et al., 2001)."

Interdisciplinary work is challenging. We would like to emphasize the novel aspects of this project. (1) The land cover dataset for maritime Southeast Asia that we use in our study is built from an existing classification based on Landsat images (Gunarso et al., 2013). This dataset represents a wall-to-wall mapping of land cover in this region, including explicit representation of plantations of oil palm (high isoprene emitter) and rubber (high monoterpene emitter). Gunarso et al. (2013) used a consistent classification methodology for each year of their analysis, which has provided an internally consistent set of land cover maps for this period for this region. Other studies have investigated the atmospheric composition impacts from land cover change in this region: Ashworth et al. (2012) considered a projection of forest to oil palm conversion based on meeting future demand for biofuels; Warwick et al. (2013) considered the total conversion of Borneo to oil palm from forest; and Silva et al. (2016) considered the impact of 2010 oil palm cover relative to an oil-palm-free landscape in addition to considering future projections of oil palm. We consider the impacts of actual historical land cover change, which is clearly different than Ashworth et al. (2012) and Warwick et al. (2013). The Silva et al. (2016) study imposes oil palm expansion by overlaying an oil palm map for 2010 on a separate 16-PFT land cover distribution; this is a different methodology than we apply here, where we apply an internally consistent set of maps developed using a wall-to-wall classification methodology. (2) We have developed the global climate model code to add four additional land cover type PFTs, focusing on land covers that are pervasive in maritime Southeast Asia, including oil palm and rubber plantations. Previous studies have focused only on the impacts of oil palm expansion. (3) We consider the impacts of land-cover-change-driven changes in emissions of both isoprene and monoterpenes. The study by Silva et al. (2016) presumably includes dynamic changes in monoterpene emissions for the land covers that are displaced by oil palm, but their one new land cover type – oil palm – only has the isoprene emission capacity altered relative to the forest land cover type. Ashworth et al. (2012) and Warwick et al. (2013) consider only isoprene emission changes. (4) We directly quantify the global radiative forcing induced by ozone and SOA changes driven by historical land cover change in this region using a coupled global land-chemistry-climate model framework with the embedded radiative transfer model developed by A. Lacis and J. Hansen in GISS ModelE2 (e.g. Schmidt et al., 2014). (5) We provide new climate policy metrics for global ozone radiative forcing per Mha land cover change in the tropics. (6) We quantitatively identify that important factors driving uncertainty in the forcing include (a) uncertainty in the magnitude of the isoprene BER for oil palm and (b) uncertainty in the areal extent of oil palm expansion. Using an analysis based on fixed SOA yields, we additionally show that the sign of the net forcing is sensitive to uncertainty in the SOA yield from BVOCs.

**Responses to Reviewer #3**

**Overall comments:**

**This paper examines the impacts of land cover change in maritime Southeast Asia induced mostly by oil palm expansion and the associated changes in BVOC emissions on surface ozone concentrations and tropospheric ozone profiles, and the subsequent impacts on radiative forcing. This is a novel piece of work that highlights the importance of considering**

atmospheric chemistry-mediated climate forcing in climate and land use change studies. The data integration and modeling approach are all scientifically sound, rigorous and valid. There are, however, insufficient or unclear exposition and explanation of the results at various places of the paper, as well as inadequate discussion of the results in relation to previous works. I recommend the publication of this paper, if the concerns raised below are addressed.

We thank the reviewer for their insightful comments and guidance.

**Specific comments:**

**(1) P1 L21: The introduction section appears too short, and do not set up a context nuanced enough to motivate the work (the findings of which are exciting). I recommend the authors to expand the introduction (by 30-50%) by discussing at greater lengths the various references cited. More suggestions in relation to this are given below.**

We have expanded the introduction to provide additional context and background for the study.

(1) A description of other environmental impacts associated with land cover changes in this region was added (Page 2, Line 1): "Land cover and land use changes in Southeast Asia perturb the Earth system in a variety of ways. Deforestation is a significant threat to Southeast Asian biodiversity (Sodhi et al., 2004), and the land-based carbon emissions associated with forest clearing have greatly contributed to Indonesia's status as one of the world's highest emitters of greenhouse gases (GHGs) (FAO, 2014; WRI, 2015). The magnitude of GHG emissions from deforestation is exacerbated by the pervasiveness of carbon-rich peat soils underlying Southeast Asian tropical forests (Carlson et al., 2012; Hooijer et al., 2010; Page et al., 2002; van der Werf et al., 2009). Peat soil drainage (i.e., drying) promotes oxidation of the sequestered carbon (Miettinen et al., 2017) and is often followed by fire clearing, despite the illegality of this practice in Indonesia (Indrarto et al., 2012). Indonesian forest and peat fires have fueled transnational air pollution episodes (Gaveau et al., 2014; Koplitz et al., 2016), potentially causing more than 100,000 premature mortalities in 2015 (Koplitz et al., 2016)."

(2) The bolded part of the following sentence was added to tie the air quality and climate impacts of isoprene and monoterpene emission changes back to the discussion of other environmental impacts of regional land cover change (Page 2, Line 19; our response to comment 5 of reviewer #3 also describes modifications to this sentence): "Both isoprene and monoterpenes are precursors to the short-lived climate pollutants tropospheric ozone (Atkinson and Arey, 2003) and secondary organic aerosols (SOA) (Carlton et al., 2009; Friedman and Farmer, 2018); **as such, perturbations in regional isoprene and monoterpene emissions serve as an additional mechanism by which regional land cover change can affect air quality and climate**."

(3) We replaced this sentence from our original introduction ("Previous investigations of atmospheric composition changes driven by land use change in MSEA have largely focused on the surface-level air quality impacts induced by BVOC emission changes from oil palm expansion (Ashworth et al., 2012; Silva et al., 2016; Warwick et al., 2013).") with an expanded description of the key findings of these cited studies. While no information was removed from the final introductory paragraph of the original version of the manuscript, the sentences in that final paragraph were rearranged and placed into the new expanded text to retain flow. The expanded text includes a description of how the land cover dataset used in the study by Silva et al. (2016) differs from that used in our study.

The expanded text (Page 2, Line 23):

"A few studies have used global modeling to investigate the atmospheric composition impacts of Southeast Asian oil palm expansion. Ashworth et al. (2012) analyzed the expected impacts from isoprene emission enhancements associated with the partial replacement of natural forest area with oil palm plantations under a land-use change scenario designed to meet a portion of the projected increase in demand for biofuels in coming years. Warwick et al. (2013) analyzed the impacts from isoprene emission enhancements associated with total conversion of Borneo from forest to oil palm. Both studies quantified the impacts of the isoprene emission changes first by applying a contemporary $NO_X$ inventory and secondly by assuming increased $NO_X$ emissions near the site of land-use change due to enhanced fertilizer application and increased on-site processing of the palm oil. Based on simulations that apply contemporary $NO_X$ inventories, both studies predict reductions in surface ozone co-located with the isoprene enhancements because the increased VOC serves as a net sink for ozone in the low-$NO_X$ atmosphere (Ashworth et al., 2012; Warwick et al., 2013). When $NO_X$ emission enhancements occur in concert with the land-use-change-driven isoprene emission enhancements, both studies simulate a net increase in ozone production and a concomitant increase in local surface ozone pollution (Ashworth et al., 2012; Warwick et al., 2013). Ashworth et al. (2012) predict local enhancements in annual-mean surface concentrations as high as 11 % for ozone and 10 % for biogenic SOA (as the maximum change in any grid cell) in response to a 5 % increase in regional isoprene emissions and simultaneous enhancements in local $NO_X$ emissions. Warwick et al. (2013) predict local changes in monthly-mean surface ozone as strong +70 % when simultaneously increasing $NO_X$ and isoprene emissions. These studies highlight the potential for significant local surface-level pollution impacts from land use change.

A recent third study quantified the air quality impacts associated with year 2010 oil palm cover relative to an oil-palm-free scenario (Silva et al., 2016). Using the GEOS-Chem chemical transport model and contemporary inventories for $NO_X$ emissions, Silva et al. (2016) simulate local enhancements in surface pollution as high as 26 % (3–4 ppbv) for ozone and 60 % (about 1 μg m$^{-3}$) for SOA. In Kuala Lumpur, Malaysia, the number of days that register ozone levels higher than the limits recommended by the World Health Organization more than doubled due to regional oil palm expansion (56 days based on 2010 oil palm coverage compared to 23 days in the absence of oil palm; Silva et al., 2016), again highlighting the strong impact of Southeast Asian land cover change on surface pollution.

The year 2000 16-PFT land cover distribution map that Silva et al. (2016) apply as the no-oil-palm case is designed for global modeling studies and, therefore, lacks information about the distribution of individual species of vegetation. For example, 7.2 Mha of oil palm existed in 2000 in Indonesia, Malaysia, and Papua New Guinea (Gunarso et al., 2013), yet many global vegetation distributions assign these plantations to one or more of a small number of PFTs. Silva et al. (2016) use a 250 m resolution satellite-based map of year 2010 oil palm plantations to define the contemporary oil palm distribution, and they overlay the oil palm on the palm-free base map, displacing the existing land covers in proportion to their fractional distributions. In reality, the prior land cover of oil palm plantations differs widely by region (Gunarso et al., 2013). Furthermore, the modified land cover distribution from Silva et al. (2016) lacks separate delineations for other pervasive land covers in Southeast Asia, including rubber plantations, which covered at least 6.4 Mha on the maritime continent in 2010 (Gunarso et al., 2013). Rubber trees (*Hevea brasiliensis*) are very weak emitters of isoprene (Geron et al., 2006; Klinger et al., 2002), but very strong emitters of monoterpenes (Baker et al., 2005; Klinger et al., 2002), which are important SOA precursors (Jokinen et al., 2015). Thus, while the study by Silva et al. (2016) provides evidence of significant surface pollution changes induced by oil palm expansion in Southeast Asia, it provides an incomplete picture of the impact of historical land cover change on atmospheric composition.

This small set of studies focuses on the atmospheric composition impacts induced by altered BVOC emissions from Southeast Asian oil palm expansion, all finding that the downwind impacts are smaller in magnitude than the local impacts near the site of land conversion and isoprene emission changes (Ashworth et al., 2012; Silva et al., 2016; Warwick et al., 2013). Ashworth et al. (2012) forecasted a small global climate impact from the increased isoprene emissions in their land-use change scenario, based on the very small simulated global changes in the tropospheric burdens of ozone and the hydroxyl radical (OH). However, no study has directly quantified the climate impacts associated with the induced changes in atmospheric composition.

Isoprene perturbations in the tropics may have a particularly powerful impact on longwave radiative forcing (Unger, 2014) because the strong vertical mixing prevalent in the tropics provides a mechanism for surface pollution perturbations to impact ozone concentrations in the upper troposphere (Thompson et al., 1997), where, on a per-molecule basis, ozone changes induce the strongest climate impact (Lacis et al., 1990). In response to isoprene emission enhancements associated with total conversion of vegetated land to oil palm on the island of Borneo, Warwick et al. (2013) simulate a 20 % increase in ozone at 500 hPa over Borneo and a 20 % increase in peroxyacetyl nitrate (PAN) at 500 hPa downwind of Borneo over the Pacific Ocean. PAN is an organic nitrate that can undergo long-range transport before releasing its reactive $NO_X$ moiety (Moxim et al., 1996), providing a means for ozone formation in remote environments (Kotchenruther et al., 2001). The results of Warwick et al. (2013) suggest that regional isoprene emission changes have the capacity to alter ozone concentrations in the free troposphere and, therefore, induce a radiative forcing.

This study uses the ModelE2-Yale Interactive Terrestrial Biosphere (ModelE2-YIBs) global chemistry–climate model, in conjunction with multiple observational datasets, to quantify the global atmospheric composition changes and, for the first time, the concomitant radiative forcings associated with BVOC emission changes from 1990–2010 land cover change in MSEA. The calculations presented here consider changes in emissions of both isoprene and monoterpenes. The applied regional land cover changes are derived from a Landsat-based classification (Gunarso et al., 2013) and account for changes in eight land covers that are prevalent in MSEA, including high-monoterpene-emitting rubber trees and high-isoprene-emitting oil palm trees."

Added references:

Hooijer, A., Page, S., Canadell, J.G., Silvius, M., Kwadijk, J., Wösten, H., and Jauhiainen, J.: Current and future CO2 emissions from drained peatlands in Southeast Asia, Biogeosciences, 7, 1505–1514, doi: 10.5194/bg-7-1505-2010, 2010.

Indrarto, G.B., Murharjanti, P., Khatarina, J., Pulungan, I., Ivalerina, F., Rahman, J., Prana, M.N., Resosudarmo, I.A.P., and Muharrom, E.: The context of REDD+ in Indonesia: Drivers, agents and institutions (Working Paper 92), CIFOR, Bogor, Indonesia, 2012.

Kotchenruther, R.A., Jaffe, D.A., and Jaeglé, L.: Ozone photochemistry and the role of peroxyacetyl nitrate in the springtime northeastern Pacific troposphere: Results from the Photochemical Ozone Budget of the Eastern North Pacific Atmosphere (PHOBEA) campaign, J. Geophys. Res., 106, 28,731–28,742, doi: 10.1029/2000JD000060, 2001.

Miettinen, J., Hooijer, A., Vernimmen, R., Liew, S.C., and Page, S.E.: From carbon sink to carbon source: Extensive peat oxidation in insular Southeast Asia since 1990, Environ. Res. Lett., 12, 024014, doi: 10.1088/1748-9326/aa5b6f, 2017.

Moxim, W.J., Levy II, H., and Kasibhatla, P.S.: Simulated global tropospheric PAN: Its transport and impact on NOx, J. Geophys. Res., 101, 12,621–12,638, doi: 10.1029/96JD00338, 1996.

Page, S.E., Siegert, F., Rieley, J.O., Boehm, H.-D.V., Jaya, A., and Limin, S.: The amount of carbon released from peat and forest fires in Indonesia during 1997, Nature, 420, 61–65, doi: 10.1038/nature01131, 2002.

van der Werf, G.R., Morton, D.C., DeFries, R.S., Olivier, J.G., Kasibhatla, P.S., Jackson, R.B., Collatz, G.J., and Randerson, J.T.: CO2 emissions from forest loss, Nat. Geosci., 2, 737–738, doi: 10.1038/ngeo671, 2009.

**(2) P1 L25: The "%" sign usually immediately follows the number without space.**

Fixed.

**(3) P1 L27: Can there be a sentence or two describing why we are concerned with oil palm planation from an environmental or ecological perspective (not just a climate perspective as included in the current second paragraph)?**

Done. Our response to comment 1 describes this addition to the introduction.

**(4) P2 L8-9: Please expand this paragraph by discussing briefly the key findings of these few papers (Ashworth et al., 2012; Silva et al., 2016; Warwick et al., 2013). How large or in what ranges are the concentration changes? Is the surface air quality changes significant relatively to, e.g., the impacts of anthropogenic emissions or warming?**

Done. Our response to comment 1 describes this addition to the introduction.

**(5) P2 L14: Why does upper tropospheric ozone have a larger climate impact than surface ozone?**

Briefly, because it is colder at higher altitudes, which increases the longwave radiative forcing efficiency (outgoing energy at colder temperatures compared to surface). The stronger climate impact of upper tropospheric ozone relative to surface-level ozone, on a per-molecule basis, is related to the temperature contrast of the two environments in which ozone is absorbing and re-emitting the longwave radiation (e.g., Lacis et al., 1990). The forcing efficiency of the longwave-absorbing ozone molecule is stronger when the ozone exists in the colder upper troposphere than when it exists in the warmer temperatures at the surface.

We added the bolded portion to this sentence (Page 4, Line 4): "Isoprene perturbations in the tropics may have a particularly powerful impact on longwave radiative forcing (Unger, 2014) because the strong vertical mixing prevalent in the tropics provides a mechanism for surface pollution perturbations to impact ozone concentrations in the upper troposphere (Thompson et al., 1997), where, on a per-molecule basis, ozone changes induce the strongest climate impact **due to the thermal contrast with the surface** (Lacis et al., 1990)."

**(6) P2 L19: How does the land cover change derived from this data source differ from or compare with that used by Silva et al. (2016)?**

We added this information to the expanded introduction. Our response to comment 1 describes this addition.

**(7) P3 L4: Please explain and justify whether the discontinuity created by using two different biomass burning datasets is acceptable, especially considering that biomass burning emissions are an important source of ozone there.**

Identical surface maps and emission factors were applied in the creation of both the MACCity and interpolated ACCMIP-RCP8.5 biomass burning emissions datasets (Heil and Schultz, 2014).

We have added the bolded portion to the following sentence (Page 5, Line 23): "Prescribed monthly anthropogenic and biomass burning emissions of reactive gas and primary aerosol species follow the MACCity emissions pathway (Angiola et al., 2010; Granier et al., 2011) for all years, except for 2010, when the interpolated ACCMIP-RCP8.5 dataset (Heil and Schultz, 2014) is applied for biomass burning emissions **(as MACCity biomass burning emissions are available only through 2008).**"

We have then added the following explanation (Page 5, Line 27): "MACCity biomass burning emissions were built from the ACCMIP (Lamarque et al., 2010), REanalysis of the TROpospheric chemical composition (RETRO; Schultz et al., 2008), and Global Fire Emissions Database (GFED-v2; van der Werf et al., 2006) datasets (Granier et al., 2011). The interpolated ACCMIP-RCP8.5 emissions were created using simple temporal interpolation of the ACCMIP and RCP8.5 datasets (Heil and Schultz, 2014). Identical surface maps and emission factors were applied in the creation of both the MACCity and interpolated ACCMIP-RCP8.5 biomass burning emissions datasets (Heil and Schultz, 2014)."

Added references:

Lamarque, J.-F., Bond, T.C., Eyring, V., Granier, C., Heil, A., Klimont, Z., Lee, D., Liousse, C., Mieville, A., Owen, B., Schultz, M.G., Shindell, D., Smith, S.J., Stehfest, E., Van Aardenne, J., Cooper, O.R., Kainuma, M., Mahowald, N., McConnell, J.R., Naik, V., Riahi, K., and van Vuuren, D.P.: Historical (1850–2000) gridded anthropogenic and biomass burning emissions of reactive gases and aerosols: methodology and application, Atmos. Chem. Phys., 10, 7017–7039, doi: 10.5194/acp-10-7017-2010, 2010.

Schultz, M.G., Heil, A., Hoelzemann, J.J., Spessa, A., Thonicke, K., Goldammer, J.G., Held, A.C., Pereira, J.M.C., and van het Bolscher, M.: Global wildland fire emissions from 1960 to 2000, Global Biogeochem. Cy., 22, GB2002, doi: 10.1029/2007GB003031, 2008.

van der Werf, G.R., Randerson, J.T., Giglio, L., Collatz, G.J., Kasibhatla, P.S., and Arellano Jr., A.F.: Interannual variability in global biomass burning emissions from 1997 to 2004, Atmos. Chem. Phys., 6, 3423–3441, doi: 10.5194/acp-6-3423-2006, 2006.

**(8) P3 L4-6: "Interactive" is a modeler's jargon, and even for modelers can mean different things for different purposes. I recommend avoiding it and state more clearly that these emission schemes are "semi-empirical", "mechanistic" or "dynamic functions of x, y, z, ...", especially for those that are not described more below.**

**(9) P3 L13: Avoid the use of "online".**

**(10) P3 L28: Avoid "online" and "model's".**

Fixed for all instances where these terms were used in the manuscript.

**(11) P3 L33: Please explain and justify the single chemical representation of monoterpene. Can all monoterpenes really be modeled as $\alpha$-pinene?**

The quantification of SOA yields for different monoterpene species is an exciting frontier for SOA modeling, but the uncertainties remaining in this field currently preclude expansion of this chemistry at the expense of the greater computational resources needed to run global model simulations. We have added a discussion on this matter (Page 7, Line 33): "Furthermore, the appropriateness of using $\alpha$-pinene to represent monoterpenes as a single lumped species in global modeling is an active area of research. Friedman and Farmer (2018) find order-of-magnitude differences in SOA yields for OH-oxidation of different monoterpene species, but a clear explanation based on isomer structure remains largely elusive. While Friedman and Farmer (2018) find that the magnitude of the SOA yield from $\alpha$-pinene is in the "mid-range" of the yields among the analyzed monoterpene species, other studies have shown that this may not be the case for other oxidation pathways (e.g., Draper et al., 2015)."

Added references:

Draper, D.C., Farmer, D.K., Desyaterik, Y., and Fry, J.L.: A qualitative comparison of secondary organic aerosol yields and composition from ozonolysis of monoterpenes at varying concentrations of NO2, Atmos. Chem. Phys., 15, 12267–12281, doi: 10.5194/acp-15-12267-2015, 2015.

Friedman, B. and Farmer, D.K.: SOA and gas phase organic acid yields from the sequential photooxidation of seven monoterpenes, Atmos. Env., 187, 335–345, 2018.

**(12) P5 L30: What about the LAI values for the new PFTs used for this study for MSEA? They are not described above. Are dynamic but grid-level LAI observed from, e.g., MODIS, used, or**

**are PFT-level LAI values used for these new PFTs? If so, where are these values from? As LAI is so important for atmospheric chemistry, these need to be better stated and explained.**

The PFT-level LAI values applied to each of the new PFTs are described in Table 1 and its footnotes in the main text (this table was previously known as Table S2 in the Supplement). We have updated the text to better draw attention to this information (Page 9, Line 20): "Table 1 shows, for the new land cover types, the assigned physical parameters (including LAI and vegetation height), photosynthetic parameters, and leaf-level basal emission rates of isoprene and monoterpenes.."

**(13) P10 L1: In the methodology section above, the authors have only discussed about model validity and model-observation comparison for the vegetation aspects (e.g., GPP, biogenic emissions). What about an evaluation of the ozone simulations by the model? How does the model's simulated ozone globally compare with observations and with other models? Is the general high biases of simulated ozone in many climate-chemistry models also seen in this model? Since ozone concentration is crucial to this paper, I strongly recommend having a paragraph somewhere (preferably in the methodology section) discussing these.**

We have inserted in the methodology section the following paragraph describing the extensive evaluation of ozone in ModelE2-YIBs (Page 7, Line 7): "ModelE2 has previously undergone extensive, rigorous validation of simulated present-day tropospheric and stratospheric chemical composition, circulation, and ozone forcing using multiple observational datasets (Shindell et al., 2006; Shindell et al., 2013). Shindell et al. (2013) compared simulated monthly zonal-mean total column ozone to that from observations (2000–2010 means) from the Total Ozone Mapping Spectrometer and the Ozone Monitoring Instrument (McPeters et al., 2008), finding: simulated zonal-mean total column ozone in the tropics shows little bias (< 5%) against measurements for each month, and, in the Northern Hemisphere middle and high latitudes, biases are smaller in the summer months (< 10%) than in the winter months (around 15–20%). Shindell et al. (2013) find only a small negative bias (-0.016 W m$^{-2}$) in the present-day global-average radiative impact of modeled tropospheric ozone relative to TES-derived tropospheric ozone. They note that the strongest biases in ozone concentrations in ModelE2 are generally located in regions where ozone exhibits little effect on radiation (Shindell et al., 2013). More recently, Harper et al. (2018) compared annual-mean ozone concentrations simulated by ModelE2 (representative of year 2005) with an ozonesonde climatology based on measurements taken over 1995–2011 (Tilmes et al., 2012), finding lower model biases at higher pressures (e.g., +2.6% at 200 hPa compared to +16.9% at 800 hPa)."

Added references:

Harper, K.L., Zheng, Y., and Unger, N.: Advances in representing interactive methane in ModelE2-YIBs (version 1.1), Geosci. Model Dev. Discuss., doi: 10.5194/gmd-2018-85 , 2018.

McPeters, R., Kroon, M., Labow, G., Brinksma, E., Balis, D., Petropavlovskikh, I., Veefkind, J., Bhartia, P., and Levelt, P.: Validation of the aura ozone monitoring instrument total column ozone product, J. Geophys. Res.-Atmos., 113, D15S14, doi:10.1029/2007JD008802, 2008.

Shindell, D.T., Pechony, O., Voulgarakis, A., Faluvegi, G., Nazarenko, L., Lamarque, J.-F., Bowman, K., Milly, G., Kovari, B., Ruedy, R., and Schmidt, G.A.: Interactive ozone and methane chemistry in GISS-E2 historical and future climate simulations, Atmos. Chem. Phys., 13, 2653–2689, doi: 10.5194/acp-13-2653-2013, 2013.

Tilmes, S., Lamarque, J.-F., Emmons, L.K., Conley, A., Schultz, M.G., Saunois, M., Thouret, V., Thompson, A.M., Oltmans, S.J., Johnson, B., and Tarasick, D.: Technical Note: Ozonesonde climatology between 1995 and 2011: Description, evaluation and applications, Atmos. Chem. Phys., 12, 7475–7497, doi: 10.5194/acp-12-7475-2012, 2012.

**(14) P10 L14: Wong et al. (2018) also examined and quantified the factors behind the sensitivity of surface ozone to vegetation changes including isoprene emission and dry deposition. They also found a large impact of background NOx. See reference list below.**

Please see response to point (15).

**(15) P11 L1: Dry deposition definitely also plays a role, and have you quantified the relative importance of isoprene emission vs. dry deposition to surface ozone in your model simulations? This appears to be a major missing part of this analysis and should be better addressed or discussed, even if the authors have already found that dry deposition plays only a minor role. For instance, Wong et al. (2018) found it necessary and developed a method to formally disentangle the contributions from isoprene emission and dry deposition when leaf density changes.**

The Wong et al. (2018) paper presents an exciting avenue for disentangling the various land cover change driven contributions to surface ozone changes; however, it is our understanding that their analysis does not take into account the effects of other biogeophysical changes, such as surface roughness and evapotranspiration. The land cover distribution changes in our model can alter such (and other) parameters, which might also be playing a role in the simulated surface ozone changes.

We have expanded our discussion on this matter (Page 16, Line 25): "The simulated changes in atmospheric composition might be a response not only to altered isoprene and monoterpene emissions, but also to changes in the deposition of atmospheric species induced by changes in leaf density (Wong et al., 2018) or related changes, such as surface roughness, stomatal conductance, and evapotranspiration, that are affected by the applied changes in land cover distribution. Here, the relative changes in regional ozone deposition rates (-19.7 to +4.3%) are similar to the relative changes in regional surface-level ozone concentrations (-18.3 to +4.3%) from 1990–2010 regional land cover change, in part because the ozone deposition rate

depends on the atmospheric concentration change. While increased isoprene emission leading to increased isoprene ozonolysis drives ozone losses near the surface, a formal quantitative attribution analysis disentangling the relative roles of emission and deposition changes requires further complex sensitivity simulations that are beyond the scope of this analysis. In their analysis of Southeast Asian oil palm expansion, Silva et al. (2016) used sensitivity studies to determine that the induced BVOC emission changes, rather than altered deposition rates from LAI changes, were almost exclusively responsible for the simulated surface ozone changes."

Added reference:

Wong, A.Y.H., Tai, A.P.K., and Ip, Y.-Y.: Attribution and statistical parameterization of the sensitivity of surface ozone to changes in leaf area index based on a chemical transport model, J. Geophys. Res.-Atmos., 123, 1883–1898, doi: 10.1002/2017JD027311, 2018.

**(16) P12 L2: The physical reasons for the enhancements (as opposed to reductions) of ozone over the ocean have to be explained. Can these enhancements be explained by, e.g., the mechanisms suggested by Hollaway et al. (2017)? A discussion in relation to this paper is recommended. See reference list below.**

Please see response to comment (18).

**(17) P12 L16: In Fig. 2a) and 2c), why is there a second peak for isoprene and HCHO enhancement near the tropopause?**

This is a signal of tropical deep convection (e.g., deep convective towers rapidly pulling up air into the upper troposphere).

**(18) P14 L5: Now I see that the oceanic enhancements are explained. But this explanation, with reference to Hollaway et al. (2017), should be mentioned early (see comment to P12 L2).**

We added reference to the Hollaway et al. (2017) paper and moved this explanation (now at Page 18, Line 1).

**(19) P17 L8-9: "This sensitivity study demonstrates that the climate forcing associated with regional land cover change is rapidly increasing." I feel that this is too strong a statement. All the results are showing is that 2005-2010 as a 5-year period is responsible for a noticeably large fraction of the total RF compared to other possible 5-year periods, but without breaking down the other years into incremental 5-year periods (e.g., 1990-1995, 1996-2000, 2001-2005), we can't really say there is a rapidly rising trend in RF.**

Agreed. We have removed this sentence.

**(20) P18 L5-7: "increase in regional surface ozone concentrations is unlikely to have a significant impact on the induced ozone forcing since, as Lacis et al. (1990) find, changes in surface ozone have a much smaller effect on climate forcing relative to equivalent ozone changes in the upper troposphere." This is contingent upon the assumption that the formation and long-range transport of isoprene nitrate will respond in the same way even as the surface environment becomes more high-NOx. This needs to be justified.**

We have removed this badly phrased sentence. Please see response to Reviewer (1) Point (66). We were originally trying to emphasize that increases in ozone near the Earth's surface do not exert appreciable longwave forcing, but we agree the original sentence does not read well and is not scientifically nuanced enough as is. Agreed regarding the assumption that the formation and long-range transport of isoprene nitrate would have to respond in the same way even as the surface environment becomes more high-NOx. To be clear, in this model, convective transport is moving isoprene, its oxidation products, and isoprene nitrate into the upper troposphere.

**(21) P18 L24-31: I think one major missing discussion is to compare the ozone-mediated RF with the biogeophysical RF (e.g., changing albedo, latent heat, sensible heat, etc.) and biogeochemical (CO2 exchange) associated with oil palm expansion. Indeed, most climatologists are still just concerned with the biogeophysical or biogeochemical RF, and having a comparison between those and the ozone-mediated forcing would give much insight into the importance of considering atmospheric chemistry in climate/land use change studies.**

Thank you for the brilliant idea, we agree with the reviewer, and these discussions did come up a few times in the project. Unfortunately, we are unable to access the albedo surface forcing diagnostics in the model output. We delved into LLGHG emission estimates for the recent land cover change in MSEA. Carlson et al. (2012b) estimated 11.4 MtC $y^{-1}$ $CO_2$ emissions from 1989–2008 (mostly from deforestation fire). This amounts to a cumulative total of 216.6 MtC $y^{-1}$ over the 19 years. Using IPCC AR5 GWPs, the $CO_2$ forcing is: 5.3 mW $m^{-2}$ on 20-yr time scale and 19.9 mW $m^{-2}$ on 100-yr time scale. However, the small region analysed by Carlson et al. (2012b) doesn't correspond to that applied here so is not a meaningful comparison. LLGHG land cover change emissions accounting is beyond the scope here, and would be a full paper in its own right. Interestingly, there is a substantial high impact published literature on future climate change impacts on the sustainability of oil palm plantations in MSEA (the opposite way around to that considered here).

Reference:

Carlson, K.M., Curran, L.M., Ratnasari, D., Pittman, A.M., Soares-Filho, B.S., Asner, G.P., Trigg, S.N., Gaveau, D.A., Lawrence, D., and Rodrigues, H.O.: Committed carbon emissions, deforestation, and community land conversion from oil palm plantations expansion in West Kalimantan, Indonesia, Proc. Natl. Acad. Sci.-USA, 109, 7559–7564, doi: 10.1073/pnas.1200452109, 2012b.

**(22) P19 L18-19: "Inclusion of a temporally variable BVOC BER in the global model would allow for an improved estimation of radiative forcing induced by land cover changes in this region." I think the current debate is exactly that we are not sure about the circadian control or not, and thus this statement is not necessarily true.**

We agree that this aspect of isoprene emissions is very uncertain and have removed this sentence to avoid confusion.

**(23) P19 L33-34: "(2) its apparent inconsequence to the surface pollution impacts of regional land cover change" Is there really no OH titration problem in MSEA in ModelE2- YIBs? Is that because the BER is low to begin with, compared to, say, the Amazon?**

In ModelE2-YIBs, OH is typically much lower than the global average in forested tropical regions in the model. We have removed this poorly phrased sentence and moved the entire discussion of chemistry uncertainties to the Methodology section (Sect. 2.1, Page 7, Line 21), where we have included a more balanced perspective on isoprene photooxidation uncertainties in global models (e.g., see introductory remarks to reviewer #2).

**References:**
**Hollaway, M. J., S. R. Arnold, W. J. Collins, G. Folberth, and A. Rap (2017), Sensitivity of mid-nineteenth century tropospheric ozone to atmospheric chemistry-vegetation interactions, J. Geophys. Res. Atmos., 122, 2452–2473, doi:10.1002/2016JD025462.**

**Wong, A. Y. H., A. P. K. Tai, and Y.-Y. Ip (2018), Attribution and statistical param- eterization of the sensitivity of surface ozone to changes in leaf area index based on a chemical transport model. J. Geophys. Res. Atmos., 123, 1883–1898, doi:10.1002/2017JD027311.**

References

Angiola, A., Mieville, A., and Granier, C.: MACCity (MACC/CityZEN EU projects) emissions dataset [Data files], Emissions of atmospheric Compounds & Compilation of Ancillary Data, http://eccad.sedoo.fr, 2010.

Archibald, A.T., Jenkin, M.E., Shallcross, D.E.: An isoprene mechanism intercomparison, Atmos. Environ., 44, 5356–5364, doi: 10.1016/j.atmosenv.2009.09.016, 2010.

Ashworth, K., Folberth, G., Hewitt, C.N., and Wild, O.: Impacts of near-future cultivation of biofuel feedstocks on atmospheric composition and local air quality, Atmos. Chem. Phys, 12, 919–939, doi: 10.5194/acp-12-919-2012, 2012.

Atkinson, R. and Arey, J.: Gas-phase tropospheric chemistry of biogenic volatile organic compounds: A review, Atmos. Environ., 37, S197–219, doi: 10.1016/S1352-2310(03)00391-1, 2003.

Baker, B., Bai, J.-H., Johnson, C., Cai, Z.-T., Li, Q.-J., Wang, Y.-F., Guenther, A., Greenberg, J., Klinger, L., Geron, C., and Rasmussen, R.: Wet and dry season ecosystem level fluxes of isoprene and monoterpenes from a southeast Asian secondary forest and rubber tree plantation, Atmos. Environ., 39, 381–390, doi: 10.1016/j.atmosenv.2004.07.033, 2005.

Bauer, S.E. and Koch, D.: Impact of heterogeneous sulfate formation at mineral dust surfaces on aerosol loads and radiative forcing in the Goddard Institute for Space Studies general circulation model, J. Geophys. Res., 110, D17202, doi: 10.1029/2005JD005870, 2005.

Bauer, S.E., Koch, D., Unger, N., Metzger, S.M., Shindell, D.T., and Streets, D.G.: Nitrate aerosols today and in 2030: a global simulation including aerosols and tropospheric ozone, Atmos. Chem. Phys., 7, 5043–5059, doi: 10.5194/acp-7-5043-2007, 2007a.

Bauer, S.E., Mishchenko, M.I., Lacis, A.A., Zhang, S., Perlwitz, J., and Metzger, S.M.: Do sulfate and nitrate coatings on mineral dust have important effects on radiative properties and climate modeling?, J. Geophys. Res., 112, D06307, doi: 10.1029/2005JD006977, 2007b.

Beer, C., Reichstein, M., Tomelleri, E., Ciais, P., Jung, M., Carvalhais, N., Rödenbeck, C., Arain, M.A., Baldocchi, D., Bonan, G.B., Bondeau, A., Cescatti, A., Lasslop, G., Lindroth, A., Lomas, M., Luyssaert, S., Margolis, H., Oleson, K.W., Roupsard, O., Veenendaal, E., Viovy, N., Williams, C., Woodward, F.I., and Papale, D.: Terrestrial gross carbon dioxide uptake: Global distribution and covariation with climate, Science, 329, 834–838, doi: 10.1126/science.1184984, 2010.

Bell, N., Koch, D., and Shindell, D.T.: Impacts of chemistry–aerosol coupling on tropospheric ozone and sulfate simulations in a general circulation model, J. Geophys. Res., 110, D14305, doi: 10.1029/2004JD005538, 2005.

Bian, H. and Prather, M.J.: Fast-J2: Accurate simulation of stratospheric photolysis in global chemical models, J. Atmos. Chem., 41, 281–296, doi: 10.1023/A:1014980619462, 2002.

Bian, H., Prather, M.J., and Takemura, T.: Tropospheric aerosol impacts on trace gas budgets through photolysis, J. Geophys. Res., 108, 4242, doi: 10.1029/2002JD002743, 2003.

Carlson, K.M., Curran, L.M., Asner, G.P., Pittman, A.M., Trigg, S.N., and Marion Adeney, J.: Carbon emissions from forest conversion by Kalimantan oil palm plantations, Nat. Clim. Change, 3, 283–287, doi: 10.1038/nclimate1702, 2012.

Carlson, K.M., Curran, L.M., Ratnasari, D., Pittman, A.M., Soares-Filho, B.S., Asner, G.P., Trigg, S.N., Gaveau, D.A., Lawrence, D., and Rodrigues, H.O.: Committed carbon emissions, deforestation, and community land conversion from oil palm plantations expansion in West Kalimantan, Indonesia, Proc. Natl. Acad. Sci.-USA, 109, 7559–7564, doi: 10.1073/pnas.1200452109, 2012b.

Carlton, A.G., Wiedinmyer, C., and Kroll, J.H.: A review of secondary organic aerosol (SOA) formation from isoprene, Atmos. Chem. Phys., 9, 4987–5005, doi: 10.5194/acp-9-4987-2009, 2009.

Dietz, J., Hölscher, D., Leuschner, C., Malik, A., and Amir, M.A.: Forest structure as influenced by different types of community forestry in a lower montane rainforest of Central Sulawesi, Indonesia, in: Stability of Tropical Rainforest Margins: Linking Ecological, Economic and Social Constraints of Land Use and Conservation, edited by: Tscharntke, T., Leuschner, C., Zeller, M., Guhardja, E., and Biden, A., Springer-Verlag, Berlin, pp. 131–146, 2007.

Draper, D.C., Farmer, D.K., Desyaterik, Y., and Fry, J.L.: A qualitative comparison of secondary organic aerosol yields and composition from ozonolysis of monoterpenes at varying concentrations of NO2, Atmos. Chem. Phys., 15, 12267–12281, doi: 10.5194/acp-15-12267-2015, 2015.

Emmerson, K. M. and Evans, M. J.: Comparison of tropospheric gas-phase chemistry schemes for use within global models, Atmos. Chem. Phys., 9, 1831–1845, doi: 10.5194/acp-9-1831-2009, 2009.

FAO: FAOSTAT Emissions Database [Electronic database], Food and Agriculture Organization of the United Nations, http://www.fao.org/faostat/en/#data, 2014.

Fowler, D., Nemitz, E., Misztal, P., Di Marco, C., Skiba, U., Ryder, J., Helfter, C., Cape, J.N., Owen, S., Dorsey, J., Gallagher, M.W., Coyle, M., Phillips, G., Davison, B., Langford, B., MacKenzie, R., Muller, J., Siong, J., Dari-Salisburgo, C., Di Carlo, P., Aruffo, E., Giammaria, F., Pyle, J.A., and Hewitt, C.N.: Effects of land use on surface-atmosphere exchanges of trace gases and energy in Borneo: Comparing fluxes over oil palm plantations and a rainforest, Philos. T. R. Soc. B, 366, 3196–3209, doi: 10.1098/rstb.2011.0055, 2011.

Friedman, B. and Farmer, D.K.: SOA and gas phase organic acid yields from the sequential photooxidation of seven monoterpenes, Atmos. Env., 187, 335–345, doi: 10.1016/j.atmosenv.2018.06.003, 2018.

Fry, M.M., Naik, V., West, J.J., Schwarzkopf, M.D., Fiore, A.M., Collins, W.J., Dentener, F.J., Shindell, D.T., Atherton, C., Bergmann, D., Duncan, B.N., Hess, P., MacKenzie, I.A., Marmer,E., Schultz, M.G., Szopa, S., Wild, O., and Zeng, G.: The influence of ozone precursor emissions from four world regions on tropospheric composition and radiative climate forcing, J. Geophys. Res., 117, D07306, doi: 10.1029/2011JD017134, 2012.

Fuglestvedt, J., Berntsen, T., Myhre, G., Rypdal, K., and Skeie, R.B.: Climate forcing from the transport sectors, P. Natl. Acad. Sci. USA, 105, 454–458, doi: 10.1073/pnas.0702958104, 2008.

Gaveau, D.L.A., Salim, M.A., Hergoualc'h, K., Locatelli, B., Sloan, S., Wooster, M., Marlier, M.E., Molidena, E., Yaen, H., DeFries, R., Verchot, L., Murdiyarso, D., Nasi, R., Holmgren, P., and Sheil, D.: Major atmospheric emissions from peat fires in Southeast Asia during non-drought years: Evidence from the 2013 Sumatran fires, Sci. Rep.-U.K., 4, 6112, doi: 10.1038/srep06112, 2014.

Geron, C., Guenther, A., Sharkey, T., and Arnts, R.R.: Temporal variability in basal isoprene emission factor, Tree Physiol., 20, 799–805, 2000.

Geron, C., Owen, S., Guenther, A., Greenberg, J., Rasmussen, R., Bai, J.H., Li, Q.-J., and Baker, B.: Volatile organic compounds from vegetation in southern Yunnan Province, China: Emission rates and some potential regional implications, Atmos. Environ., 40, 1759–1773, doi: 10.1016/j.atmosenv.2005.11.022, 2006.

Gery, M.W., Whitten, G.Z., Killus, J.P., and Dodge, M.C.: A photochemical kinetics mechanism for urban and regional scale computer modeling, J. Geophys. Res.-Atmos., 94, 12,925–12,956, doi: 10.1029/JD094iD10p12925, 1989.

Granier, C., Bessagnet, B., Bond, T., D'Angiola, A., Denier van der Gon, H., Frost, G.J., Heil, A., Kaiser, J.W., Kinne, S., Klimont, Z., Kloster, S., Lamarque, J.-F., Liousse, C., Masui, T., Meleux, F., Mieville, A., Ohara, T., Raut, J.-C., Riahi, K., Schultz, M.G., Smith, S.J., Thompson, A., van Aardenne, J., van der Werf, G.R., and van Vuuren, D.P.: Evolution of anthropogenic and biomass burning emissions of air pollutants at global and regional scales during the 1980–2010 period, Climatic Change, 109, 163–190, doi: 10.1007/s10584-011-0154-1, 2011.

Guenther, A., Hewitt, C.N., Erickson, D., Fall, R., Geron, C., Graedel, T., Harley, P., Klinger, L., Lerdau, M., McKay, W.A., Pierce, T., Scholes, B., Steinbrecher, R., Tallamraju, R., Taylor, J., and Zimmerman, P.: A global model of natural volatile organic compound emissions, J. Geophys. Res.-Atmos., 100, 8873–8892, doi: 10.1029/94JD02950, 1995.

Guenther, A.B., Jiang, X., Heald, C.L., Sakulyanontvittaya, T., Duhl, T., Emmons, L.K., and Wang, X.: The Model of Emissions of Gases and Aerosols from Nature version 2.1 (MEGAN2.1): An extended and updated framework for modeling biogenic emissions, Geosci. Model Dev., 5, 1471–1492, doi: 10.5194/gmd-5-1471-2012, 2012.

Gunarso, P., Hartoyo, M.E., Agus, F., and Killeen, T.J.: Oil palm and land use change in Indonesia, Malaysia, and Papua New Guinea: Reports from the Technical Panels of the 2nd Greenhouse Gas Working Group of the Roundtable on Sustainable Palm Oil (RSPO), Roundtable on Sustainable Palm Oil, Kuala Lumpur, Malaysia, 2013.

Hallquist, M., Stewart, D.J., Stephenson, S.K., and Cox, R.A.: Hydrolysis of $N_2O_5$ on sub-micron sulfate aerosols, Phys. Chem. Chem. Phys., 5, 3453–3463, doi: 10.1039/b301827j, 2003.

Hanson, D. and Mauersberger, K.: Laboratory studies of the nitric acid trihydrate: Implications for the south polar stratosphere, Geophys. Res. Lett., 15, 855–858, doi: 10.1029/GL015i008p00855, 1988.

Harper, K.L., Zheng, Y., and Unger, N.: Advances in representing interactive methane in ModelE2-YIBs (version 1.1), Geosci. Model Dev. Discuss., doi: 10.5194/gmd-2018-85 , 2018.

Heald, C.L. and Geddes, J.A.: The impact of historical land use change from 1850 to 2000 on secondary particulate matter and ozone, Atmos. Chem. Phys., 16, 14997–15010, doi: 10.5194/acp-16-14997-2016, 2016.

Heil, A. and Schultz, M.G.: Interpolated ACCMIP and RCP emission dataset [Data files], Available from: http://accmip-emis.iek.fz-juelich.de/data/accmip/gridded_netcdf/accmip_interpolated/, 2014.

Hewitt, C.N., Lee, J.D., MacKenzie, A.R., Barkley, M.P., Carslaw, N., Carver, G.D., Chappell, N.A., Coe, H., Collier, C., Commane, R., Davies, F., Davison, B., DiCarlo, P., Di Marco, C.F., Dorsey, J.R., Edwards, P.M., Evans, M.J., Fowler, D., Furneaux, K.L., Gallagher, M., Guenther, A., Heard, D.E., Helfter, C., Hopkins, J., Ingham, T., Irwin, M., Jones, C., Karunaharan, A., Langford, B., Lewis, A.C., Lim, S.F., MacDonald, S.M., Mahajan, A.S., Malpass, S., McFiggans, G., Mills, G., Misztal, P., Moller, S., Monks, P.S., Nemitz, E., Nicolas-Perea, V., Oetjen, H., Oram, D.E., Palmer, P.I., Phillips, G.J., Pike, R., Plane, J.M.C., Pugh, T., Pyle, J.A., Reeves, C.E., Robinson, N.H., Stewart, D., Stone, D., Whalley, L.K., and Yang, X.: Overview: oxidant and particle photochemical processes above a south-east Asian tropical rainforest (the OP3 project): Introduction, rationale, location characteristics and tools, Atmos. Chem. Phys., 10, 169–199, doi: 10.5194/acp-10-169-2010, 2010.

Hewitt, C.N., MacKenzie, A.R., Di Carlo, P., Di Marco, C.F., Dorsey, J.R., Evans, M., Fowler, D., Gallagher, M.W., Hopkins, J.R., Jones, C.E., Langford, B., Lee, J.D., Lewis, A.C., Lim, S.F., McQuaid, J., Misztal, P., Moller, S.J., Monks, P.S., Nemitz, E., Oram, D.E., Owen, S.M., Phillips, G.J., Pugh, T.A.M., Pyle, J.A., Reeves, C.E., Ryder, J., Siong, J., Skiba, U., and Stewart, D.J.: Nitrogen management is essential to prevent tropical oil palm plantations from causing ground-level ozone pollution, P. Natl. Acad. Sci. USA, 106, 18447–18451, doi: 10.1073/pnas.0907541106, 2009.

Hollaway, M.J., Arnold, S.R., Collins, W.J., Folberth, G., and Rap, A.: (2017), Sensitivity of midnineteenth century tropospheric ozone to atmospheric chemistry-vegetation interactions, J. Geophys. Res.-Atmos., 122, 2452–2473, doi:10.1002/2016JD025462, 2017.

Hooijer, A., Page, S., Canadell, J.G., Silvius, M., Kwadijk, J., Wösten, H., and Jauhiainen, J.: Current and future CO2 emissions from drained peatlands in Southeast Asia, Biogeosciences, 7, 1505–1514, doi: 10.5194/bg-7-1505-2010, 2010.

Houweling, S., Dentener, F., and Lelieveld, J.: The impact of nonmethane hydrocarbon compounds on tropospheric photochemistry, J. Geophys. Res., 103, 10,673–10,696, doi: 10.1029/97JD03582, 1998.

Hoyle, C.R., Berntsen, T., Myhre, G., and Isaksen, I.S.A.: Secondary organic aerosol in the global aerosol–chemical transport model Oslo CTM2, Atmos. Chem. Phys., 7, 5675–5694, doi: 10.5194/acp-7-5675-2007, 2007.

Indonesian Ministry of Agriculture: Tree Crop Estate Statistics of Indonesia 2015–2017, Directorate General of Estate Crops, Ministry of Agriculture, Indonesia, 2017.

Indrarto, G.B., Murharjanti, P., Khatarina, J., Pulungan, I., Ivalerina, F., Rahman, J., Prana, M.N., Resosudarmo, I.A.P., and Muharrom, E.: The context of REDD+ in Indonesia: Drivers, agents and institutions (Working Paper 92), CIFOR, Bogor, Indonesia, 2012.

Jacob, D.J.: Introduction to Atmospheric Chemistry, Princeton University Press, 1999.

Jardine, A.B., Jardine, K.J., Fuentes, J.D., Martin, S.T., Martins, G., Durgante, F., Carneiro, V., Higuchi, N., Manzi, A.O., and Chambers, J.Q.: Highly reactive light-dependent monoterpenes in the Amazon, Geophys. Res. Lett., 42, 1576–1583, doi: 10.1002/2014GL062573, 2015.

Jardine, K.J., Jardine, A.B., Holm, J.A., Lombardozzi, D.L., Negron-Juarez, R.I., Martin, S.T., Beller, H.R., Gimenez, B.O., Higuchi, N., and Chambers, J.Q.: Monoterpene 'thermometer' of tropical forest-atmosphere response to climate warming, Plant Cell Environ., 40, 441–452, doi: 10.1111/pce.12879, 2017.

Jokinen, T., Berndt, T., Makkonen, R., Kerminen, V.-M., Junninen, H., Paasonen, P., Stratmann, F., Herrmann, H., Guenther, A.B., Worsnop, D.R., Kulmala, M., Ehn, M., and Sipilä, M.: Production of extremely low volatile organic compounds from biogenic emissions: Measured yields and atmospheric implications, P. Natl. Acad. Sci. USA, 112, 7123–7128, doi: 10.1073/pnas.1423977112, 2015.

Kane, S.M., Caloz, F., and Leu, M.-T.: Heterogeneous uptake of gaseous $N_2O_5$ by $(NH_4)_2SO_4$, $NH_4HSO_4$, and $H_2SO_4$ aerosols, J. Phys. Chem. A, 105, 6465–6470, doi: 10.1021/jp010490x , 2001.

Kelly, J.M., Doherty, R.M., O'Connor, F., and Mann, G.W.: The impact of biogenic, anthropogenic, and biomass burning volatile organic compound emissions on regional and seasonal variations in secondary organic aerosol, Atmos. Chem. Phys., 18, 7393–7422, doi: 10.5194/acp-18-7393-2018, 2018.

Klinger, L.F., Li, Q.-J., Guenther, A.B., Greenberg, J.P., Baker, B., and Bai, J.-H.: Assessment of volatile organic compound emissions from ecosystems of China, J. Geophys. Res.-Atmos., 107, 4603, doi: 10.1029/2001JD001076, 2002.

Knote, C., Tuccella, P., Curci, G., Emmons, L., Orlando, J. J., Madronich, S., Baró, R., Jiménez-Guerrero, P., Luecken, D., Hogrefe, C., Forkel, R., Werhahn, J., Hirtl, M., Perez, J. L., San José, R., Giordano, L., Brunner, D., Yahya, K., and Zhang, Y.: Influence of the choice of gas-phase mechanism on predictions of key gaseous pollutants during the AQMEII phase-2 intercomparison, Atmos. Environ., 115, 553–568, doi: 10.1016/j.atmosenv.2014.11.066, 2015.

Koch, D., Schmidt, G.A., and Field, C.V.: Sulfur, sea salt, and radionuclide aerosols in GISS ModelE, J. Geophys. Res., 111, D06206, doi: 10.1029/2004JD005550, 2006.

Koplitz, S.N., Mickley, L.J., Marlier, M.E., Buonocore, J.J., Kim, P.S., Liu, T., Sulprizio, M.P., DeFries, R.S., Jacob, D.J., Schwartz, J., Pongsiri, M., and Myers, S.S.: Public health impacts of the severe haze in Equatorial Asia in September–October 2015: Demonstration of a new framework for informing fire management strategies to reduce downwind smoke exposure, Environ. Res. Lett., 11, 094023, doi: 10.1088/1748-9326/11/9/094023, 2016.

Kotchenruther, R.A., Jaffe, D.A., and Jaeglé, L.: Ozone photochemistry and the role of peroxyacetyl nitrate in the springtime northeastern Pacific troposphere: Results from the Photochemical Ozone Budget of the Eastern North Pacific Atmosphere (PHOBEA) campaign, J. Geophys. Res., 106, 28,731–28,742, doi: 10.1029/2000JD000060, 2001.

Kroll, J.H., Ng, N.L., Murphy, S.M., Flagan, R.C., and Seinfeld, J.H.: Secondary organic aerosol formation from isoprene photooxidation under high-$NO_X$ conditions, Geophys. Res. Lett., 32, L18808, doi: 10.1029/2005GL023637, 2005.

Lacis, A.A., Wuebbles, D.J., and Logan, J.A.: Radiative forcing of climate by changes in the vertical distribution of ozone, J. Geophys. Res., 95, 9971–9981, doi: 10.1029/JD095iD07p09971, 1990.

Lamarque, J.-F., Bond, T.C., Eyring, V., Granier, C., Heil, A., Klimont, Z., Lee, D., Liousse, C., Mieville, A., Owen, B., Schultz, M.G., Shindell, D., Smith, S.J., Stehfest, E., Van Aardenne, J., Cooper, O.R., Kainuma, M., Mahowald, N., McConnell, J.R., Naik, V., Riahi, K., and van Vuuren, D.P.: Historical (1850–2000) gridded anthropogenic and biomass burning emissions of reactive gases and aerosols: methodology and application, Atmos. Chem. Phys., 10, 7017–7039, doi: 10.5194/acp-10-7017-2010, 2010.

Langford, B., Misztal, P.K., Nemitz, E., Davison, B., Helfter, C., Pugh, T.A.M., MacKenzie, A.R., Lim, S.F., and Hewitt, C.N.: Fluxes and concentrations of volatile organic compounds from a South-East Asian tropical rainforest, Atmos. Chem. Phys., 10, 8391–8412, doi: 10.5194/acp-10-8391-2010, 2010.

Lathière, J., Hauglustaine, D.A., Friend, A.D., De Noblet-Ducoudré, N., Viovy, N., and Folberth, G.A.: Impact of climate variability and land use changes on global biogenic volatile organic compound emissions, Atmos. Chem. Phys., 6, 2129–2146, doi: 10.5194/acp-6-2129-2006, 2006.

Malaysian Palm Oil Board (MPOB), Economics and Industry Development Division, Statistics: bepi.mpob.gov.my,  accessed: 25 August 2018.

Margono, B.A., Potapov, P.V., Turubanova, S., Stolle, F., and Hansen, M.C.: Primary forest cover loss in Indonesia over 2000–2012, Nat. Clim. Change, 4, 730–735, doi: 10.1038/nclimate2277, 2014.

McPeters, R., Kroon, M., Labow, G., Brinksma, E., Balis, D., Petropavlovskikh, I., Veefkind, J., Bhartia, P., and Levelt, P.: Validation of the aura ozone monitoring instrument total column ozone product, J. Geophys. Res.-Atmos., 113, D15S14, doi:10.1029/2007JD008802, 2008.

Miettinen, J., Hooijer, A., Vernimmen, R., Liew, S.C., and Page, S.E.: From carbon sink to carbon source: Extensive peat oxidation in insular Southeast Asia since 1990, Environ. Res. Lett., 12, 024014, doi: 10.1088/1748-9326/aa5b6f, 2017.

Misztal, P.K., Nemitz, E., Langford, B., Di Marco, C.F., Phillips, G.J., Hewitt, C.N., MacKenzie, A.R., Owen, S.M., Fowler, D., Heal, M.R., and Cape, J.N.: Direct ecosystem fluxes of volatile organic compounds from oil palms in South-East Asia, Atmos. Chem. Phys., 11, 8995–9017, doi: 10.5194/acp-11-8995-2011, 2011.

Moxim, W.J., Levy II, H., and Kasibhatla, P.S.: Simulated global tropospheric PAN: Its transport and impact on NOx, J. Geophys. Res., 101, 12,621–12,638, doi: 10.1029/96JD00338, 1996.

Myhre, G., Shindell, D., Bréon, F.-M., Collins, W., Fuglestvedt, J., Huang, J., Koch, D., Lamarque, J.-F., Lee, D., Mendoza, B., Nakajima, T., Robock, A., Stephens, G., Takemura, T., and Zhang, H.: Anthropogenic and natural radiative forcing, in: Climate Change 2013: The Physical Science Basis. Contribution of Working Group I to the Fifth Assessment Report of the Intergovernmental Panel on Climate Change, edited by: Stocker, T.F., Qin, D., Plattner, G.-K., Tignor, M., Allen, S.K., Boschung, J., Nauels, A., Xia, Y., Bex, V., and Midgley, P.M., Cambridge University Press, Cambridge, United Kingdom and New York, 2013.

Page, S.E., Siegert, F., Rieley, J.O., Boehm, H.-D.V., Jaya, A., and Limin, S.: The amount of carbon released from peat and forest fires in Indonesia during 1997, Nature, 420, 61–65, doi: 10.1038/nature01131, 2002.

Rap, A., Scott, C.E., Reddington, C.L., Mercado, L., Ellis, R.J., Garraway, S., Evans, M.J., Beerling, D.J., MacKenzie, A.R., Hewitt, C.N., and Spracklen, D.V.: Enhanced global primary production by biogenic aerosol via diffuse radiation fertilization, Nat. Geosci.,  et al., Nat Geoscience, doi: 10.1038/s41561-018-0208-3, 2018.

Schmidt, G.A., Kelley, M., Nazarenko, L., Ruedy, R., Russell, G.L., Aleinov, I., Bauer, M., Bauer, S.E., Bhat, M.K., Bleck, R., Canuto, V., Chen, Y.-H., Cheng, Y., Clune, T.L., Del Genio, A., de Fainchtein, R., Faluvegi, G., Hansen, J.E., Healy, R.J., Kiang, N.Y., Koch, D., Lacis, A.A., LeGrande, A.N., Lerner, J., Lo, K.K., Matthews, E.E., Menon, S., Miller, R.L., Oinas, V., Oloso, A.O., Perlwitz, J.P., Puma, M.J., Putman, W.M., Rind, D., Romanou, A., Sato, M., Shindell, D.T., Sun, S., Syed, R.A., Tausnev, N., Tsigaridis, K., Unger, N., Voulgarakis, A., Yao, M.-S., and Zhang, J.: Configuration and assessment of the GISS ModelE2 contributions to the CMIP5 archive, J. Adv. Model. Earth Sy., 6, 141–184, doi: 10.1002/2013MS000265, 2014.

Schultz, M.G., Heil, A., Hoelzemann, J.J., Spessa, A., Thonicke, K., Goldammer, J.G., Held, A.C., Pereira, J.M.C., and van het Bolscher, M.: Global wildland fire emissions from 1960 to 2000, Global Biogeochem. Cy., 22, GB2002, doi: 10.1029/2007GB003031, 2008.

Scott, C.E., Arnold, S.R., Monks, S.A., Asmi, A., Paasonen, P., and Spracklen, D.V.: Substantial large-scale feedbacks between natural aerosols and climate, Nat Geoscience, 11, 44–48, doi: 10.1038/s41561-017-0020-5, 2018.

Scott, C.E., Monks, S.A., Spracklen, D.V., Arnold, S.R., Forster, P.M., Rap, A., Äijälä, M., Artaxo, P., Carslaw, K.S., Chipperfield, M.P., Ehn, M., Gilardoni, S., Heikkinen, L., Kulmala, M., Petäjä, T., Reddington, C.L.S., Rizzo, L.V., Swietlicki, E., Vignati, E., and Wilson, C.: Impact on short-lived climate forcers increases projected warming due to deforestation,  Nat. Commun., 9:157, 1–9, doi: 10.1038/s41467-017-02412-4, 2018.

Scott, C.E., Monks, S.A., Spracklen, D.V., Arnold, S.R., Forster, P.M., Rap, A., Carslaw, K.S., Chipperfield, M.P., Reddington, C.L.S., and Wilson, C.: Impact on short-lived climate forcers (SLCFs) from a realistic land-use change scenario via changes in biogenic emissions, Faraday Discuss., 200, 101–120, doi: 10.1039/c7fd00028f, 2017.

Shindell, D.T., Faluvegi, G., and Bell, N.: Preindustrial-to-present-day radiative forcing by tropospheric ozone from improved simulations with the GISS chemistry-climate GCM, Atmos. Chem. Phys., 3, 1675–1702, doi: 10.5194/acp-3-1675-2003, 2003.

Shindell, D.T., Faluvegi, G., Unger, N., Aguilar, E., Schmidt, G.A., Koch, D.M., Bauer, S.E., and Miller, R.L.: Simulations of preindustrial, present-day, and 2100 conditions in the NASA GISS composition and climate model G-PUCCINI, Atmos. Chem. Phys., 6, 4427–4459, doi: 10.5194/acp-6-4427-2006, 2006.

Shindell, D.T., Pechony, O., Voulgarakis, A., Faluvegi, G., Nazarenko, L., Lamarque, J.-F., Bowman, K., Milly, G., Kovari, B., Ruedy, R., and Schmidt, G.A.: Interactive ozone and methane

chemistry in GISS-E2 historical and future climate simulations, Atmos. Chem. Phys., 13, 2653–2689, doi: 10.5194/acp-13-2653-2013, 2013.

Silva, S.J., Heald, C.L., Geddes, J.A., Austin, K.G., Kasibhatla, P.S., and Marlier, M.E.: Impacts of current and projected oil palm plantation expansion on air quality over Southeast Asia, Atmos. Chem. Phys., 16, 10621–10635, doi: 10.5194/acp-16-10621-2016, 2016.

Sodhi, N.S., Koh, L.P., Brook, B.W., and Ng, P.K.L.: Southeast Asian biodiversity: An impending disaster, Trends Ecol. Evol., 19, 654–660, doi: 10.1016/j.tree.2004.09.006, 2004.

Stavrakou, T., Müller, J.-F., Bauwens, M., De Smedt, I., Van Roozendael, M., Guenther, A., Wild, M., and Xia, X.: Isoprene emissions over Asia 1979-2012: Impact of climate and land-use changes, Atmos. Chem. Phys., 14, 4587–4605, doi: 10.5194/acp-14-4587-2014, 2014.

Stockwell, W.R., Kirchner, F., Kuhn, M., and Seefeld, S.: A new mechanism for regional atmospheric chemistry modeling, J. Geophys. Res., 102, 25,847–25,879, doi: 10.1029/97JD00849, 1997.

Surratt, J.D., Chan, A.W.H., Eddingsaas, N.C., Chan, M., Loza, C.L., Kwan, A.J., Hersey, S.P., Flagan, R.C., Wennberg, P.O., and Seinfeld, J.H.: Reactive intermediates revealed in secondary organic aerosol formation from isoprene, P. Natl. Acad. Sci. USA, 107, 6640–6645, doi: 10.1073/pnas.0911114107, 2010.

Thompson, A.M., Tao, W.-K., Pickering, K.E., Scala, J.R., and Simpson, J.: Tropical deep convection and ozone formation, B. Am. Meteorol. Soc., 78, 1043–1054, doi: 10.1175/1520-0477(1997)078<1043:TDCAOF>2.0.CO;2, 1997.

Tilmes, S., Lamarque, J.-F., Emmons, L.K., Conley, A., Schultz, M.G., Saunois, M., Thouret, V., Thompson, A.M., Oltmans, S.J., Johnson, B., and Tarasick, D.: Technical Note: Ozonesonde climatology between 1995 and 2011: Description, evaluation and applications, Atmos. Chem. Phys., 12, 7475–7497, doi: 10.5194/acp-12-7475-2012, 2012.

Tsigaridis, K., Daskalakis, N., Kanakidou, M., Adams, P.J., Artaxo, P., Bahadur, R., Balkanski, Y., Bauer, S.E., Bellouin, N., Benedetti, A., Bergman, T., Berntsen, T.K., Beukes, J.P., Bian, H., Carslaw, K.S., Chin, M., Curci, G., Diehl, T., Easter, R.C., Ghan, S.J., Gong, S.L., Hodzic, A., Hoyle, C.R., Iversen, T., Jathar, S., Jimenez, J.L., Kaiser, J.W., Kirkevåg, A., Koch, D., Kokkola, H., Lee, Y.H., Lin, G., Liu, X., Luo, G., Ma, X., Mann, G.W., Mihalopoulos, N., Morcrette, J.-J., Müller, J.-F., Myhre, G., Myriokefalitakis, S., Ng, N.L., O'Donnell, D., Penner, J.E., Pozzoli, L., Pringle, K.J., Russell, L.M., Schulz, M., Sciare, J., Seland, Ø., Shindell, D.T., Sillman, S., Skeie, R. B., Spracklen, D., Stavrakou, T., Steenrod, S.D., Takemura, T., Tiitta, P.,Tilmes, S., Tost, H., van Noije, T., van Zyl, P.G., von Salzen, K., Yu, F., Wang, Z., Wang, Z., Zaveri, R. A., Zhang, H., Zhang, K., Zhang, Q., and Zhang, X.: The AeroCom evaluation and intercomparison of organic aerosol in global models, Atmos. Chem. Phys, 14, 10845–10895, doi: 10.5194/acp-14-10845-2014, 2014.

Tsigaridis, K. and Kanakidou, M.: Secondary organic aerosol importance in the future atmosphere, Atmos. Environ., 41, 4682–4692, doi: 10.1016/j.atmosenv.2007.03.045, 2007.

Turnock, S.T., Wild, O., Dentener, F.J., Davila, Y., Emmons, L.K., Flemming, J., Folberth, G.A., Henze, D.K., Jonson, J.E., Keating, T.J., Kengo, S., Lin, M., Lund, M., Tilmes, S., and O'Connor, F.M.: The impact of future emission policies on tropospheric ozone using a parameterized approach, Atmos. Chem. Phys., 18, 8953–8978, doi: 10.5194/acp-18-8953-2018, 2018.

Unger, N.: Human land-use-driven reduction of forest volatiles cools global climate, Nat. Clim. Change, 4, 907–910, doi: 10.1038/NCLIMATE2347, 2014a.

Unger, N.: On the role of plant volatiles in anthropogenic global climate change, Geophys. Res. Lett., 41, 8563–8569, doi: 10.1002/2014GL061616, 2014b.

Unger, N., Bond, T.C., Wang, J.S., Koch, D.M., Menon, S., Shindell, D.T., and Bauer, S.: Attribution of climate forcing to economic sectors, P. Natl. Acad. Sci. USA, 107, 3382–3387, doi: 10.1073/pnas.0906548107, 2010.

van der Werf, G.R., Morton, D.C., DeFries, R.S., Olivier, J.G., Kasibhatla, P.S., Jackson, R.B., Collatz, G.J., and Randerson, J.T.: $CO_2$ emissions from forest loss, Nat. Geosci., 2, 737–738, doi: 10.1038/ngeo671, 2009.

van der Werf, G.R., Randerson, J.T., Giglio, L., Collatz, G.J., Kasibhatla, P.S., and Arellano Jr., A.F.: Interannual variability in global biomass burning emissions from 1997 to 2004, Atmos. Chem. Phys., 6, 3423–3441, doi: 10.5194/acp-6-3423-2006, 2006.

Vermeulen, S. and Goad, N.: Towards better practice in smallholder palm oil production, Natural Resource Issues Series (No. 5), International Institute for Environment and Development, London, UK, 2006.

Warwick, N.J., Archibald, A.T., Ashworth, K., Dorsey, J., Edwards, P.M., Heard, D.E., Langford, B., Lee, J., Misztal, P.K., Whalley, L.K., and Pyle, J.A.: A global model study of the impact of land-use change in Borneo on atmospheric composition, Atmos. Chem. Phys., 13, 9183–9194, doi: 10.5194/acp-13-9183-2013, 2013.

Wild, O., Fiore, A.M., Shindell, D.T., Doherty, R.M., Collins, W.J., Dentener, F.J., Schultz, M.G., Gong, S., MacKenzie, I.A., Zeng, G., Hess, P., Duncan, B.N., Bergmann, D.J., Szopa, S., Jonson, J.E., Keating, T.J., and Zuber, A.: Modelling future changes in surface ozone: A parameterized approach, Atmos. Chem. Phys., 12, 2037–2054, doi: 10.5194/acp-12-2037-2012, 2012.

Wolfe, G.M., Kaiser, J., Hanisco, T.F., Keutsch, F.N., de Gouw, J.A., Gilman, J.B., Graus, M., Hatch, C.D., Holloway, J., Horowitz, L.W., Lee, B.H., Lerner, B.M., Lopez-Hilifiker, F., Mao, J., Marvin, M.R., Peischl, J., Pollack, I.B., Roberts, J.M., Ryerson, T.B., Thornton, J.A., Veres, P.R.,

and Warneke, C.: Formaldehyde production from isoprene oxidation across NOx regimes, Atmos. Chem. Phys., 16, 2597–2610, doi: 10.5194/acp-16-2597-2016, 2016.

Wong, A.Y.H., Tai, A.P.K., and Ip, Y.-Y.: Attribution and statistical parameterization of the sensitivity of surface ozone to changes in leaf area index based on a chemical transport model, J. Geophys. Res.-Atmos., 123, 1883–1898, doi: 10.1002/2017JD027311, 2018.

WRI: CAIT Climate Data Explorer [Electronic database], World Resources Institute, http://cait.wri.org, 2015.

Yu, F.: A secondary organic aerosol formation model considering successive oxidation aging and kinetic condensation of organic compounds: Global scale implications, Atmos. Chem. Phys., 11, 1083–1099, doi: 10.5194/acp-11-1083-2011, 2011.

Zhang, H., Yee, L.D., Lee, B.H., Curtis, M.P., Worton, D.R., Isaacman-VanWertz, G., Offenberg, J.H., Lewandowski, M., Kleindienst, T.E., Beaver, M.R., Holder, A.L., Lonneman, W.A., Docherty, K.S., Jaoui, M., Pye, H.O.T., Hu, W., Day, D.A., Campuzano-Jost, P., Jimenez, J.L., Guo, H., Weber, R.J., de Gouw, J., Koss, A.R., Edgerton, E.S., Brune, W., Mohr, C., Lopez-Hilfiker, F.D., Lutz, A., Kreisberg, N.M., Spielman, S.R., Hering, S.V., Wilson, K.R., Thornton, J.A., and Goldstein, A.H.: Monoterpenes are the largest source of summertime organic aerosol in the southeastern United States, P. Natl. Acad. Sci. USA, 115, 2038–2043, doi: 10.1073/pnas.1717513115, 2018.

---

## Author Response (AR1)

We thank the reviewers for their helpful comments, which have led us to a substantially improved version of the paper. Here, the reviewers' comments are shown in boldfaced black text, and our responses are shown in non-boldfaced blue text. The page and line numbers to which we refer in our responses correspond to the updated manuscript (the comments of all reviewers are taken into account in this updated manuscript).

First and foremost, we confirm that the tropospheric chemical mechanism in GISS ModelE2 is not CBM04. The original manuscript version contained an incorrect oversimplified description of the tropospheric chemistry scheme in GISS ModelE2 that has caused our Reviewers confusion and understandable concerns. We understand that using an old-fashioned chemical mechanism developed 25 years ago for urban polluted high-NOx environments would be an inappropriate tool to apply to a study of large-scale isoprene emission perturbation in the tropics. The chemical mechanism in GISS ModelE2 has been substantially updated and improved over the past 15 years, for example, to account for important reactions, pathways, and species under low-NOx conditions (e.g. Shindell et al., 2003; 2006; 2013; Schmidt et al., 2014).

We now include a more detailed description of the chemical mechanism in Section 2.1 (ModelE2-YIBs description) (Page 4, Line 32):

"The troposphere features $NO_X$-$O_X$-$HO_X$-CO-$CH_4$ chemistry; an explicit representation of isoprene; and a lumped hydrocarbon scheme involving terpenes, peroxyacyl nitrates (PANs), alkyl nitrates, aldehydes, alkenes, and alkanes. The representation of hydrocarbons generally follows Houweling et al. (1998), which is originally derived from the Carbon Bond Mechanism-4 (Gery et al., 1989) and the Regional Atmospheric Chemistry Model (RACM; Stockwell et al., 1997), but includes several modifications aimed at representing the wide range of chemical conditions found in Earth's atmosphere, such as the addition of reactions important in low-$NO_X$ conditions including representation of organic peroxy radical chemistry under low-$NO_X$ conditions and introduction of organic nitrate chemistry. Shindell et al. (2013) describe in detail the recent updates to the tropospheric chemistry scheme, including the incorporation of acetone chemistry (Houweling et al., 1998) and the addition of terpene oxidation (Tsigaridis and Kanakidou, 2007). SOA formation is driven by $NO_X$-dependent oxidation of emissions of isoprene, monoterpenes, and other reactive VOCs following a volatility-based two-product scheme (Tsigaridis and Kanakidou, 2007). The formation of secondary inorganic aerosols, including sulfate (Bell et al., 2005; Koch et al., 2006) and nitrate (Bauer et al., 2007a), depend on both modeled oxidant levels and the availability of source gases. Primary aerosol types include dust (which provides a surface for heterogeneous chemistry; Bauer and Koch, 2005; Bauer et al., 2007b), black carbon, organic carbon, and sea salt (Koch et al., 2006). Stratospheric chemistry, introduced to the chemical mechanism by Shindell et al. (2006), includes nitrous oxide ($N_2O$) and halogen (bromine and chlorine) chemistry. Recent updates to stratospheric chemistry are summarized by Shindell et al. (2013) and include changes in the representations of polar stratospheric cloud formation (Hanson and Mauersberger, 1988) and heterogeneous hydrolysis of $N_2O_5$ on sulfate (Hallquist et al., 2003; Kane et al., 2001)."

Interdisciplinary work is challenging. We would like to emphasize the novel aspects of this project. (1) The land cover dataset for maritime Southeast Asia that we use in our study is built from an existing classification based on Landsat images (Gunarso et al., 2013). This dataset represents a wall-to-wall mapping of land cover in this region, including explicit representation of plantations of oil palm (high isoprene emitter) and rubber (high monoterpene emitter). Gunarso et al. (2013) used a consistent classification methodology for each year of their analysis, which has provided an internally consistent set of land cover maps for this period for this region. Other studies have investigated the atmospheric composition impacts from land cover change in this region: Ashworth et al. (2012) considered a projection of forest to oil palm conversion based on meeting future demand for biofuels; Warwick et al. (2013) considered the total conversion of Borneo to oil palm from forest; and Silva et al. (2016) considered the impact of 2010 oil palm cover relative to an oil-palm-free landscape in addition to considering future projections of oil palm. We consider the impacts of actual historical land cover change, which is clearly different than Ashworth et al. (2012) and Warwick et al. (2013). The Silva et al. (2016) study imposes oil palm expansion by overlaying an oil palm map for 2010 on a separate 16-PFT land cover distribution; this is a different methodology than we apply here, where we apply an internally consistent set of maps developed using a wall-to-wall classification methodology. (2) We have developed the global climate model code to add four additional land cover type PFTs, focusing on land covers that are pervasive in maritime Southeast Asia, including oil palm and rubber plantations. Previous studies have focused only on the impacts of oil palm expansion. (3) We consider the impacts of land-cover-change-driven changes in emissions of both isoprene and monoterpenes. The study by Silva et al. (2016) presumably includes dynamic changes in monoterpene emissions for the land covers that are displaced by oil palm, but their one new land cover type – oil palm – only has the isoprene emission capacity altered relative to the forest land cover type. Ashworth et al. (2012) and Warwick et al. (2013) consider only isoprene emission changes. (4) We directly quantify the global radiative forcing induced by ozone and SOA changes driven by historical land cover change in this region using a coupled global land-chemistry-climate model framework with the embedded radiative transfer model developed by A. Lacis and J. Hansen in GISS ModelE2 (e.g. Schmidt et al., 2014). (5) We provide new climate policy metrics for global ozone radiative forcing per Mha land cover change in the tropics. (6) We quantitatively identify that important factors driving uncertainty in the forcing include (a) uncertainty in the magnitude of the isoprene BER for oil palm and (b) uncertainty in the areal extent of oil palm expansion. Using an analysis based on fixed SOA yields, we additionally show that the sign of the net forcing is sensitive to uncertainty in the SOA yield from BVOCs.

**Responses to Reviewer #1**

**The authors present the findings of a global modeling study probing the impacts of historical land cover change on the islands of maritime SE Asia with a particular focus on the expansion of oil palm plantations at the expense of natural forest. They apply a chemistry-climate model with interactive land surface to investigate the resulting changes in BVOC emissions and atmospheric composition in the region. In line with previous studies they conclude that changes in surface concentrations of the air pollutants / short-lived climate forcers, ozone and secondary organic aerosols (SOA), are negligible. However, they demonstrate that due to**

strong convection in the tropics, upper tropospheric concentrations are more strongly affected and calculate the radiative forcing associated with these changes, showing that land cover change over this 20-year period in this region may have resulted in local changes in radiative balance.

On the whole this is a carefully implemented study with a reasonable selection of simulations designed to probe some of sensitivities of the model to their assumptions of land cover and vegetation characteristics. Their findings are generally well-presented. There is no doubt that the issue of tropical forest loss and / or degradation is of major global importance with both air quality and climate, and maritime SE Asia is a region which is experiencing rapid and extensive changes in land use.

However, I do have a number of reservations regarding their methodology, some of the assumptions made and the style in which they have presented some of their results. At present I feel that these are of sufficient concern to preclude publication.

Principal among these is the coarse resolution of the model used; a global model at 2x2.5 degree is not sufficiently fine resolution to adequately resolve the complex terrain or the heterogeneity of land cover, emission sources and chemical background conditions. NOx emissions have also rapidly increased in this region and the land cover changes included in this study will also introduce further changes. Given the sensitivity of ozone production and loss and SOA formation to the relative abundance of NOx and VOCs finely resolved spatial distributions are required to correctly diagnose both the direction and the magnitude of the changes in ozone concentration in particular.

My second major concern is the chemistry mechanism included in ModelE2-YIBs which the authors describe as based on CBM-4. CBM-4 was developed in the late 1980s and early 1990s at a time when the atmospheric chemistry community was principally focused on urban air quality and inorganic pollutants. The limited detail that the authors provide here suggests that the mechanism has not been updated to include the recent (i.e. post-2008) improvements in our understanding of isoprene oxidation under conditions of high BVOC and relatively low NOx concentrations, conditions that must apply to large parts of the region under study. The same applies to monoterpene chemistry and the formation of biogenic SOA from both isoprene and monoterpene oxidation products. Without these updates it is hard to have confidence in the modeled changes in atmospheric composition arising from changes in BVOC emissions and concentrations.

Finally, I find that the manuscript is highly skewed to changes in isoprene and ozone, with monoterpene and SOA impacts rarely mentioned in the main text. However the final figures of radiative forcing include the forcing due to changes in SOA. A full discussion of monoterpenes and SOA is therefore needed in the main text.

More detailed comments are given below.

**1. p1 L22 - Could the authors provide a map of the region to indicate what they are describing as "maritime SE Asia" and "the maritime continent"**

We provide Figure S2 (previously known as Figure S1), which shows the applied land cover changes. In Figure 1a, we analyze the surface ozone impacts over a wider region. Additional maps of the region are shown in Figures S3–S6 .

**2. L22 - It would be useful if the authors could provide some sense of scale. What proportion of Indonesia as whole is 4.5Mha? Or perhaps more relevant, what proportion of the natural forest does this represent?**

Good suggestion. We have modified this sentence (addition is bolded; Page 1, Line 24): "More than 4.5 Mha of natural forest were cleared in Indonesia alone over 2000–2010, **which is a loss of 4.6 % of year 2000 Indonesian natural forest cover** (Margono et al., 2014)."

Reference:
Margono, B.A., Potapov, P.V., Turubanova, S., Stolle, F., and Hansen, M.C.: Primary forest cover loss in Indonesia over 2000–2012, Nat. Clim. Change, 4, 730–735, doi: 10.1038/nclimate2277, 2014. (Their Table 1 reports total natural forest area in Indonesia in 2000 as 98.4 Mha.)

**3. L26 - It may have quadrupled but it started from a very low base; this is one of a number of times that the authors have tended toward dramatising the results.**

The reviewer appears to be somewhat missing the point here. Firstly, the areal cover in 1990 is implicit in the sentence: "The amount of land area planted in oil palm in Indonesia and Malaysia nearly quadrupled over 1990–2010, reaching 13 Mha by 2010 (Gunarso et al., 2013)." We modify the sentence to make the areal cover in 1990 now explicit (Page 1, Line 27): "The amount of land area planted in oil palm in Indonesia and Malaysia increased from 3.5 Mha in 1990 to 13 Mha by 2010 (Gunarso et al., 2013)." Secondly, we don't agree that this statement can be "one of a number of times that the authors have tended toward dramatizing the results" because (1) we are not discussing any project results in the Introduction Section and (2) in the Introduction section, we are describing the background motivation for the study as an opportunity for a real world case study during which a large-scale human-induced land cover change happened in the Earth system that has driven a regional-scale increase in isoprene emission.

**4. p2 L7-9 - In fact, Ashworth et al. reported the change in the total tropospheric burden of ozone and SOA before focusing on surface changes and Warwick et al. present altitudinal plots of the changes in some trace gases.**

We have expanded the Introduction Section, including highlighting the important findings of both the Ashworth et al. (2012) and Warwick et al. (2013) studies, in addition to another relevant study (Silva et al., 2016).

The Ashworth et al. (2012) study reports tropospheric burden changes for ozone and OH, but does not report changes for any specific altitude other than the surface. We do not find any mention of non-surface-level changes in SOA in Ashworth et al. (2012). The Warwick et al. (2013) study plots PAN and OH changes as a vertical cross-section at the equator from the surface to 90 hPa and the spatial distribution of PAN changes at 500 hPa; in addition, they report the peak ozone change at 500 hPa over Borneo, but they do not report any other non-surface-level ozone changes, which are particularly important for our study.

We have added:

(1) Page 4, Line 7: "In response to isoprene emission enhancements associated with total conversion of vegetated land to oil palm on the island of Borneo, Warwick et al. (2013) simulate a 20% increase in ozone at 500 hPa over Borneo and a 20% increase in peroxyacetyl nitrate (PAN) at 500 hPa downwind of Borneo over the Pacific Ocean. PAN is an organic nitrate that can undergo long-range transport before releasing its reactive $NO_X$ moiety (Moxim et al., 1996), providing a means for ozone formation in remote environments (Kotchenruther et al., 2001). The results of Warwick et al. (2013) suggest that regional isoprene emission changes have the capacity to alter ozone concentrations in the free troposphere and, therefore, induce a radiative forcing."

(2) Page 3, Line 33: "Ashworth et al. (2012) speculated a small global forcing impact from the increased isoprene emissions in their land-use change scenario, based on the small simulated global changes in the tropospheric burdens of ozone and the hydroxyl radical (OH). However, no study has directly quantified the global radiative impacts associated with the induced changes in atmospheric composition."

**5. L20-21 - This is the first mention of monoterpenes (aside from the abstract). I suggest for the authors also to discuss the atmospheric chemistry and composition effects of monoterpenes as per isoprene in the previous paragraph. For instance the surface flux measurements reported from the OP3 field study (Langford, Misztal) showed that natural forests are much stronger emitters of monoterpenes than oil palm plantations. And the previous investigations also included changes in monoterpene emissions which is not clear in L7-9 as the preceding lines had focused exclusively on isoprene.**

**The changes in monoterpenes and SOA seem to very much be of lesser importance to the authors than changes in isoprene and ozone here and throughout the manuscript. While I accept that the changes are smaller they still contribute to the overall radiative forcing reported by the authors and should be given full coverage in the main text and not just the SI**

The reviewer is correct in that we mainly focus on isoprene emission changes because of the larger change in isoprene (+6.5 TgC $y^{-1}$) relative to the change in monoterpenes (-0.5 TgC $y^{-1}$). Likewise, we deliberately focus on ozone more than SOA because of the stronger simulated radiative forcing from the ozone perturbations. Hence, the paper is not skewed. For example, the paper would be skewed if the primary focus was monoterpenes-SOA.

That said, we agree with the reviewer that the monoterpenes and SOA need to be given appropriate coverage in the main text and not just the SI (additions in bold, deletions crossed out; Page 2, Line 11): "Above-canopy flux measurements **taken in Borneo in 2008** indicate that, compared to the natural forests of maritime Southeast Asia (MSEA), oil palm plantations are much stronger emitters of the biogenic volatile organic compound (BVOC) isoprene ($C_5H_8$), **with mean midday fluxes about five times stronger from oil palm** (Langford et al., 2010; Misztal et al., 2011). The simultaneous large-scale contraction of low-isoprene-emitting natural forest area and expansion of high-isoprene-emitting oil palm plantations suggests a land-cover-change-driven increase in regional isoprene emissions over recent decades (Silva et al., 2016; Stavrakou et al., 2014). **Measurements indicate that the forests of MSEA emit monoterpenes, a class of BVOCs with chemical formula $C_{10}H_{16}$, but find negligible monoterpene emissions from oil palm (Langford et al., 2010; Misztal et al., 2011). Both** isoprene **and monoterpenes are**  precursors to the short-lived climate pollutants tropospheric ozone (Atkinson and Arey, 2003) and secondary organic aerosols (SOA) (Carlton et al., 2009; **Friedman and Farmer, 2018**); **as such, perturbations in regional isoprene and monoterpene emissions serve as an additional mechanism by which regional land cover change can affect air quality and climate**."

Added reference:
Friedman, B. and Farmer, D.K.: SOA and gas phase organic acid yields from the sequential photooxidation of seven monoterpenes, Atmos. Env., 187, 335–345, doi: 10.1016/j.atmosenv.2018.06.003, 2018.

With respect to the previous studies, it is our understanding that: Ashworth et al. (2012) only consider emission changes for isoprene, but do consider the impact that the resulting changes in atmospheric composition have on monoterpene processing; Warwick et al. (2013) likewise consider only isoprene emission changes for forest to oil palm conversion (their paper does not explicitly state how the atmospheric composition changes from the isoprene emission perturbations impact the simulated monoterpene chemistry, although this is presumably taken into account); and Silva et al. (2016) alter only the isoprene emission capacity (and not the monoterpene emission capacity) of their new oil palm land cover type relative to the forest PFT, but their results presumably take into account the effect of monoterpene emission changes associated with the loss of the various land covers to oil palm.

**6. L24 - Does this mean that the authors have only conducted a series of atmosphere-only model simulations? So there are no climate / land surface feedbacks included on-line?**

Atmosphere-only run is a standard technical term widely used by the World Climate Research Program (WCRP) Coupled Model Intercomparison Project (CMIP). It means that the global climate model uses prescribed observed sea surface temperatures and sea ice fields. Thus, the climate model does not have a fully coupled dynamic ocean. The term is in common usage in the global climate modeling community. Atmosphere-only simulations can be dynamically coupled to atmospheric chemistry and the land surface, as in our work.

**7. L24-27 - Actually I am now confused as to exactly what model simulations were performed. The authors have referred to atmosphere-only, chemistry-climate, and land surface models here. Exactly what configuration is being used?**

The reviewer is unfamiliar with standard terminology in the global climate and atmospheric chemistry modeling communities. See response to Point (6). Atmosphere-only means prescribed sea surface temperatures and sea ice fields. Atmosphere-only simulations can be dynamically coupled to atmospheric chemistry and the land surface, as in our work. Our description of the model set-up and configuration is clear, complete, and appropriate. No further changes are needed here.

**8. L27 - 2 deg x 2.5 deg is too coarse to adequately resolve the complexity and heterogeneity of the land mass and land cover in this region particularly given the sensitivity of BVOC oxidation and ozone production rate to VOC:NOx ratio.**

The reviewer is raising a longstanding query that concerns the entire large-scale global climate and chemistry mathematical modeling communities, way beyond the scope of this study, regarding what is actually needed in a global model (with associated limited computational resources) to reproduce large-scale composition impacts versus a highly localized mathematical representation of every real process on the ground tending to continuous resolution, many of which do not actually matter to the global radiative impact of ozone and SOA. This conflict commonly emerges between communities engaged in large-scale mathematical modeling versus communities engaged in local ecosystem-scale measurements. Nobody is ever surprised when it comes up in interdisciplinary work.

The reviewer's comment applies to the use of all global chemistry-climate (CCM) and global chemistry-transport (CTM) models for the study of the global radiative impacts of regional-scale changes in short-lived emission precursors. The "complexity and heterogeneity of the land mass and land cover and the sensitivity of BVOC oxidation and ozone production rate to VOC:NOx ratio" are NOT issues unique to the MSEA region. Undeniably, these issues are important in all chemical regimes and regions of the world where the large-scale atmospheric responses to short-lived emission precursor perturbations are being studied. State-of-the-science global CCMs and CTMs typically have horizontal resolution 1-2° latitude/longitude. The model horizontal and vertical resolution applied in this study is comparable to global CCMs and CTMs currently being used in the WCRP CMIP6 Aerchem-MIP and RF-MIP in support of the

forthcoming IPCC AR6 assessment, and the Task Force on Hemispheric Transport of Air Pollutants (HTAP). These international assessment programs each employ dozens of global models with 1-2° latitude/longitude resolution to quantify the impacts of local and regional short-lived precursor emission changes, including VOCs, in very different regions and regimes. It is a moot point that global CCMs and CTMs do not fully resolve the complexity and heterogeneity of land mass and land cover and other sub-grid phenomena. The models parameterize these conditions and processes.

By the reviewer's own logic, thousands of peer-reviewed publications in high-caliber journals, HTAP, Aerchem-MIP and RF-MIP, and short-lived climate forcers in IPCC AR6/AR5/AR4 are invalid. We do not agree. The goals of this work are to quantify the global radiative forcing of ozone and SOA changes due to the isoprene emission injection and altered BVOC fluxes as a result of recent human-induced land cover change in MSEA. Therefore, we have used a model framework that has been designed to simulate the global radiative forcing impacts from local and regional short-lived emission precursor perturbations, including, but not limited to, assessments by HTAP, Aerchem-MIP, RF-MIP, and a wide range of other international multi-model assessment programs over the past 20 years.

**9. L31 - I have reservations whether the chemistry mechanism employed here is suitable for the conditions encountered in this region. Although it is rapidly developing with the concomitant increases in anthropogenic emissions, much of island areas of the region are still low-NOx, high-VOC regimes. Older chemical mechanisms were designed for the typical chemistry encountered in mid- to high-NOx urban / industrial areas and considerable understanding has been gained of the very different oxidation pathways of (particularly) isoprene under lower NOx conditions. Have any of these updates been included in the chemistry here?**

Please see comment at the top of this document. In the original manuscript version, we neglected to provide a detailed enough description of the current chemical mechanism. Certainly, we too would have major reservations about a study using CBM04 to quantify composition impacts of a large isoprene emission injection in the tropics. The revised manuscript now includes a more detailed and accurate description of the chemical mechanism.

**10. p3 L8 - Are the monoterpenes emitted as a single lumped monoterpene species? Or at least in part speciated (e.g to specifically include a-pinene, b-pinene, d-limonene and others as is often done)? It should be noted that the monoterpene emissions algorithms included in Lathiere et al are in fact the Guenther et al. algorithms from 1995; at the very least this paper should be referenced here. Also, these algorithms assume that monoterpene emissions are entirely temperature dependent whereas more recent field measurements have shown that many species emit a proportion of monoterpenes directly (i.e. monoterpene emissions exhibit both light and temperature dependency, see e.g. Steinbrecher et al, Guenther et al. 2012). Are the authors confident that this is not the case for SE Asian plant species?**

We have added the Guenther et al. (1995) reference and a brief discussion of light-dependency. ModelE2 applies a lumped monoterpene species (Page 6, Line 28): "Temperature-dependent leaf-level monoterpene emissions, functionally $\alpha$-pinene, likewise vary by ecosystem type, similarly through prescription of PFT-specific basal emission rates (Guenther et al., 1995; Lathière et al., 2006). Recent work suggests that tropical monoterpene emissions exhibit both light and temperature dependency (Guenther et al., 2012; Jardine et al., 2015, 2017) that is not included in the emission algorithm here but may be explored in future work."

Added references:

Guenther, A., Hewitt, C.N., Erickson, D., Fall, R., Geron, C., Graedel, T., Harley, P., Klinger, L., Lerdau, M., McKay, W.A., Pierce, T., Scholes, B., Steinbrecher, R., Tallamraju, R., Taylor, J., and Zimmerman, P.: A global model of natural volatile organic compound emissions, J. Geophys. Res.-Atmos., 100, 8873–8892, doi: 10.1029/94JD02950, 1995.

Jardine, A.B., Jardine, K.J., Fuentes, J.D., Martin, S.T., Martins, G., Durgante, F., Carneiro, V., Higuchi, N., Manzi, A.O., and Chambers, J.Q.: Highly reactive light-dependent monoterpenes in the Amazon, Geophys. Res. Lett., 42, 1576–1583, doi: 10.1002/2014GL062573, 2015.

Jardine, K.J., Jardine, A.B., Holm, J.A., Lombardozzi, D.L., Negron-Juarez, R.I., Martin, S.T., Beller, H.R., Gimenez, B.O., Higuchi, N., and Chambers, J.Q.: Monoterpene 'thermometer' of tropical forest-atmosphere response to climate warming, Plant Cell Environ., 40, 441–452, doi: 10.1111/pce.12879, 2017.

**11. L15-19 - How were emission factors assigned to these additional land cover types? How do they differ from the standard land cover in this region in the default land surface map? Again this is critical to the results and should be included in the main text and not just the SI.**

On (Page 6, Line 22), we state: "For each PFT, the isoprene emission rate depends linearly on the fraction of electrons available to undergo isoprene synthesis, **the calculation of which requires prescription of a PFT-specific leaf-level isoprene basal emission rate (BER) at standard conditions.**"

We have moved the Table of model parameters (previously known as Table S2 in the Supplement) to the main text (now known as Table 1).

On Page 9, Line 20, we removed the list of references from this sentence (as this information is found in the footnotes of Table 1, now in the main text) and re-phrased to better describe what is found in the table: "Table 1 shows, for the new land cover types, the assigned physical parameters (including LAI and vegetation height), photosynthetic parameters, and leaf-level basal emission rates of isoprene and monoterpenes."

We describe the relationship between the isoprene BERs for the standard and new rainforest PFTs where this information is critical (Page 13, Line 10): "In this sensitivity analysis, the dipterocarp forest **isoprene** BER is increased by a factor of 12, **making it equivalent to the isoprene BER assigned to the standard evergreen broadleaf forest PFT in YIBs**."

**12. L31-32 - Of real importance to this study is the previous performance of the YIBs model in this region; the 2013 study was global. Do the referenced comparisons include field sites in maritime SE Asia?**

The 2013 global evaluation paper did include time-varying OP3 Borneo field measurements (e.g. Langford et al., 2010). The point of referencing the global-scale evaluation against a wide range of different ecosystems and regions is to demonstrate that the model has reasonable isoprene emission performance over a range of different ecosystems and regions. The present manuscript provides a comparison of oil palm and forest isoprene fluxes to those from the OP3 campaign in Borneo (Page 22, Line 8; Page 22, Line 25).

**13.  p4 L14-15 - It seems odd to go the lengths of using 30m x 30m resolution land cover data in a model running at 2 deg/ x 2.5 deg**

The 30 m x 30 m resolution land cover dataset (i.e., the dataset of Gunarso et al., 2013) is re-gridded to the model resolution of 2° latitude x 2.5° longitude before application to the model (Page 8, Line 27). The purpose of applying the high-resolution dataset is because, as far as we know, this is the only land cover dataset available for this region that employs a wall-to-wall classification methodology that provides multiple years of data (using a consistent classification methodology for each year) and includes several land covers that are prevalent in Southeast Asia (e.g., oil palm and rubber plantations). The availability of such a dataset prevents us from needing to build a single dataset out of multiple datasets that were potentially derived using different classification methodologies or are from different time periods.

**14. L20-26 - Please clarify. Are you saying that in 1990 the only land cover data available shows natural forest (or whatever) in locations that were shown as oil palm in 2000 and 2010? What fraction of data is missing? Of the 1990 data what fraction is converted to oil palm by 2000 and 2010? Of the missing data what fraction is "converted" to oil palm by 2000 and 2010? It would be useful to have a feel for how substantial the "likely underestimation" might be.**

We have clarified the language (modifications bolded, Page 8, Line 19): "The Gunarso et al. (2013) classification for 1990 for Indonesia is likewise incomplete; in this dataset, the pixels classified for 1990 are principally those that are **oil palm in 1990 or eventually become oil palm**."

Thus (Page 8, Line 23): "Indonesian oil palm cover in 1990 is accurate within the limits of the classification methodology."

The rest of Indonesia is largely unclassified in 1990, as described, which is why we apply the year 2000 land cover to these pixels.

**15. L30 - Is dirt equivalent to the "bare ground" classification included in other land surface schemes?**

Fixed. We have amended the text to state "bare ground" rather than "dirt."

**16. p5 L3 - The authors say that these are "minor land cover types". Again it would be useful to be provided with sufficient information to judge just how minor. What fraction of land is included in these 5 types in Gunsaro et al classification?**

These minor land cover types account for 2.8% of pixels in both 1990 and 2010.

**17. L8-13 - Again it would be good to have an idea of the likely underestimation, and this should be relatively easy for the authors to achieve by applying an LAI reduction, as described, to the areas classified as "disturbed" in Gunsaro.**

For the forest class, around 43% is disturbed, while around 57% is undisturbed (these fractions are largely consistent across years). Dietz et al. (2007) report LAIs for various disturbance levels: undisturbed (6.2 $m^2$ $m^{-2}$), removal of small-diameter trees (5.3 $m^2$ $m^{-2}$), and removal of large-diameter trees (5.0 $m^2$ $m^{-2}$). Based on these LAIs, the mean LAI for the forest class (47% disturbed, 57% undisturbed) would be: (1) 0.57 x 6.2 $m^2$ $m^{-2}$ + 0.43 x 5.3 $m^2$ $m^{-2}$ = 5.8 $m^2$ $m^{-2}$ (assuming that the disturbed forest falls closer to the small-diameter removal category) and (2) 0.57 x 6.2 $m^2$ $m^{-2}$ + 0.43 x 5 $m^2$ $m^{-2}$ = 5.7 $m^2$ $m^{-2}$ (assuming that the disturbed forest falls closer to the large-diameter removal category). In our simulations, we assign a forest LAI of 6.0 $m^2$ $m^{-2}$, based on measurement of a natural forest plot in Malaysian Borneo (Fowler et al., 2011), which is an area included in our land cover change analysis. Thus, our assigned value is only about 3–5% higher than these rough estimates, which is a good approximation considering that we do not have any information about the level of disturbance of the "disturbed" forest patches in the land cover change dataset that we apply (that is, the classification of Gunarso et al. (2013)). In the Gunarso et al. (2013) classification, a "disturbed" forest patch has a reduced basal area with evidence of clearing or logging. Such classifications are not uncommon; for example, Margono et al., Nature Climate Change, 2014, use a "primary degraded forest" class, in which the forest has been fragmented or experienced selective logging or other disturbance.

**18. L21 - As noted above, Table S2 should be in the main text as these parameter values are critical to the results. The notes regarding their derivation can be left in SI. The values for the "standard" PFTs in YIBs for this region should also be shown in this table for comparison.**

We have moved this table and the footnotes, which are an integral part of the table, to the main text (now known as Table 1).

**19. L27-34 - There is a real mishmash of years for the various datasets. As the simulations are being conducted for a nominal 2010 (i.e. that is the climatology) with 1990 or 2010 SE Asia landcover why introduce further limitations / discrepancies by using Y2000 landcover for the rest of the world with vegetation characteristics derived using Y2000 meteorology only to change to 2010?**

We assume that the global radiative forcing impacts by ozone and SOA due to oil palm expansion in MSEA are insensitive to changes in the background land cover state outside of the MSEA region over the 1990–2010 period. Unfortunately, we do not always have available all observational or modeled data for each year for each boundary condition for model runs, which means that we sometimes need to combine datasets in appropriate ways to run simulations. Here, we use a dataset for 2000, which falls within the era of interest (1990–2010). We use the year 2000 rest-of-world land cover dataset specifically because we already had available the PFT-specific vegetation height parameters for the set of PFTs used in ModelE2-YIBs at the resolution used in ModelE2-YIBs. As we describe in the paper, we obtained the PFT-specific height parameters applied to the rest of the world vegetation from an existing 140-year simulation run with our model. This 140-year simulation was run using dynamic carbon allocation and applied the same rest-of-world land cover distribution that we apply here. Using this configuration, a 140-year simulation requires at least a few months of run time, which accounts only for the actual time that the model is integrating and does not include time spent in the simulation queue between re-submissions (since our cluster allows only one week of run time before the simulation must be re-submitted, the additional time spent in the run queue can be substantial). With unlimited computational resources, we could run an additional century-long simulation for year 1990 or 2010, but we don't have access to these resources, and, more importantly, it is unlikely that switching to 1990 and 2010 background land cover datasets has any meaningful influence on the results here. The benefit of doing so is not clear because we hold static the rest-of-world land cover map and physical vegetation characteristics (because we need to isolate the impacts of MSEA regional land cover change) and this dataset is a reasonable approximation of land cover and vegetation characteristics for this 1990–2010 era.

**20. p6 L33-34 - Could the authors please clarify how the simulations were driven with the meteorology? Was the same climate / meteorology applied for 13 years? Because there will be an effect of inter-annual variability on emissions, chemistry and therefore O3 and SOA formation; how has this been accounted for? Is this what the authors have attempted to do via the additional simulations?**

We have removed the incorrect description of the nudged winds. The quantified standard deviations (e.g., radiative forcing in Table 5 on Page 21) are based on internal interannual variability in the climate model. We have additionally performed a sensitivity simulation to assess the impact of using a different background climate (including emissions year) on the forcing results (Table 5 on Page 21).

**21. p7 L4-6 - This is the case for all current isoprene emissions models which are linked to PAR, T, CO2, soil moisture, etc. Please clarify what aspect the authors mean is the case "because" it is interactively linked OR remove the word "because"**

To improve clarity, we have re-phrased this (additions bolded, deletions crossed out; Page 12, Line 14): " Isoprene production in ModelE2-YIBs is  **calculated as a semi-mechanistic function of** photosynthetic carbon assimilation (Unger et al., 2013). Isoprene emissions are sensitive to simulated changes in the parameters that affect photosynthesis, including the background climate state (**e.g., temperature, PAR, and soil moisture**) and **the** atmospheric $CO_2$ concentration."

**22. L6-7 - Please give more detail of how monoterpene emissions are sensitive to climate as per isoprene**

We have changed "climate" to "temperature" (Page 12, Line 17): "Simulated monoterpene emissions are likewise sensitive to temperature shifts (Lathière et al., 2006)."

**23. L9-11 - But changing the landcover will also affect e.g. NOx emissions, either due to changes in fertiliser application or to changes in natural soil emissions. How have the authors accounted for this?**

**Because again the resolution of the global model will not be sufficient to pick up changes such as this simply by running a sensitivity test with a different background atmosphere.**

The reviewer raises a good point. We account for anthropogenic changes in NOx emissions and all other short-lived emission precursors by applying the MACCity inventory for anthropogenic emissions of carbonaceous aerosols and reactive gases (Granier et al., 2011). The MACCity inventory is partially based on the ACCMIP inventory, which is based on a multitude of global- and regional-scale emission inventories (Lamarque et al., 2010). MACCity includes agricultural NOx emissions. Climate-sensitive lightning NOx and soil NOx emissions are included in the simulations. Atmospheric NOx measurements in the region are extremely limited. A possible future work direction beyond the scope here is to exploit satellite NOx data to learn more about the NOx levels in the region.

**24. L28-30 - Why have the authors not used the measured emission rate in the first instance?**

Our model requires a leaf-level BER (YIBs has its own canopy up-scaling scheme consistent for carbon, water, energy, and BVOCs). Therefore, we adopted a strategy that maximizes use of the

limited available BVOC flux data in the region. First, we implemented published leaf-level isoprene BERs to all PFTs including oil palm (Table 1; now in the main text). Then, we used the raw measured fluxes (not canopy-level BERs) from the OP3 campaign to evaluate and validate the model's simulated above-canopy fluxes (Page 22, Line 8; Page 22, Line 25). This strategy is more physically realistic, and provides a better benchmark than, for example, artificially forcing the model to reproduce the OP3 canopy-scale BERs as a boundary condition.

**25. L30-32 - Why? The authors specifically introduced this land class because measurements had shown that the global emission factors were not suitable. The work reported in Langford and Misztal suggested that emission factors were out by a factor of 3 so using 12 seems rather extreme.**

The rationale is simply a sensitivity study to examine the impacts when the forest emits with a default isoprene BER for tropical rainforest, that is where the factor of 12 comes from. See response to point (11).

**26. p8 Table 2 - this seems to imply that the isoprene emission factor applied to oil palm is as measured in the standard run but half measured in the OPber sensitivity tests which appears to contradict what the authors have described in the previous page.**

The values reflect the difference between the canopy-scale and leaf-level BERs for oil palm isoprene emission.

**27. p9 L3-4 - It would also be good to see how well 1990land_base and 1990land_1990atm GPP compare with measured GPP**

We amended the text to state (Page 14, Line 9): "Simulated global gross primary production (GPP) for 2010 is 124 PgC y$^{-1}$ (simulation 2010land_base), which almost precisely matches an estimate derived from flux-tower measurements that is representative of 1998–2005: 123 ± 8 PgC y$^{-1}$ (mean ± 1 standard deviation; Beer et al., 2010). The simulated 1990 global GPP of 108 PgC y$^{-1}$ (simulation 1990land_1990atm) is outside of the 1-standard-deviation range of the observation-based mean, but falls within the 95% confidence interval (102–135 PgC y$^{-1}$; Beer et al., 2010)."

The small change in GPP from 1990–2010 land cover change (2010land_base minus 1990land_base) is described later in the text, so providing the value for 1990land_base here would be repetitive.

**28. L5 - please define contemporary, because Table 5 in the Guenther paper contains estimates from early 90s to around 2008. Again it would be useful to see the emissions estimates for both 1990 and 2010 land cover and climates here.**

The estimates from Guenther et al. (2012) Table 5 are from references that were published over the period 1995–2011. The table does not indicate the year represented by each estimate. However, the forcing datasets (e.g., those for "weather" and "LAI") listed in the table for each of the estimates suggest that the estimates are from the contemporary (modern day) period as opposed to future projections or the pre-industrial era. We use these estimates to show that the global emissions of isoprene and monoterpenes that are simulated in our study are reasonable. We also compare our 2010 isoprene estimate to another estimate representative of the 1971–2000 mean.

We have expanded the text to include our 1990 estimates (Page 14, Line 15): "The model estimates for 1990 (325 TgC $y^{-1}$ isoprene and 90 TgC $y^{-1}$ monoterpenes for simulation 1990land_1990atm) and 2010 (363 TgC $y^{-1}$ isoprene and 77 TgC $y^{-1}$ monoterpenes for simulation 2010land_base) fall within these ranges. Using the same process-based, leaf-level isoprene production algorithm employed in ModelE2-YIBs, although driven with different forcing datasets, Hantson et al. (2017) predict contemporary isoprene emissions (385 TgC $y^{-1}$; 1971–2000 mean) that are 18% higher than those predicted here for 1990 and only 6% higher than those predicted here for 2010."

**29. L10-19 - Here and throughout, although the authors describe this as a study of how BVOC emissions changes have affected the region the manuscript is entirely dominated by consideration of isoprene. While this is understandable given that total regional isoprene emissions are more than 5x those of monoterpenes I think the paper would benefit from more consideration of monoterpene emissions and impacts as monoterpenes and isoprene have different effects on atmospheric composition and chemistry. I suggest the authors also pay careful consideration to their use of the catch-all BVOC as this study appears only to include isoprene and monoterpenes.**

See response to Point (5). We will continue with the use of "BVOC" to describe isoprene and monoterpenes. There is a growing body of literature on the impacts of isoprene and monoterpenes on regional and global radiation budgets and short-lived climate forcers (e.g., Heald and Geddes, 2016; Hollaway et al., 2017; Scott et al., 2017, 2018; Unger, 2014a,b). To our knowledge, there is no current published evidence that any other BVOC species have statistically significant large-scale global and regional radiative effects. Because of their extremely short-lifetimes, it is likely that other highly reactive emitted compounds have much more localized impacts.

The reviewer's comments would be more relevant to a surface air quality study rather than a study focused on global radiative forcing. The global mean annual average radiative forcing metric is used because it is a linear predictor of global mean surface air temperature response at steady state. Therefore, we focus on annual average analyses in this study (e.g. IPCC AR5, Myhre et al., 2013). Global and regional radiative forcing effects of perturbations to short-lived precursor emissions are typically reported on an annual-mean basis. The paper is already getting rather too long and therefore we do not include seasonal surface changes in the manuscript.

The PhD thesis "Forcings and feedbacks in the climate system: The role of reactive compounds in the atmosphere, Yale University, K. L. Harper, 2018" reports seasonal changes: "Surface ozone reductions are simulated over Peninsular Malaysia, Sumatra, and Borneo in all seasons for ΔLC (Figure 3.12). The changes in circulation and precipitation associated with the boreal winter (DJF) and boreal summer (JJA) monsoons are the likely sources of the small seasonal variations in the distribution of the surface ozone changes. The location of peak ozone enhancement over the marine environment shifts from west of Sumatra (in DJF and MAM) to north of Borneo (in JJA). Negligible changes are simulated over New Guinea in all seasons."

[Figure]

**Figure 3.12.** Changes in seasonal-average surface ozone mixing ratios (ppbv) for ΔLC.

**38. L16-19 - What is the resolution of the NOx emissions input? As previously noted I do have concerns over the capability of the model to resolve the heterogeneity of this region and Hewitt et al. 2010 demonstrated the sensitivity of the atmospheric chemistry in this region to NOx levels over a range of BVOC emissions**

The emissions input resolution corresponds to the global model resolution. See response to Point (23). Hewitt et al. (2010) used a box model. We have reservations about using a box model to simulate regional ozone air quality changes. Box models are appropriate tools to use to understand reactive radicals unaffected by transport processes and can be assumed to be in steady state in the atmosphere. Ozone has a relatively long lifetime and is strongly determined by transport and physical processes in the atmosphere. Using a box model designed to understand radical reaction pathways and kinetics to project changes in regional ozone surface air quality is just plain wrong. One may obtain some insights into key reaction pathways and important chemical species, but the projected changes in ozone concentrations aren't particularly useful in the absence of atmospheric physics and transport.

**39. L28 - Please re-phrase; "inflated" sounds as if the authors applied an arbitrary increase whereas in both cases the scenarios in which NOx emissions were increased were based on the differences observed between forest and plantation.**

We have re-phrased this sentence (Page 16, Line 16): "Both studies found that increasing the $NO_X$ emissions in the region of land conversion (to account for enhanced fertilizer application and industrial processing of the oil palm) enhanced surface ozone concentrations (Ashworth et al., 2012; Warwick et al., 2013)."

Our extended introduction (which is detailed in our response to comment 1 for reviewer #3) also mentions the reason for the enhanced $NO_X$ emissions in these studies.

As a point of clarification, the increased $NO_X$ emissions in the Warwick et al. (2013) and Ashworth et al. (2012) studies were not entirely based on the observed differences between forest and plantation. In both studies, they increase $NO_X$ emissions to represent increased fertilizer application of the oil palm (it appears that this is based on the observations), but they also include emissions based on increased industrial processing and, in the case of the Warwick et al. (2013) study, transportation. The transportation and processing emissions are based on estimates of energy requirements for these activities. In the Warwick et al. (2013) study, in the simulation where they apply enhanced $NO_X$ emissions in the oil palm landscape, the applied $NO_X$ emissions are more than 3.5x those inferred by Hewitt et al. (2009) for the oil palm landscape (0.07 mg(N) $m^{-2}$ $h^{-1}$ in their simulation vs. 0.019 mg(N) $m^{-2}$ $h^{-1}$ from the Hewitt et al. (2009) study). Hewitt et al. (2009) inferred fluxes for the forest landscape of 0.009 mg(N) $m^{-2}$ $h^{-1}$, indicating that the $NO_X$ fluxes were only about 2x as high for oil palm relative to forest. Warwick et al. (2013) apply a factor of 7 increase in $NO_X$ emissions relative to their baseline case.

**40. L34 - A likely key difference between the work of Silva and that presented here is that Silva applied the GEOS-Chem model at a resolution of 0.5deg x 0.667deg, a far more appropriate resolution for this highly complex region.**

Perhaps more appropriate if the goal is to quantify regional surface air quality impacts associated with regional oil palm expansion. Again, that is not our goal here. Our study is not a regional air quality study, rather, we quantify the global radiative perturbation associated with atmospheric composition changes. As such, we apply a model with the typical resolution used by IPCC CMIP6 and HTAP for studying the global radiative impacts of regional perturbations to the short-lived precursor emissions. See response to point (8). The reviewer may consider that simply increasing horizontal resolution without changing the model's sub-grid parameterizations, processes and mechanisms does not imply an improvement in simulation accuracy. The reviewer seems to assume an automatic increase in accuracy. It depends on the linearity of the processes and impacts involved. For example, in the NOx-limited regime, ozone production has a linear dependence on NOx concentrations (Introduction to Atmospheric Chemistry, Daniel J. Jacob). Therefore, the coarse resolution grid is simply an average of the higher resolution version. Certainly, increased resolution does give more output information because the grid cell numbers have increased and that is important for regional air quality applications. GEOS-Chem is an excellent model to study regional air quality and large-scale composition changes at all the horizontal resolutions at which is it available.

Please see response to point (5) above. In the revised manuscript we discuss the monoterpenes and SOA more upfront in the manuscript.

We discuss the land-cover-change-driven monoterpene emissions changes in Sect. 3.1, and these are plotted in Figure S3. Regional changes in surface SOA are plotted in Figure S6.

We now state the change in the global SOA burden (Page 20, Line 32): "The global ozone perturbation induced a positive forcing of +9.2 ± 0.7 mW m$^{-2}$, offset only slightly by a negative forcing (-0.8 ± 0.1 mW m$^{-2}$) induced by a 1.4% enhancement (+6.5 Gg) in the global burden of largely reflective SOA particles. (The regional change in SOA is plotted in Figure S6.)"

The simulated global annual-mean burden of biogenic SOA is 0.46 Tg in the 2010 base simulation (2010land_base). A recent study using the UKCA model calculates the annual-mean SOA burden, considering isoprene and monoterpene precursors, as 0.41 Tg (Kelly et al., ACP 2018). In the MSEA region (here, the region shown in Figure 1a), the maximum surface SOA concentration in 4.1 µg m$^{-3}$, occurring over central Sumatra, with most grid cells showing concentrations of < 2 µg m$^{-3}$. Previous global model simulations have reported SOA concentrations of the same order of magnitude in this region (Hoyle et al., 2007; Yu, 2011). Yu (2011) simulated regional SOA concentrations of < 2 µg m$^{-3}$, similar to the results of the 2010land_base simulation.

"Small" and "large" are to some extent value judgments and not purely objective. It is our job to use mathematical modeling to provide quantitative values for Earth system and global change processes involving the short-lived climate forcers. 9 mW m$^{-2}$ or 37 mW m$^{-2}$ is small and even negligible compared to > 1800 mW m$^{-2}$ CO$_2$ global forcing. Is 9 (4–16) mW m$^{-2}$ from a regional BVOC injection due to recent human-induced land cover change in the tropics "small" compared to 30 mW m$^{-2}$ due to all anthropogenic VOC increases since the preindustrial; or 50 mW m$^{-2}$ due to global road transportation emissions? Social scientists are better equipped to answer this question. We offer a perspective on the sensitivity of the tropical atmosphere to human land cover change.

**Responses to Reviewer #2**

Harper and Unger present a study of the radiative forcing brought about via differences in isoprene emission under different land use configurations in the maritime Southeast Asia (MSEA) region. These land use changes comprise the move towards more oil palm plantations, which emit more biogenic volatile organic compounds than the native natural forests. The changes in isoprene emitted to the atmosphere as a result of the increased oil palm leads to changes in ozone. Of particular interest is that the Enhanced BVOCs caused bigger changes globally to ozone in the upper troposphere (0.6 ppb) than lower troposphere (>0.1 ppb), which would seem an important result. The novelty of this study is that the authors then go on to calculate the radiative forcing expected by these ozone changes, finding a small increase of +1 mW m-2 Mha-1.

This shows that impacts of land use changes in tropical regions, which are subject to stronger convective patterns that elsewhere, are very important.

My feeling is that this is a really nice idea, but the wrong tool has been used to carry out the study. The small changes in ozone seen at the top of the troposphere are probably lost in the noise of uncertainty of the chemical scheme chosen, and thus I question the impacts on radiative forcing.

The authors use the carbon bond 4 chemical mechanism to represent the oxidation of isoprene in the atmosphere. This scheme is very old and does not include some of the recent discoveries brought about via questioning the discrepancies between isoprene predicted by models, and observed mixing ratios. These particularly relate to additional OH recycling, which directly impact the influence of isoprene on O3 (eg Lelieveld et al., 2008; Peeters et al., 2009).

The authors do mention the uncertainty in the isoprene chemistry regarding increased oxidant cycling, right at the end of the paper in the conclusions, but I think there are other problems with this choice of chemistry scheme. High isoprene atmospheres, such as that found in this MSEA region, have caused more differences in chemical mechanisms than most others. Unfortunately, the carbon bond scheme has never fared well when tested alongside other chemistry schemes under similar isoprene rich atmospheres. I wonder why there has been no model development in the chemistry scheme in this work when the science behind this paper depends so highly upon it?

For example Knote et al (2015) tested two variants of the newer carbon bond 5 (CB05) scheme (neither of which contained updates to the isoprene chemistry) and found they "tended to be biased low in O3 under low NOx/high VOC conditions (e.g. biogenic emissions rich) as well as under very high NOx conditions. In general, the CB05 schemes produced 'lower than average 8 hourly O3' produced by other schemes. Mechanisms were 'found to differ more strongly in their predictions of O3 levels and other pollutants in regions with strong biogenic VOC emissions".

Archibald et al (2010) tested 8 chemical schemes in isoprene rich regions and found that the CB05 mechanism was 'unable to generate/recycle HOx at the rates needed to match recently reported observations at locations characterized by low levels of NOx.'

An older study - Emmerson and Evans (2009) tested the carbon bond 4 scheme against 6 other schemes. However the carbon bond 4 results disagreed with the majority of the other schemes, in even the sign of the changes in ozone (e.g. loss instead of production - see figure 3 panel e). Differences (and thus uncertainty) of 14 ppb were found between the resulting ozone from the Master Chemical Mechanism and the carbon bond 4 scheme, which is 14 times more than the ~1 ppb of ozone changes found in Harper and Unger's study at the top of the troposphere, and upon which the radiative forcing calculations are based.

Thus I don't agree with the authors' comment that no updates to the chemistry have occurred because of "its apparent inconsequence to the surface pollution impacts of regional land cover change". I think if a different chemistry scheme had been implemented that the changes in ozone found by Harper and Unger as a result of including more oil palm plantations in the model would lead to more significant differences in the radiative forcing than found by their study.

I'd recommend updating the chemistry scheme. Perhaps even to include a sensitivity study with a more up to date representation of just the isoprene chemistry – particularly one that agrees with the sign of ozone changes driven by our current understanding. The chemical aspect of Harper and Unger's work is my only criticism, which if rectified I would then recommend publication in ACP.

Thank you for the thoughtful comments and guidance. We confirm that the tropospheric chemical mechanism in GISS ModelE2 is not CBM04. Please see response at the top of this document at response to Reviewer #1 point (9). Certainly, we too would have major concerns about a study using CBM04 to quantify composition impacts of a large isoprene emission injection in the tropics. The revised manuscript now includes a more detailed and accurate description of the chemical mechanism.

We have removed this sentence: "its apparent inconsequence to the surface pollution impacts of regional land cover change." We now provide a more balanced assessment of uncertainties due to isoprene oxidation chemistry, which we have moved to the methodology (Sect. 2.1) as advised in point (9) below. In our expanded assessment, we have included reference to the studies noted above (Archibald et al., 2010; Emmerson and Evans (2009); Knote et al. (2015)).

**General comments**

(1) A map figure would be good, showing the study area with the areal extent of regions growing oil palm in 1990 and where/how these regions have increased by 2010.

The original version of the manuscript includes Figure S1 in the Supplementary Information; this figure is now labeled Figure S2 in the updated manuscript. This figure shows, for each of eight land cover types (including oil palm), the regional change in land cover for (1) 1990–2005 and (2) 1990–2010. In the manuscript, we refer to this figure on Page 11, Line 16.

We have included a new figure (known as Figure S1 in the updated manuscript) that shows the areal extent of these same eight land cover types in year 1990. We have added the following sentence to the manuscript to point readers to this figure (Page 16, Line 14): "Figure S1 in the Supplement shows the regional land cover distribution for 1990."

**(2) Page 2 line 2. "Compared to natural forests oil palm plantations are much stronger emitters of BVOCs" Some numbers would be good here. How much stronger?**

Updated text (additions in bold; Page 2, Line 11): "Above-canopy flux measurements **taken in Borneo in 2008** indicate that, compared to the natural forests of maritime Southeast Asia (MSEA), oil palm plantations are much stronger emitters of the biogenic volatile organic compound (BVOC) isoprene ($C_5H_8$), **with mean midday fluxes about five times stronger from oil palm** (Langford et al., 2010; Misztal et al., 2011)."

The factor difference may be even stronger if comparing the canopy-level BERs, but the values of the BERs can depend on the model parameterization applied (e.g., Misztal et al., 2011), so we use the above comparison instead.

**(3) Page 2 line 20. Try placing the (Baker et al., 2005; Klinger et al., 2002) references at the end of sentence to avoid breaking the flow of the sentence up too much.**

Fixed. (We introduce the high-monoterpene-emitting capacity of rubber trees earlier in the revised introduction.)

**(4) Page 2 line 29. How is photolysis treated in the model?**

We have provided an expanded description of photolysis in the manuscript (Page 5, Line 18): "Photolysis rate calculations follow the Fast-J2 scheme of Bian and Prather (2002). At each 30 minute time step, the simulated distributions of clouds, ozone, and aerosols are passed to the photolysis code, providing a mechanism for simulated changes in aerosols to impact atmospheric chemistry through modification of photolysis rates (Bian et al., 2003)."

We have added these references:

Bian, H. and Prather, M.J.: Fast-J2: Accurate simulation of stratospheric photolysis in global chemical models, J. Atmos. Chem., 41, 281–296, doi: 10.1023/A:1014980619462, 2002.

Bian, H., Prather, M.J., and Takemura, T.: Tropospheric aerosol impacts on trace gas budgets through photolysis, J. Geophys. Res., 108, 4242, doi: 10.1029/2002JD002743, 2003.

**(5) Page 3 line 27. 'the' calculation**

Fixed.

**(6) Page 5 line 12. It is not clear where this LAI dataset has come from?**

For the four new land cover types that have been added to ModelE2-YIBs for this study, the assigned LAI values are from published literature and are shown in Table 1 in the main text (this table was previously Table S2 in the Supplement), with the references noted in the footnotes of that table. To better point readers to this information, we have added the phrase (Page 9, Line 20) "including LAI and vegetation height" to the sentence in the manuscript that describes the information that can be found in Table 1; in this sentence, we additionally replace the phrase "BVOC emissions" with the more specific phrase "leaf-level basal emission rates for isoprene and monoterpenes." (The assigned LAI applied to other vegetation is described in the following paragraph.)

**(7) Page 5 line 14 (onwards in this paragraph). LAI has units of m2 m-2**

Fixed in all three instances in this paragraph (Page 9, Line 11): "An analysis of the leaf area index (LAI) of rainforest plots in Central Sulawesi, Indonesia, under different land use regimes found that disturbance of the forest by selective logging reduced the LAI below the 6.2 $m^2$ [leaf] $m^{-2}$ [ground] value measured for the undisturbed natural forest (Dietz et al., 2007). Removal of "small-diameter" trees reduced LAI to 5.3 $m^2$ $m^{-2}$, while removal of "large-diameter" trees reduced LAI to 5.0 $m^2$ $m^{-2}$ (Dietz et al., 2007)."

We also added the units to two places in the new Table 1 (previously Table S2 in the Supplement) – once in the table and once in the footnote.

**(8) Page 5 line 21. Table S2 – mention this is in the supplementary section.**

We have moved this table and its footnotes to the main text (now known as Table 1).

**(9) Page 19 line 21. This whole discussion of uncertainties in the chemistry scheme would be better placed in section 2.1 which introduces the method used.**

Fixed. We expanded this discussion and moved it to Section 2.1.

**(8) P3 L4-6: "Interactive" is a modeler's jargon, and even for modelers can mean different things for different purposes. I recommend avoiding it and state more clearly that these emission schemes are "semi-empirical", "mechanistic" or "dynamic functions of x, y, z, …", especially for those that are not described more below.**

**(9) P3 L13: Avoid the use of "online".**

**(10) P3 L28: Avoid "online" and "model's".**

Fixed for all instances where these terms were used in the manuscript.

**(11) P3 L33: Please explain and justify the single chemical representation of monoterpene. Can all monoterpenes really be modeled as $\alpha$-pinene?**

The quantification of SOA yields for different monoterpene species is an exciting frontier for SOA modeling, but the uncertainties remaining in this field currently preclude expansion of this chemistry at the expense of the greater computational resources needed to run global model simulations. We have added a discussion on this matter (Page 7, Line 33): "Furthermore, the appropriateness of using $\alpha$-pinene to represent monoterpenes as a single lumped species in global modeling is an active area of research. Friedman and Farmer (2018) find order-of-magnitude differences in SOA yields for OH-oxidation of different monoterpene species, but a clear explanation based on isomer structure remains largely elusive. While Friedman and Farmer (2018) find that the magnitude of the SOA yield from $\alpha$-pinene is in the "mid-range" of the yields among the analyzed monoterpene species, other studies have shown that this may not be the case for other oxidation pathways (e.g., Draper et al., 2015)."

Added references:

Draper, D.C., Farmer, D.K., Desyaterik, Y., and Fry, J.L.: A qualitative comparison of secondary organic aerosol yields and composition from ozonolysis of monoterpenes at varying

concentrations of NO2, Atmos. Chem. Phys., 15, 12267–12281, doi: 10.5194/acp-15-12267-2015, 2015.

Friedman, B. and Farmer, D.K.: SOA and gas phase organic acid yields from the sequential photooxidation of seven monoterpenes, Atmos. Env., 187, 335–345, 2018.

**(12) P5 L30: What about the LAI values for the new PFTs used for this study for MSEA? They are not described above. Are dynamic but grid-level LAI observed from, e.g., MODIS, used, or are PFT-level LAI values used for these new PFTs? If so, where are these values from? As LAI is so important for atmospheric chemistry, these need to be better stated and explained.**

The PFT-level LAI values applied to each of the new PFTs are described in Table 1 and its footnotes in the main text (this table was previously known as Table S2 in the Supplement). We have updated the text to better draw attention to this information (Page 9, Line 20): "Table 1 shows, for the new land cover types, the assigned physical parameters (including LAI and vegetation height), photosynthetic parameters, and leaf-level basal emission rates of isoprene and monoterpenes.."

**(13) P10 L1: In the methodology section above, the authors have only discussed about model validity and model-observation comparison for the vegetation aspects (e.g., GPP, biogenic emissions). What about an evaluation of the ozone simulations by the model? How does the model's simulated ozone globally compare with observations and with other models? Is the general high biases of simulated ozone in many climate-chemistry models also seen in this model? Since ozone concentration is crucial to this paper, I strongly recommend having a paragraph somewhere (preferably in the methodology section) discussing these.**

We have inserted in the methodology section the following paragraph describing the extensive evaluation of ozone in ModelE2-YIBs (Page 7, Line 7): "ModelE2 has previously undergone extensive, rigorous validation of simulated present-day tropospheric and stratospheric chemical composition, circulation, and ozone forcing using multiple observational datasets (Shindell et al., 2006; Shindell et al., 2013). Shindell et al. (2013) compared simulated monthly zonal-mean total column ozone to that from observations (2000–2010 means) from the Total Ozone Mapping Spectrometer and the Ozone Monitoring Instrument (McPeters et al., 2008), finding: simulated zonal-mean total column ozone in the tropics shows little bias (< 5%) against measurements for each month, and, in the Northern Hemisphere middle and high latitudes, biases are smaller in the summer months (< 10%) than in the winter months (around 15–20%). Shindell et al. (2013) find only a small negative bias (-0.016 W m$^{-2}$) in the present-day global-average radiative impact of modeled tropospheric ozone relative to TES-derived tropospheric ozone. They note that the strongest biases in ozone concentrations in ModelE2 are generally located in regions where ozone exhibits little effect on radiation (Shindell et al., 2013). More recently, Harper et al. (2018) compared annual-mean ozone concentrations simulated by

ModelE2 (representative of year 2005) with an ozonesonde climatology based on measurements taken over 1995–2011 (Tilmes et al., 2012), finding lower model biases at higher pressures (e.g., +2.6% at 200 hPa compared to +16.9% at 800 hPa)."

Added references:

Harper, K.L., Zheng, Y., and Unger, N.: Advances in representing interactive methane in ModelE2-YIBs (version 1.1), Geosci. Model Dev. Discuss., doi: 10.5194/gmd-2018-85 , 2018.

McPeters, R., Kroon, M., Labow, G., Brinksma, E., Balis, D., Petropavlovskikh, I., Veefkind, J., Bhartia, P., and Levelt, P.: Validation of the aura ozone monitoring instrument total column ozone product, J. Geophys. Res.-Atmos., 113, D15S14, doi:10.1029/2007JD008802, 2008.

Shindell, D.T., Pechony, O., Voulgarakis, A., Faluvegi, G., Nazarenko, L., Lamarque, J.-F., Bowman, K., Milly, G., Kovari, B., Ruedy, R., and Schmidt, G.A.: Interactive ozone and methane chemistry in GISS-E2 historical and future climate simulations, Atmos. Chem. Phys., 13, 2653–2689, doi: 10.5194/acp-13-2653-2013, 2013.

Tilmes, S., Lamarque, J.-F., Emmons, L.K., Conley, A., Schultz, M.G., Saunois, M., Thouret, V., Thompson, A.M., Oltmans, S.J., Johnson, B., and Tarasick, D.: Technical Note: Ozonesonde climatology between 1995 and 2011: Description, evaluation and applications, Atmos. Chem. Phys., 12, 7475–7497, doi: 10.5194/acp-12-7475-2012, 2012.

**(14) P10 L14: Wong et al. (2018) also examined and quantified the factors behind the sensitivity of surface ozone to vegetation changes including isoprene emission and dry deposition. They also found a large impact of background NOx. See reference list below.**

Please see response to point (15).

**(15) P11 L1: Dry deposition definitely also plays a role, and have you quantified the relative importance of isoprene emission vs. dry deposition to surface ozone in your model simulations? This appears to be a major missing part of this analysis and should be better addressed or discussed, even if the authors have already found that dry deposition plays only a minor role. For instance, Wong et al. (2018) found it necessary and developed a method to formally disentangle the contributions from isoprene emission and dry deposition when leaf density changes.**

The Wong et al. (2018) paper presents an exciting avenue for disentangling the various land cover change driven contributions to surface ozone changes; however, it is our understanding that their analysis does not take into account the effects of other biogeophysical changes, such as surface roughness and evapotranspiration. The land cover distribution changes in our model

can alter such (and other) parameters, which might also be playing a role in the simulated surface ozone changes.

We have expanded our discussion on this matter (Page 16, Line 25): "The simulated changes in atmospheric composition might be a response not only to altered isoprene and monoterpene emissions, but also to changes in the deposition of atmospheric species induced by changes in leaf density (Wong et al., 2018) or related changes, such as surface roughness, stomatal conductance, and evapotranspiration, that are affected by the applied changes in land cover distribution. Here, the relative changes in regional ozone deposition rates (-19.7 to +4.3%) are similar to the relative changes in regional surface-level ozone concentrations (-18.3 to +4.3%) from 1990–2010 regional land cover change, in part because the ozone deposition rate depends on the atmospheric concentration change. While increased isoprene emission leading to increased isoprene ozonolysis drives ozone losses near the surface, a formal quantitative attribution analysis disentangling the relative roles of emission and deposition changes requires further complex sensitivity simulations that are beyond the scope of this analysis. In their analysis of Southeast Asian oil palm expansion, Silva et al. (2016) used sensitivity studies to determine that the induced BVOC emission changes, rather than altered deposition rates from LAI changes, were almost exclusively responsible for the simulated surface ozone changes."

Added reference:

Wong, A.Y.H., Tai, A.P.K., and Ip, Y.-Y.: Attribution and statistical parameterization of the sensitivity of surface ozone to changes in leaf area index based on a chemical transport model, J. Geophys. Res.-Atmos., 123, 1883–1898, doi: 10.1002/2017JD027311, 2018.

**(16) P12 L2: The physical reasons for the enhancements (as opposed to reductions) of ozone over the ocean have to be explained. Can these enhancements be explained by, e.g., the mechanisms suggested by Hollaway et al. (2017)? A discussion in relation to this paper is recommended. See reference list below.**

Please see response to comment (18).

**(17) P12 L16: In Fig. 2a) and 2c), why is there a second peak for isoprene and HCHO enhancement near the tropopause?**

This is a signal of tropical deep convection (e.g., deep convective towers rapidly pulling up air into the upper troposphere).

**(18) P14 L5: Now I see that the oceanic enhancements are explained. But this explanation, with reference to Hollaway et al. (2017), should be mentioned early (see comment to P12 L2).**

We added reference to the Hollaway et al. (2017) paper and moved this explanation (now at Page 18, Line 1).

**(19) P17 L8-9: "This sensitivity study demonstrates that the climate forcing associated with regional land cover change is rapidly increasing." I feel that this is too strong a statement. All the results are showing is that 2005-2010 as a 5-year period is responsible for a noticeably large fraction of the total RF compared to other possible 5-year periods, but without breaking down the other years into incremental 5-year periods (e.g., 1990-1995, 1996-2000, 2001-2005), we can't really say there is a rapidly rising trend in RF.**

Agreed. We have removed this sentence.

**(20) P18 L5-7: "increase in regional surface ozone concentrations is unlikely to have a significant impact on the induced ozone forcing since, as Lacis et al. (1990) find, changes in surface ozone have a much smaller effect on climate forcing relative to equivalent ozone changes in the upper troposphere." This is contingent upon the assumption that the formation and long-range transport of isoprene nitrate will respond in the same way even as the surface environment becomes more high-NOx. This needs to be justified.**

We have removed this badly phrased sentence. Please see response to Reviewer (1) Point (66). We were originally trying to emphasize that increases in ozone near the Earth's surface do not exert appreciable longwave forcing, but we agree the original sentence does not read well and is not scientifically nuanced enough as is. Agreed regarding the assumption that the formation and long-range transport of isoprene nitrate would have to respond in the same way even as the surface environment becomes more high-NOx. To be clear, in this model, convective transport is moving isoprene, its oxidation products, and isoprene nitrate into the upper troposphere.

**(21) P18 L24-31: I think one major missing discussion is to compare the ozone-mediated RF with the biogeophysical RF (e.g., changing albedo, latent heat, sensible heat, etc.) and biogeochemical (CO2 exchange) associated with oil palm expansion. Indeed, most climatologists are still just concerned with the biogeophysical or biogeochemical RF, and having a comparison between those and the ozone-mediated forcing would give much insight into the importance of considering atmospheric chemistry in climate/land use change studies.**

Thank you for the brilliant idea, we agree with the reviewer, and these discussions did come up a few times in the project. Unfortunately, we are unable to access the albedo surface forcing diagnostics in the model output. We delved into LLGHG emission estimates for the recent land cover change in MSEA. Carlson et al. (2012b) estimated 11.4 MtC $y^{-1}$ $CO_2$ emissions from 1989–2008 (mostly from deforestation fire). This amounts to a cumulative total of 216.6 MtC $y^{-1}$ over

the 19 years. Using IPCC AR5 GWPs, the $CO_2$ forcing is: 5.3 mW m$^{-2}$ on 20-yr time scale and 19.9 mW m$^{-2}$ on 100-yr time scale. However, the small region analysed by Carlson et al. (2012b) doesn't correspond to that applied here so is not a meaningful comparison. LLGHG land cover change emissions accounting is beyond the scope here, and would be a full paper in its own right. Interestingly, there is a substantial high impact published literature on future climate change impacts on the sustainability of oil palm plantations in MSEA (the opposite way around to that considered here).

Reference:

Carlson, K.M., Curran, L.M., Ratnasari, D., Pittman, A.M., Soares-Filho, B.S., Asner, G.P., Trigg, S.N., Gaveau, D.A., Lawrence, D., and Rodrigues, H.O.: Committed carbon emissions, deforestation, and community land conversion from oil palm plantations expansion in West Kalimantan, Indonesia, Proc. Natl. Acad. Sci.-USA, 109, 7559–7564, doi: 10.1073/pnas.1200452109, 2012b.

**(22) P19 L18-19: "Inclusion of a temporally variable BVOC BER in the global model would allow for an improved estimation of radiative forcing induced by land cover changes in this region." I think the current debate is exactly that we are not sure about the circadian control or not, and thus this statement is not necessarily true.**

We agree that this aspect of isoprene emissions is very uncertain and have removed this sentence to avoid confusion.

**(23) P19 L33-34: "(2) its apparent inconsequence to the surface pollution impacts of regional land cover change" Is there really no OH titration problem in MSEA in ModelE2- YIBs? Is that because the BER is low to begin with, compared to, say, the Amazon?**

In ModelE2-YIBs, OH is typically much lower than the global average in forested tropical regions in the model. We have removed this poorly phrased sentence and moved the entire discussion of chemistry uncertainties to the Methodology section (Sect. 2.1, Page 7, Line 21), where we have included a more balanced perspective on isoprene photooxidation uncertainties in global models (e.g., see introductory remarks to reviewer #2).

10  composition changes and, for the first time, the associated radiative forcing induced by the land-cover-change-driven biogenic volatile organic compound (BVOC) emission changes (+6.5 TgC $y^{-1}$ isoprene, -0.5 TgC $y^{-1}$ monoterpenes). Regionally, surface-level ozone concentrations largely decreased (-3.8 to +0.8 ppbv). The tropical land cover changes occurred in a region of strong convective transport, providing a mechanism for the BVOC perturbations to affect the composition of the upper troposphere. Enhanced concentrations of isoprene and its degradation products are simulated in the
15  upper troposphere, and, on a global-mean basis, land cover change had a stronger impact on ozone in the upper troposphere (+0.5 ppbv) than in the lower troposphere (< 0.1 ppbv increase). The positive climate forcing from ozone changes (+9.2 mW $m^{-2}$) was partially offset by a negative forcing (-0.8 mW $m^{-2}$) associated with an enhancement in secondary organic aerosol (SOA). The sign of the net forcing is sensitive to uncertainty in the SOA yield from BVOCs. The global-mean ozone forcing per unit of regional oil palm expansion is +1 mW $m^{-2}$ $Mha^{-1}$. In light of expected continued expansion of oil palm
20  plantations, regional land cover changes may play an increasingly important role in driving future global ozone radiative forcing.

**1 Introduction**

Recent decades have witnessed large-scale land cover and land use changes on the maritime continent. More than 4.5 Mha of natural forest were cleared in Indonesia alone over 2000–2010, which is a loss of 4.6% of year 2000 Indonesian natural
25  forest cover (Margono et al., 2014). Increasing demand for palm oil, produced by oil palm trees (*Elaeis guineensis*), has simultaneously driven widespread expansion of this agro-industrial tree crop (USDA, 2010). Indonesia and Malaysia cumulatively produce 85% of the current global palm oil supply (USDA, 2017). The amount of land area planted in oil palm in Indonesia and Malaysia increased from 3.5 Mha in 1990 to 13 Mha by 2010 (Gunarso et al., 2013).

Land cover and land use changes in Southeast Asia perturb the Earth system in a variety of ways. Deforestation is a significant threat to Southeast Asian biodiversity (Sodhi et al., 2004), and the land-based carbon emissions associated with forest clearing have greatly contributed to Indonesia's status as one of the world's highest emitters of greenhouse gases (GHGs) (FAO, 2014; WRI, 2015). The magnitude of GHG emissions from deforestation is exacerbated by the pervasiveness of carbon-rich peat soils underlying Southeast Asian tropical forests (Carlson et al., 2012; Hooijer et al., 2010; Page et al., 2002; van der Werf et al., 2009). Peat soil drainage (i.e., drying) promotes oxidation of the sequestered carbon (Miettinen et al., 2017) and is often followed by fire clearing, despite the illegality of this practice in Indonesia (Indrarto et al., 2012). Indonesian forest and peat fires have fueled transnational air pollution episodes (Gaveau et al., 2014; Koplitz et al., 2016), potentially causing more than 100,000 premature mortalities in 2015 (Koplitz et al., 2016).

Above-canopy flux measurements taken in Borneo indicate that, compared to the natural forests of maritime Southeast Asia (MSEA), oil palm plantations are much stronger emitters of the biogenic volatile organic compound (BVOC) isoprene ($C_5H_8$), with mean midday fluxes about five times stronger from oil palm (Langford et al., 2010; Misztal et al., 2011). The simultaneous large-scale contraction of low-isoprene-emitting natural forest area and expansion of high-isoprene-emitting oil palm plantations suggests a land-cover-change-driven increase in regional isoprene emissions over recent decades (Silva et al., 2016; Stavrakou et al., 2014). Measurements indicate that the forests of MSEA emit monoterpenes, a class of BVOCs with chemical formula $C_{10}H_{16}$, but find negligible monoterpene emissions from oil palm (Langford et al., 2010; Misztal et al., 2011). Both isoprene and monoterpenes are precursors to the short-lived climate pollutants tropospheric ozone (Atkinson and Arey, 2003) and secondary organic aerosols (SOA) (Carlton et al., 2009; Friedman and Farmer, 2018); as such, perturbations in regional isoprene and monoterpene emissions serve as an additional mechanism by which regional land cover change can affect air quality and climate.

A few studies have used global modeling to investigate the atmospheric composition impacts of Southeast Asian oil palm expansion. Ashworth et al. (2012) analyzed the expected impacts from isoprene emission enhancements associated with the partial replacement of natural forest area with oil palm plantations under a land-use change scenario designed to meet a portion of the projected increase in demand for biofuels in coming years. Warwick et al. (2013) analyzed the impacts from isoprene emission enhancements associated with total conversion of Borneo from forest to oil palm. Both studies quantified the impacts of the isoprene emission changes first by applying a contemporary $NO_X$ inventory and secondly by assuming increased $NO_X$ emissions near the site of land-use change due to enhanced fertilizer application and increased on-site processing of the palm oil. Based on simulations that apply contemporary $NO_X$ inventories, both studies predict reductions in surface ozone co-located with the isoprene enhancements because the increased VOC serves as a net sink for ozone in the low-$NO_X$ atmosphere (Ashworth et al., 2012; Warwick et al., 2013). When $NO_X$ emission enhancements occur in concert with the land-use-change-driven isoprene emission enhancements, both studies simulate a net increase in ozone production and a concomitant increase in local surface ozone pollution (Ashworth et al., 2012; Warwick et al., 2013). Ashworth et al.

Harper, Kandice 8/23/2018 1:05 PM

Harper, Kandice 8/23/2018 1:05 PM

(2012) predict local enhancements in annual-mean surface concentrations as high as 11% for ozone and 10% for biogenic SOA (as the maximum change in any grid cell) in response to a 5% increase in regional isoprene emissions and simultaneous enhancements in local $NO_X$ emissions. Warwick et al. (2013) predict local changes in monthly-mean surface ozone as strong +70% when simultaneously increasing $NO_X$ and isoprene emissions. These studies highlight the potential for significant local surface-level pollution impacts from land use change.

A recent third study quantified the air quality impacts associated with year 2010 oil palm cover relative to an oil-palm-free scenario (Silva et al., 2016). Using the GEOS-Chem chemical transport model and contemporary inventories for $NO_X$ emissions, Silva et al. (2016) simulate local enhancements in surface pollution as high as 26% (3–4 ppbv) for ozone and 60% (about 1 $\mu g\ m^{-3}$) for SOA. In Kuala Lumpur, Malaysia, the number of days that register ozone levels higher than the limits recommended by the World Health Organization more than doubled due to regional oil palm expansion (56 days based on 2010 oil palm coverage compared to 23 days in the absence of oil palm; Silva et al., 2016), again highlighting the strong impact of Southeast Asian land cover change on surface pollution.

The year 2000 16-plant functional type (PFT) land cover distribution map that Silva et al. (2016) apply as the no-oil-palm case is designed for global modeling studies and, therefore, lacks information about the distribution of individual species of vegetation. For example, 7.2 Mha of oil palm existed in 2000 in Indonesia, Malaysia, and Papua New Guinea (Gunarso et al., 2013), yet many global vegetation distributions assign these plantations to one or more of a small number of PFTs. Silva et al. (2016) use a 250 m resolution satellite-based map of year 2010 oil palm plantations to define the contemporary oil palm distribution, and they overlay the oil palm on the palm-free base map, displacing the existing land covers in proportion to their fractional distributions. In reality, the prior land cover of oil palm plantations differs widely by region (Gunarso et al., 2013). Furthermore, the modified land cover distribution from Silva et al. (2016) lacks separate delineations for other pervasive land covers in Southeast Asia, including rubber plantations, which covered at least 6.4 Mha on the maritime continent in 2010 (Gunarso et al., 2013). Rubber trees (*Hevea brasiliensis*) are very weak emitters of isoprene (Geron et al., 2006; Klinger et al., 2002), but very strong emitters of monoterpenes (Baker et al., 2005; Klinger et al., 2002), which are important SOA precursors (Jokinen et al., 2015). Thus, while the study by Silva et al. (2016) provides evidence of significant surface pollution changes induced by oil palm expansion in Southeast Asia, it provides an incomplete picture of the impact of historical land cover change on atmospheric composition.

This small set of studies focuses on the atmospheric composition impacts induced by altered BVOC emissions from Southeast Asian oil palm expansion, all finding that the downwind impacts are smaller in magnitude than the local impacts near the site of land conversion and isoprene emission changes (Ashworth et al., 2012; Silva et al., 2016; Warwick et al., 2013). Ashworth et al. (2012) speculated a small global forcing impact from the increased isoprene emissions in their land-use change scenario, based on the small simulated global changes in the tropospheric burdens of ozone and the hydroxyl

radical (OH). However, no study has directly quantified the global radiative impacts associated with the induced changes in atmospheric composition.

Isoprene perturbations in the tropics may have a particularly powerful impact on longwave radiative forcing (Unger, 2014) because the strong vertical mixing prevalent in the tropics provides a mechanism for surface pollution perturbations to impact ozone concentrations in the upper troposphere (Thompson et al., 1997), where, on a per-molecule basis, ozone changes induce the strongest climate impact due to the thermal contrast with the surface (Lacis et al., 1990). In response to isoprene emission enhancements associated with total conversion of vegetated land to oil palm on the island of Borneo, Warwick et al. (2013) simulate a 20% increase in ozone at 500 hPa over Borneo and a 20% increase in peroxyacetyl nitrate (PAN) at 500 hPa downwind of Borneo over the Pacific Ocean. PAN is an organic nitrate that can undergo long-range transport before releasing its reactive $NO_X$ moiety (Moxim et al., 1996), providing a means for ozone formation in remote environments (Kotchenruther et al., 2001). The results of Warwick et al. (2013) suggest that regional isoprene emission changes have the capacity to alter ozone concentrations in the free troposphere and, therefore, induce a radiative forcing.

This study uses the ModelE2-Yale Interactive Terrestrial Biosphere (ModelE2-YIBs) global chemistry–climate model, in conjunction with multiple observational datasets, to quantify the global atmospheric composition changes and, for the first time, the concomitant radiative forcings associated with BVOC emission changes from 1990–2010 land cover change in MSEA. The calculations presented here consider changes in emissions of both isoprene and monoterpenes. The applied regional land cover changes are derived from a Landsat-based classification (Gunarso et al., 2013) and account for changes in eight land covers that are prevalent in MSEA, including high-monoterpene-emitting rubber trees and high-isoprene-emitting oil palm trees.

**2 Data and methods**

**2.1 ModelE2-YIBs description**

Atmosphere-only simulations employ the NASA GISS ModelE2-YIBs global chemistry–climate model. The YIBs model (Yue and Unger, 2015) is a land surface model embedded in the NASA GISS ModelE2 global chemistry–climate model (Schmidt et al., 2014). The model features 2°-latitude × 2.5°-longitude horizontal resolution, 40 vertical layers (surface to 0.1 hPa), and a physical and chemical time step of 30 minutes.

The chemical mechanism includes 156 reactions involving 51 chemical species with full coupling of tropospheric and stratospheric chemistry (Schmidt et al., 2014; Shindell et al., 2006). The troposphere features $NO_X$-$O_X$-$HO_X$-CO-$CH_4$
* * *
**Comments / Deletions (margin):**

Harper, Kandice 8/23/2018 12:51 PM

Harper, Kandice 8/23/2018 6:14 PM

Harper, Kandice 8/26/2018 10:34 AM

Harper, Kandice 8/26/2018 10:34 AM

Harper, Kandice 8/23/2018 12:59 PM

Harper, Kandice 8/24/2018 2:17 PM

Harper, Kandice 8/23/2018 1:00 PM

chemistry; an explicit representation of isoprene; and a lumped hydrocarbon scheme involving terpenes, peroxyacyl nitrates (PANs), alkyl nitrates, aldehydes, alkenes, and alkanes. The representation of hydrocarbons generally follows Houweling et al. (1998), which is originally derived from the Carbon Bond Mechanism-4 (Gery et al., 1989) and the Regional Atmospheric Chemistry Model (RACM; Stockwell et al., 1997), but includes several modifications aimed at representing the

5 wide range of chemical conditions found in Earth's atmosphere, such as the addition of reactions important in low-$NO_X$ conditions including representation of organic peroxy radical chemistry under low-$NO_X$ conditions and introduction of organic nitrate chemistry. Shindell et al. (2013) describe in detail the recent updates to the tropospheric chemistry scheme, including the incorporation of acetone chemistry (Houweling et al., 1998) and the addition of terpene oxidation (Tsigaridis and Kanakidou, 2007). SOA formation is driven by $NO_X$-dependent oxidation of emissions of isoprene, monoterpenes, and

10 other reactive VOCs following a volatility-based two-product scheme (Tsigaridis and Kanakidou, 2007). The formation of secondary inorganic aerosols, including sulfate (Bell et al., 2005; Koch et al., 2006) and nitrate (Bauer et al., 2007a), depend on both modeled oxidant levels and the availability of source gases. Primary aerosol types include dust (which provides a surface for heterogeneous chemistry; Bauer and Koch, 2005; Bauer et al., 2007b), black carbon, organic carbon, and sea salt (Koch et al., 2006). Stratospheric chemistry, introduced to the chemical mechanism by Shindell et al. (2006), includes

15 nitrous oxide ($N_2O$) and halogen (bromine and chlorine) chemistry. Recent updates to stratospheric chemistry are summarized by Shindell et al. (2013) and include changes in the representations of polar stratospheric cloud formation (Hanson and Mauersberger, 1988) and heterogeneous hydrolysis of $N_2O_5$ on sulfate (Hallquist et al., 2003; Kane et al., 2001). Photolysis rate calculations follow the Fast-J2 scheme of Bian and Prather (2002). At each 30 minute time step, the simulated distributions of clouds, ozone, and aerosols are passed to the photolysis code, providing a mechanism for

20 simulated changes in aerosols to impact atmospheric chemistry through modification of photolysis rates (Bian et al., 2003).

Global annual-mean mixing ratios are prescribed for the well-mixed greenhouse gases carbon dioxide ($CO_2$), methane ($CH_4$), nitrous oxide ($N_2O$), and chlorofluorocarbons (CFCs) (Meinshausen et al., 2011; Riahi et al., 2007). Prescribed monthly anthropogenic and biomass burning emissions of reactive gas and primary aerosol species follow the MACCity emissions

25 pathway (Angiola et al., 2010; Granier et al., 2011) for all years, except for 2010, when the interpolated ACCMIP-RCP8.5 dataset (Heil and Schultz, 2014) is applied for biomass burning emissions (as MACCity biomass burning emissions are available only through 2008). MACCity biomass burning emissions were built from the ACCMIP (Lamarque et al., 2010), REanalysis of the TROpospheric chemical composition (RETRO; Schultz et al., 2008), and Global Fire Emissions Database (GFED-v2; van der Werf et al., 2006) datasets (Granier et al., 2011). The interpolated ACCMIP-RCP8.5 emissions were

30 created using simple temporal interpolation of the ACCMIP and RCP8.5 datasets (Heil and Schultz, 2014). Identical surface maps and emission factors were applied in the creation of both the MACCity and interpolated ACCMIP-RCP8.5 biomass burning emissions datasets (Heil and Schultz, 2014). The model dynamically calculates climate-sensitive emissions of reactive compounds for a number of natural sources. These emission sectors, with their primary driving meteorological variables, include: lightning $NO_X$ (moist convection; Price and Rind, 1992; Price et al., 1997); soil $NO_X$ (precipitation and

temperature; Yienger and Levy II, 1995), dust (wind speed; Miller et al., 2006), sea salt particles (wind speed; Koch et al., 2006), marine dimethyl sulfide (wind speed; Koch et al., 2006), and the BVOCs isoprene (radiation, temperature, and soil moisture; Arneth et al., 2007; Unger et al., 2013) and monoterpenes (temperature; Lathière et al., 2006), which are described in more detail below.

Leaf-level gas exchange in ModelE2 (Collatz et al., 1991) couples the Farquhar–von Caemmerer kinetic model of photosynthetic $CO_2$ uptake (Farquhar et al., 1980; Farquhar and von Caemmerer, 1982) to the Ball–Berry model of stomatal conductance (Ball et al., 1987). The environmental inputs used to drive the vegetation biophysics, isoprene emissions, and monoterpene emissions are the values simulated by the general circulation model. The standard YIBs vegetation comprises

10 eight plant functional types: C3-grassland, C4-grassland, crop, deciduous broadleaf forest, evergreen broadleaf forest, evergreen needleleaf forest, shrubland, and tundra. The model code was modified to include four additional land cover types: (1) oil palm plantations; (2) rubber tree plantations; (3) other tree plantations; and (4) dipterocarp evergreen broadleaf forest. Relative to the forests of the Amazon, the tropical forests of MSEA contain a larger proportion of evergreen dipterocarp forests (Hewitt et al., 2010). The dipterocarp evergreen broadleaf forest PFT was added to account for the comparatively

15 lower isoprene emission capacity (Langford et al., 2010; Stavrakou et al., 2014) of the natural forests of MSEA. In each model grid cell, an individual canopy is simulated for each PFT. The canopy radiative transfer scheme divides each canopy into a flexible number of vertical layers (generally 2–16). In each layer, sunlit leaves use direct photosynthetically active radiation (PAR) for leaf-level photosynthesis, while shaded leaves use diffuse PAR (Spitters et al., 1986).

20 YIBs features a process-based biochemical model of isoprene emission in which the rate of isoprene production dynamically depends on the electron-transport-limited rate of photosynthetic carbon assimilation (Arneth et al., 2007; Unger et al., 2013). For each PFT, the isoprene emission rate depends linearly on the fraction of electrons available to undergo isoprene synthesis, the calculation of which requires prescription of a PFT-specific leaf-level isoprene basal emission rate (BER) at standard conditions. The leaf-level isoprene emission rate additionally depends on the atmospheric $CO_2$ concentration and

25 the simulated canopy temperature and PAR. The ability of the model to simulate isoprene emissions has been evaluated extensively against multiple above-canopy flux datasets from tropical and temperate ecosystems (Unger et al., 2013). The model simulated the local flux magnitude within a factor of two at nine specific measurement sites, some of which correspond to short (weeks-long) measurement campaigns (Unger et al., 2013). Temperature-dependent leaf-level monoterpene emissions, functionally α-pinene, likewise vary by ecosystem type, similarly through prescription of PFT-

30 specific basal emission rates (Guenther et al., 1995; Lathière et al., 2006). Recent work suggests that tropical monoterpene emissions exhibit both light and temperature dependency (Guenther et al. 2012; Jardine et al., 2015, 2017) that is not included in the emission algorithm here but may be explored in future work.
* * *
**Harper, Kandice 8/24/2018 2:57 PM**

**Harper, Kandice 8/24/2018 2:58 PM**

**Harper, Kandice 8/23/2018 6:15 PM**

**Harper, Kandice 8/24/2018 2:58 PM**

**Harper, Kandice 8/26/2018 11:16 AM**

**Harper, Kandice 8/24/2018 2:59 PM**

**Harper, Kandice 8/24/2018 2:59 PM**

**Harper, Kandice 8/24/2018 2:59 PM**

**Harper, Kandice 8/24/2018 2:59 PM**

**Harper, Kandice 8/23/2018 12:40 PM**

Aerosol and gas-phase chemistry are fully coupled, and the chemical mechanism is fully coupled to the climate modules (e.g., radiation and dynamics). All simulations apply ozone and aerosol climatologies to the radiation code (Schmidt et al., 2014), but simulated changes in ozone and aerosols can impact calculated photolysis rates. In this study, aerosols do not affect cloud properties. Observation-based, monthly-varying, five-year-average sea surface temperature and sea ice fields are prescribed according to the Hadley Centre Sea Ice and Sea Surface Temperature dataset (Rayner et al., 2003).

ModelE2 has previously undergone extensive, rigorous validation of simulated present-day tropospheric and stratospheric chemical composition, circulation, and ozone forcing using multiple observational datasets (Shindell et al., 2006; Shindell et al., 2013). Shindell et al. (2013) compared simulated monthly zonal-mean total column ozone to that from observations (2000–2010 means) from the Total Ozone Mapping Spectrometer and the Ozone Monitoring Instrument (McPeters et al., 2008), finding: simulated zonal-mean total column ozone in the tropics shows little bias (< 5%) against measurements for each month, and, in the Northern Hemisphere middle and high latitudes, biases are smaller in the summer months (< 10%) than in the winter months (around 15–20%). Shindell et al. (2013) find only a small negative bias (-0.016 W m$^{-2}$) in the present-day global-average radiative impact of modeled tropospheric ozone relative to TES-derived tropospheric ozone. They note that the strongest biases in ozone concentrations in ModelE2 are generally located in regions where ozone exhibits little effect on radiation (Shindell et al., 2013). More recently, Harper et al. (2018) compared annual-mean ozone concentrations simulated by ModelE2 (representative of year 2005) with an ozonesonde climatology based on measurements taken over 1995–2011 (Tilmes et al., 2012), finding lower model biases at higher pressures (e.g., +2.6% at 200 hPa compared to +16.9% at 800 hPa).

A number of uncertainties exist related to both isoprene oxidation chemistry and SOA formation. For example, field measurements, including one campaign over Borneo (Stone et al., 2011), indicate that OH concentrations over the pristine tropical rainforest are much higher than predicted by known isoprene chemistry (Lelieveld et al., 2008; Martinez et al., 2010; Stone et al., 2011). A number of OH-recycling mechanisms associated with isoprene oxidation have been proposed (e.g., da Silva et al., 2010; Lelieveld et al., 2008; Paulot et al., 2009; Peeters et al., 2009). Other researchers have argued that the high measured OH concentrations may be an artifact of the type of instrumentation employed, which results in an artificial inflation of the measured concentrations (Mao et al., 2012). No OH recycling is applied in the OH-initiated isoprene oxidation pathway in the chemical mechanism of ModelE2-YIBs because of the significant uncertainties associated with this aspect of isoprene chemistry. Importantly, Warwick et al. (2013) found that inclusion of OH recycling had little impact on the magnitude or distribution of surface ozone changes induced by biogenic emission changes from total conversion of Bornean tropical forest to oil palm plantations. However, some studies suggest that the simulation of ozone and other oxidants is sensitive to the isoprene chemical mechanism that is applied (e.g., Archibald et al., 2010; Emmerson and Evans (2009); Knote et al. (2015)). Furthermore, the appropriateness of using α-pinene to represent monoterpenes as a single lumped species in global modeling is an active area of research. Friedman and Farmer (2018) find order-of-magnitude

differences in SOA yields for OH-oxidation of different monoterpene species, but a clear explanation based on isomer structure remains largely elusive. While Friedman and Farmer (2018) find that the magnitude of the SOA yield from α-pinene is in the "mid-range" of the yields among the analyzed monoterpene species, other studies have shown that this may not be the case for other oxidation pathways (e.g., Draper et al., 2015). The appropriateness of using the two-product scheme for SOA production in global models is likewise an open question (e.g., Tsigaridis et al., 2014) and is discussed in more detail in Sect. 4. Future work would benefit from an exploration of the impact on radiative forcing induced through application of different mechanisms of (1) isoprene photooxidation and (2) SOA formation (e.g., Surratt et al., 2010; Zhang et al., 2018).

**2.2 Vegetation datasets**

[revised manuscript text omitted]

The Landsat-based YIBs-compatible land cover distributions are applied to the 57 model grid cells covering Peninsular Malaysia, Sumatra, Borneo, and New Guinea (Figure S1). The simulations apply non-zero areal extents of the four new land
20   cover types only in MSEA. Table 1 shows, for the new land cover types, the assigned physical parameters (including LAI and vegetation height), photosynthetic parameters, and leaf-level basal emission rates of isoprene and monoterpenes.

The static land cover distribution applied to the rest of the world is the year 2000 land cover distribution developed for the Community Land Model (CLM; Oleson et al., 2010) using multiple satellite datasets, including retrievals from both MODIS
25   (Hansen et al., 2003) and AVHRR (DeFries et al., 2000). The 16-PFT data were aggregated into the standard set of eight YIBs PFTs (Yue and Unger, 2015). Gridded PFT-specific LAI and vegetation height parameters are prescribed. For the rest-of-world vegetation, prescribed LAI are derived from CLM (Oleson et al., 2010); and prescribed heights are the output of a 140-year ModelE2-YIBs simulation (Yue and Unger, 2015) that simulated dynamic carbon allocation, used the same CLM land cover distribution described here, and was forced with year 2000 meteorology from the WFDEI (WATCH Forcing Data
30   methodology applied to ERA-Interim data; Weedon et al., 2014) dataset.

Harper, Kandice 8/23/2018 3:50 PM

Harper, Kandice 8/23/2018 3:52 PM

Harper, Kandice 8/23/2018 3:51 PM

Harper, Kandice 8/23/2018 3:51 PM

Harper, Kandice 8/23/2018 4:06 PM

Harper, Kandice 8/23/2018 3:51 PM

**Table 1**. Physical and photosynthetic parameters and isoprene and monoterpene basal emission rates assigned to four new land cover types in the ModelE2-YIBs source code. Plts. = plantations.

| Parameter | Dipterocarp forest[a] | Oil palm plts[b] | Rubber plts[c] | Other tree plts[d] |
|---|---|---|---|---|
| Vcmax25 ($\mu$mol $CO_2$ m$^{-2}$ s$^{-1}$) | 40 | 42 | 44 | 40 |
| $I_S$ ($\mu$gC g$^{-1}$ (leaf dry weight) h$^{-1}$) | 2 | 153 | 0.17 | 2 |
| $M_S$ ($\mu$gC g$^{-1}$ (leaf dry weight) h$^{-1}$) | 0.6 | 0 | 25 | 0.6 |
| SLA (m$^2$ (leaf) kg$^{-1}$ (leaf)) | 9.9 | 10.5 | 9.9 | 9.9 |
| PAR absorptance | 0.9 | 0.93 | 0.9 | 0.9 |
| Height (m) | 35 | 12 | 18 | 18 |
| LAI (m$^2$ [leaf] m$^{-2}$ [ground]) | 6 | 6 | 6 | 5.3 |

Parameters: (1) Vcmax25: maximum photosynthetic capacity at 25°C; (2) $I_S$: leaf-level isoprene basal emission rate (BER) at standard conditions of incident photosynthetically active radiation (PAR; 1000 $\mu$mol photons m$^{-2}$ s$^{-1}$) and leaf temperature (30°C); (3) $M_S$: leaf-level monoterpene BER at standard conditions of incident PAR (1000 $\mu$mol photons m$^{-2}$ s$^{-1}$) and leaf temperature (30°C); (4) SLA: specific leaf area; (5) PAR absorptance: the fraction of PAR photons incident on the leaf that are absorbed by the leaf; (6) vegetation height; and (7) leaf area index (LAI).

a) Vcmax25: Value assigned to the standard evergreen broadleaf forest PFT in YIBs. This value is supported by the average of measurements from five common tree species in the forest of Sulawesi, Indonesia (38.4 $\mu$mol m$^{-2}$ s$^{-1}$; Rakkibu, 2008). $I_S$: Upper limit of the BER (< 2 $\mu$gC g$^{-1}$ h$^{-1}$) reported for the species *Dipterocarpus obtusifolia* (Geron et al., 2006). Three additional tree species in the *Dipterocarpaceae* family are likewise reported to have low leaf-level isoprene emission rates, where low indicates an emission rate on the order of 0.1 $\mu$gC g$^{-1}$ h$^{-1}$ (actual numerical rates are not provided; Klinger et al., 2002). $M_S$: Calculated as the mean of the measured leaf-level emission rates for 11 dipterocarp species (Llusia et al., 2014). Height: Measurement from natural forest plot in Malaysian Borneo (Fowler et al., 2011). Within the range reported by Dietz et al. (2007) for natural forest plots. LAI: Measurement from natural forest plot in Malaysian Borneo (Fowler et al., 2011). Close to the LAI of 6.2 m$^2$ m$^{-2}$ estimated for undisturbed forest stands by Dietz et al. (2007). SLA and PAR absorptance: Values assigned to the standard evergreen broadleaf forest PFT in YIBs.

b) Vcmax25: Corresponds to mature (12-year-old) plantations (Meijide et al., 2017). $I_S$: Cronn and Nutmagul, 1982; Kesselmeier and Staudt, 1999. $M_S$: Measured emissions for six different monoterpenes (Geron et al., 2006). SLA: Average of two measurements (Fan et al., 2015; Legros et al., 2009). PAR absorptance: For measurements of mature leaves (Ritchie and Runcie, 2014). Height and LAI: Measurements from 12-year-old commercial plantation in Malaysian Borneo (Misztal et al., 2011).

c) Vcmax25: Kositsup et al., 2009. $I_S$: Geron et al., 2006; Klinger et al., 2002. $M_S$: Baker et al., 2005; Klinger et al., 2002. SLA and PAR absorptance: Values assigned to the standard evergreen broadleaf forest PFT in YIBs. Height: Mean height of rubber trees measured in 49 stands in Peninsular Malaysia (Suratman et al., 2004). LAI: Rubber trees are evergreen trees in the humid tropics (Li et al., 2016). Assigned LAI measured for a mature oil palm plantation in Malaysian Borneo (Misztal et al., 2011).

d) Description: This PFT is a combination of the timber plantation and mixed tree crop / agroforest cover types from the Gunarso et al. (2013) land cover classification scheme that is used to build the maritime Southeast Asian land cover distribution maps that are applied to the ModelE2-YIBs simulations. Typical species grown on timber plantations include *Gmelina sp.*, *Paraserianthes falcataria*, and *Acacia mangium* (Gunarso et al., 2013). Southeast Asian agroforest plots can contain a diverse array of vegetation, including oil palm, rubber trees, herbaceous crops, and many other tree species used as cash or subsistence crops (e.g., fruit and timber trees) (Scales and Marsden, 2008). Many agroforest plots in Southeast Asia maintain forest-like structural characteristics, despite the fact that they are cultivated, rather than natural, systems (Scales and

Marsden, 2008). Vcmax25: Value assigned to both the evergreen broadleaf forest and crops classes in YIBs. $I_S$: Same as dipterocarp forest PFT. Low-isoprene-emitting rubber trees are prevalent in Indonesian agroforest systems (Scales and Marsden, 2008), warranting a low BER for this agroforest-containing PFT. The average $I_S$ (4.7 µgC g$^{-1}$ h$^{-1}$) based on two common timber species – *Acacia mangium* (Klinger et al., 2002) and *Gmelina arborea* (Singh et al., 2014) – is similar in magnitude to the $I_S$ for the dipterocarp forest PFT, providing further justification for assignment of a low $I_S$. $M_S$: Same as dipterocarp forest PFT. Similar $M_S$ is reported for *Acacia mangium* (0.66 µgC g$^{-1}$ h$^{-1}$; Klinger et al., 2002). SLA and PAR absorptance: Values assigned to the standard evergreen broadleaf forest PFT in YIBs. Height: Reported height measured for plots of an agroforest (Dietz et al., 2007) and a timber plantation (Krisnawati et al., 2011). LAI: Mean LAI reported for three agroforest plots in Central Sulawesi, Indonesia (Dietz et al., 2007). Timber plantation class from Gunarso et al. (2013) has a partially open canopy, indicating a lower LAI than for the natural forest PFT.

Table 2 shows the areal extents of YIBs land covers in MSEA for 1990, 2005, and 2010. Figure S1 in the Supplement shows the regional land cover distribution for 1990. Over 1990–2010, 11.3 Mha of natural rainforest was lost (-8% relative to 1990). Forest loss was widespread on Borneo, Sumatra, and Peninsular Malaysia (Figure S2). Contraction of rubber plantations (-1.4 Mha) was primarily confined to Sumatra and Peninsular Malaysia. The high-isoprene-emitting oil palm class experienced the largest absolute increase in areal extent over the study era (+9.6 Mha, +267%). Widespread expansion occurred on Sumatra, Borneo, and Peninsular Malaysia.

**Table 2:** Areal extents (Mha) of eight YIBs land cover types in maritime Southeast Asia. Extents encompass only the 57 grid cells that apply the land cover distributions derived from the Gunarso et al. (2013) analysis. The changes in areal extent (in Mha) relative to 1990 are listed in parentheses for 2005 and 2010.

| YIBs land cover | 1990 | 2005 | 2010 |
|---|---|---|---|
| Shrubland | 29.7 | 30.5 (+0.8) | 30.8 (+1.1) |
| Crops | 10.6 | 11.3 (+0.7) | 13.1 (+2.6) |
| C4-grassland | 3.2 | 2.9 (-0.3) | 2.9 (-0.3) |
| Bare | 2.8 | 3.3 (+0.5) | 3.6 (+0.8) |
| Oil palm plantations | 3.6 | 9.8 (+6.3) | 13.2 (+9.6) |
| Rubber plantations | 7.8 | 6.7 (-1.1) | 6.4 (-1.4) |
| Other tree plantations | 14.0 | 13.7 (-0.3) | 12.9 (-1.1) |
| Dipterocarp rainforest | 140.6 | 134.0 (-6.7) | 129.3 (-11.3) |

**2.3 Simulation configurations**

Table 3 summarizes the configurations of nine global chemistry–climate simulations. Two principal time-slice simulations – 2010land_base and 1990land_base – are used to diagnose the global-mean radiative perturbation associated with the

Harper, Kandice 8/23/2018 3:56 PM

Harper, Kandice 8/25/2018 5:51 PM

Harper, Kandice 8/23/2018 3:56 PM

Harper, Kandice 8/23/2018 3:56 PM

atmospheric composition changes induced by 1990–2010 land cover change in MSEA. The two simulations differ only in terms of the year of the applied maritime Southeast Asian land cover distribution, which is indicated in the simulation names. The regional land cover changes are imposed on a background climate and atmosphere representative of year 2010; that is, both the 2010 land cover distribution in the perturbation simulation and the 1990 land cover distribution in the control

5  simulation are exposed to the climate state, atmospheric $CO_2$ concentration, and background atmosphere representative of year 2010. The present-day rest-of-world land cover distribution is identical for all simulations. All simulations were run for 13 years, and averages over the final 10 years of output were used for analysis. The impact of 1990–2010 maritime Southeast Asian land cover change, denoted ΔLC, on atmospheric composition and radiative balance is diagnosed as 2010land_base minus 1990land_base (Table 4). Seven additional simulations (Tables 3 and 4) probe the sensitivity of the radiative forcing

10  results to: (1) the applied background atmosphere; (2) the degree of regional land cover change; (3) the magnitude of the isoprene BER assigned to the oil palm plantation PFT; and (4) the magnitude of the isoprene BER assigned to the dipterocarp forest PFT.

Isoprene production in ModelE2-YIBs is calculated as a semi-mechanistic function of photosynthetic carbon assimilation

15  (Unger et al., 2013). Isoprene emissions are sensitive to simulated changes in the parameters that affect photosynthesis, including the background climate state (e.g., temperature, PAR, and soil moisture) and the atmospheric $CO_2$ concentration. Simulated monoterpene emissions are likewise sensitive to temperature shifts (Lathière et al., 2006). Following emission, the atmospheric processing of isoprene and monoterpenes is influenced by the background atmospheric composition; for example, the availability of $NO_X$ affects the VOC-driven production of both ozone (Sillman, 1999) and SOA (Tsigaridis and

20  Kanakidou, 2007). The calculation 2010land_1990atm minus 1990land_1990atm, denoted ΔLC-1990atm, is designed to test the influence of the prescribed background state upon which the maritime Southeast Asian land cover changes are imposed (Table 4). Relative to the main set of simulations probing Southeast Asian land cover change (i.e., ΔLC), the ΔLC-1990atm simulations prescribe identical changes in Southeast Asian land cover, but impose the changes on a background state representative of year 1990 rather than year 2010 (Table 3). The different background states can lead to different isoprene-

25  and monoterpene-driven impacts on ozone and SOA concentrations from the identical prescribed land cover changes by influencing (1) the magnitude and distribution of the isoprene and monoterpene emission changes associated with the land cover changes and (2) the atmospheric processing of the emitted isoprene and monoterpenes.

[revised manuscript text omitted]

10 precisely matches an estimate derived from flux-tower measurements that is representative of 1998–2005: 123 ± 8 PgC $y^{-1}$ (mean ± 1 standard deviation; Beer et al., 2010). The simulated 1990 global GPP of 108 PgC $y^{-1}$ (simulation 1990land_1990atm) is outside of the 1-standard-deviation range of the observation-based mean, but falls within the 95% confidence interval (102–135 PgC $y^{-1}$; Beer et al., 2010). Guenther et al. (2012) collated contemporary global annual BVOC emission estimates from the literature, finding ranges of 309–706 TgC $y^{-1}$ for isoprene and 26–156 TgC $y^{-1}$ for

15 monoterpenes. The model estimates for 1990 (325 TgC $y^{-1}$ isoprene and 90 TgC $y^{-1}$ monoterpenes for simulation 1990land_1990atm) and 2010 (363 TgC $y^{-1}$ isoprene and 77 TgC $y^{-1}$ monoterpenes) fall within these ranges. Using the same process-based, leaf-level isoprene production algorithm employed in ModelE2-YIBs, although driven with different forcing datasets, Hantson et al. (2017) predict contemporary isoprene emissions (385 TgC $y^{-1}$; 1971–2000 mean) that are 18% higher than those predicted here for 1990 and only 6% higher than those predicted here for 2010.

Harper, Kandice 8/23/2018 9:12 PM

Harper, Kandice 8/23/2018 9:12 PM

Harper, Kandice 8/25/2018 8:03 PM

Harper, Kandice 8/24/2018 2:42 PM

In MSEA, 2010 isoprene emission rates are generally higher on Sumatra, Peninsular Malaysia, and Borneo than on the island of New Guinea (Figure S3), which maintains high areal coverage of low-isoprene-emitting dipterocarp rainforests. Rubber plantations contributed 56% (1.6 TgC $y^{-1}$) of the regional monoterpene emissions in 2010, while oil palm (8.8 TgC $y^{-1}$) and

Harper, Kandice 8/23/2018 6:22 PM

Harper, Kandice 8/25/2018 5:52 PM

Harper, Kandice 8/24/2018 2:43 PM

shrubs (5.8 TgC y$^{-1}$) dominated regional isoprene emissions (55% and 36%, respectively). The low-isoprene-emitting dipterocarp rainforests, which covered 61% of the region's land surface in 2010, were responsible for only 8% of regional isoprene emissions. The strong contributions made by rubber and oil palm plantations to the regional monoterpene and isoprene budgets, respectively, underscore the importance of explicitly accounting for these land covers in regional land use

5   and land cover change analyses.

Regional 1990–2010 land cover change induced a negligible decrease in regional GPP (-0.1 PgC y$^{-1}$), which is principally attributed to an increase from oil palm (+0.3 PgC y$^{-1}$) and a decrease from dipterocarp rainforests (-0.4 PgC y$^{-1}$). Regional land cover change induced annual emissions changes of +6.5 TgC y$^{-1}$ isoprene and -0.5 TgC y$^{-1}$ monoterpenes. The land-

10   cover-change-driven net increase in regional isoprene emissions is almost entirely due to expansion of industrial oil palm plantations (+6.4 TgC y$^{-1}$, +271% relative to 1990 oil palm isoprene emissions). Regional isoprene emissions from shrubs increased by 4.2% (+0.2 TgC y$^{-1}$), associated with a 3.7% increase in shrubland. The large loss of dipterocarp rainforest had little impact on isoprene emissions (-0.1 TgC y$^{-1}$), as this PFT is a weak isoprene emitter. Contraction of rubber plantation extent was largely responsible for the reduction in monoterpene emissions (-0.4 TgC y$^{-1}$).

15   **3.2 Atmospheric composition**

Low surface ozone concentrations are simulated for the pristine atmospheres of the tropical forests. In the Southeast Asia study region in 2010 (2010land_base), the less disturbed landscapes of Borneo and New Guinea exhibit lower surface ozone concentrations than the comparatively more industrialized regions of Sumatra and Peninsular Malaysia (Figure S4). Considering only the grid cells in the 57-grid-cell study area that are majority (> 50%) forest by area, simulated annual-mean

20   surface ozone concentrations in 2010 are 9.0 ppbv in New Guinea (n=10) and 9.5 ppbv in Borneo (n=7). Considering only the grid cells that are > 85% forest by area, simulated ozone concentrations are 7.8 ppbv in New Guinea (n=4) and 7.2 ppbv in Borneo (n=2). Measurements at a rainforest site in Malaysian Borneo in 2008 found daytime surface-level ozone mixing ratios of 5–8 ppbv (Hewitt et al., 2010), providing support for the low ozone concentrations simulated over the highly forested regions of the study area.

1990–2010 land cover change in MSEA drove a reduction in annual-mean surface ozone concentrations over Borneo, Peninsular Malaysia, and Sumatra (Figure 1). Negligible changes occurred over New Guinea. Isoprene oxidation is more implicated in ozone production and loss rather than SOA formation (whereas monoterpenes are more implicated in SOA formation). Regionally, small surface ozone enhancements are simulated over the marine environment, with maximum

30   enhancements occurring over the ocean to the west of Sumatra. Because this region exhibits low surface ozone concentrations, the small absolute changes (-3.8 to +0.8 ppbv) translate into comparatively large relative changes (-18.3% to +4.3%).

Harper, Kandice 8/23/2018 4:58 PM

Harper, Kandice 8/23/2018 4:58 PM

Harper, Kandice 8/25/2018 5:53 PM

Harper, Kandice 8/23/2018 5:00 PM

Harper, Kandice 8/24/2018 2:43 PM

Simulated reductions in surface ozone largely occur in the regions of enhanced isoprene emissions, specifically Peninsular Malaysia, Sumatra, and Borneo (Figure S5). Surface ozone reductions occurring in response to enhanced VOC emissions in low-$NO_X$ conditions are associated with an increase in the relative importance of ozone destruction reactions (e.g., direct reaction of the VOC with ozone) (Ashworth et al., 2012; Sillman, 1999). $NO_X$ surface emissions in the analysis region were,

5 on average, 0.036 mgN m$^{-2}$ h$^{-1}$ in 2010. Based on atmospheric chemical modeling in conjunction with aircraft and ground measurements in Borneo, $NO_X$ fluxes for 2008 were inferred as 0.009 mgN m$^{-2}$ h$^{-1}$ over a rainforest site and 0.019 mgN m$^{-2}$ h$^{-1}$ over an oil palm plantation (Hewitt et al., 2009). The applied $NO_X$ emissions are slightly higher than, but the same order of magnitude as, the observation-based inferred fluxes, providing support for the "$NO_X$-sensitive regime" (Sillman, 1999) that is modeled in this study.

Previous studies have shown that the sign and strength of the regional surface ozone response to increased isoprene strongly depend on availability of $NO_X$ (Ashworth et al., 2012; Silva et al., 2016; Warwick et al., 2013). When applying contemporary $NO_X$ emissions from published inventories, both Warwick et al. (2013) and Ashworth et al. (2012) simulated local surface ozone reductions in response to increased isoprene emissions from regional oil palm expansion. The Warwick

15 et al. (2013) study was an idealized simulation that carpeted Borneo in oil palm, while the Ashworth et al. (2012) study applied future projections of oil palm expansion for biofuel production. Both studies found that increasing the $NO_X$ emissions in the region of land conversion (to account for enhanced fertilizer application and industrial processing of the oil palm) enhanced surface ozone concentrations (Ashworth et al., 2012; Warwick et al., 2013). In a study focused on estimating the air quality impacts associated with 2010 oil palm cover compared to a no-oil-palm landscape, Silva et al. (2016)

20 simulated increased surface ozone concentrations over much of the region. They found that some low-$NO_X$ regions (e.g., parts of Borneo) exhibited surface ozone reductions in response to increased isoprene emissions (Silva et al., 2016). The differences in the simulated impacts on regional surface ozone between this study and the Silva et al. (2016) study are likely due to the magnitude of the applied regional $NO_X$ emissions.

25 The simulated changes in atmospheric composition might be a response not only to altered isoprene and monoterpene emissions, but also to changes in the deposition of atmospheric species induced by changes in leaf density (Wong et al., 2018) or related changes, such as surface roughness, stomatal conductance, and evapotranspiration, that are affected by the applied changes in land cover distribution. Here, the relative changes in regional ozone deposition rates (-19.7 to +4.3%) are similar to the relative changes in regional surface-level ozone concentrations (-18.3 to +4.3%) from 1990–2010 regional land

30 cover change, in part because the ozone deposition rate depends on the atmospheric concentration change. While increased isoprene emission leading to increased isoprene ozonolysis drives ozone losses near the surface, a formal quantitative attribution analysis disentangling the relative roles of emission and deposition changes requires further complex sensitivity simulations that are beyond the scope of this analysis. In their analysis of Southeast Asian oil palm expansion, Silva et al.

Harper, Kandice 8/25/2018 5:54 PM

Harper, Kandice 8/23/2018 5:23 PM

Harper, Kandice 8/23/2018 6:24 PM

Harper, Kandice 8/24/2018 5:31 PM

(2016) used sensitivity studies to determine that the induced BVOC emission changes, rather than altered deposition rates from LAI changes, were almost exclusively responsible for the simulated surface ozone changes.

Harper, Kandice 8/24/2018 5:32 PM

[Figure]

**Figure 1:** Changes in annual-mean ozone mixing ratio (ppbv) due to 1990–2010 maritime Southeast Asian land cover change: a) Southeast Asian surface-level ozone; b) global surface-level ozone; c) tropospheric ozone at 335 hPa; and d) global-mean ozone profile.

While the strongest impacts of regional land cover change on annual-mean surface ozone are confined to Southeast Asia, weak long-range enhancements are simulated, particularly over the tropical ocean (< 1 ppbv over Indian Ocean, < 0.5 ppbv

elsewhere) (Figure 1). Enhanced isoprene oxidation drives an increase in the formation of alkyl nitrates, which are $NO_X$ reservoirs. Alkyl nitrates sequester reactive $NO_X$ upon formation and can undergo long-range transport, eventually releasing the $NO_X$ far from the source region (Atherton, 1989). Thus, alkyl nitrate perturbations provide a mechanism for ozone perturbations to occur far from the site of the hydrocarbon emission change (Hollaway et al., 2017); here, resulting in ozone enhancements over the tropical ocean.

With decreasing atmospheric pressure, the long-range impact on ozone spreads beyond the tropics and generally grows in magnitude (Figure 1). The global-mean ozone enhancement increases in magnitude as altitude increases from the surface to ~100 hPa (Figure 1). Considering the troposphere, the global-mean ozone enhancement from regional land cover change is on the order of 0.5 ppbv in the upper troposphere (e.g., at 237 hPa), compared to < 0.1 ppbv in the lower troposphere (at pressures > 875 hPa). The maximum relative change in the global-mean ozone mixing ratio is +0.6%, occurring at 160 hPa. While enhanced alkyl nitrate formation can likewise play a role in the free tropospheric ozone changes (as for surface ozone), additional mechanisms of ozone change (including in situ production from transported isoprene and its degradation products) are also implicated, as described below.

[revised manuscript text omitted]

Harper, Kandice 8/23/2018 6:50 PM

Harper, Kandice 8/26/2018 12:49 AM

Harper, Kandice 8/26/2018 12:15 AM

Harper, Kandice 8/26/2018 12:50 AM

had little impact on the ozone and SOA forcings induced by land cover change (Table 5). The fact that local surface ozone reductions, rather than enhancements, are simulated in response to enhanced isoprene emissions (Figure 1) suggests a $NO_X$-sensitive chemical environment.

5  Taking into account the results of the sensitivity simulations, the best estimate of the global-mean forcing from ozone changes induced by regional 1990–2010 land cover change is +9 mW m$^{-2}$, with a range of +4 to +16 mW m$^{-2}$. The quantified range accounts only for uncertainties probed by the sensitivity studies.

**4 Conclusions and future work**

The best estimate of global-mean forcing from isoprene and monoterpene emission perturbations driven by regional land
10  cover change – quantified here using simulations that apply satellite-derived land cover distributions and measured leaf-level isoprene and monoterpene BERs – indicates a positive forcing (+8.4 ± 0.7 mW m$^{-2}$), which is a warming impact. In absolute terms, the quantified forcing from 1990–2010 maritime Southeast Asian land cover change is small, particularly in comparison to the forcing associated with industrial-era perturbations of well-mixed greenhouse gases (e.g., Myhre et al., 2013). However, the ozone perturbations associated with changes in global anthropogenic emissions of non-methane VOCs
15  over the industrial era (1750–2011) induced a global-mean stratospherically adjusted forcing on the order of +30 mW m$^{-2}$ (Myhre et al., 2013), which is only around 3× the magnitude of the instantaneous ozone forcing associated with 1990–2010 land cover change in MSEA (+9.2 mW m$^{-2}$). For comparison, the global ozone forcing driven by the 1990–2010 land cover change in MSEA is at the low end of the range of estimates for ozone forcing from global anthropogenic emission source sectors in year 2000 (+5 to +80 mW m$^{-2}$): for example, industry = +15 mW m$^{-2}$; household biofuel +28 mW m$^{-2}$; road
20  transport = +50 mW m$^{-2}$; power = +53 mW m$^{-2}$; biomass burning = +71 mW m$^{-2}$ (Fuglestvedt et al., 2008; Unger et al., 2010). A multi-model study found that 20% reductions in NMVOCs (about 2–4 TgC y$^{-1}$) in four large world regions (North America, East Asia, Europe, and South Asia) in 2001 led to global ozone forcings around -1 mW m$^{-2}$ (Fry et al., 2012).

[revised manuscript text omitted]

Bauer, S.E., Mishchenko, M.I., Lacis, A.A., Zhang, S., Perlwitz, J., and Metzger, S.M.: Do sulfate and nitrate coatings on mineral dust have important effects on radiative properties and climate modeling?, J. Geophys. Res., 112, D06307, doi: 10.1029/2005JD006977, 2007b.

Beer, C., Reichstein, M., Tomelleri, E., Ciais, P., Jung, M., Carvalhais, N., Rödenbeck, C., Arain, M.A., Baldocchi, D., Bonan, G.B., Bondeau, A., Cescatti, A., Lasslop, G., Lindroth, A., Lomas, M., Luyssaert, S., Margolis, H., Oleson, K.W., Roupsard, O., Veenendaal, E., Viovy, N., Williams, C., Woodward, F.I., and Papale, D.: Terrestrial gross carbon dioxide uptake: Global distribution and covariation with climate, Science, 329, 834–838, doi: 10.1126/science.1184984, 2010.

Beirle, S., Huntrieser, H., and Wagner, T.: Direct satellite observation of lightning-produced NOx, Atmos. Chem. Phys., 10, 10965–10986, doi: 10.5194/acp-10-10965-2010, 2010.

Bell, N., Koch, D., and Shindell, D.T.: Impacts of chemistry–aerosol coupling on tropospheric ozone and sulfate simulations in a general circulation model, J. Geophys. Res., 110, D14305, doi: 10.1029/2004JD005538, 2005.

Bian, H. and Prather, M.J.: Fast-J2: Accurate simulation of stratospheric photolysis in global chemical models, J. Atmos. Chem., 41, 281–296, doi: 10.1023/A:1014980619462, 2002.

Bian, H., Prather, M.J., and Takemura, T.: Tropospheric aerosol impacts on trace gas budgets through photolysis, J. Geophys. Res., 108, 4242, doi: 10.1029/2002JD002743, 2003.

Boersma, K.F., Eskes, H.J., Meijer, E.W., and Kelder, H.M.: Estimates of lightning NOx production from GOME satellite observations, Atmos. Chem. Phys., 5, 2311–2331, doi: 10.5194/acp-5-2311-2005, 2005.

Carlson, K.M., Curran, L.M., Asner, G.P., Pittman, A.M., Trigg, S.N., and Marion Adeney, J.: Carbon emissions from forest conversion by Kalimantan oil palm plantations, Nat. Clim. Change, 3, 283–287, doi: 10.1038/nclimate1702, 2012.

5 Carlton, A.G., Wiedinmyer, C., and Kroll, J.H.: A review of secondary organic aerosol (SOA) formation from isoprene, Atmos. Chem. Phys., 9, 4987–5005, doi: 10.5194/acp-9-4987-2009, 2009.

Collatz, G.J., Ball, J.T., Grivet, C., and Berry, J.A.: Physiological and environmental regulation of stomatal conductance, photosynthesis and transpiration: A model that includes a laminar boundary layer, Agr. Forest Meteorol., 54, 107–136, doi: 10.1016/0168-1923(91)90002-8, 1991.

Collins, W.J., Stevenson, D.S., Johnson, C.E., and Derwent, R.G.: Role of convection in determining the budget of odd hydrogen in the upper troposphere, J. Geophys. Res., 104, 26927–26941, doi: 10.1029/1999JD900143, 1999.

Cronn, D.R. and Nutmagul, W.: Analysis of atmospheric hydrocarbons during winter MONEX, Tellus, 34, 159–165, doi: 15 10.1111/j.2153-3490.1982.tb01803.x, 1982.

da Silva, G., Graham, C., and Wang, Z.-F.: Unimolecular β-hydroxyperoxy radical decomposition with OH recycling in the photochemical oxidation of isoprene, Environ. Sci. Technol., 44, 250–256, doi: 10.1021/es900924d, 2010.

20 DeFries, R.S., Hansen, M.C., Townshend, J.R.G., Janetos, A.C., and Loveland, T.R.: A new global 1-km dataset of percentage tree cover derived from remote sensing, Global Change Biol., 6, 247–254, doi: 10.1046/j.1365-2486.2000.00296.x, 2000.

Dietz, J., Hölscher, D., Leuschner, C., Malik, A., and Amir, M.A.: Forest structure as influenced by different types of 25 community forestry in a lower montane rainforest of Central Sulawesi, Indonesia, in: Stability of Tropical Rainforest Margins: Linking Ecological, Economic and Social Constraints of Land Use and Conservation, edited by: Tscharntke, T., Leuschner, C., Zeller, M., Guhardja, E., and Biden, A., Springer-Verlag, Berlin, pp. 131–146, 2007.

Draper, D.C., Farmer, D.K., Desyaterik, Y., and Fry, J.L.: A qualitative comparison of secondary organic aerosol yields and 30 composition from ozonolysis of monoterpenes at varying concentrations of NO2, Atmos. Chem. Phys., 15, 12267–12281, doi: 10.5194/acp-15-12267-2015, 2015.

Emmerson, K. M. and Evans, M. J.: Comparison of tropospheric gas-phase chemistry schemes for use within global models, Atmos. Chem. Phys., 9, 1831–1845, doi: 10.5194/acp-9-1831-2009, 2009.

Fan, Y., Roupsard, O., Bernoux, M., Le Maire, G., Panferov, O., Kotowska, M.M., and Knohl, A.: A sub-canopy structure for simulating oil palm in the Community Land Model (CLM-Palm): Phenology, allocation and yield, Geosci. Model Dev., 8, 3785–3800, doi: 10.5194/gmd-8-3785-2015, 2015.

FAO: FAOSTAT Emissions Database [Electronic database], Food and Agriculture Organization of the United Nations, http://www.fao.org/faostat/en/#data, 2014.

Farquhar, G.D. and von Caemmerer, S.: Modelling of photosynthetic response to environmental conditions, in: Physiological Plant Ecology II, edited by: Lange, O.L., Nobel, P.S., Osmond, C.B., and Ziegler, H., Springer-Verlag, Berlin, pp. 549–587, 1982.

Farquhar, G.D., von Caemmerer, S., and Berry, J.A.: A biochemical model of photosynthetic $CO_2$ assimilation in leaves of C3 species, Planta, 149, 78–90, doi: 10.1007/BF00386231, 1980.

Folkins, I., Chatfield, R., Baumgardner, D., and Proffitt, M.: Biomass burning and deep convection in southeastern Asia: Results from ASHOE/MAESA, J. Geophys. Res., 102, 13,291–13,299, doi: 10.1029/96JD03711, 1997.

Fowler, D., Nemitz, E., Misztal, P., Di Marco, C., Skiba, U., Ryder, J., Helfter, C., Cape, J.N., Owen, S., Dorsey, J., Gallagher, M.W., Coyle, M., Phillips, G., Davison, B., Langford, B., MacKenzie, R., Muller, J., Siong, J., Dari-Salisburgo, C., Di Carlo, P., Aruffo, E., Giammaria, F., Pyle, J.A., and Hewitt, C.N.: Effects of land use on surface-atmosphere exchanges of trace gases and energy in Borneo: Comparing fluxes over oil palm plantations and a rainforest, Philos. T. R. Soc. B, 366, 3196–3209, doi: 10.1098/rstb.2011.0055, 2011.

Friedman, B. and Farmer, D.K.: SOA and gas phase organic acid yields from the sequential photooxidation of seven monoterpenes, Atmos. Env., 187, 335–345, doi: 10.1016/j.atmosenv.2018.06.003, 2018.

Fry, M.M., Naik, V., West, J.J., Schwarzkopf, M.D., Fiore, A.M., Collins, W.J., Dentener, F.J., Shindell, D.T., Atherton, C., Bergmann, D., Duncan, B.N., Hess, P., MacKenzie, I.A., Marmer,E., Schultz, M.G., Szopa, S., Wild, O., and Zeng, G.: The influence of ozone precursor emissions from four world regions on tropospheric composition and radiative climate forcing, J. Geophys. Res., 117, D07306, doi: 10.1029/2011JD017134, 2012.

Harper, Kandice 8/26/2018 8:39 AM

Fuglestvedt, J., Berntsen, T., Myhre, G., Rypdal, K., and Skeie, R.B.: Climate forcing from the transport sectors, P. Natl. Acad. Sci. USA, 105, 454–458, doi: 10.1073/pnas.0702958104, 2008.

Gaveau, D.L.A., Salim, M.A., Hergoualc'h, K., Locatelli, B., Sloan, S., Wooster, M., Marlier, M.E., Molidena, E., Yaen, H., DeFries, R., Verchot, L., Murdiyarso, D., Nasi, R., Holmgren, P., and Sheil, D.: Major atmospheric emissions from peat fires in Southeast Asia during non-drought years: Evidence from the 2013 Sumatran fires, Sci. Rep.-U.K., 4, 6112, doi: 10.1038/srep06112, 2014.

Geron, C., Guenther, A., Sharkey, T., and Arnts, R.R.: Temporal variability in basal isoprene emission factor, Tree Physiol., 20, 799–805, 2000.

Geron, C., Owen, S., Guenther, A., Greenberg, J., Rasmussen, R., Bai, J.H., Li, Q.-J., and Baker, B.: Volatile organic compounds from vegetation in southern Yunnan Province, China: Emission rates and some potential regional implications, Atmos. Environ., 40, 1759–1773, doi: 10.1016/j.atmosenv.2005.11.022, 2006.

Gery, M.W., Whitten, G.Z., Killus, J.P., and Dodge, M.C.: A photochemical kinetics mechanism for urban and regional scale computer modeling, J. Geophys. Res.-Atmos., 94, 12,925–12,956, doi: 10.1029/JD094iD10p12925, 1989.

Granier, C., Bessagnet, B., Bond, T., D'Angiola, A., Denier van der Gon, H., Frost, G.J., Heil, A., Kaiser, J.W., Kinne, S., Klimont, Z., Kloster, S., Lamarque, J.-F., Liousse, C., Masui, T., Meleux, F., Mieville, A., Ohara, T., Raut, J.-C., Riahi, K., Schultz, M.G., Smith, S.J., Thompson, A., van Aardenne, J., van der Werf, G.R., and van Vuuren, D.P.: Evolution of anthropogenic and biomass burning emissions of air pollutants at global and regional scales during the 1980–2010 period, Climatic Change, 109, 163–190, doi: 10.1007/s10584-011-0154-1, 2011.

Guenther, A., Hewitt, C.N., Erickson, D., Fall, R., Geron, C., Graedel, T., Harley, P., Klinger, L., Lerdau, M., McKay, W.A., Pierce, T., Scholes, B., Steinbrecher, R., Tallamraju, R., Taylor, J., and Zimmerman, P.: A global model of natural volatile organic compound emissions, J. Geophys. Res.-Atmos., 100, 8873–8892, doi: 10.1029/94JD02950, 1995.

Guenther, A., Karl, T., Harley, P., Wiedinmyer, C., Palmer, P.I., and Geron, C.: Estimates of global terrestrial isoprene emissions using MEGAN (Model of Emissions of Gases and Aerosols from Nature), Atmos. Chem. Phys., 6, 3181–3210, doi: 10.5194/acp-6-3181-2006, 2006.

Guenther, A.B., Jiang, X., Heald, C.L., Sakulyanontvittaya, T., Duhl, T., Emmons, L.K., and Wang, X.: The Model of Emissions of Gases and Aerosols from Nature version 2.1 (MEGAN2.1): An extended and updated framework for modeling biogenic emissions, Geosci. Model Dev., 5, 1471–1492, doi: 10.5194/gmd-5-1471-2012, 2012.

5  Gunarso, P., Hartoyo, M.E., Agus, F., and Killeen, T.J.: Oil palm and land use change in Indonesia, Malaysia, and Papua New Guinea: Reports from the Technical Panels of the 2nd Greenhouse Gas Working Group of the Roundtable on Sustainable Palm Oil (RSPO), Roundtable on Sustainable Palm Oil, Kuala Lumpur, Malaysia, 2013.

Hallquist, M., Stewart, D.J., Stephenson, S.K., and Cox, R.A.: Hydrolysis of $N_2O_5$ on sub-micron sulfate aerosols, Phys.
10  Chem. Chem. Phys., 5, 3453–3463, doi: 10.1039/b301827j, 2003.

Hansen, M.C., DeFries, R.S., Townshend, J.R.G., Carroll, M., Dimiceli, C., and Sohlberg, R.A.: Global percent tree cover at a spatial resolution of 500 meters: First results of the MODIS vegetation continuous fields algorithm, Earth Interact., 7, 1–15, doi: 10.1175/1087-3562(2003)007<0001:GPTCAA>2.0.CO;2, 2003.

Hanson, D. and Mauersberger, K.: Laboratory studies of the nitric acid trihydrate: Implications for the south polar stratosphere, Geophys. Res. Lett., 15, 855–858, doi: 10.1029/GL015i008p00855, 1988.

Hantson, S., Knorr, W., Schurgers, G., Pugh, T.A.M., and Arneth, A.: Global isoprene and monoterpene emissions under
20  changing climate, vegetation, CO2 and land use, Atmos. Environ., 155, 35–45, doi: 10.1016/j.atmosenv.2017.02.010, 2017.

Harper, K.L., Zheng, Y., and Unger, N.: Advances in representing interactive methane in ModelE2-YIBs (version 1.1), Geosci. Model Dev. Discuss., doi: 10.5194/gmd-2018-85 , 2018.

25  Heil, A. and Schultz, M.G.: Interpolated ACCMIP and RCP emission dataset [Data files], Available from: http://accmip-emis.iek.fz-juelich.de/data/accmip/gridded_netcdf/accmip_interpolated/, 2014.

Hewitt, C.N., Ashworth, K., Boynard, A., Guenther, A., Langford, B., MacKenzie, A.R., Misztal, P.K., Nemitz, E., Owen, S.M., Possell, M., Pugh, T.A.M., Ryan, A.C., and Wild, O.: Ground-level ozone influenced by circadian control of isoprene
30  emissions, Nat. Geosci., 4, 671–674, doi: 10.1038/ngeo1271, 2011.

Hewitt, C.N., Lee, J.D., MacKenzie, A.R., Barkley, M.P., Carslaw, N., Carver, G.D., Chappell, N.A., Coe, H., Collier, C., Commane, R., Davies, F., Davison, B., DiCarlo, P., Di Marco, C.F., Dorsey, J.R., Edwards, P.M., Evans, M.J., Fowler, D., Furneaux, K.L., Gallagher, M., Guenther, A., Heard, D.E., Helfter, C., Hopkins, J., Ingham, T., Irwin, M., Jones, C.,

Karunaharan, A., Langford, B., Lewis, A.C., Lim, S.F., MacDonald, S.M., Mahajan, A.S., Malpass, S., McFiggans, G., Mills, G., Misztal, P., Moller, S., Monks, P.S., Nemitz, E., Nicolas-Perea, V., Oetjen, H., Oram, D.E., Palmer, P.I., Phillips, G.J., Pike, R., Plane, J.M.C., Pugh, T., Pyle, J.A., Reeves, C.E., Robinson, N.H., Stewart, D., Stone, D., Whalley, L.K., and Yang, X.: Overview: oxidant and particle photochemical processes above a south-east Asian tropical rainforest (the OP3 project): Introduction, rationale, location characteristics and tools, Atmos. Chem. Phys., 10, 169–199, doi: 10.5194/acp-10-169-2010, 2010.

Hewitt, C.N., MacKenzie, A.R., Di Carlo, P., Di Marco, C.F., Dorsey, J.R., Evans, M., Fowler, D., Gallagher, M.W., Hopkins, J.R., Jones, C.E., Langford, B., Lee, J.D., Lewis, A.C., Lim, S.F., McQuaid, J., Misztal, P., Moller, S.J., Monks, P.S., Nemitz, E., Oram, D.E., Owen, S.M., Phillips, G.J., Pugh, T.A.M., Pyle, J.A., Reeves, C.E., Ryder, J., Siong, J., Skiba, U., and Stewart, D.J.: Nitrogen management is essential to prevent tropical oil palm plantations from causing ground-level ozone pollution, P. Natl. Acad. Sci. USA, 106, 18447–18451, doi: 10.1073/pnas.0907541106, 2009.

Hollaway, M.J., Arnold, S.R., Collins, W.J., Folberth, G., and Rap, A.: (2017), Sensitivity of midnineteenth century tropospheric ozone to atmospheric chemistry-vegetation interactions, J. Geophys. Res.-Atmos., 122, 2452–2473, doi:10.1002/2016JD025462, 2017.

Hooijer, A., Page, S., Canadell, J.G., Silvius, M., Kwadijk, J., Wösten, H., and Jauhiainen, J.: Current and future CO2 emissions from drained peatlands in Southeast Asia, Biogeosciences, 7, 1505–1514, doi: 10.5194/bg-7-1505-2010, 2010.

Houweling, S., Dentener, F., and Lelieveld, J.: The impact of nonmethane hydrocarbon compounds on tropospheric photochemistry, J. Geophys. Res., 103, 10,673–10,696, doi: 10.1029/97JD03582, 1998.

Indonesian Ministry of Agriculture: Tree Crop Estate Statistics of Indonesia 2015–2017, Directorate General of Estate Crops, Ministry of Agriculture, Indonesia, 2017.

Indrarto, G.B., Murharjanti, P., Khatarina, J., Pulungan, I., Ivalerina, F., Rahman, J., Prana, M.N., Resosudarmo, I.A.P., and Muharrom, E.: The context of REDD+ in Indonesia: Drivers, agents and institutions (Working Paper 92), CIFOR, Bogor, Indonesia, 2012.

Jardine, A.B., Jardine, K.J., Fuentes, J.D., Martin, S.T., Martins, G., Durgante, F., Carneiro, V., Higuchi, N., Manzi, A.O., and Chambers, J.Q.: Highly reactive light-dependent monoterpenes in the Amazon, Geophys. Res. Lett., 42, 1576–1583, doi: 10.1002/2014GL062573, 2015.

Harper, Kandice 8/26/2018 11:03 AM

Jardine, K.J., Jardine, A.B., Holm, J.A., Lombardozzi, D.L., Negron-Juarez, R.I., Martin, S.T., Beller, H.R., Gimenez, B.O., Higuchi, N., and Chambers, J.Q.: Monoterpene 'thermometer' of tropical forest-atmosphere response to climate warming, Plant Cell Environ., 40, 441–452, doi: 10.1111/pce.12879, 2017.

5  Jokinen, T., Berndt, T., Makkonen, R., Kerminen, V.-M., Junninen, H., Paasonen, P., Stratmann, F., Herrmann, H., Guenther, A.B., Worsnop, D.R., Kulmala, M., Ehn, M., and Sipilä, M.: Production of extremely low volatile organic compounds from biogenic emissions: Measured yields and atmospheric implications, P. Natl. Acad. Sci. USA, 112, 7123–7128, doi: 10.1073/pnas.1423977112, 2015.

10  Kane, S.M., Caloz, F., and Leu, M.-T.: Heterogeneous uptake of gaseous $N_2O_5$ by $(NH_4)_2SO_4$, $NH_4HSO_4$, and $H_2SO_4$ aerosols, J. Phys. Chem. A, 105, 6465–6470, doi: 10.1021/jp010490x , 2001.

Keenan, T.F. and Niinemets, Ü.: Circadian control of global isoprene emissions, Nat. Geosci., 5, 435, doi: 10.1038/ngeo1500, 2012.

Kesselmeier, J. and Staudt, M.: Biogenic volatile organic compounds (VOC): An overview on emission, physiology and ecology, J. Atmos. Chem., 33, 23–88, doi: 10.1023/A:1006127516791, 1999.

Klinger, L.F., Li, Q.-J., Guenther, A.B., Greenberg, J.P., Baker, B., and Bai, J.-H.: Assessment of volatile organic compound
20  emissions from ecosystems of China, J. Geophys. Res.-Atmos., 107, 4603, doi: 10.1029/2001JD001076, 2002.

Knote, C., Tuccella, P., Curci, G., Emmons, L., Orlando, J. J., Madronich, S., Baró, R., Jiménez-Guerrero, P., Luecken, D., Hogrefe, C., Forkel, R., Werhahn, J., Hirtl, M., Perez, J. L., San José, R., Giordano, L., Brunner, D., Yahya, K., and Zhang, Y.: Influence of the choice of gas-phase mechanism on predictions of key gaseous pollutants during the AQMEII phase-2
25  intercomparison, Atmos. Environ., 115, 553–568, doi: 10.1016/j.atmosenv.2014.11.066, 2015.

Koch, D., Schmidt, G.A., and Field, C.V.: Sulfur, sea salt, and radionuclide aerosols in GISS ModelE, J. Geophys. Res., 111, D06206, doi: 10.1029/2004JD005550, 2006.

30  Koplitz, S.N., Mickley, L.J., Marlier, M.E., Buonocore, J.J., Kim, P.S., Liu, T., Sulprizio, M.P., DeFries, R.S., Jacob, D.J., Schwartz, J., Pongsiri, M., and Myers, S.S.: Public health impacts of the severe haze in Equatorial Asia in September–October 2015: Demonstration of a new framework for informing fire management strategies to reduce downwind smoke exposure, Environ. Res. Lett., 11, 094023, doi: 10.1088/1748-9326/11/9/094023, 2016.

Kositsup, B., Montpied, P., Kasemsap, P., Thaler, P., Améglio, T., and Dreyer, E.: Photosynthetic capacity and temperature responses of photosynthesis of rubber trees (*Hevea brasiliensis Müll. Arg.*) acclimate to changes in ambient temperatures, Trees, 23, 357–365, doi: 10.1007/s00468-008-0284-x, 2009.

Kotchenruther, R.A., Jaffe, D.A., and Jaeglé, L.: Ozone photochemistry and the role of peroxyacetyl nitrate in the springtime northeastern Pacific troposphere: Results from the Photochemical Ozone Budget of the Eastern North Pacific Atmosphere (PHOBEA) campaign, J. Geophys. Res., 106, 28,731–28,742, doi: 10.1029/2000JD000060, 2001.

Krisnawati, H., Varis, E., Kallio, M., and Kanninen, M.: Paraserianthes falcataria (L.) Nielsen: Ecology, silviculture and productivity, Center for International Forestry Research, Bogor, Indonesia, 2011.

Kroll, J.H., Ng, N.L., Murphy, S.M., Flagan, R.C., and Seinfeld, J.H.: Secondary organic aerosol formation from isoprene photooxidation under high-$NO_x$ conditions, Geophys. Res. Lett., 32, L18808, doi: 10.1029/2005GL023637, 2005.

Lacis, A.A., Wuebbles, D.J., and Logan, J.A.: Radiative forcing of climate by changes in the vertical distribution of ozone, J. Geophys. Res., 95, 9971–9981, doi: 10.1029/JD095iD07p09971, 1990.

Lamarque, J.-F., Bond, T.C., Eyring, V., Granier, C., Heil, A., Klimont, Z., Lee, D., Liousse, C., Mieville, A., Owen, B., Schultz, M.G., Shindell, D., Smith, S.J., Stehfest, E., Van Aardenne, J., Cooper, O.R., Kainuma, M., Mahowald, N., McConnell, J.R., Naik, V., Riahi, K., and van Vuuren, D.P.: Historical (1850–2000) gridded anthropogenic and biomass burning emissions of reactive gases and aerosols: methodology and application, Atmos. Chem. Phys., 10, 7017–7039, doi: 10.5194/acp-10-7017-2010, 2010.

[revised manuscript text omitted]

Stone, D., Evans, M.J., Edwards, P.M., Commane, R., Ingham, T., Rickard, A.R., Brookes, D.M., Hopkins, J., Leigh, R.J., Lewis, A.C., Monks, P.S., Oram, D., Reeves, C.E., Stewart, D., and Heard, D.E.: Isoprene oxidation mechanisms:

Measurements and modelling of OH and HO2 over a South-East Asian tropical rainforest during the OP3 field campaign, Atmos. Chem. Phys., 11, 6749–6771, doi: 10.5194/acp-11-6749-2011, 2011.

Suratman, M.N., Bull, G.Q., Leckie, D.G., Lemay, V.M., Marshall, P.L., and Mispan, M.R.: Prediction models for estimating the area, volume, and age of rubber (*Hevea brasiliensis*) plantations in Malaysia using Landsat TM data, Int. Forest. Rev., 6, 1–12, doi: 10.1505/ifor.6.1.1.32055, 2004.

Surratt, J.D., Chan, A.W.H., Eddingsaas, N.C., Chan, M., Loza, C.L., Kwan, A.J., Hersey, S.P., Flagan, R.C., Wennberg, P.O., and Seinfeld, J.H.: Reactive intermediates revealed in secondary organic aerosol formation from isoprene, P. Natl. Acad. Sci. USA, 107, 6640–6645, doi: 10.1073/pnas.0911114107, 2010.

Thompson, A.M., Tao, W.-K., Pickering, K.E., Scala, J.R., and Simpson, J.: Tropical deep convection and ozone formation, B. Am. Meteorol. Soc., 78, 1043–1054, doi: 10.1175/1520-0477(1997)078<1043:TDCAOF>2.0.CO;2, 1997.

Tilmes, S., Lamarque, J.-F., Emmons, L.K., Conley, A., Schultz, M.G., Saunois, M., Thouret, V., Thompson, A.M., Oltmans, S.J., Johnson, B., and Tarasick, D.: Technical Note: Ozonesonde climatology between 1995 and 2011: Description, evaluation and applications, Atmos. Chem. Phys., 12, 7475–7497, doi: 10.5194/acp-12-7475-2012, 2012.

Tsigaridis, K., Daskalakis, N., Kanakidou, M., Adams, P.J., Artaxo, P., Bahadur, R., Balkanski, Y., Bauer, S.E., Bellouin, N., Benedetti, A., Bergman, T., Berntsen, T.K., Beukes, J.P., Bian, H., Carslaw, K.S., Chin, M., Curci, G., Diehl, T., Easter, R.C., Ghan, S.J., Gong, S.L., Hodzic, A., Hoyle, C.R., Iversen, T., Jathar, S., Jimenez, J.L., Kaiser, J.W., Kirkevåg, A., Koch, D., Kokkola, H., Lee, Y.H., Lin, G., Liu, X., Luo, G., Ma, X., Mann, G.W., Mihalopoulos, N., Morcrette, J.-J., Müller, J.-F., Myhre, G., Myriokefalitakis, S., Ng, N.L., O'Donnell, D., Penner, J.E., Pozzoli, L., Pringle, K.J., Russell, L.M., Schulz, M., Sciare, J., Seland, Ø., Shindell, D.T., Sillman, S., Skeie, R. B., Spracklen, D., Stavrakou, T., Steenrod, S.D., Takemura, T., Tiitta, P.,Tilmes, S., Tost, H., van Noije, T., van Zyl, P.G., von Salzen, K., Yu, F., Wang, Z., Wang, Z., Zaveri, R. A., Zhang, H., Zhang, K., Zhang, Q., and Zhang, X.: The AeroCom evaluation and intercomparison of organic aerosol in global models, Atmos. Chem. Phys, 14, 10845–10895, doi: 10.5194/acp-14-10845-2014, 2014.

Tsigaridis, K. and Kanakidou, M.: Secondary organic aerosol importance in the future atmosphere, Atmos. Environ., 41, 4682–4692, doi: 10.1016/j.atmosenv.2007.03.045, 2007.

Unger, N.: On the role of plant volatiles in anthropogenic global climate change, Geophys. Res. Lett., 41, 8563–8569, doi: 10.1002/2014GL061616, 2014.

Unger, N., Bond, T.C., Wang, J.S., Koch, D.M., Menon, S., Shindell, D.T., and Bauer, S.: Attribution of climate forcing to economic sectors, P. Natl. Acad. Sci. USA, 107, 3382–3387, doi: 10.1073/pnas.0906548107, 2010.

Unger, N., Harper, K., Zheng, Y., Kiang, N.Y., Aleinov, I., Arneth, A., Schurgers, G., Amelynck, C., Goldstein, A., Guenther, A., Heinesch, B., Hewitt, C.N., Karl, T., Laffineur, Q., Langford, B., McKinney, K.A., Misztal, P., Potosnak, M., Rinne, J., Pressley, S., Schoon, N., and Serça, D.: Photosynthesis-dependent isoprene emission from leaf to planet in a global carbon–chemistry–climate model, Atmos. Chem. Phys., 13, 10243–10269, doi: 10.5194/acp-13-10243-2013, 2013.

USDA: Indonesia: Rising global demand fuels palm oil expansion, Commodity Intelligence Report, United States Department of Agriculture Foreign Agricultural Service, 2010.

USDA: Oilseeds: World markets and trade, United States Department of Agriculture Foreign Agricultural Service, 2017.

van der Werf, G.R., Morton, D.C., DeFries, R.S., Olivier, J.G., Kasibhatla, P.S., Jackson, R.B., Collatz, G.J., and Randerson, J.T.: CO2 emissions from forest loss, Nat. Geosci., 2, 737–738, doi: 10.1038/ngeo671, 2009.

van der Werf, G.R., Randerson, J.T., Giglio, L., Collatz, G.J., Kasibhatla, P.S., and Arellano Jr., A.F.: Interannual variability in global biomass burning emissions from 1997 to 2004, Atmos. Chem. Phys., 6, 3423–3441, doi: 10.5194/acp-6-3423-2006, 2006.

Vermeulen, S. and Goad, N.: Towards better practice in smallholder palm oil production, Natural Resource Issues Series (No. 5), International Institute for Environment and Development, London, UK, 2006.

Warwick, N.J., Archibald, A.T., Ashworth, K., Dorsey, J., Edwards, P.M., Heard, D.E., Langford, B., Lee, J., Misztal, P.K., Whalley, L.K., and Pyle, J.A.: A global model study of the impact of land-use change in Borneo on atmospheric composition, Atmos. Chem. Phys., 13, 9183–9194, doi: 10.5194/acp-13-9183-2013, 2013.

Weedon, G.P., Balsamo, G., Bellouin, N., Gomes, S., Best, M.J., and Viterbo, P.: The WFDEI meteorological forcing data set: WATCH Forcing Data methodology applied to ERA-Interim reanalysis data, Water Resour. Res., 50, 7505–7514, doi: 10.1002/2014WR015638, 2014.

Wilkinson, M.J., Owen, S.M., Possell, M., Hartwell, J., Gould, P., Hall, A., Vickers, C., and Hewitt, C.N.: Circadian control of isoprene emissions from oil palm (*Elaeis guineensis*), Plant J., 47, 960–968, doi: 10.1111/j.1365- 313X.2006.02847.x, 2006.

Wolfe, G.M., Kaiser, J., Hanisco, T.F., Keutsch, F.N., de Gouw, J.A., Gilman, J.B., Graus, M., Hatch, C.D., Holloway, J., Horowitz, L.W., Lee, B.H., Lerner, B.M., Lopez-Hilifiker, F., Mao, J., Marvin, M.R., Peischl, J., Pollack, I.B., Roberts, J.M., Ryerson, T.B., Thornton, J.A., Veres, P.R., and Warneke, C.: Formaldehyde production from isoprene oxidation across NOx regimes, Atmos. Chem. Phys., 16, 2597–2610, doi: 10.5194/acp-16-2597-2016, 2016.

Wong, A.Y.H., Tai, A.P.K., and Ip, Y.-Y.: Attribution and statistical parameterization of the sensitivity of surface ozone to changes in leaf area index based on a chemical transport model, J. Geophys. Res.-Atmos., 123, 1883–1898, doi: 10.1002/2017JD027311, 2018.

WRI: CAIT Climate Data Explorer [Electronic database], World Resources Institute, http://cait.wri.org, 2015.

Yienger, J.J. and Levy II, H.: Empirical model of global soil-biogenic NOx emissions, J. Geophys. Res., 100, 11,447–11,464, doi: 10.1029/95JD00370, 1995.

Yue, X., and Unger, N.: The Yale Interactive terrestrial Biosphere model version 1.0: Description, evaluation and implementation into NASA GISS ModelE2, Geosci. Model Dev., 8, 2399–2417, doi: 10.5194/gmd-8-2399-2015, 2015.

Zhang, H., Yee, L.D., Lee, B.H., Curtis, M.P., Worton, D.R., Isaacman-VanWertz, G., Offenberg, J.H., Lewandowski, M., Kleindienst, T.E., Beaver, M.R., Holder, A.L., Lonneman, W.A., Docherty, K.S., Jaoui, M., Pye, H.O.T., Hu, W., Day, D.A., Campuzano-Jost, P., Jimenez, J.L., Guo, H., Weber, R.J., de Gouw, J., Koss, A.R., Edgerton, E.S., Brune, W., Mohr, C., Lopez-Hilfiker, F.D., Lutz, A., Kreisberg, N.M., Spielman, S.R., Hering, S.V., Wilson, K.R., Thornton, J.A., and Goldstein, A.H.: Monoterpenes are the largest source of summertime organic aerosol in the southeastern United States, P. Natl. Acad. Sci. USA, 115, 2038–2043, doi: 10.1073/pnas.1717513115, 2018.

**Table S1**. Algorithm used to map Gunarso et al. (2013) land cover types to seven PFTs and bare land.

| YIBs cover type | Gunarso et al. (2013) cover type(s) |
| --- | --- |
| Shrubland | Upland shrubland + swamp shrubland |
| Crops | Rice fields + dry cultivated land |
| C4-grassland | Upland grassland + swamp grassland |
| Dirt | Bare soil |
| Oil palm plantations | Oil palm plantations |
| Rubber plantations | Rubber plantations |
| Other tree plantations | Timber plantation + mixed tree crops / agroforest |
| Dipterocarp forest | Undisturbed upland forest + undisturbed mangrove + undisturbed swamp forest + disturbed upland forest + disturbed mangrove + disturbed swamp forest |

[Figure]

**Figure S1**. Land cover distribution for 1990 (shown only for the grid cells for which the applied land cover is derived from the classification of Gunarso et al. (2013); other grid cells are shown in gray). Cover types include dipterocarp evergreen broadleaf forest (DPT), shrubland (SHR), C4-grassland (C4G), crops (CRP), oil palm plantations (OIL), rubber plantations (RUB), other tree plantations (OTP), and bare land (BARE).

Harper, Kandice 8/25/2018 5:43 PM
**Dipterocarp forest**[a]
[Figure]
 ... [1]

[Figure]

**Figure S2.** Regional land cover change for 2005 and 2010 relative to 1990 (shown only for the grid cells for which the applied land cover is derived from the classification of Gunarso et al. (2013); other grid cells are shown in gray). Cover types include dipterocarp evergreen broadleaf forest (DPT), shrubland (SHR), C4-grassland (C4G), crops (CRP), oil palm plantations (OIL), rubber plantations (RUB), other tree plantations (OTP), and bare land (BARE).

Harper, Kandice 8/25/2018 5:47 PM

[Figure]

**Figure S3.** Annual emissions of a) isoprene and b) monoterpenes in 2010 in maritime Southeast Asia (simulation 2010land_base).

Harper, Kandice 8/25/2018 5:47 PM

[Figure]

**Figure S4.** Annual-mean surface ozone mixing ratio for 2010 (simulation 2010land_base).

Harper, Kandice 8/25/2018 5:47 PM

[Figure]

[Figure]

**Figure S5.** Change in annual emissions of a) isoprene and b) monoterpenes due to 1990–2010 maritime Southeast Asian land cover change (2010land_base – 1990land_base).

Harper, Kandice 8/25/2018 5:47 PM

[Figure]

**Figure S6.** Change in annual-mean surface SOA concentration (μg m$^{-3}$) due to 1990–2010 maritime Southeast Asian land cover change (2010land_base minus 1990land_base).

Harper, Kandice 8/25/2018 5:47 PM

**References**

Gunarso, P., Hartoyo, M.E., Agus, F., and Killeen, T.J.: Oil palm and land use change in Indonesia, Malaysia, and Papua New Guinea: Reports from the Technical Panels of the 2nd Greenhouse Gas Working Group of the Roundtable on Sustainable Palm Oil (RSPO), Roundtable on Sustainable Palm Oil, Kuala Lumpur, Malaysia, 2013.